# TRANSFORMERS LEARN TO IMPLEMENT MULTI-STEP GRADIENT DESCENT WITH CHAIN OF THOUGHT

**Jianhao Huang**[1]*, **Zixuan Wang**[2]*, **Jason D. Lee**[2]
[1]Shanghai Jiaotong University, [2]Princeton University
`huang_jh2021@sjtu.edu.cn`, {`wangzx, jasonlee`}`@princeton.edu`

## ABSTRACT

Chain of Thought (CoT) prompting has been shown to significantly improve the performance of large language models (LLMs), particularly in arithmetic and reasoning tasks, by instructing the model to produce intermediate reasoning steps. Despite the remarkable empirical success of CoT and its theoretical advantages in enhancing expressivity, the mechanisms underlying CoT training remain largely unexplored. In this paper, we study the training dynamics of transformers over a CoT objective on an in-context weight prediction task for linear regression. We prove that while a one-layer linear transformer without CoT can only implement a single step of gradient descent (GD) and fails to recover the ground-truth weight vector, a transformer with CoT prompting can learn to perform multi-step GD autoregressively, achieving near-exact recovery. Furthermore, we show that the trained transformer effectively generalizes on the unseen data. Empirically, we demonstrate that CoT prompting yields substantial performance improvements.

## 1 INTRODUCTION

Transformer-based Large Language Models (LLMs) have demonstrated significant success across various language modeling tasks, achieving state-of-the-art performance in numerous domains (OpenAI, 2023). Remarkably, these models have also unlocked complex reasoning abilities, particularly in mathematical problem-solving and coding tasks (Chowdhery et al., 2023; Anil et al., 2022; Achiam et al., 2023). A key method driving this advancement is the Chain of Thought (CoT), which enables LLMs to generate intermediate reasoning steps autoregressively rather than providing a direct answer. This process effectively improves the model's capacity to solve complex problems. In practice, CoT reasoning can be elicited either by providing few-shot CoT examples or by appending prompts like "let's think step by step" to bootstrap the model's response (Kojima et al., 2022; Wei et al., 2022; Suzgun et al., 2022; Nye et al., 2021).

Theoretically, CoT enables LLMs to perform multi-step sequential computations by generating intermediate results, thereby significantly improving the expressive power of transformers (Li et al., 2024b; Feng et al., 2024; Merrill & Sabharwal, 2023a) compared to standard decoder transformers that generate direct outputs without intermediate reasoning (Liu et al., 2022; Merrill & Sabharwal, 2023b). Despite these theoretical insights, it remains unclear how transformers are **trained** on CoT data to effectively execute multi-step reasoning. Furthermore, it is unknown whether a transformer trained specifically with an auto-regressive objective with multi-step CoT can substantially outperform one trained to directly output answers without CoT.

This paper takes an initial step beyond expressiveness to study the training dynamics of transformers when trained on CoT data. Specifically, following the modified in-context learning (ICL) setting on linear regression proposed by Ahn et al. (2023); Zhang et al. (2023), we use it as a testbed to analyze the training process with the CoT framework implemented. We name the task **in-context weight prediction** where the goal is to predict the linear weight vector from the sequence of input prompts. Instead of performing direct ICL and outputting a prediction, the transformer with CoT prompting is allowed to generate multiple intermediate steps before arriving at the final answer. We theoretically investigate the transformer's training trajectory on the CoT objective and show the expressiveness

---

*Equal Contribution (alphabetical order).

gap between transformers trained with CoT and those without. Our main results show this separation is **learnable**: gradient-based algorithm can learn the constructed transformer with CoT.

We summarize our contributions as follows:

- **Expressiveness Gap.** We characterize the global optimum of the population loss for the in-context weight prediction task on linear regression using a one-layer transformer without CoT prompting. Our results show that, without CoT, the transformer at the global minimizer effectively performs a single step of gradient descent (GD)(Theorem 3.1), leading to significant errors in predicting the $d$-dimensional weight vector $\boldsymbol{w}^* \in \mathbb{R}^d$ when the number of examples for ICL is $n = \widetilde{\Theta}(d)$ (Corollary 3.1). In contrast, we demonstrate that a one-layer transformer with CoT prompting can achieve near-exact recovery by executing multi-step GD (Theorem 3.2).

- **Convergence.** We prove the convergence results of running gradient flow on the population CoT loss under mild assumptions (Theorem 4.1). Our analysis uses a novel stage-wise approach combining dynamics analysis and landscape properties: the parameters initially approach the global minimizer, followed by local convergence toward the final solution. Our proof technique involves a novel characterization of the complicated population gradient. Furthermore, we prove that the trained transformer can exhibit both in-distribution and out-of-distribution generalization (Theorem 4.2) at inference time. We are the first to establish the learnable separation between transformers with and without CoT under the in-context linear regression setting. We empirically validate that the trained transformer converges to the minimizer predicted by our theory, with a distinct performance gap between models trained with and without CoT prompting.

**Outline.** In Section 2, we formalize the problem setting including the data model, the one-layer transformer architecture, and the CoT prompting format. In Section 3, we theoretically show the performance gap between the transformer with and without CoT. Section 4 consists of our main results, including our dynamics analysis and out-of-distribution (OOD) generalization result. Section 5 empirically validates the advantage of CoT.

## 1.1 RELATED WORKS

**Training dynamics of transformers.** Several works have studied the training process of specific transformer architectures. Jelassi et al. (2022); Li et al. (2023) examined the training process and sample complexity of Vision Transformer (Dosovitskiy et al., 2020). Tarzanagh et al. (2023); Ataee Tarzanagh et al. (2023); Li et al. (2024a) explored the connection between the optimization landscape of self-attention mechanisms and the Support Vector Machine problem. Tian et al. (2023a;c) provided insights into the training dynamics of the self-attention and MLP layers during the training process respectively.

A related line of research focuses on Markov-like data models. Bietti et al. (2024) studied the *induction head* mechanism from the perspective of associative memory. Nichani et al. (2024) demonstrated that a simplified two-layer transformer provably learns a generalized induction head on latent causal graphs. Chen et al. (2024b) further proved that a modified two-layer multi-head transformer can learn in-context generalized $n$-gram. Edelman et al. (2024) investigated the multi-stage phase transitions during training on bigram and $n$-gram ($n \geq 3$). Additionally, Makkuva et al. (2024) studied the loss landscape of transformers trained on sequences from a Markov Chain.

Another growing body of literature aims to understand the training dynamics of in-context learning (ICL). Garg et al. (2022) first empirically studied the ICL capabilities of transformers over a variety of function classes. Akyürek et al. (2022); Von Oswald et al. (2023) investigated the behavior of transformers on random ICL instances of linear regression. Several works have also established the existence of deep transformers capable of implementing multi-step gradient descent (GD) across different domains (Fu et al., 2023; Bai et al., 2023; Giannou et al., 2023). Mahankali et al. (2023); Ahn et al. (2024) analyzed the loss landscape of the linear regression ICL task and Zhang et al. (2023) proved global convergence on a one-layer linear self-attention layer using gradient flow. Gatmiry et al. (2024) demonstrated that a linear looped transformer with specific update procedures can learn to implement multi-step GD for linear regression. Further analyses of training dynamics under more realistic assumptions about data models and architectures have been conducted by Huang et al. (2023); Kim & Suzuki (2024); Chen et al. (2024a). For a detailed discussion see Appendix A.1.

Compared to prior works, our study and Huang et al. (2023); Ahn et al. (2024); Zhang et al. (2023); Tarzanagh et al. (2023); Nichani et al. (2024); Kim & Suzuki (2024); Wang et al. (2024); Chen et al. (2024b); Ren et al. (2024) all use similar reparameterizations that combine key and query matrices to simplify the training dynamics. Moreover, many previous studies (Tian et al., 2023a; Zhang et al., 2023; Huang et al., 2023; Nichani et al., 2024; Kim & Suzuki, 2024; Chen et al., 2024a; Gatmiry et al., 2024) adopted the population loss to facilitate the analysis of these dynamics.

A closely related work is Gatmiry et al. (2024), which shows that a looped transformer can implement multi-step GD on the ICL linear regression task to directly predict the query answer in context. In comparison, the goal of our setting is to predict the weight vector from the input examples using a realistic CoT autoregressive generation process. Theoretically, we also establish a performance gap between transformers with CoT and those without. See Appendix A.2 for a more detailed discussion.

**Chain of Thought and Scratchpad** The CoT prompting method was first introduced by Wei et al. (2022) to enhance the multi-step reasoning capability of LLMs. Before the formalization of CoT, Nye et al. (2021) demonstrated that allowing language models to generate intermediate results on "*scratchpads*" dramatically boosts the multi-step computation ability of LLMs. Wang et al. (2022b); Yao et al. (2024); Creswell et al. (2022); Zhou et al. (2022) further proposed variants of the CoT/scratchpad method to improve the efficiency and reliability of generation.

Recently, several works have attempted to understand CoT from both experimental and theoretical perspectives. Wang et al. (2022a); Saparov & He (2022); Shi et al. (2022); Paul et al. (2023) empirically studied the capability of CoT, providing valuable insights on its reasoning processes. Meanwhile, Wu et al. (2023); Tutunov et al. (2023); Hou et al. (2023); Cabannes et al. (2024) investigated CoT through the lens of mechanistic interpretability. On the theoretical side, Liu et al. (2022); Merrill & Sabharwal (2023a); Li et al. (2024b); Feng et al. (2024) explored the expressive power of transformers with CoT, showing that CoT can significantly extend the expressivity of transformers in the context of circuit complexity. Hu et al. (2024) investigated the statistical foundations of CoT. However, the training dynamics of CoT remain largely unexplored. To the best of our knowledge, this work is among the first theoretical analyses of training dynamics on CoT/scratchpad objectives.

## 2 PRELIMINARIES

In this section, we describe the modified in-context learning linear regression task, i.e. **in-context weight prediction**, the one-layer linear self-attention architecture, and the Chain of Thought (CoT) prompting formulation.

**Notation** We use $[T]$ to denote the set $\{1, 2, ..., T\}$. Scalars are in lower-case unbolded letters ($y$, $\alpha$, etc.). Matrices and vectors are denoted in upper-case bold letters ($\boldsymbol{W}$, $\boldsymbol{V}$, etc.) and lower-case bold letters ($\boldsymbol{x}$, $\boldsymbol{w}$, etc.), respectively. $\boldsymbol{W}_{[i,j]}, \boldsymbol{W}_{[i,:]}, \boldsymbol{W}_{[:,j]}$ respectively denotes the $(i, j)$-th entry, $i$-th row, and $j$-th column of the matrix $\boldsymbol{W}$. $\boldsymbol{W}_{[:,-1]}$ means the last column of the matrix $\boldsymbol{W}$. The notation $\boldsymbol{W}_{ij}$ denotes block matrices/vectors on the $i$-th row and $j$-th column according to context. For norm, $\|\cdot\|$ denotes $\ell_2$ norm and $\|\cdot\|_F$ denotes the Frobenius norm. We use $\mathbb{1}\{\cdot\}$ to denote the indicator function. We use $\tilde{O}(\cdot)$ to hide logarithmic factors in the asymptotic notations.

### 2.1 IN-CONTEXT WEIGHT PREDICTION

Previous works (Zhang et al., 2023; Ahn et al., 2023; 2024; Akyürek et al., 2022; Mahankali et al., 2023) focus on the in-context learning (ICL) task on linear regression. We suppose the data sequence is sampled from a linear regression task where the ground-truth

$$\boldsymbol{w}^* \sim \mathcal{N}(0, \boldsymbol{I}_d) \quad \boldsymbol{x}_i \sim \mathcal{N}(0, \boldsymbol{I}_d) \quad y_i = \boldsymbol{w}^{*\top} \boldsymbol{x}_i \text{ for all } i \in [n]. \tag{1}$$

The goal of in-context learning is to predict the correct label $\boldsymbol{w}^{*\top} \boldsymbol{x}_{\text{query}}$ given a query $\boldsymbol{x}_{\text{query}}$ and the previous example pairs $(\boldsymbol{x}_i, y_i)$. Most previous works (Zhang et al., 2023; Ahn et al., 2024; Mahankali et al., 2023) show the transformer predicts the query label $y_{\text{query}}$ by implicitly doing a one-step gradient descent without predicting the linear classifier $\boldsymbol{w}^*$.

In this work, we go one step further: instead of directly outputting the query label, we require the transformers to implement gradient descent to learn the ground-truth weight vector $\boldsymbol{w}^*$. We call this task **in-context weight prediction** for linear regression. Specifically, the data sequence is in the following format:

$$\boldsymbol{Z}_0 = \begin{bmatrix} \boldsymbol{x}_1 & \cdots & \boldsymbol{x}_n & 0 \\ y_1 & \cdots & y_n & 0 \\ 0 & \cdots & 0 & \boldsymbol{w}_0 \\ 0 & \cdots & 0 & 1 \end{bmatrix} := \begin{bmatrix} \boldsymbol{X} & 0 \\ \boldsymbol{y} & 0 \\ \boldsymbol{0}_{d\times n} & \boldsymbol{w}_0 \\ \boldsymbol{0}_{1\times n} & 1 \end{bmatrix} \in \mathbb{R}^{d_e \times (n+1)}, \tag{2}$$

where $\boldsymbol{X} := [\boldsymbol{x}_1, \cdots, \boldsymbol{x}_n]$ is the data matrix and $\boldsymbol{w}_0$ is the initialization of the linear parameter $\hat{\boldsymbol{w}}$. We assume $\boldsymbol{w}_0 = \boldsymbol{0}_d$ for simplicity, and define $d_e = 2d + 2$. Our setting is similar to the setting in Bai et al. (2023) where multi-layer transformers are constructed to do explicit multi-step GD on the weight vector $\hat{\boldsymbol{w}}$. We separate the input example space and the weight vector space as in Bai et al. (2023) (the $\{\mathbf{p}_i\}_{i\in[N+1]}$) in order to facilitate training. Moreover, we add a dummy token (an extra 1) at the end of each token similar to what Bai et al. (2023) did in their input sequence format.

## 2.2 Linear Self-attention Layer

We consider a one-layer linear self-attention (LSA) module with residual connection, following the setting in Zhang et al. (2023); Ahn et al. (2023); Gatmiry et al. (2024): we remove the softmax($\cdot$) non-linearity, consolidate the projection and value matrix into a single matrix $\boldsymbol{V} \in \mathbb{R}^{d_e \times d_e}$, and merge the key and query matrices into $\boldsymbol{W} \in \mathbb{R}^{d_e \times d_e}$. We denote

$$f_{\text{LSA}}(\boldsymbol{Z}; \boldsymbol{V}, \boldsymbol{W}) = \boldsymbol{Z} + \boldsymbol{V}\boldsymbol{Z} \cdot \frac{\boldsymbol{Z}^\top \boldsymbol{W} \boldsymbol{Z}}{n} \tag{3}$$

The prediction of the transformer will be the last token of the output sequence, namely

$$f_{\text{LSA}}(\boldsymbol{Z}; \boldsymbol{V}, \boldsymbol{W})_{[:,-1]} = \boldsymbol{Z}_{[:,-1]} + \boldsymbol{V}\boldsymbol{Z} \cdot \frac{\boldsymbol{Z}^\top \boldsymbol{W} \boldsymbol{Z}_{[:,-1]}}{n} \tag{4}$$

Since the first $(d+1)$ entries of the full weight tokens $(\boldsymbol{0}, 0, \boldsymbol{w}, 1)$ are zero, only part of the $\boldsymbol{W}$ and $\boldsymbol{V}$ affect the prediction. We can rewrite the parameter $\boldsymbol{V}, \boldsymbol{W}$ into block matrices

$$\boldsymbol{V} = \begin{bmatrix} \boldsymbol{V}_{11} & \boldsymbol{V}_{12} & \boldsymbol{V}_{13} & \boldsymbol{V}_{14} \\ \boldsymbol{V}_{21} & v_{22} & \boldsymbol{V}_{23} & v_{24} \\ \boldsymbol{V}_{31} & \boldsymbol{V}_{32} & \boldsymbol{V}_{33} & \boldsymbol{V}_{34} \\ \boldsymbol{V}_{41} & v_{42} & \boldsymbol{V}_{43} & v_{44} \end{bmatrix}, \boldsymbol{W} = \begin{bmatrix} \boldsymbol{W}_{11} & \boldsymbol{W}_{12} & \boldsymbol{W}_{13} & \boldsymbol{W}_{14} \\ \boldsymbol{W}_{21} & w_{22} & \boldsymbol{W}_{23} & w_{24} \\ \boldsymbol{W}_{31} & \boldsymbol{W}_{32} & \boldsymbol{W}_{33} & \boldsymbol{W}_{34} \\ \boldsymbol{W}_{41} & w_{42} & \boldsymbol{W}_{43} & w_{44} \end{bmatrix} \in \mathbb{R}^{(2d+2)\times(2d+2)}$$

where the block matrices are in the following shape ($i, j \in \{1, 2\}$):

$$\boldsymbol{V}_{2i-1,2j-1}, \boldsymbol{W}_{2i-1,2j-1} \in \mathbb{R}^{d\times d}; \boldsymbol{V}_{2i-1,2j}, \boldsymbol{W}_{2i-1,2j}, \boldsymbol{V}_{2i,2j-1}^\top, \boldsymbol{W}_{2i,2j-1}^\top \in \mathbb{R}^{d\times 1}; v_{2i,2j}, w_{2i,2j} \in \mathbb{R}.$$

In the following sections, we will show only $\boldsymbol{V}_{31}$, $\boldsymbol{W}_{13}$, and $w_{24}$ affects the prediction. We will further prove that all other entries are always zero along the training trajectory if initialized at zero.

## 2.3 Chain-of-Thought Prompting

In language modeling tasks, transformers have been proven to be versatile in various downstream tasks. However, transformers struggle to solve mathematical or scientific problems with one single generation, where several reasoning steps are required. CoT was then proposed to make transformers learn to generate intermediate results auto-regressively before reaching the answer.

With CoT, we allow the transformer to generate $k$ steps before it outputs the final prediction $\hat{\boldsymbol{w}}_k$ for the ground-truth $\boldsymbol{w}^*$. Specifically, given the generated input sequence $\hat{\boldsymbol{Z}}_i$ at the $i$-th step of generation, we have $f_{\text{LSA}}(\hat{\boldsymbol{Z}}_i)_{[:,-1]}$ as the prediction of the next token (($i$+1)-th token), and append it to the end of the current sequence s.t. $\hat{\boldsymbol{Z}}_{i+1} = \left[\hat{\boldsymbol{Z}}_i, f_{\text{LSA}}(\hat{\boldsymbol{Z}}_i)_{[:,-1]}\right]$. After $k$ generation steps, the CoT process induces $k$ intermediate sequences $\{\hat{\boldsymbol{Z}}_i\}_{i=1}^k$ in the following form:

$$\hat{\boldsymbol{Z}}_i = \begin{bmatrix} \boldsymbol{x}_1 & \cdots & \boldsymbol{x}_n & 0 & \star & \cdots & \star \\ y_1 & \cdots & y_n & 0 & \star & \cdots & \star \\ 0 & \cdots & 0 & \boldsymbol{w}_0 & \hat{\boldsymbol{w}}_1 & \cdots & \hat{\boldsymbol{w}}_i \\ 0 & \cdots & 0 & 1 & 1 & \cdots & 1 \end{bmatrix} \in \mathbb{R}^{d_e\times(n+i+1)}, i \in [k] \tag{Inference}$$

Here, we define $\hat{\boldsymbol{w}}_i := f_{\text{LSA}}(\hat{\boldsymbol{Z}}_{i-1})_{[d+2:2d+1,-1]}$ as the $i$-th step prediction for the weight vector. The other entries in the same column are irrelevant and we denote them as $\star$. Finally, the transformer inputs the last generated sequence $\hat{\boldsymbol{Z}}_k$ back to the transformer once again to generate the final output $\hat{\boldsymbol{w}}_{k+1} := f_{\text{LSA}}(\hat{\boldsymbol{Z}}_k)_{[d+2:2d+1,-1]}$ as the prediction of the weight vector $\boldsymbol{w}^*$.

Different from the inference time generation, the training process is similar to pre-training on the ground-truth sequence to predict the next token. Specifically, we input the transformer with CoT ground-truth sequences $\boldsymbol{Z}_i$:

$$\boldsymbol{Z}_i = \begin{bmatrix} \boldsymbol{x}_1 & \cdots & \boldsymbol{x}_n & 0 & 0 & \cdots & 0 \\ y_1 & \cdots & y_n & 0 & 0 & \cdots & 0 \\ 0 & \cdots & 0 & \boldsymbol{w}_0 & \boldsymbol{w}_1 & \cdots & \boldsymbol{w}_i \\ 0 & \cdots & 0 & 1 & 1 & \cdots & 1 \end{bmatrix} \in \mathbb{R}^{d_e \times (n+i+1)}, i \in [k] \qquad \text{(Training)}$$

where $\boldsymbol{w}_i = \boldsymbol{w}_{i-1} - \eta \cdot \frac{\boldsymbol{X}(\boldsymbol{X}^\top \boldsymbol{w}_{i-1} - \boldsymbol{y}^\top)}{n}$ is the ground-truth intermediate weight vector after $i$ gradient steps on the linear regression objective. Each gradient step adopts a fixed learning rate $\eta$ for all possible training instances $\{\boldsymbol{X}, \boldsymbol{w}\}$ when generating the ground-truth sequence $\boldsymbol{Z}_i$. Note that $\boldsymbol{Z}_i$ is the corresponding ground-truth sequence of $\hat{\boldsymbol{Z}}_i$.

In the training objective for the $i$-th step, the transformer is required to predict the next token $\boldsymbol{Z}_{i+1[:,-1]} := (\boldsymbol{0}_d, 0, \boldsymbol{w}_{i+1}, 1)$ given the $i$-th ground-truth intermediate sequence $\boldsymbol{Z}_i$. Finally, we predict the final ground-truth weight vector $\boldsymbol{w}^*$ with the final intermediate sequence $\boldsymbol{Z}_k$. The CoT **training** objective given a sample prompt $\boldsymbol{X}, \boldsymbol{y}$ then becomes:

$$\ell^{\text{CoT}}(\boldsymbol{X}, \boldsymbol{w}^*; \boldsymbol{V}, \boldsymbol{W}) = \frac{1}{2} \sum_{i=0}^{k} \left\| f_{\text{LSA}}(\boldsymbol{Z}_i)_{[:,-1]} - (\boldsymbol{0}_d, 0, \boldsymbol{w}_{i+1}, 1) \right\|^2 \qquad (5)$$

Here we denote $\boldsymbol{w}_{k+1} := \boldsymbol{w}^*$ for clarity. Following Zhang et al. (2023); Nichani et al. (2024); Kim & Suzuki (2024); Tian et al. (2023b); Chen et al. (2024a); Gatmiry et al. (2024), we consider the gradient flow dynamics over the population loss of the CoT objective:

$$\mathcal{L}^{\text{CoT}}(\boldsymbol{V}, \boldsymbol{W}) = \mathbb{E}_{\boldsymbol{x}_i \sim \mathcal{N}(0, \boldsymbol{I}_d), \boldsymbol{w}^* \sim \mathcal{N}(0, \boldsymbol{I}_d)} \left[ \ell^{\text{CoT}}(\boldsymbol{X}, \boldsymbol{w}^*; \boldsymbol{V}, \boldsymbol{W}) \right] \qquad (6)$$

For clarity, we write the expectation as $\mathbb{E}_{\boldsymbol{X}, \boldsymbol{w}^*}[\cdot]$. The following differential equation gives the gradient flow dynamics of the parameters:

$$\frac{\mathrm{d}\boldsymbol{\theta}}{\mathrm{d}t} = -\nabla \mathcal{L}^{\text{CoT}}(\boldsymbol{\theta}), \quad \boldsymbol{\theta} := (\boldsymbol{V}, \boldsymbol{W}).$$

When measuring the performance after training, we apply the CoT **inference** procedure to generate $k$ intermediate sequences $\{\hat{\boldsymbol{Z}}_i\}_{i=1}^k$ and consider the final output token $f(\hat{\boldsymbol{Z}}_k)_{[:,-1]}$ by inputting the last generated sequence $\hat{\boldsymbol{Z}}_k$. The performance evaluation is measured on the error between the final output $f(\hat{\boldsymbol{Z}}_k)_{[:,-1]}$ and the ground-truth $\boldsymbol{w}^*$:

$$\mathcal{L}^{\text{Eval}}(\boldsymbol{V}, \boldsymbol{W}) = \frac{1}{2} \mathbb{E}_{\boldsymbol{X}, \boldsymbol{w}^*} \left[ \left\| f_{\text{LSA}}(\hat{\boldsymbol{Z}}_k)_{[:,-1]} - (\boldsymbol{0}_d, 0, \boldsymbol{w}^*, 1) \right\|^2 \right] \qquad (7)$$

When CoT prompting is not used ($k = 0$), the evaluation loss $\mathcal{L}^{\text{Eval}}$ is equivalent to $\mathcal{L}^{\text{CoT}}$.

## 3    EXPRESSIVENESS IMPROVEMENT WITH CHAIN OF THOUGHT

In this section, we theoretically explore the performance gap on our data model between transformers with CoT and those without. We first prove that a one-layer transformer without CoT can only implement a one-step GD and cannot recover the ground-truth, while it can near-exactly predict the ground-truth parameter with CoT by implementing multi-step GD.

### 3.1    ONE-LAYER TRANSFORMER CANNOT RECOVER GROUND-TRUTH

For the ICL linear regression task, the optimal prediction given by a one-layer linear transformer is equivalent to a single step of GD on the MSE objective of linear regression (Mahankali et al., 2023). What about our task on predicting the ground-truth weight vector $\boldsymbol{w}^*$ in context? The following theorem proves that the optimal solution is still a one-step GD solution.

**Theorem 3.1** (Lower bound without CoT). *If the global minimizer of $\mathcal{L}^{\text{Eval}}(\boldsymbol{V}, \boldsymbol{W})$ is $(\boldsymbol{V}^*, \boldsymbol{W}^*)$, the corresponding one-layer transformer $f_{\text{LSA}}(\boldsymbol{Z}_0)_{[:,-1]}$ implements one step GD on a linear model with some learning rate $\eta^* = \frac{n}{n+d+1}$ and the transformer outputs $(\boldsymbol{0}_d, 0, \frac{\eta^*}{n} \boldsymbol{X} \boldsymbol{y}^\top, 1)$.*

We briefly present the high-level intuitions in the proof and the detailed proof is deferred to Appendix B.1. We use a similar technique in Mahankali et al. (2023) when proving the optimality of one-step GD in the ICL task. The key strategy of the proof is to replace $(\mathbf{0}_d, 0, \boldsymbol{w}^*, 1)$ in the evaluation loss $\mathcal{L}^{\mathrm{Eval}}(\boldsymbol{V}, \boldsymbol{W})$ (Equation (7)) with $(\mathbf{0}_d, 0, \frac{\eta^*}{n}\boldsymbol{X}\boldsymbol{y}^\top, 1)$ in the following form.

$$\mathcal{L}^{\mathrm{Eval}}(\boldsymbol{V}, \boldsymbol{W}) = \frac{1}{2}\mathbb{E}\left[\left\|f_{\mathrm{LSA}}(\boldsymbol{Z}_0)_{[:,-1]} - \left(\mathbf{0}_d, 0, \frac{\eta^*}{n}\boldsymbol{X}\boldsymbol{y}^\top, 1\right)\right\|^2\right] + C$$

In order to prove this equation above, we show the gradient of the original loss Equation (7) and this formula are identical. We first obtain the closed-form formula of the expected gradient for both sides with regard to $\boldsymbol{X}, \boldsymbol{w}^*$. Then we use the symmetric property of the distribution of $\boldsymbol{X}, \boldsymbol{w}^*$ to simplify the gradient expressions, and eventually prove them equal.

The equivalent form of loss indicates that the evaluation loss only depends on the $\ell_2$ distance between the output of the linear self-attention module and $\left(\mathbf{0}_d, 0, \frac{\eta^*}{n}\boldsymbol{X}\boldsymbol{y}^\top, 1\right)$. Therefore, any $(\boldsymbol{V}, \boldsymbol{W})$ is a global minimizer of this loss function if and only if the output of $f_{\mathrm{LSA}}(\boldsymbol{Z}_k)_{[:,-1]}$ is $(\mathbf{0}_d, 0, \frac{\eta^*}{n}\boldsymbol{X}\boldsymbol{y}^\top, 1)$. Meanwhile, one can assign

$$\boldsymbol{V}^* = \begin{bmatrix} \mathbf{0} & \mathbf{0} & \mathbf{0} & \mathbf{0} \\ \mathbf{0} & \mathbf{0} & \mathbf{0} & \mathbf{0} \\ -\eta^*\boldsymbol{I} & \mathbf{0} & \mathbf{0} & \mathbf{0} \\ \mathbf{0} & \mathbf{0} & \mathbf{0} & \mathbf{0} \end{bmatrix}, \boldsymbol{W}^* = \begin{bmatrix} \mathbf{0} & \mathbf{0} & \boldsymbol{I} & \mathbf{0} \\ \mathbf{0} & \mathbf{0} & \mathbf{0} & -1 \\ \mathbf{0} & \mathbf{0} & \mathbf{0} & \mathbf{0} \\ \mathbf{0} & \mathbf{0} & \mathbf{0} & \mathbf{0} \end{bmatrix} \tag{8}$$

and the one-layer transformer achieves the optimal solution, which concludes the proof.

Is a one-step gradient solution good enough? Most of the previous ICL work Zhang et al. (2023); Ahn et al. (2023); Gatmiry et al. (2024) consider the number of examples $n \to +\infty$ when $d$ is fixed. In this case, the one-step GD solution can perfectly find the ground-truth weight vector $\boldsymbol{w}^*$. However, a simple corollary of this theorem indicates that the one-step solution has a non-negligible error when there are limited samples, e.g. $n = \widetilde{\Theta}(d)$. This number of examples $n$ is required to guarantee the reconstruction of $\boldsymbol{w}^* \in \mathbb{R}^d$.

**Corollary 3.1.** *For any parameters $(\boldsymbol{V}, \boldsymbol{W})$ in the one-layer transformer, $\mathcal{L}^{\mathrm{Eval}}(\boldsymbol{V}, \boldsymbol{W}) \geq \Theta\left(\frac{d^2}{n}\right)$. Moreover, if $n = \tilde{\Theta}(d), \mathcal{L}^{\mathrm{Eval}}(\boldsymbol{V}, \boldsymbol{W}) = \tilde{\Theta}(d) \xrightarrow{d \to +\infty} +\infty$.*

*Proof.* By Theorem 3.1, we directly calculate the evaluation loss on the global optimum:

$$\mathcal{L}^{\mathrm{Eval}}(\boldsymbol{V}, \boldsymbol{W}) \geq \frac{1}{2}\mathbb{E}_{\boldsymbol{X},\boldsymbol{w}^*}\left\|\frac{\eta^*}{n}\boldsymbol{X}\boldsymbol{X}^\top\boldsymbol{w}^* - \boldsymbol{w}^*\right\|^2 = \frac{1}{2}\mathbb{E}_{\boldsymbol{X}}\operatorname{tr}\left(\boldsymbol{I} - \frac{\eta^*}{n}\boldsymbol{X}\boldsymbol{X}^\top\right)^2$$

since $\mathbb{E}_{\boldsymbol{w}^*}\left[\boldsymbol{w}^*\boldsymbol{w}^{*\top}\right] = \boldsymbol{I}$. Apply $\mathbb{E}\left[\boldsymbol{X}\boldsymbol{X}^\top\right] = n\boldsymbol{I}$ and $\mathbb{E}\left[(\boldsymbol{X}\boldsymbol{X}^\top)^2\right] = n(n+d+1)\boldsymbol{I}$,

$$\frac{1}{2}\mathbb{E}_{\boldsymbol{X}}\operatorname{tr}\left(\boldsymbol{I} - \frac{\eta^*}{n}\boldsymbol{X}\boldsymbol{X}^\top\right)^2 = \frac{1}{2}\left(d - 2\eta^*d + \frac{\eta^{*2}}{n}(n+d+1)d\right) = \Theta\left(\frac{d^2}{n}\right)$$

and we finish the proof by substituting $n$ with $\tilde{\Theta}(d)$. $\square$

## 3.2 One-layer Transformer with CoT Can Implement Multi-step GD

The previous subsection shows that the one-step solution by the one-layer transformer without CoT is not the endgame. Nevertheless, CoT can become the savior for this simple transformer because it enables the transformer to generate several intermediate computation steps to improve the final performance. The following theorem shows that with the reinforcement of CoT, there exists a one-layer transformer that can perform multi-step GD using intermediate generations. We show that $\Theta(\log d)$ steps of CoT can remarkably improve the performance, reducing the error from $\Theta(\frac{d}{\mathrm{poly}\log d})$ to $O(1/\mathrm{poly}\,d)$. With constant learning rate, $\Theta(\log d)$ steps of GD is also necessary to reconstruct $\boldsymbol{w}^*$ accurately. The proof is deferred to Appendix B.2.

**Theorem 3.2** (Informal). *There exists $V^*$ and $W^*$ s.t. $f_{\mathrm{LSA}}(Z_k)_{[:,-1]}$ outputs $(0_d, 0, w_k, 1)$ where $w_k := \left(I - (I - \frac{\eta}{n} XX^\top)^k\right) w^*$ is the $k$-step GD solution with learning rate $\eta$ on a linear regression model. Moreover, if $n = \tilde{\Omega}(d)$, $k = \Omega(\log d)$, $\eta \in (0.1, 1)$, then the evaluation loss*

$$\mathcal{L}^{\mathrm{Eval}}(V^*, W^*) = \frac{1}{2} \mathbb{E}_{X, w^*} \left[ \left\| \left(I - \frac{\eta}{n} XX^\top\right)^{k+1} w^* \right\|^2 \right] \le O\left(\frac{1}{\mathrm{poly}(d)}\right) \tag{9}$$

With the one-step GD solution in Theorem 3.1, the proof is straightforward: we assign the parameters $(V, W)$ in the same form of Equation (8), with the $\eta^*$ replaced by $\eta$. However, now the transformer is allowed to generate $k$ steps before reaching the final output. We can inductively calculate the $i$-th step of generation, showing that the output is exactly the $i$-th gradient step:

$$f_{\mathrm{LSA}}(Z_{i-1})_{[:,-1]} = (0_d, 0, w_i, 1), \quad i = 1, 2, ..., k+1$$

After $k+1$ steps, we have the final output $\left(I - (I - \frac{\eta}{n} XX^\top)^{k+1}\right) w^*$ by induction and the evaluation loss becomes Equation (9). By Lemma D.4, the final loss is upper bounded by $O\left(\frac{1}{\mathrm{poly}(d)}\right)$. This is strictly better than a one-step GD solution by comparing with Corollary 3.1.

Now we theoretically display the expressivity improvement of transformers brought by CoT. In the following sections, we will further prove that **this separation is learnable** simply by gradient flow.

## 4 GRADIENT DYNAMICS OVER CHAIN OF THOUGHT

In this section, we go beyond the construction and prove our convergence result on the CoT objective. We show that the final solution found by gradient flow is approximately our construction in Theorem 3.2, which is significantly better than the one-step gradient descent solution without CoT.

### 4.1 MAIN RESULTS

According to our construction in Theorem 3.2, we use the following specific initialization to zero out the irrelevant blocks while keeping the essential blocks $W_{13}$, $V_{31}$, and $w_{24}$.

**Assumption 4.1** (Initialization). *Let $\sigma > 0$ be a parameter. We assume the initialization of the parameters satisfies that*

$$V = \begin{bmatrix} 0 & 0 & 0 & 0 \\ 0 & 0 & 0 & 0 \\ V_{31}(0) & 0 & 0 & 0 \\ 0 & 0 & 0 & 0 \end{bmatrix}, W = \begin{bmatrix} 0 & 0 & W_{13}(0) & 0 \\ 0 & 0 & 0 & w_{24} \\ 0 & 0 & 0 & 0 \\ 0 & 0 & 0 & 0 \end{bmatrix}$$

*Here $W_{13}(0) = \sum_{i=1}^{d} \lambda_i^W u_i u_i^\top$ and $V_{31}(0) = \sum_{i=1}^{d} \lambda_i^V u_i u_i^\top$ are symmetric and simultaneously diagonalizable, $\lambda_i^V \le -\sigma, \lambda_i^W \in [\sigma, \frac{1}{2}]$. Further, we fix $w_{24} = -1$ for all $t > 0$.*

This initialization follows Chen et al. (2024a) by assuming $V_{31}$ and $W_{13}$ share the same set of eigenvectors. It is close to the particular symmetric random initialization schemes discussed in Zhang et al. (2023) with a scaling factor $\sigma$. We use this specific initialization to zero out the irrelevant blocks, and we fix $w_{24} = -1$ to break the homogeneity of the model and avoid multiple global minimizers. Those simplification facilitates the analysis of the complex dynamical system.

Now we prove that under appropriate initialization, gradient flow will nearly converge to the global minimizer. We provide a proof sketch in the next subsection. See Appendix C.3 for details.

**Theorem 4.1** (Informal, Global Convergence). *Suppose $n = \tilde{\Omega}(d)$, $\eta \in (0.1, 0.9)$, $k = \Theta(\log d)$. Under Assumption 4.1 with $\sigma = \Theta(1)$, if we run gradient flow on the population loss in Equation (5), then after time $t = O\left(\log d + \log \frac{1}{\epsilon}\right)$, we have $\mathcal{L}^{\mathrm{CoT}}(t) \le \epsilon$ for any $\epsilon \in \left(\frac{1}{\mathrm{poly}(d)}, 1\right)$.*

### 4.2 PROOF IDEAS

In this subsection, we briefly outline the proof of Theorem 4.1.

Before analyzing the dynamics, we first prove that under Assumption 4.1, the gradient dynamics only depend on the parameter blocks $W_{13}(t), V_{31}(t), w_{24}$, while other blocks stay zero (Lemma C.2). This is because the Gaussian data assumption makes sure the gradients of these

blocks are zero once initialized at zero, except for $\boldsymbol{W}_{13}(t), \boldsymbol{V}_{31}(t), w_{24}$. By this lemma, we can simplify the linear self-attention formula and consider the following equivalent yet simplified loss (we denote $\widetilde{\boldsymbol{W}} := \boldsymbol{W}_{13}, \widetilde{\boldsymbol{V}} := \boldsymbol{V}_{31}$, and $w_{24}$ is fixed as $-1$.):

$$\mathcal{L}^{\text{CoT}}(\boldsymbol{\theta}) = \frac{1}{2} \mathbb{E}_{\boldsymbol{X}, \boldsymbol{w}^*} \sum_{i=0}^{k-1} \left\| \frac{1}{n} (\widetilde{\boldsymbol{V}} \boldsymbol{X} \boldsymbol{X}^\top \widetilde{\boldsymbol{W}} + \eta \boldsymbol{X} \boldsymbol{X}^\top) \boldsymbol{w}_i - \frac{1}{n} (\widetilde{\boldsymbol{V}} + \eta \boldsymbol{I}) \boldsymbol{X} \boldsymbol{X}^\top \boldsymbol{w}^* \right\|_2^2$$

$$+ \frac{1}{2} \mathbb{E}_{\boldsymbol{X}, \boldsymbol{w}^*} \left\| \left( \boldsymbol{I} + \frac{1}{n} \widetilde{\boldsymbol{V}} \boldsymbol{X} \boldsymbol{X}^\top \widetilde{\boldsymbol{W}} \right) \boldsymbol{w}_k - \left( \frac{1}{n} \widetilde{\boldsymbol{V}} \boldsymbol{X} \boldsymbol{X}^\top + \boldsymbol{I} \right) \boldsymbol{w}^* \right\|_2^2$$

For ease of presentation, we denote $\boldsymbol{S} := \frac{1}{n} \boldsymbol{X} \boldsymbol{X}^\top$. To analyze the gradient dynamics, we first need to compute the exact closed-form gradient instead of keeping the expectation. However, the formula involves the $i$-th step weight vector $\boldsymbol{w}_i = \left( \boldsymbol{I} - (\boldsymbol{I} - \eta \boldsymbol{S})^i \right) \boldsymbol{w}^*$, involving the higher order moments of the Wishart matrix $\boldsymbol{S}$[1] whose closed form is hard to obtain. In our paper, we provide a tighter estimate compared to previous work (Gatmiry et al., 2024) using the concentration of the Wishart matrix $\boldsymbol{S}$ (Vershynin, 2018) when $n = \Theta(d \operatorname{poly} \log d)$ to estimate the expectation. In particular, we use the exponential decaying tail probability bound for the operator norm of the error $\delta \boldsymbol{S} := \boldsymbol{S} - \boldsymbol{I}$. For example, when estimating the expectation $\mathbb{E}[(\boldsymbol{I} - \eta \boldsymbol{S})^i]$, we can decompose the expectation into two cases: when $\|\delta \boldsymbol{S}\|_{op}$ is small, $(\boldsymbol{I} - \eta \boldsymbol{S})^i \approx (1 - \eta)^i \boldsymbol{I}$; when $\|\delta \boldsymbol{S}\|$ is larger than a threshold, the rest part of the expectation can be controlled by integrating the exponential decaying tail probability.[2] The concentration lemmas are provided in Appendix D.

The motivation behind a better concentration estimation is to ensure nearly independent dynamics along different eigenspaces $\{\boldsymbol{u}_i\}_{i=1}^d$ of $\widetilde{\boldsymbol{W}}$ and $\widetilde{\boldsymbol{V}}$. As an extreme case, we consider $n \to \infty$ and $\boldsymbol{S}$ converges to $\boldsymbol{I}$ almost surely. Now the gradient component on the $\boldsymbol{u}_i \boldsymbol{u}_i^\top$ subspace is only dependent on $\lambda_i^{\widetilde{\boldsymbol{V}}}$ and $\lambda_i^{\widetilde{\boldsymbol{W}}}$ without any other $\lambda_j^{\widetilde{\boldsymbol{V}}}, \lambda_j^{\widetilde{\boldsymbol{W}}}, j \neq i$ involved. That means there is no interaction between two different subspaces, i.e. the dynamics are independent. However, some interactions are introduced since the concentration error $\delta \boldsymbol{S} \neq 0$ when $n$ is finite. Therefore, the improved characterization of the expected gradient is essential to upper bound the interaction between the dynamics of different eigenspaces $\{\boldsymbol{u}_i\}_{i=1}^d$, leading to a nearly independent evolution at initialization. This independence property motivates us to conduct the following stage-wise analysis:

**Stage 1: $\widetilde{\boldsymbol{W}}, \widetilde{\boldsymbol{V}}$ converges to near-optimal.** In this stage, the dynamics along each direction $\boldsymbol{u}_i$ stay nearly independent. Specifically, we can expand the gradient flow dynamics for $\widetilde{\boldsymbol{V}}, \widetilde{\boldsymbol{W}}$ and project them into the eigenspaces $\boldsymbol{u}_i \boldsymbol{u}_i^\top$ to get the dynamics of the eigenvalues $\lambda_i^{\widetilde{\boldsymbol{V}}} := \boldsymbol{u}_i^\top \widetilde{\boldsymbol{V}} \boldsymbol{u}_i$, $\lambda_i^{\widetilde{\boldsymbol{W}}} := \boldsymbol{u}_i^\top \widetilde{\boldsymbol{W}} \boldsymbol{u}_i$. The dynamics of eigenvalues are characterized by the following Lemma 4.1 where we can prove that the interaction terms between different subspaces are bounded by $O(1/\log^2 d)$.

**Lemma 4.1** (Informal version of Lemma C.6). *The dynamics of $\lambda_i^{\widetilde{\boldsymbol{V}}}$ and $\lambda_i^{\widetilde{\boldsymbol{W}}}$ are given by the following equations with $\left| \delta_j^{\widetilde{\boldsymbol{V}}} \right| \leq O\left( \frac{1}{\log^2 d} \right), \left| \delta_j^{\widetilde{\boldsymbol{W}}} \right| \leq O\left( \frac{1}{\log^2 d} \right)$:*

$$\frac{\mathrm{d} \lambda_j^{\widetilde{\boldsymbol{V}}}}{\mathrm{d} t} = - \left[ (k+1) \left( 1 - \lambda_j^{\widetilde{\boldsymbol{W}}} \right)^2 + \frac{2}{\eta} \lambda_j^{\widetilde{\boldsymbol{W}}} \left( 1 - \lambda_j^{\widetilde{\boldsymbol{W}}} \right) + \frac{\lambda_j^{\widetilde{\boldsymbol{W}}^2}}{\eta(2-\eta)} \right] \lambda_j^{\widetilde{\boldsymbol{V}}} + \frac{1-\eta}{2-\eta} \lambda_j^{\widetilde{\boldsymbol{W}}} - 1 + \delta_j^{\widetilde{\boldsymbol{V}}}$$

$$\frac{\mathrm{d} \lambda_j^{\widetilde{\boldsymbol{W}}}}{\mathrm{d} t} = \left[ k + 1 - \frac{1}{\eta} \right] \lambda_j^{\widetilde{\boldsymbol{V}}^2} \left( 1 - \lambda_j^{\widetilde{\boldsymbol{W}}} \right) + \frac{1-\eta}{\eta(2-\eta)} \lambda_j^{\widetilde{\boldsymbol{V}}^2} \lambda_j^{\widetilde{\boldsymbol{W}}} + \frac{1-\eta}{2-\eta} \lambda_j^{\widetilde{\boldsymbol{V}}} - \delta_j^{\widetilde{\boldsymbol{W}}}.$$

This nearly independent evolution along each eigenvector $\boldsymbol{u}_i$ enables us to analyze the individual dynamics of $\lambda_i^{\widetilde{\boldsymbol{V}}}$ and $\lambda_i^{\widetilde{\boldsymbol{W}}}$ at the beginning of training. Under Assumption 4.1, $\lambda_j^{\widetilde{\boldsymbol{V}}}, \lambda_j^{\widetilde{\boldsymbol{W}}}$ are initialized $\Theta(1)$. By Lemma 4.1, we prove by induction that the eigenvalues will go through two phases: (1) $\lambda_j^{\widetilde{\boldsymbol{V}}}$

---

[1]To deal with the similar problem, Gatmiry et al. (2024) proposed a simple combinatorial method to estimate the expectation. We use the same technique to get a certain form of the expectation (see Appendix D), but the bound is not tight enough to get the desired results. See discussion in Appendix A.2.

[2]This method can keep the $(1-\eta)^i$ factor to prevent introducing unwanted estimation errors when $i$ is large.

increases yet stay smaller than $-O\left(\frac{1}{k(1-\lambda_j^{\widetilde{W}})}\right)$, while $\lambda_j^{\widetilde{W}}$ increases to $1-o(1)$. (2) $\lambda_j^{\widetilde{W}}$ stays $o(1)$-close to 1, and $\lambda_j^{\widetilde{V}}$ also converges to $o(1)$-close to $-\eta$. Here all $o(1)$ terms are some $O(1/\log^c d)$ terms for some constant $c > 0$. That indicates that the distance between the eigenvalues and the target $\left|\lambda_j^{\widetilde{V}} + \eta\right|, \left|\lambda_j^{\widetilde{W}} - 1\right|$ converge to $O(1/\log^c d)$ for all $j \in [d]$ at the end of Stage 1.

**Stage 2: Local convergence.** One may expect that after Stage 1, the transformer can approximate gradient steps quite accurately since the parameter $\widetilde{V}, \widetilde{W}$ are both $o(1)$-close to ground-truth along each direction $u_i$. Unfortunately, the sum of error in $d$ directions can still be $\widetilde{\Theta}(d)$ since we can only reduce the error to $O(1/\operatorname{poly}\log d)$ in each direction. So for now, the solution cannot recover the weight vector $w^*$ at this stage. To address this issue, we further consider the exact form of the interaction terms $\delta_j^{\widetilde{W}}, \delta_j^{\widetilde{V}}$ and analyze the local convergence. By fine-grained expansion of the error terms, we notice that $\delta_j^{\widetilde{W}}$ and $\delta_j^{\widetilde{V}}$ are always coupled with some individual residual like $(1 - \lambda_j^{\widetilde{W}})$, $(\eta + \lambda_j^{\widetilde{V}})$, or some weighted average or those individual residuals. Meanwhile, the coefficient of the residual in the interaction terms is still upper bounded by $O(1/\operatorname{poly}\log d)$. That enables us to derive some gradient lower bound similar to PL-conditions (Lemma C.12) when $\widetilde{V}, \widetilde{W}$ are close to the ground-truth, leading to local convergence to near-optimal at a linear rate.

The final training error is some $O(\frac{1}{\operatorname{poly} d})$, which depends on the inference step $k$ and ground-truth $\eta$. Note that the optimal loss value is also at least polynomially small in $d$ given $\Theta(\log d)$ CoT steps. Therefore, now we can conclude that the transformer can learn to implement multi-step GD when given intermediate ground-truth states after optimizing the CoT loss with gradient flow.

### 4.3 OUT-OF-DISTRIBUTION GENERALIZATION AT INFERENCE

In this section, we prove that after training, the transformer not only correctly predicts the weight vector in context with CoT generation, but also can generalize out-of-distribution (OOD). The following theorem shows that the trained transformer obtained from Theorem 4.1 with CoT generalizes over other problem instances when the input example sequence has an OOD covariance, as long as the covariance is not too ill-conditioned. Here $\mathcal{L}_{\Sigma}^{\text{Eval}}$ is defined as the OOD evaluation loss in eq. (7) with the in-context examples $x_i \sim \mathcal{N}(0, \Sigma)$ and weight vector $w^* \sim \mathcal{N}(0, I)$:

$$\mathcal{L}_{\Sigma}^{\text{Eval}}(V, W) = \frac{1}{2}\mathbb{E}_{x_i \sim \mathcal{N}(0,\Sigma), w^*}\left[\left\|f_{\text{LSA}}(\hat{Z}_k)_{[:,-1]} - (\mathbf{0}_d, 0, w^*, 1)\right\|^2\right]$$

**Theorem 4.2** (Informal, Theorem C.2). *Suppose $n = \tilde{\Omega}(d)$, $\eta \in (0.1, 0.9)$, $k' = \Theta(\log d)$. Assume the out-of-distribution covariance is well-conditioned: $\frac{\delta}{\eta} \leq \lambda_{\min}(\Sigma) \leq \lambda_{\max}(\Sigma) \leq \frac{2-\delta}{\eta}$ for some constant $\delta > 0$. Then after training in Theorem 4.1, we have $\mathcal{L}_{\Sigma}^{\text{Eval}}(t) \leq \epsilon$ for any $\epsilon \in \left(\frac{1}{\operatorname{poly}(d)}, 1\right)$.*

Note that this theorem covers both in-distribution (when $\eta = \delta$) and OOD tasks at evaluation, indicating that the transformer is trained to implement a general iterative optimization algorithm. Moreover, the inference step number $k'$ in this theorem can go beyond the training CoT steps $k$, achieving better estimation for $w^*$.

One may think once the next-token-prediction training loss $\mathcal{L}^{\text{CoT}}$ converges to zero based on ground-truth CoT data, the transformer naturally learns to do multi-step reasoning at inference, i.e. $\mathcal{L}^{\text{Eval}}$ is small. However, at the $i$-th generation step, the transformer is predicting the next weight token $\hat{w}_{i+1}$ based on the previous generation $\hat{w}_i$ instead of the ground-truth intermediate step $w_i$. It is possible that prediction error for each step accumulates or even increases exponentially. In this theorem, we expand the sum of all the prediction errors at each step and show a converging series of errors throughout the inference process. That ensures we can achieve any $O(\frac{1}{\operatorname{poly}(d)})$-small evaluation loss when we have $k' = \Theta(\log d)$ reasoning steps. The detailed proof is provided in Appendix C.4.

## 5 EXPERIMENTS

In this section, we introduce our experimental setup on our in-context weight vector prediction task to numerically validate our theoretical results. Specifically, we show that parameters of the

transformer match the prediction of our theory when optimized over the CoT loss. Furthermore, we present the gap of evaluation loss $\mathcal{L}^{\mathrm{Eval}}$ in Equation (7) between transformers with and without CoT.

**Experimental Setup** We train the transformer architecture in Equation (3) on the synthetic data. The data distribution follows our in-context weight prediction task in Equation (1). In particular, we choose the token dimensions $d = 10$, number of in-context examples $n = 20$, and GD learning rate $\eta = 0.4$ for generating the ground-truth intermediate states. We use a batch size $B = 1000$ and run Adam with learning rate $\alpha = 0.001$ for $\tau = 750$ iterations. More details refer to Appendix E.

**Global convergence** Our experiments show that the structure that weights of the full model exhibit is consistent with Theorem 3.2. At final convergence, all of the entries of $\boldsymbol{W}$ converge to zero except the elements on the diagonal in the top-right corner block (the red box in the heatmap of $\boldsymbol{W}$, Figure 1), while all the entries of $\boldsymbol{V}$ are near zero except elements on the diagonal in the bottom-left corner (the red box in the heatmap of $\boldsymbol{V}$, Figure 1). Also, the pattern shows $\boldsymbol{W}_{13} = \alpha\boldsymbol{I}, w_{24} = -\alpha$, and $\boldsymbol{V}_{31} = -\frac{\eta}{\alpha}\boldsymbol{I}$ with some scaling factor $\alpha$,[3] which is equivalent to the construction stated in Theorem 3.2 and Theorem 4.1. That means the transformer implements one step of gradient descent $\left(\boldsymbol{0}_d, 0, -\frac{\eta}{n}\boldsymbol{X}\boldsymbol{X}^{\top}(\boldsymbol{w}_i - \boldsymbol{w}^*), 0\right)$ before the residual connection, and the autoregressive CoT process enables model to perform multi-step GD.

**Performance improvement** We empirically verify the evaluation loss gap between transformers with and without CoT shown by Theorem 3.1 and Theorem 3.2. Our experiments in Figure 2 demonstrate that the evaluation loss of transformers with CoT converges to near zero even when $k = 10$. In comparison, the optimal expected loss that the one-layer linear transformer can achieve (the pink dashed line, from Corollary 3.1) is much larger than any of the model that applies multiple steps of computation. We also observe that evaluation loss at convergence keeps decreasing when the number of reasoning steps $k$ increases from $10$ to $40$, which is consistent with Theorem C.1 where larger $k$ allows for smaller error $\epsilon$.

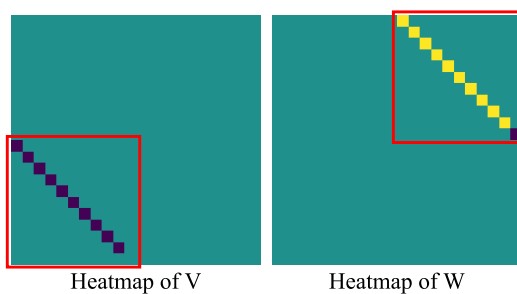

Heatmap of V        Heatmap of W

Figure 1: **Model weights:** We present the heatmap of the weights of the trained transformer. We initialize $\boldsymbol{V}, \boldsymbol{W}$ randomly at $t = 0$, where $n = 20$, $d = 10$ and $k = 20$. After training, all entries of $\boldsymbol{V}$ and $\boldsymbol{W}$ converge to zero except the two blocks highlighted in the red box. Moreover, the pattern matches the theoretical results.

Figure 2: $k$-**step v.s. 1-step:** We plot the evaluation loss $\mathcal{L}^{\mathrm{Eval}}$ when $n = 20$, $d = 10$. We randomly initialize the transformer. For transformers with CoT, loss converges to near zero while transformers without CoT cannot. Moreover, the loss at convergence decreases when $k$ increases.

# 6 CONCLUSION

This paper investigates the training dynamics of transformers when the Chain of Thought (CoT) prompting is introduced. By focusing on the in-context weight prediction task, our theoretical results demonstrate that transformers can learn to implement iterative algorithms like multi-step GD with the enhancement of CoT, highlighting the essential role of CoT in multi-step reasoning tasks. Our empirical findings corroborate these theoretical insights, indicating that CoT prompting provides significant performance benefits.

There are still many open problems. Can we move beyond population loss on the in-context weight prediction task and show a sample complexity guarantee? Can CoT empower the transformer to acquire compositional reasoning capability instead of doing the same iterative steps?

---

[3]In Figure 1, $\alpha > 0$ while all $\alpha \neq 0$ works for the construction. Empirically, the sign of $\alpha$ depends on the random initialization, and both positive and negative solutions exist.

ACKNOWLEDGEMENT

JDL acknowledges support of the NSF CCF 2002272, NSF IIS 2107304, NSF CIF 2212262, ONR Young Investigator Award, and NSF CAREER Award 2144994.

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

# A    DISCUSSION AND LIMITATION

## A.1    RELATED WORKS ON EXPRESSIVENESS

Our work is closely related to the previous works in multi-step GD using multi-layer attention layers, including Bai et al. (2023); Fu et al. (2023); Ding et al. (2023); Ahn et al. (2024); Giannou et al. (2023); Gatmiry et al. (2024). These works guarantee that transformers are **expressive enough** to do in-context learning by implementing gradient descent, and they serve as the foundation of our work which focuses on **optimization**. Most of them focus on the in-context learning setup as the testbed so we naturally follow the setup to understand the advantage of CoT.

Most of the above works on **expressiveness** focus on those iterative algorithms, e.g. (pre-conditioned) gradient descent on various objectives (Bai et al., 2023; Ahn et al., 2024; Ding et al., 2023), Newton methods/matrix inverse (Giannou et al., 2023), etc. Those papers have similar constructive proof techniques using multi-layer transformers: they construct a basic block(s) to represent one step of some iterative algorithm and stack them up to do multi-steps of that algorithm. Sometimes the blocks can be even the same, which means a "looped" transformer, i.e. implementing the same transformer blocks several times as a loop, can express those algorithms. In our warm-up construction for a better understanding of the setup, we use similar techniques to construct the linear transformer that allows auto-regressive generation to iteratively implement the block. However, we require the practical auto-regressive setting, which is novel in the literature.

Most importantly, despite the close relation between our work and those previous expressiveness papers, our work mainly focuses on the **optimization** perspective. It is a big step beyond expressiveness because there is no guarantee that one can algorithmically find the constructed solutions in the previous work. Ahn et al. (2024); Gatmiry et al. (2024) are the only two papers related to optimization of multi-layer transformers over in-context linear regression setup. Ahn et al. (2024) analyzed the global optimizer/critical points for multi-layer transformers, but they didn't prove that any gradient-based algorithm can reach those solutions. Compared to all the works above, our proof techniques for the main theorems are completely orthogonal and **not** straightforward extensions of the previous papers like Bai et al. (2023).

Gatmiry et al. (2024) is the most related work to us. They also proved some results on **learning** to implement multi-step GD by looped transformer. We will highlight the differences and **our novel contributions** of our work in the next section.

## A.2    DISCUSSION ON GATMIRY ET AL. (2024)

In this section, we compare our work with Gatmiry et al. (2024). We begin by outlining the similarities and connections between the two works before highlighting our theoretical contributions in contrast to Gatmiry et al. (2024).

Both Gatmiry et al. (2024) and our study analyze the dynamics of a one-layer linear transformer in the context of a linear regression task, demonstrating that transformers can implement multi-step gradient descent. We adopt similar architectural frameworks to those in Zhang et al. (2023); Ahn et al. (2024; 2023); Mahankali et al. (2023), as well as several other works. The key connection between our work and Gatmiry et al. (2024) lies in the observation that both looped transformers and transformers with CoT prompting through autoregressive generation are capable of naturally implementing iterative algorithms like gradient descent.

However, our data model and training objective are intrinsically different from those in Gatmiry et al. (2024), leading to distinct insights. While Gatmiry et al. (2024) focuses on an ICL setting for linear regression tasks involving examples and a query, our task is centered on predicting the ground-truth weight vector $w^*$ within context, i.e. in-context weight prediction. The final converging solutions are totally different, even though they both are equivalent to some type of GD. From the perspective of the training objective, Gatmiry et al. (2024) uses a standard squared loss over the ICL objective. In contrast, we use a sum of squared losses across all intermediate steps, corresponding to the CoT loss defined in Equation (6). Therefore, we highlight the effectiveness in improving the performance of the CoT prompting on a shallow transformer, while Gatmiry et al. (2024) stress a multi-layer transformer with shared weights (looped transformer) can do multi-step GD through the layers.

From a technical perspective, Gatmiry et al. (2024) fix the outer layer and train only the matrix $\boldsymbol{A}$, which is analogous to our matrix $\boldsymbol{W}$. In contrast, our work allows for training both layers of the transformer, providing a stronger analysis of training dynamics. Our proof strategy is also novel, given that our training dynamics are more complicated: obtaining our final solution requires solving a challenging $d$-dimensional dynamical system, whereas prior work in ICL reduces the outer layer to a scalar.

**As a more profound theoretical contribution, we rigorously establish a clear performance gap between the one-layer transformer without CoT and the ones with CoT.** Specifically, the one-layer transformer without CoT is restricted to a single step of GD, with the final error $\Theta(d/\operatorname{poly}\log d)$, while a one-layer transformer with CoT can achieve a $O(1/\operatorname{poly} d)$ loss with only $\Theta(\log d)$ steps. On the other hand, Gatmiry et al. (2024) do not show their transformer implementing the multi-step GD can outperform the transformer with one-step GD. According to Theorem 4.2, the looped transformer in their setting can only provably get the final loss down to $\frac{d^{5/2} L \cdot 4^L}{\sqrt{n}}$, where $L$ is the number of loops. However, a one-layer transformer can achieve $\Theta(d^2/n)$ loss by implementing one-step of GD, **asymptotically better than the multi-step solution in Gatmiry et al. (2024)**.

### A.3 Looped Transformer Learns to Implement Multi-step Gradient Descent (Improved Version)

In this section, we also discuss how our estimation technique improves the result in Gatmiry et al. (2024) for looped transformers. The gap between our analysis lies in our different methods of calculating the terms in the gradient concerning Wishart matrices. For intuition, we introduce **the novel expectation calculation method** in Section 4, which asymptotically improves the estimation of higher moments of Wishart matrices in Gatmiry et al. (2024). We adopt the combinatorial technique in Gatmiry et al. (2024) to compute the form of $\mathbb{E}\left[\boldsymbol{S}\boldsymbol{\Lambda}\boldsymbol{S}^k\boldsymbol{\Gamma}\boldsymbol{S}^{k'}\right]$, but when we calculate the expected gradient we use the concentration tail bound technique to calculate the expectation.

We will demonstrate that applying our techniques proves looped transformers outperform their counterparts without looping in the ICL setting. Here, we follow the notations, models and loss defined in Gatmiry et al. (2024). We focus on the simple setting with $\boldsymbol{\Sigma} = \boldsymbol{I}$, but the same technique should also be applied to cases with other covariance matrices.

**Definition A.1** (Corollary A.4 in Gatmiry et al. (2024))**.** *The loss for loop Transformer is*

$$\mathcal{L}(\boldsymbol{A}, \boldsymbol{0}) = \mathbb{E}\left[\operatorname{tr}\left(\boldsymbol{I} - \boldsymbol{A}^{\frac{1}{2}}\boldsymbol{S}\boldsymbol{A}^{\frac{1}{2}}\right)^{2L}\right].$$

By using our techniques, we strengthen the conclusion and obtain the following theorem.

**Theorem A.1.** *Suppose $n = \Omega(dL^2\log^2 d)$ and $\|\boldsymbol{A}(0)\| = O(1)$. Consider the gradient flow with respect to the loss $\mathcal{L}(\boldsymbol{A}, \boldsymbol{0})$:*

$$\frac{\mathrm{d}}{\mathrm{d}t}\boldsymbol{A}(t) = -\nabla_{\boldsymbol{A}}\mathcal{L}(\boldsymbol{A}(t), \boldsymbol{0}).$$

*Then, for any $\xi > \Theta\left(\left(\frac{L^2 d\log^2 d}{n}\right)^L\right)$, after time $t \geq \Omega\left(\frac{1}{L^2}\left(\frac{d}{\xi}\right)^{\frac{L-1}{L}}\right)$, we have $\mathcal{L}(\boldsymbol{A}(t), \boldsymbol{0}) \leq \xi$. In particular, given any polynomially small $\xi > \Theta(\frac{1}{\operatorname{poly}(d)})$, there exists $L = \Theta(\log d)$, the final loss $\mathcal{L}(\boldsymbol{A}(t), \boldsymbol{0}) \leq \xi$.*

**Remark.** This result also gets arbitrary polynomially small loss as the case in the CoT setting, establishing the separation between looped transformer and the transformer without loop.

*Proof.* Denote $\lambda_i$ to be the $i$-th eigenvalue of $\boldsymbol{A}$ and $\boldsymbol{u}_i$ to be the corresponding eigenvector. Let

$$k = \arg\max_i |1 - \lambda_i|, \quad \lambda := \lambda_k.$$

Suppose $\boldsymbol{u} \in \arg\max_{\boldsymbol{u}_i, i \in [d]} |1 - \lambda_i|$ is the corresponding eigenvector. Apply Lemma A.3, we have

$$\frac{\mathrm{d}\boldsymbol{A}}{\mathrm{d}t} = 2L(\boldsymbol{I} - \boldsymbol{A})^{2L-1} + \|\boldsymbol{I} - \boldsymbol{A}\|^{2L-1}\Delta_{2L}.$$

Multiply $\boldsymbol{u}$ on both sides (note that $\boldsymbol{u}$ is a fixed direction, so the time-differential is zero), we obtain

$$\frac{\mathrm{d}(\boldsymbol{A}\boldsymbol{u})}{\mathrm{d}t} = 2L(\boldsymbol{I} - \boldsymbol{A})^{2L-1}\boldsymbol{u} + \|\boldsymbol{I} - \boldsymbol{A}\|^{2L-1}\Delta_{2L}\boldsymbol{u}.$$

Note that the gradient of $\boldsymbol{A}$ has the same eigenvectors $\boldsymbol{u}_i$ as $\boldsymbol{A}$. Therefore, $\Delta_{2L}$ have the same eigenvector as $\boldsymbol{A}$ and we have

$$\frac{\mathrm{d}(1 - \lambda)}{\mathrm{d}t} = \frac{\mathrm{d}\boldsymbol{u}^\top(\boldsymbol{I} - \boldsymbol{A})\boldsymbol{u}}{\mathrm{d}t} = -2L(1 - \lambda)^{2L-1} - (1 - \lambda)^{2L-1}\boldsymbol{u}^\top\Delta_{2L}\boldsymbol{u}.$$

By Lemma A.3, we have $\|\Delta_{2L}\| \leq O(\frac{L}{\log d}) \leq L$. The dynamics of $(1 - \lambda)^2$ become:

$$\frac{\mathrm{d}(1 - \lambda)^2}{\mathrm{d}t} = -4L(1 - \lambda)^{2L} - 2(1 - \lambda)^{2L}\boldsymbol{u}^\top\Delta_{2L}\boldsymbol{u} \leq -3L(1 - \lambda)^{2L}.$$

We can therefore upper bound the difference between $\lambda$ and $1$ by solving the ODE:

$$(1 - \lambda(t))^2 \leq \left((1 - \lambda(0))^{2-2L} + 3L(2L - 1)t\right)^{-\frac{1}{L-1}}.$$

When the training time $t \geq \Omega\left(\frac{1}{L^2}\left(\frac{d}{\xi}\right)^{\frac{L-1}{L}}\right)$, the largest eigenvalue of $\boldsymbol{I} - \boldsymbol{A}$, i.e.

$$\|\boldsymbol{I} - \boldsymbol{A}\|^2 = O\left(\left(\frac{\xi}{d}\right)^{1/L}\right).$$

To satisfy the condition on Lemma A.3, the $\xi$ should be lower bounded by $\Theta\left(d\left(\frac{L^2 d \log^2 d}{n}\right)^L\right)$.

Now consider the loss expression:

$$\begin{aligned}
\mathcal{L}(\boldsymbol{A}, \boldsymbol{0}) &= \mathbb{E}\left[\mathrm{tr}\left(\boldsymbol{I} - \boldsymbol{A}^{\frac{1}{2}}\boldsymbol{S}\boldsymbol{A}^{\frac{1}{2}}\right)^{2L}\right] \\
&= \mathbb{E}\left[\mathrm{tr}\left((\boldsymbol{I} - \boldsymbol{S}\boldsymbol{A})^{2L}\right)\right] \\
&= \mathrm{tr}\,\mathbb{E}\left((\boldsymbol{I} - \boldsymbol{S}\boldsymbol{A})^{2L}\right) \\
&= \mathrm{tr}\left((\boldsymbol{I} - \boldsymbol{A})^{2L} + \|\boldsymbol{I} - \boldsymbol{A}\|^{2L}\Delta\right) \qquad \text{(By Lemma A.2)} \\
&\leq 2d\|\boldsymbol{I} - \boldsymbol{A}\|^{2L} \leq O(\xi).
\end{aligned}$$

By the computation above, $\mathcal{L}(\boldsymbol{A}, \boldsymbol{0}) \leq \Theta(\xi)$. In particular, when $L = c\log d$ and $n \geq 2(L^2 d \log^2 d)$ with $c > 1$, the loss is smaller than $O(d^{c-1})$. Hence, only $O(\log d)$ steps of looping can achieve arbitrary polynomially small loss $O(\frac{1}{\mathrm{poly}(d)})$. $\qquad\square$

**Remark.** If we have $\Theta(\log d)$ steps of GD using this looped transformer, we can get an arbitrary polynomially small loss. It is a huge improvement compared to Gatmiry et al. (2024), and this result successfully establishes a separation between the looped transformer and the ones without the loop.

**Lemma A.1** (Equation (24) in Gatmiry et al. (2024)). *The derivative of the loss can be written as*

$$\nabla_{\boldsymbol{A}}\mathcal{L}(\boldsymbol{A}, \boldsymbol{0}) = -\sum_{i=0}^{2L-1}\mathbb{E}\left[(\boldsymbol{I} - \boldsymbol{S}\boldsymbol{A})^i\boldsymbol{S}(\boldsymbol{I} - \boldsymbol{S}\boldsymbol{A})^{2L-1-i}\right].$$

**Lemma A.2.** *Assume $n = \tilde{\Omega}(dL^2)$, $\|\boldsymbol{I} - \boldsymbol{A}\|_{op} \geq \Theta\left(\sqrt{\frac{L^2 d \log^2 d}{n}}\right)$.*

$$\mathbb{E}\left[(\boldsymbol{I} - \boldsymbol{S}\boldsymbol{A})^i\boldsymbol{S}(\boldsymbol{I} - \boldsymbol{S}\boldsymbol{A})^{2L-1-i}\right] = (\boldsymbol{I} - \boldsymbol{A})^{2L-1} + \|\boldsymbol{I} - \boldsymbol{A}\|^{2L-1}\Delta$$

*where $\Delta$ has $O\left(\frac{1}{\log d}\right)$-operator norm.*

*Proof.* Denote $\delta \boldsymbol{S} = \boldsymbol{S} - \boldsymbol{I}$. Then we expand the term in the expectation:

$$(\boldsymbol{I} - \boldsymbol{S}\boldsymbol{A})^i \boldsymbol{S} (\boldsymbol{I} - \boldsymbol{S}\boldsymbol{A})^{2L-1-i} = (\boldsymbol{I} - \boldsymbol{A} - \delta \boldsymbol{S} \cdot \boldsymbol{A})^i (\boldsymbol{I} + \delta \boldsymbol{S})(\boldsymbol{I} - \boldsymbol{A} - \delta \boldsymbol{S} \cdot \boldsymbol{A})^{2L-1-i}$$

Now take expectation to both sides. Note that $\mathbb{E}[\delta \boldsymbol{S}] = \boldsymbol{0}$, so all terms containing first order $\delta \boldsymbol{S}$ vanish. We denote

$$\|\boldsymbol{I} - \boldsymbol{A}\|^{2L-1} \widetilde{\Delta} = (\boldsymbol{I} - \boldsymbol{S}\boldsymbol{A})^i \boldsymbol{S} (\boldsymbol{I} - \boldsymbol{S}\boldsymbol{A})^{2L-1-i} - (\boldsymbol{I} - \boldsymbol{A})^i (\boldsymbol{I} + \delta \boldsymbol{S})(\boldsymbol{I} - \boldsymbol{A})^{2L-1-i}$$
$$- i(\boldsymbol{I} - \boldsymbol{A})^{i-1} \cdot \delta \boldsymbol{S} \cdot \boldsymbol{A}(\boldsymbol{I} - \boldsymbol{A})^{2L-1-i} - (2L-1-i)(\boldsymbol{I} - \boldsymbol{A})^{2L-2} \cdot \delta \boldsymbol{S} \cdot \boldsymbol{A}$$

to be the sum of all higher order terms (the degree of $\delta \boldsymbol{S} \geq 2$). We can estimate the expectation using similar technique as in Lemma D.1.

$$\|\widetilde{\Delta}\|_{op} \leq \sum_{k=2}^{2L-1} \binom{2L-1}{k} \frac{\left\|(\boldsymbol{I} - \boldsymbol{A})^{2L-1-k}(\delta \boldsymbol{S} \cdot \boldsymbol{A})^k\right\|}{\|\boldsymbol{I} - \boldsymbol{A}\|^{2L-1}} \qquad \text{(Term 1)}$$

$$+ \sum_{k=1}^{2L-1-i} \binom{2L-1-i}{k} \frac{\left\|(\boldsymbol{I} - \boldsymbol{A})^i \delta \boldsymbol{S}(\boldsymbol{I} - \boldsymbol{A})^{2L-1-i-k}(\delta \boldsymbol{S} \cdot \boldsymbol{A})^k\right\|}{\|\boldsymbol{I} - \boldsymbol{A}\|^{2L-1}} \qquad \text{(Term 2)}$$

$$+ \sum_{k=1}^{i} \sum_{l=0}^{2L-1-i} \binom{i}{k}\binom{2L-1-i}{l} \frac{\left\|(\boldsymbol{I} - \boldsymbol{A})^{i-k}(\delta \boldsymbol{S} \cdot \boldsymbol{A})^k \delta \boldsymbol{S}(\boldsymbol{I} - \boldsymbol{A})^{2L-1-i-l}(\delta \boldsymbol{S} \cdot \boldsymbol{A})^l\right\|}{\|\boldsymbol{I} - \boldsymbol{A}\|^{2L-1}}$$

$$\text{(Term 3)}$$

To get an estimate of the operator norm, we bound each term (Term 1 to Term 3) respectively.

**Term 1:**

$$\sum_{k=2}^{2L-1} \binom{2L-1}{k} \frac{\left\|(\boldsymbol{I} - \boldsymbol{A})^{2L-1-k}(\delta \boldsymbol{S} \cdot \boldsymbol{A})^k\right\|}{\|\boldsymbol{I} - \boldsymbol{A}\|^{2L-1}} \leq \sum_{k=2}^{2L-1} \left((2L-1)\frac{\|\delta \boldsymbol{S}\| \cdot \|\boldsymbol{A}\|}{\|\boldsymbol{I} - \boldsymbol{A}\|}\right)^k$$

Note that $\|\delta \boldsymbol{S}\|$ is of order $O(\sqrt{\frac{d}{n}})$ with high probability, the term in the middle is less than 1 and the dominating term of the error is $\left((2L-1)\frac{\|\delta \boldsymbol{S}\| \cdot \|\boldsymbol{A}\|}{\|\boldsymbol{I} - \boldsymbol{A}\|}\right)^2 = O(\frac{1}{\log d})$.

**Term 2:**

$$\sum_{k=1}^{2L-1-i} \binom{2L-1-i}{k} \frac{\left\|(\boldsymbol{I} - \boldsymbol{A})^i \delta \boldsymbol{S}(\boldsymbol{I} - \boldsymbol{A})^{2L-1-i-k}(\delta \boldsymbol{S} \cdot \boldsymbol{A})^k\right\|}{\|\boldsymbol{I} - \boldsymbol{A}\|^{2L-1}}$$
$$\leq \frac{\|\boldsymbol{I} - \boldsymbol{A}\|}{\|\boldsymbol{A}\|} \sum_{k=1}^{2L-1-i} \left((2L-1-i)\frac{\|\delta \boldsymbol{S}\| \cdot \|\boldsymbol{A}\|}{\|\boldsymbol{I} - \boldsymbol{A}\|}\right)^{k+1}$$

**Term 3:**

$$\sum_{k=1}^{i} \sum_{l=0}^{2L-1-i} \binom{i}{k}\binom{2L-1-i}{l} \frac{\left\|(\boldsymbol{I} - \boldsymbol{A})^{i-k}(\delta \boldsymbol{S} \cdot \boldsymbol{A})^k \delta \boldsymbol{S}(\boldsymbol{I} - \boldsymbol{A})^{2L-1-i-l}(\delta \boldsymbol{S} \cdot \boldsymbol{A})^l\right\|}{\|\boldsymbol{I} - \boldsymbol{A}\|^{2L-1}}$$
$$\leq \frac{\|\boldsymbol{I} - \boldsymbol{A}\|}{\|\boldsymbol{A}\|} \sum_{k=1}^{i} \left(i\frac{\|\delta \boldsymbol{S}\| \cdot \|\boldsymbol{A}\|}{\|\boldsymbol{I} - \boldsymbol{A}\|}\right)^k \cdot \sum_{l=0}^{2L-1-i} \left((2L-1-i)\frac{\|\delta \boldsymbol{S}\| \cdot \|\boldsymbol{A}\|}{\|\boldsymbol{I} - \boldsymbol{A}\|}\right)^{l+1}$$

Now we upper bound the operator norm of the error term $\Delta := \mathbb{E}\left[\widetilde{\Delta}\right]$:

$$\|\Delta\| = \left\|\mathbb{E}\left[\widetilde{\Delta}\right]\right\| = \mathbb{E}\left[\|\widetilde{\Delta}\|\right]$$
$$= \mathbb{E}\left[\|\widetilde{\Delta}\|\left(\mathbb{1}\left\{\|\widetilde{\Delta}\| \leq \frac{C'}{\log d}\right\} + \mathbb{1}\left\{\|\widetilde{\Delta}\| > \frac{C'}{\log d}\right\}\right)\right]$$
$$\leq \frac{C'}{\log d} + \int_{\frac{C'}{\log d}}^{\infty} \Pr\left[\|\widetilde{\Delta}\| \geq s\right] \mathrm{d}s.$$

When $\|\widetilde{\Delta}\| \geq s$ where $s \geq \frac{C'}{\log d}$, we can first upper bound the $\|\widetilde{\Delta}\|$ with $\|\delta S\|$ using : there exists some constant $C_1 > 0$ s.t.

$$\|\widetilde{\Delta}\| \leq \max\left((C_1 L\|\delta S\|)^2, (C_1 L\|\delta S\|)^{2L+4}\right).$$

Therefore, when $\|\widetilde{\Delta}\| \geq s$, $\|\delta S\| \geq \min\{\frac{s^{1/2}}{C_1 L}, \frac{s^{1/(2L+4)}}{C_1 L}\}$. To apply the tail bound, we need to make sure we pick some $s'$ such that $\max\left(\delta, \delta^2\right) \leq \min\{\frac{s^{1/2}}{C_1 L}, \frac{s^{1/(2L+4)}}{C_1 L}\}$ to upper bound the integral of probability, where $\delta = C(\sqrt{\frac{d}{n}} + \frac{s'}{\sqrt{n}})$. Now since $s > \frac{C'}{\log d}$, $\min\{\frac{s^{1/2}}{C_1 L}, \frac{s^{1/(2L+4)}}{C_1 L}\} \geq C_\alpha \frac{1}{\log^{\frac{3}{2}} d}$ for some constant $C_\alpha$. Therefore, we just need $\max\{\frac{s'}{\sqrt{n}}, \frac{s'^2}{n}\} \leq \min\{\frac{s^{1/2}}{C_1 L}, \frac{s^{1/(2L+4)}}{C_1 L}\}$, i.e. $s' \leq \min\left\{C_2 \frac{s^{1/(2L+4)}\sqrt{n}}{L}, C_3 \frac{s^{1/(4L+8)}\sqrt{n}}{\sqrt{L}}, C_4 \frac{\sqrt{sn}}{L}, C_5 \frac{s^{1/4}\sqrt{n}}{\sqrt{L}}\right\}.$

Applying the tail bound (18) with $s' = \min\left\{C_2 \frac{s^{1/(2L+4)}\sqrt{n}}{L}, C_3 \frac{s^{1/(4L+8)}\sqrt{n}}{\sqrt{L}}, C_4 \frac{\sqrt{sn}}{L}, C_5 \frac{s^{1/4}\sqrt{n}}{\sqrt{L}}\right\}$ where $C_2, C_3, C_4, C_5$ are some constant, we have the error term for the tail expectation,

$$\int_{\frac{C'}{\log d}}^{\infty} \Pr\left[\|\widetilde{\Delta}\| \geq s\right] \mathrm{d}s \leq \int_{\frac{C'}{\log d}}^{\infty} \Pr\left[\|\delta S\| \geq \min\left\{\frac{s^{1/2}}{C_1 L}, \frac{s^{1/(2L+4)}}{C_1 L}\right\}\right] \mathrm{d}s$$

$$\leq 2\int_{\frac{C'}{\log d}}^{\infty} \exp\{-s'^2\} \mathrm{d}s.$$

Now we estimate the upper bound of error with

$$s'^2 = \min\left\{C_2^2 \cdot \frac{s^{1/(L+2)}}{L^2}n, C_3^2 \cdot \frac{s^{1/(2L+4)}}{L}n, C_4^2 \cdot \frac{sn}{L^2}, C_5^2 \cdot \frac{\sqrt{s}n}{L}\right\}.$$

For the first term, let $x = \frac{C_2^2 n}{L^2} s^{1/(L+2)}$:

$$2\int_{\frac{C'}{\log d}}^{\infty} \exp\left\{-C_2^2 \cdot \frac{s^{1/(L+2)}}{L^2}n\right\} \mathrm{d}s$$

$$= 2(L+2)\int_{\frac{C_2^2 n}{L^2}\left(\frac{C'}{\log d}\right)^{\frac{1}{L+2}}}^{\infty} \left(\frac{L^2}{C_2^2 n}\right)^{L+2} \exp\{-x\}x^{L+1} \mathrm{d}x$$

$$\leq 2(L+2)\cdot \left(\frac{L^2}{C_2^2 n}\right)^{L+2} \cdot \left(\frac{C_2^2 n}{L^2}\left(\frac{C'}{\log d}\right)^{\frac{1}{L+2}}\right)^{L+1} \exp\left\{-\frac{C_2^2 n}{L^2}\left(\frac{C'}{\log d}\right)^{\frac{1}{L}}\right\} \leq \frac{1}{\log d}.$$

The second term, let $x = C_3^2 \cdot \frac{s^{1/(2L+4)}}{L}n$:

$$2\int_{\frac{C'}{\log d}}^{\infty} \exp\left\{-C_3^2 \cdot \frac{s^{1/(2L+4)}}{L}n\right\} \mathrm{d}s$$

$$= 4(L+2)\int_{\frac{C_3^2 n}{L}\left(\frac{C'}{\log d}\right)^{\frac{1}{2L+4}}}^{\infty} \left(\frac{L}{C_3^2 n}\right)^{2L+4} \exp\{-x\}x^{2L+3} \mathrm{d}x$$

$$\leq 4(L+2)\cdot \left(\frac{L}{C_3^2 n}\right)^{2L+4} \cdot \left(\frac{C_3^2 n}{L}\left(\frac{C'}{\log d}\right)^{\frac{1}{2L+4}}\right)^{2L+3} \exp\left\{-\frac{C_3^2 n}{L}\left(\frac{C'}{\log d}\right)^{\frac{1}{2L+4}}\right\} \leq \frac{1}{\log d}.$$

For the third term, let $x = \frac{C_4^2 sn}{L^2}$:

$$2\int_{\frac{C'}{\log d}}^{\infty} \exp\left\{-\frac{C_4^2 sn}{k^2}\right\} \mathrm{d}s = 2\int_{\frac{C'}{\log d}\cdot\frac{C_4^2 n}{L^2}}^{\infty} \frac{L^2}{C_4^2 n} \exp\{-x\} \mathrm{d}x$$

$$\leq \frac{2L^2}{C_4^2 n} \exp\left\{-\frac{C'}{\log d}\cdot\frac{C_4^2 n}{L^2}\right\} \leq \frac{1}{\log d}.$$

The fourth term, let $x = C_5^2 \cdot \frac{s^{1/2}}{L} n$:

$$2 \int_{\frac{C'}{\log d}}^{\infty} \exp\left\{-C_5^2 \cdot \frac{s^{1/2}}{L} n\right\} ds$$

$$= \frac{4L^2}{n^2 C_5^4} \int_{C_5^2 \frac{n}{L}\left(\frac{C'}{\log d}\right)^{1/2}}^{\infty} \exp\{-x\} x\, dx$$

$$\leq \frac{4L^2}{n^2 C_5^4} \cdot C_5^2 \frac{n}{L}\left(\frac{C'}{\log d}\right)^{1/2} \exp\left\{-C_5^2 \frac{n}{L}\left(\frac{C'}{\log d}\right)^{1/2}\right\} \leq \frac{1}{\log d}.$$

Therefore, we plug this error back into the upper bound of $\|\Delta\|$:

$$\|\Delta\| \leq \frac{C'}{\log d} + \int_{\frac{C'}{\log d}}^{\infty} \Pr\left[\|\widetilde{\Delta}\| \geq s\right] ds$$

$$\leq \frac{C'}{\log d} + \int_{\frac{C'}{\log d}}^{\infty} \Pr\left[\|\delta \boldsymbol{S}\| \geq \min\left\{\frac{s^{1/2}}{C_1 k}, \frac{s^{1/(2L+4)}}{C_1 k}\right\}\right] ds \leq O\left(\frac{1}{\log d}\right).$$

$\square$

**Lemma A.3.** *Assume $n = \tilde{\Omega}(dL^2)$, for any $\boldsymbol{A}$ that $\|\boldsymbol{I} - \boldsymbol{A}\| \geq \Theta\left(\sqrt{\frac{L^2 d \log^2 d}{n}}\right)$, we have the following gradient estimate:*

$$\nabla_{\boldsymbol{A}} \mathcal{L}(\boldsymbol{A}, \boldsymbol{0}) = -2L(\boldsymbol{I} - \boldsymbol{A})^{2L-1} - \|\boldsymbol{I} - \boldsymbol{A}\|^{2L-1} \Delta_{2L}.$$

*where $\|\Delta_{2L}\| \leq O\left(\frac{L}{\log d}\right)$.*

*Proof.* We use our technique to estimate the derivative of the loss. By Lemma A.1, we have

$$\nabla_{\boldsymbol{A}} \mathcal{L}(\boldsymbol{A}, \boldsymbol{0}) = -\sum_{i=0}^{2L-1} \mathbb{E}\left[(\boldsymbol{I} - \boldsymbol{S}\boldsymbol{A})^i \boldsymbol{S}(\boldsymbol{I} - \boldsymbol{S}\boldsymbol{A})^{2L-1-i}\right].$$

Apply Lemma A.2 to each term in the summation, we have

$$\mathbb{E}\left[(\boldsymbol{I} - \boldsymbol{S}\boldsymbol{A})^i \boldsymbol{S}(\boldsymbol{I} - \boldsymbol{S}\boldsymbol{A})^{2L-1-i}\right] = (\boldsymbol{I} - \boldsymbol{A})^{2L-1} + \|\boldsymbol{I} - \boldsymbol{A}\|^{2L-1} \Delta_i.$$

where $\Delta_i$ has $O\left(\frac{1}{\log d}\right)$-operator norm. Denote $\Delta_{2L} = \sum_{i=0}^{2L-1} \Delta_i$ and add all terms together, we obtain

$$\nabla_{\boldsymbol{A}} \mathcal{L}(\boldsymbol{A}, \boldsymbol{0}) = -2L(\boldsymbol{I} - \boldsymbol{A})^{2L-1} - \|\boldsymbol{I} - \boldsymbol{A}\|^{2L-1} \Delta_{2L}.$$

where $\Delta_{2L}$ has $O\left(\frac{L}{\log d}\right)$-operator norm. $\square$

## A.4 LIMITATION AND FUTURE DIRECTIONS

**Architecture and parameterization** In this work, we use the single-layer linear transformer to analyze the training dynamics. Moreover, we adopt the same reparameterization and similar initialization in previous works (Zhang et al., 2023; Tian et al., 2023a; Chen et al., 2024a; Mahankali et al., 2023; Ahn et al., 2024). It deviates from the practical softmax attention with $\boldsymbol{Q}, \boldsymbol{K}, \boldsymbol{V}$ parameterization and random initialization, which is a limitation of this work.

However, analyzing the linear counterpart of the model before targeting the more difficult practical models is common in the development of learning theory. As for linear attention, the connection between linear attention and softmax attention is also partially justified by the empirical observations in Ahn et al. (2023). Analyzing the dynamics using more practical architectures will be a very important and fundamental future direction.

**Population loss and sample complexity** Following most of the previous work, we use population loss when analyzing the training trajectory instead of using finite sample loss. This modification is to simplify the analysis and focus on the population dynamics without noise. A possible future step is to generalize this analysis to a finite sample setting and train the model with online SGD.

**CoT on iterative tasks** In this work, we mainly focus on **iterative** tasks, one of the simplest forms where multi-step CoT can help yield better performance. That serves as the initial step towards understanding why CoT helps reasoning following the first principle. As a limitation, though CoT can empower the transformer to acquire compositional reasoning capability instead of doing the same iterative step, it is a much harder question beyond our paper's scope. It is a very important future direction and definitely worth further exploring.

# B PROOFS OF THEOREMS IN SECTION 3

In this section, we prove the expressiveness results of the linear transformers with and without CoT. In Appendix B.1, we prove that a one-layer linear transformer without CoT can only obtain the one-step gradient descent solution. In Appendix B.2, we prove that there exists a one-layer linear transformer that implements multi-step gradient descent with the CoT prompting. As corollaries, there exists a separation between the one-step and multi-step solutions.

## B.1 PROOF OF THEOREM 3.1

We first restate the theorem:

**Theorem B.1** (Lower bound without CoT). *If the global minimizer of $\mathcal{L}^{\mathrm{Eval}}(\boldsymbol{V}, \boldsymbol{W})$ is $(\boldsymbol{V}^*, \boldsymbol{W}^*)$, the corresponding one-layer transformer $f_{\mathrm{LSA}}(\boldsymbol{Z}_0)_{[:,-1]}$ implements one step GD on a linear model with some learning rate $\eta = \frac{n}{n+d+1}$ and the transformer outputs $\frac{\eta}{n}\boldsymbol{X}\boldsymbol{y}^\top$.*

*Proof.* Recall the loss expression in Equation (5) when $k = 0$,

$$\mathcal{L}(\boldsymbol{V}, \boldsymbol{W}) = \frac{1}{2}\mathbb{E}_{\boldsymbol{X}, \boldsymbol{w}^*} \left\| f_{\mathrm{LSA}}(\boldsymbol{Z}_0)_{[:,-1]} - (\boldsymbol{0}_d, 0, \boldsymbol{w}^*, 1) \right\|^2$$

$$= \frac{1}{2}\mathbb{E}_{\boldsymbol{X}, \boldsymbol{w}^*} \left\| \boldsymbol{V}\boldsymbol{Z}_0 \cdot \frac{\boldsymbol{Z}_0^\top \boldsymbol{W}\boldsymbol{Z}_{0[:,-1]}}{n} - (\boldsymbol{0}_d, 0, \boldsymbol{w}^*, 0)^\top \right\|^2 \qquad \text{(since } \boldsymbol{w}_0 = \boldsymbol{0}_d.\text{)}$$

The key insight of the proof is to replace the $\boldsymbol{w}^*$ with the one-step GD solution $\frac{\eta}{n}\boldsymbol{X}\boldsymbol{y}^\top$,

$$\mathcal{L}(\boldsymbol{V}, \boldsymbol{W}) = \frac{1}{2}\mathbb{E}\left[ \left\| \boldsymbol{V}\boldsymbol{Z}_0 \cdot \frac{\boldsymbol{Z}_0^\top \boldsymbol{W}\boldsymbol{Z}_{0[:,-1]}}{n} - \left(\boldsymbol{0}_d, 0, \frac{\eta}{n}\boldsymbol{X}\boldsymbol{y}^\top, 0\right)^\top \right\|^2 \right] + C$$

After proving this property, we can conclude that the optimal solution without CoT is exactly the one-step solution $\frac{\eta}{n}\boldsymbol{X}\boldsymbol{y}^\top$. We prove this result by showing the gradient of those two loss functions are the same.

First, before calculating the gradient, we extract the identical parts of the loss. Notice that the ground-truth entries are all zero in $i = 1, 2, \cdots, d, d+1, 2d+2$ positions in both expressions. Therefore, that part of error is the norm of the output $f_{\mathrm{LSA}}(\boldsymbol{Z}_0)_{[:,-1]}$ in those corresponding entries:

$$\frac{1}{2}\mathbb{E}\left[ \left\| \boldsymbol{V}\boldsymbol{Z}_0 \cdot \frac{\boldsymbol{Z}_0^\top \boldsymbol{W}\boldsymbol{Z}_{0[1:d+1,-1]}}{n} \right\|^2 \right] + \frac{1}{2}\mathbb{E}\left[ \left\| \boldsymbol{V}\boldsymbol{Z}_0 \cdot \frac{\boldsymbol{Z}_0^\top \boldsymbol{W}\boldsymbol{Z}_{0[2d+2,-1]}}{n} \right\|^2 \right]$$

which is the same for both expressions. Therefore, we just need to consider

$$f_{\mathrm{LSA}}(\boldsymbol{Z}_0)_{[d+2:2d+1,-1]} = \boldsymbol{V}_{[d+2:2d+1,:]}\boldsymbol{Z}_0 \cdot \frac{\boldsymbol{Z}_0^\top \boldsymbol{W}\boldsymbol{Z}_{0[:,-1]}}{n},$$

which corresponds to the ground-truth signals. Here $\boldsymbol{V}_{[d+2:2d+1,:]} = [\boldsymbol{V}_{31}, \boldsymbol{V}_{32}, \boldsymbol{V}_{33}, \boldsymbol{V}_{34}]$. We only need to prove that

$$\mathbb{E}\left\| f_{\mathrm{LSA}}(\boldsymbol{Z}_0)_{[d+2:2d+1,-1]} - \boldsymbol{w}^* \right\|^2 = \mathbb{E}\left\| f_{\mathrm{LSA}}(\boldsymbol{Z}_0)_{[d+2:2d+1,-1]} - \frac{\eta}{n}\boldsymbol{X}\boldsymbol{X}^\top \boldsymbol{w}^* \right\|^2 + C$$

for some constant $C$.

We show the gradients of both sides are the same, and equivalently the differential of both sides should be the same. The differential of L.H.S. is

$$
\mathrm{d}\Big(\mathbb{E}\left\|f_{\mathrm{LSA}}(\boldsymbol{Z}_0)_{[d+2:2d+1,-1]} - \boldsymbol{w}^*\right\|^2\Big)
$$
$$
= 2\mathbb{E}\big[(f_{\mathrm{LSA}}(\boldsymbol{Z}_0)_{[d+2:2d+1,-1]} - \boldsymbol{w}^*)^\top \mathrm{d}f_{\mathrm{LSA}}(\boldsymbol{Z}_0)_{[d+2:2d+1,-1]}\big]
$$

and the differential of R.H.S. is

$$
\mathrm{d}\bigg(\mathbb{E}\left\|f_{\mathrm{LSA}}(\boldsymbol{Z}_0)_{[d+2:2d+1,-1]} - \frac{\eta}{n}\boldsymbol{X}\boldsymbol{X}^\top\boldsymbol{w}^*\right\|^2\bigg)
$$
$$
= 2\mathbb{E}\bigg[(f_{\mathrm{LSA}}(\boldsymbol{Z}_0)_{[d+2:2d+1,-1]} - \frac{\eta}{n}\boldsymbol{X}\boldsymbol{X}^\top\boldsymbol{w}^*)^\top \mathrm{d}f_{\mathrm{LSA}}(\boldsymbol{Z}_0)_{[d+2:2d+1,-1]}\bigg]
$$

Therefore, we only need to prove that

$$
\mathbb{E}\Big[\boldsymbol{w}^{*\top}\mathrm{d}f_{\mathrm{LSA}}(\boldsymbol{Z}_0)_{[d+2:2d+1,-1]}\Big] = \mathbb{E}\bigg[\Big(\frac{\eta}{n}\boldsymbol{X}\boldsymbol{X}^\top\boldsymbol{w}^*\Big)^\top \mathrm{d}f_{\mathrm{LSA}}(\boldsymbol{Z}_0)_{[d+2:2d+1,-1]}\bigg] \qquad (10)
$$

We expand this expression $f_{\mathrm{LSA}}(\boldsymbol{Z}_0)_{[d+2:2d+1,-1]}$ (Note that now we don't have the assumption of initialization):

$$
\boldsymbol{V}_{[d+2:2d+1,:]}\boldsymbol{Z}_0 \cdot \frac{\boldsymbol{Z}_0^\top \boldsymbol{W}\boldsymbol{Z}_{0[:,-1]}}{n}
$$
$$
= \frac{1}{n}\begin{bmatrix}\boldsymbol{V}_{31} & \boldsymbol{V}_{32} & \boldsymbol{V}_{33} & \boldsymbol{V}_{34}\end{bmatrix}\begin{bmatrix}\boldsymbol{X} & \boldsymbol{0} \\ \boldsymbol{y} & 0 \\ \boldsymbol{0}_{d\times n} & \boldsymbol{w}_0 \\ \boldsymbol{0}_{1\times n} & 1\end{bmatrix}\begin{bmatrix}\boldsymbol{X}^\top & \boldsymbol{y}^\top & \boldsymbol{0}_{n\times d} & \boldsymbol{0}_n \\ \boldsymbol{0}_{1\times d} & 0 & \boldsymbol{w}_0^\top & 1\end{bmatrix}\boldsymbol{W}\begin{bmatrix}\boldsymbol{0} \\ 0 \\ \boldsymbol{w}_0 \\ 1\end{bmatrix}
$$
$$
= \frac{1}{n}\begin{bmatrix}\boldsymbol{V}_{31} & \boldsymbol{V}_{32} & \boldsymbol{V}_{33} & \boldsymbol{V}_{34}\end{bmatrix}\begin{bmatrix}\boldsymbol{X}\boldsymbol{X}^\top & \boldsymbol{X}\boldsymbol{y}^\top & \boldsymbol{0}_{d\times d} & \boldsymbol{0}_d \\ \boldsymbol{y}\boldsymbol{X}^\top & \boldsymbol{y}\boldsymbol{y}^\top & \boldsymbol{0}_{1\times d} & 0 \\ \boldsymbol{0}_{d\times d} & \boldsymbol{0}_d & \boldsymbol{0}_{d\times d} & \boldsymbol{0}_d \\ \boldsymbol{0}_{1\times d} & 0 & \boldsymbol{0}_{1\times d} & 1\end{bmatrix}\begin{bmatrix}\boldsymbol{W}_{14} \\ w_{24} \\ \boldsymbol{W}_{34} \\ w_{44}\end{bmatrix} \qquad (\text{since } \boldsymbol{w}_0 = \boldsymbol{0}_d)
$$
$$
= \frac{1}{n}\begin{bmatrix}\boldsymbol{V}_{31} & \boldsymbol{V}_{32} & \boldsymbol{V}_{33} & \boldsymbol{V}_{34}\end{bmatrix}\begin{bmatrix}\boldsymbol{X}\boldsymbol{X}^\top\boldsymbol{W}_{14} + w_{24}\boldsymbol{X}\boldsymbol{y}^\top \\ \boldsymbol{y}\boldsymbol{X}^\top\boldsymbol{W}_{14} + w_{24}\boldsymbol{y}\boldsymbol{y}^\top \\ \boldsymbol{0}_d \\ w_{44}\end{bmatrix}
$$
$$
= \frac{1}{n}\Big(\boldsymbol{V}_{31} + \boldsymbol{V}_{32}\boldsymbol{w}^{*\top}\Big)\boldsymbol{X}\boldsymbol{X}^\top(\boldsymbol{W}_{14} + w_{24}\boldsymbol{w}^*) + \frac{\boldsymbol{V}_{34}w_{44}}{n} \qquad (\boldsymbol{y} = \boldsymbol{X}^\top\boldsymbol{w}^*.)
$$

and the differential of $f_{\mathrm{LSA}}(\boldsymbol{Z}_0)_{[d+2:2d+1,-1]}$ is

$$
\mathrm{d}f_{\mathrm{LSA}}(\boldsymbol{Z}_0)_{[d+2:2d+1,-1]}
$$
$$
= \mathrm{d}\Big(\frac{1}{n}\Big(\boldsymbol{V}_{31} + \boldsymbol{V}_{32}\boldsymbol{w}^{*\top}\Big)\boldsymbol{X}\boldsymbol{X}^\top(\boldsymbol{W}_{14} + w_{24}\boldsymbol{w}^*)\Big) + \mathrm{d}\frac{\boldsymbol{V}_{34}w_{44}}{n}
$$
$$
= \frac{1}{n}\Big(\mathrm{d}\boldsymbol{V}_{31} + \mathrm{d}\boldsymbol{V}_{32}\boldsymbol{w}^{*\top}\Big)\boldsymbol{X}\boldsymbol{X}^\top(\boldsymbol{W}_{14} + w_{24}\boldsymbol{w}^*) + \frac{1}{n}(\mathrm{d}\boldsymbol{V}_{34} \cdot w_{44} + \boldsymbol{V}_{34}\mathrm{d}w_{44})
$$
$$
+ \frac{1}{n}\Big(\boldsymbol{V}_{31} + \boldsymbol{V}_{32}\boldsymbol{w}^{*\top}\Big)\boldsymbol{X}\boldsymbol{X}^\top(\mathrm{d}\boldsymbol{W}_{14} + \mathrm{d}w_{24}\boldsymbol{w}^*)
$$

Now, to prove Equation (10), we compare the differential for each parameter on both sides. For all parameter, we start from the left side and prove it equal to the right.

$\boldsymbol{V}_{31}$: The $\boldsymbol{V}_{31}$ term of differential in $\mathrm{d}f_{\mathrm{LSA}}(\boldsymbol{Z}_0)_{[d+2:2d+1,-1]}$ is $\frac{1}{n}\mathrm{d}\boldsymbol{V}_{31}\boldsymbol{X}\boldsymbol{X}^\top(\boldsymbol{W}_{14} + w_{24}\boldsymbol{w}^*)$,

$$
\mathbb{E}\bigg[\boldsymbol{w}^{*\top} \cdot \frac{1}{n}\mathrm{d}\boldsymbol{V}_{31}\boldsymbol{X}\boldsymbol{X}^\top(\boldsymbol{W}_{14} + w_{24}\boldsymbol{w}^*)\bigg]
$$
$$
= \mathbb{E}\bigg[\mathrm{tr}\Big(\boldsymbol{w}^{*\top} \cdot \frac{1}{n}\mathrm{d}\boldsymbol{V}_{31}\boldsymbol{X}\boldsymbol{X}^\top(\boldsymbol{W}_{14} + w_{24}\boldsymbol{w}^*)\Big)\bigg] \qquad (\text{It is a scalar in the trace.})
$$

$$= \mathbb{E}\left[\text{tr}\left(\frac{1}{n}\text{d}\boldsymbol{V}_{31}\boldsymbol{X}\boldsymbol{X}^\top(\boldsymbol{W}_{14} + w_{24}\boldsymbol{w}^*)\boldsymbol{w}^{*\top}\right)\right]$$

$$= \mathbb{E}[\text{tr}\,(\text{d}\boldsymbol{V}_{31}w_{24})] \qquad (\mathbb{E}[\boldsymbol{X}\boldsymbol{X}^\top] = n\boldsymbol{I}_d, \mathbb{E}[\boldsymbol{w}^*] = 0, \mathbb{E}[\boldsymbol{w}^*\boldsymbol{w}^{*\top}] = \boldsymbol{I}_d.)$$

$$= \mathbb{E}\left[\text{tr}\left(\frac{\eta}{n^2}\cdot\text{d}\boldsymbol{V}_{31}w_{24}\boldsymbol{X}\boldsymbol{X}^\top\boldsymbol{X}\boldsymbol{X}^\top\right)\right] \qquad (\mathbb{E}[(\boldsymbol{X}\boldsymbol{X}^\top)^2] = n(n + d + 1)\boldsymbol{I}_d, \eta = \frac{n}{n+d+1}.)$$

$$= \mathbb{E}\left[\text{tr}\left(\frac{\eta}{n^2}\cdot\boldsymbol{X}\boldsymbol{X}^\top\text{d}\boldsymbol{V}_{31}\boldsymbol{X}\boldsymbol{X}^\top(\boldsymbol{W}_{14} + w_{24}\boldsymbol{w}^*)\boldsymbol{w}^{*\top}\right)\right] \qquad (\mathbb{E}[\boldsymbol{w}^*] = 0, \mathbb{E}[\boldsymbol{w}^*\boldsymbol{w}^{*\top}] = \boldsymbol{I}_d.)$$

$$= \mathbb{E}\left[(\frac{\eta}{n}\boldsymbol{X}\boldsymbol{X}^\top\boldsymbol{w}^*)^\top\cdot\frac{1}{n}\text{d}\boldsymbol{V}_{31}\boldsymbol{X}\boldsymbol{X}^\top(\boldsymbol{W}_{14} + w_{24}\boldsymbol{w}^*)\right]$$

So those two $\text{d}\boldsymbol{V}_{31}$ terms are identical.

$\boldsymbol{V}_{32}$: The $\boldsymbol{V}_{32}$ term of differential in $\text{d}f_{\text{LSA}}(\boldsymbol{Z}_0)_{[d+2:2d+1,-1]}$ is $\frac{\text{d}\boldsymbol{V}_{32}}{n}\boldsymbol{w}^{*\top}\boldsymbol{X}\boldsymbol{X}^\top(\boldsymbol{W}_{14} + w_{24}\boldsymbol{w}^*)$,

$$\mathbb{E}\left[\boldsymbol{w}^{*\top}\cdot\frac{\text{d}\boldsymbol{V}_{32}}{n}\boldsymbol{w}^{*\top}\boldsymbol{X}\boldsymbol{X}^\top(\boldsymbol{W}_{14} + w_{24}\boldsymbol{w}^*)\right]$$

$$= \mathbb{E}\left[\text{tr}\left(\boldsymbol{w}^{*\top}\cdot\frac{\text{d}\boldsymbol{V}_{32}}{n}\boldsymbol{w}^{*\top}\boldsymbol{X}\boldsymbol{X}^\top(\boldsymbol{W}_{14} + w_{24}\boldsymbol{w}^*)\right)\right] \qquad \text{(It is a scalar in the trace.)}$$

$$= \mathbb{E}\left[\text{tr}\left(\frac{\text{d}\boldsymbol{V}_{32}}{n}\boldsymbol{w}^{*\top}\boldsymbol{X}\boldsymbol{X}^\top(\boldsymbol{W}_{14} + w_{24}\boldsymbol{w}^*)\boldsymbol{w}^{*\top}\right)\right]$$

$$= \mathbb{E}\left[\text{tr}\left(\frac{\text{d}\boldsymbol{V}_{32}}{n}\boldsymbol{w}^{*\top}\boldsymbol{X}\boldsymbol{X}^\top\boldsymbol{W}_{14}\boldsymbol{w}^{*\top}\right)\right] \qquad (\mathbb{E}[\boldsymbol{w}^*] = \boldsymbol{0} \text{ and } \boldsymbol{w}^{*\top}\boldsymbol{X}\boldsymbol{X}^\top\boldsymbol{w}^*\boldsymbol{w}^{*\top} \text{ is odd})$$

$$= \mathbb{E}\left[\text{tr}\left(\frac{\text{d}\boldsymbol{V}_{32}}{n}\boldsymbol{W}_{14}^\top\boldsymbol{X}\boldsymbol{X}^\top\boldsymbol{w}^*\boldsymbol{w}^{*\top}\right)\right] \qquad (\boldsymbol{W}_{14}^\top\boldsymbol{X}\boldsymbol{X}^\top\boldsymbol{w}^* \text{ is a scalar.})$$

$$= \mathbb{E}\left[\text{tr}\,(\text{d}\boldsymbol{V}_{32}\boldsymbol{W}_{14}^\top)\right] \qquad (\mathbb{E}[\boldsymbol{X}\boldsymbol{X}^\top] = n\boldsymbol{I}_d, \mathbb{E}[\boldsymbol{w}^*\boldsymbol{w}^{*\top}] = \boldsymbol{I}_d.)$$

$$= \mathbb{E}\left[\text{tr}\left(\frac{\eta}{n^2}\cdot\text{d}\boldsymbol{V}_{32}\boldsymbol{W}_{14}^\top\boldsymbol{X}\boldsymbol{X}^\top\boldsymbol{X}\boldsymbol{X}^\top\right)\right] \qquad (\mathbb{E}[(\boldsymbol{X}\boldsymbol{X}^\top)^2] = n(n + d + 1)\boldsymbol{I}_d, \eta = \frac{n}{n+d+1}.)$$

$$= \mathbb{E}\left[\text{tr}\left(\frac{\eta}{n^2}\cdot\boldsymbol{X}\boldsymbol{X}^\top\text{d}\boldsymbol{V}_{32}\boldsymbol{w}^{*\top}\boldsymbol{X}\boldsymbol{X}^\top(\boldsymbol{W}_{14} + w_{24}\boldsymbol{w}^*)\boldsymbol{w}^{*\top}\right)\right] \qquad (\mathbb{E}[\boldsymbol{w}^*] = 0, \mathbb{E}[\boldsymbol{w}^*\boldsymbol{w}^{*\top}] = \boldsymbol{I}_d.)$$

$$= \mathbb{E}\left[(\frac{\eta}{n}\boldsymbol{X}\boldsymbol{X}^\top\boldsymbol{w}^*)^\top\cdot\frac{1}{n}\text{d}\boldsymbol{V}_{32}\boldsymbol{w}^{*\top}\boldsymbol{X}\boldsymbol{X}^\top(\boldsymbol{W}_{14} + w_{24}\boldsymbol{w}^*)\right]$$

So those two $\text{d}\boldsymbol{V}_{32}$ terms are identical.

$\boldsymbol{V}_{34}$: The $\boldsymbol{V}_{34}$ term of differential in $\text{d}f_{\text{LSA}}(\boldsymbol{Z}_0)_{[d+2:2d+1,-1]}$ is $\frac{1}{n}\text{d}\boldsymbol{V}_{34}w_{44}$,

$$\mathbb{E}\left[\boldsymbol{w}^{*\top}\frac{1}{n}\text{d}\boldsymbol{V}_{34}w_{44}\right] = 0 = \mathbb{E}\left[(\frac{\eta}{n}\boldsymbol{X}\boldsymbol{X}^\top\boldsymbol{w}^*)^\top\frac{1}{n}\text{d}\boldsymbol{V}_{34}w_{44}\right]$$

since $\mathbb{E}[\boldsymbol{w}^*] = \boldsymbol{0}_d$. Therefore those two are equal.

$\boldsymbol{W}_{14}$: The $\boldsymbol{W}_{14}$ term of differential in $\text{d}f_{\text{LSA}}(\boldsymbol{Z}_0)_{[d+2:2d+1,-1]}$ is $\frac{1}{n}\left(\boldsymbol{V}_{31} + \boldsymbol{V}_{32}\boldsymbol{w}^{*\top}\right)\boldsymbol{X}\boldsymbol{X}^\top\text{d}\boldsymbol{W}_{14}$,

$$\mathbb{E}\left[\boldsymbol{w}^{*\top}\cdot\frac{1}{n}\left(\boldsymbol{V}_{31} + \boldsymbol{V}_{32}\boldsymbol{w}^{*\top}\right)\boldsymbol{X}\boldsymbol{X}^\top\text{d}\boldsymbol{W}_{14}\right]$$

$$= \mathbb{E}\left[\text{tr}\left(\boldsymbol{w}^{*\top}\cdot\frac{1}{n}\left(\boldsymbol{V}_{31} + \boldsymbol{V}_{32}\boldsymbol{w}^{*\top}\right)\boldsymbol{X}\boldsymbol{X}^\top\text{d}\boldsymbol{W}_{14}\right)\right] \qquad \text{(It is a scalar in the trace.)}$$

$$= \mathbb{E}\left[\text{tr}\left(\frac{1}{n}\left(\boldsymbol{w}^{*\top}\boldsymbol{V}_{32}\boldsymbol{w}^{*\top}\right)\boldsymbol{X}\boldsymbol{X}^\top\text{d}\boldsymbol{W}_{14}\right)\right] \qquad (\mathbb{E}[\boldsymbol{w}^*] = \boldsymbol{0}_d.)$$

$$= \mathbb{E}\left[\text{tr}\left(\frac{1}{n}\left(\boldsymbol{V}_{32}^\top\boldsymbol{w}^*\boldsymbol{w}^{*\top}\right)\boldsymbol{X}\boldsymbol{X}^\top\text{d}\boldsymbol{W}_{14}\right)\right] \qquad (\boldsymbol{V}_{32}^\top\boldsymbol{w}^* \text{ is a scalar.})$$

$$= \mathbb{E}\left[\text{tr}\left(\boldsymbol{V}_{32}^\top\text{d}\boldsymbol{W}_{14}\right)\right] \qquad (\mathbb{E}[\boldsymbol{X}\boldsymbol{X}^\top] = n\boldsymbol{I}_d, \mathbb{E}[\boldsymbol{w}^*\boldsymbol{w}^{*\top}] = \boldsymbol{I}_d.)$$

$$= \mathbb{E}\left[\text{tr}\left(\frac{\eta}{n^2}\cdot\boldsymbol{X}\boldsymbol{X}^\top\boldsymbol{V}_{32}^\top\boldsymbol{X}\boldsymbol{X}^\top\text{d}\boldsymbol{W}_{14}\right)\right] \qquad (\mathbb{E}[(\boldsymbol{X}\boldsymbol{X}^\top)^2] = n(n + d + 1)\boldsymbol{I}_d, \eta = \frac{n}{n+d+1}.)$$

$$= \mathbb{E}\left[(\frac{\eta}{n}\boldsymbol{X}\boldsymbol{X}^\top\boldsymbol{w}^*)^\top \cdot \frac{1}{n}\left(\boldsymbol{V}_{31} + \boldsymbol{V}_{32}\boldsymbol{w}^{*\top}\right)\boldsymbol{X}\boldsymbol{X}^\top \mathrm{d}\boldsymbol{W}_{14}\right]$$

Thus the two $\mathrm{d}\boldsymbol{W}_{14}$ terms are the same.

$w_{24}$: The $w_{24}$ term in $\mathrm{d}f_{\mathrm{LSA}}(\boldsymbol{Z}_0)_{[d+2:2d+1,-1]}$ is $\frac{1}{n}\left(\boldsymbol{V}_{31} + \boldsymbol{V}_{32}\boldsymbol{w}^{*\top}\right)\boldsymbol{X}\boldsymbol{X}^\top \mathrm{d}w_{24}\boldsymbol{w}^*$,

$$\begin{aligned}
&\mathbb{E}\left[\boldsymbol{w}^{*\top} \cdot \frac{1}{n}\left(\boldsymbol{V}_{31} + \boldsymbol{V}_{32}\boldsymbol{w}^{*\top}\right)\boldsymbol{X}\boldsymbol{X}^\top\boldsymbol{w}^*\mathrm{d}w_{24}\right] \\
&= \mathbb{E}\left[\mathrm{tr}\left(\boldsymbol{w}^{*\top} \cdot \frac{1}{n}\left(\boldsymbol{V}_{31} + \boldsymbol{V}_{32}\boldsymbol{w}^{*\top}\right)\boldsymbol{X}\boldsymbol{X}^\top\boldsymbol{w}^*\mathrm{d}w_{24}\right)\right] &&\text{(It is a scalar in the trace.)} \\
&= \mathbb{E}\left[\mathrm{tr}\left(\frac{1}{n}\left(\boldsymbol{w}^{*\top}\boldsymbol{V}_{31}\right)\boldsymbol{X}\boldsymbol{X}^\top\boldsymbol{w}^*\mathrm{d}w_{24}\right)\right] &&(\mathbb{E}[\boldsymbol{w}^*] = \boldsymbol{0}_d.) \\
&= \mathbb{E}[\mathrm{tr}\left(\boldsymbol{V}_{31}\mathrm{d}w_{24}\right)] &&(\mathbb{E}[\boldsymbol{X}\boldsymbol{X}^\top] = n\boldsymbol{I}_d, \mathbb{E}[\boldsymbol{w}^*\boldsymbol{w}^{*\top}] = \boldsymbol{I}_d.) \\
&= \mathbb{E}\left[\mathrm{tr}\left(\frac{\eta}{n^2} \cdot \boldsymbol{X}\boldsymbol{X}^\top\boldsymbol{V}_{31}\boldsymbol{X}\boldsymbol{X}^\top\mathrm{d}w_{24}\right)\right] &&(\mathbb{E}[(\boldsymbol{X}\boldsymbol{X}^\top)^2] = n(n+d+1)\boldsymbol{I}_d, \eta = \tfrac{n}{n+d+1}.) \\
&= \mathbb{E}\left[(\frac{\eta}{n}\boldsymbol{X}\boldsymbol{X}^\top\boldsymbol{w}^*)^\top \cdot \frac{1}{n}\left(\boldsymbol{V}_{31} + \boldsymbol{V}_{32}\boldsymbol{w}^{*\top}\right)\boldsymbol{X}\boldsymbol{X}^\top\boldsymbol{w}^*\mathrm{d}w_{24}\right]
\end{aligned}$$

Therefore the differential for $w_{24}$ are the same.

$w_{44}$: The $w_{44}$ term of differential in $\mathrm{d}f_{\mathrm{LSA}}(\boldsymbol{Z}_0)_{[d+2:2d+1,-1]}$ is $\frac{1}{n}\boldsymbol{V}_{34}\mathrm{d}w_{44}$,

$$\mathbb{E}\left[\boldsymbol{w}^{*\top}\frac{1}{n}\boldsymbol{V}_{34}\mathrm{d}w_{44}\right] = 0 = \mathbb{E}\left[(\frac{\eta}{n}\boldsymbol{X}\boldsymbol{X}^\top\boldsymbol{w}^*)^\top\frac{1}{n}\boldsymbol{V}_{34}\mathrm{d}w_{44}\right]$$

since $\mathbb{E}[\boldsymbol{w}^*] = \boldsymbol{0}_d$. Therefore those two are also equal.

In conclusion, Equation (10) holds since all the differential terms are equal. Therefore, $\exists C$

$$\mathbb{E}\left\|f_{\mathrm{LSA}}(\boldsymbol{Z}_0)_{[d+2:2d+1,-1]} - \boldsymbol{w}^*\right\|^2 = \mathbb{E}\left\|f_{\mathrm{LSA}}(\boldsymbol{Z}_0)_{[d+2:2d+1,-1]} - \frac{\eta}{n}\boldsymbol{X}\boldsymbol{X}^\top\boldsymbol{w}^*\right\|^2 + C$$

which finishes our proof. $\qquad\square$

### B.2 PROOF OF THEOREM 3.2

Here we restate the Theorem 3.2 and provide the detailed proof.

**Theorem B.2.** *Suppose $n = \Theta(d\log^5 d)$, $k \geq C\log d$, $\eta \in (0.1, 0.9)$. There exists $\boldsymbol{V}^*$ and $\boldsymbol{W}^*$ s.t. $f_{\mathrm{LSA}}(\boldsymbol{Z}_k)_{[:,-1]}$ outputs $(\boldsymbol{0}_d, 0, \boldsymbol{w}_{k+1}, 1)$ where $\boldsymbol{w}_i := \left(\boldsymbol{I} - (\boldsymbol{I} - \frac{\eta}{n}\boldsymbol{X}\boldsymbol{X}^\top)^i\right)\boldsymbol{w}^*$ is the $k$-step GD solution with learning rate $\eta$ on a linear regression model. Moreover, the evaluation loss*

$$\mathcal{L}^{\mathrm{Eval}}(\boldsymbol{V}^*, \boldsymbol{W}^*) = \frac{1}{2}\mathbb{E}_{\boldsymbol{X}, \boldsymbol{w}^*}\left[\left\|\left(\boldsymbol{I} - \frac{\eta}{n}\boldsymbol{X}\boldsymbol{X}^\top\right)^{k+1}\boldsymbol{w}^*\right\|^2\right] \leq \frac{1}{d^{C\log\left(\frac{1}{1-\eta}\right)}} \tag{11}$$

*Proof.* We construct $\boldsymbol{V}^*$ and $\boldsymbol{W}^*$ in the following way,

$$\boldsymbol{V}^* = \begin{bmatrix} \boldsymbol{0} & \boldsymbol{0} & \boldsymbol{0} & \boldsymbol{0} \\ \boldsymbol{0} & \boldsymbol{0} & \boldsymbol{0} & \boldsymbol{0} \\ -\eta\boldsymbol{I} & \boldsymbol{0} & \boldsymbol{0} & \boldsymbol{0} \\ \boldsymbol{0} & \boldsymbol{0} & \boldsymbol{0} & \boldsymbol{0} \end{bmatrix}, \boldsymbol{W}^* = \begin{bmatrix} \boldsymbol{0} & \boldsymbol{0} & \boldsymbol{I} & \boldsymbol{0} \\ \boldsymbol{0} & \boldsymbol{0} & \boldsymbol{0} & -1 \\ \boldsymbol{0} & \boldsymbol{0} & \boldsymbol{0} & \boldsymbol{0} \\ \boldsymbol{0} & \boldsymbol{0} & \boldsymbol{0} & \boldsymbol{0} \end{bmatrix} \tag{12}$$

Now the transformer is allowed to generate $k$ steps before reaching the final output. We can inductively calculate the $i$-th step of generation, showing that the output is exactly the parameter after $i$-th gradient step ($i = 1, 2, ..., k+1$):

$$\begin{aligned}
f_{\mathrm{LSA}}(\boldsymbol{Z}_i)_{[:,-1]} &= (\boldsymbol{0}_d, 0, \boldsymbol{w}_i, 1) + \boldsymbol{V}\boldsymbol{Z}_i \cdot \frac{\boldsymbol{Z}_i^\top \boldsymbol{W}\boldsymbol{Z}_{i[:,-1]}}{n} \\
&= (\boldsymbol{0}_d, 0, \boldsymbol{w}_i, 1) + \frac{1}{n}\left(\boldsymbol{0}_d, 0, \boldsymbol{V}_{31}(t)\boldsymbol{X}\boldsymbol{X}^\top(\boldsymbol{W}_{13}(t)\boldsymbol{w}_i - \boldsymbol{w}^*), 0\right)
\end{aligned}$$

$$= (\mathbf{0}_d, 0, \boldsymbol{w}_i, 1) + (\mathbf{0}_d, 0, -\frac{\eta}{n} \boldsymbol{X} \boldsymbol{X}^\top (\boldsymbol{w}_i - \boldsymbol{w}^*), 0)$$

$$= (\mathbf{0}_d, 0, \boldsymbol{w}_{i+1}, 1)$$

After $k{+}1$ steps, we have the final output $\left(\boldsymbol{I} - (\boldsymbol{I} - \frac{\eta}{n} \boldsymbol{X} \boldsymbol{X}^\top)^{k+1}\right) \boldsymbol{w}^*$ by induction and the evaluation loss becomes Equation (9). By Lemma D.4, the final loss is

$$\frac{1}{2} \mathbb{E}_{\boldsymbol{X}, \boldsymbol{w}^*} \left[ \left\| \left(\boldsymbol{I} - \frac{\eta}{n} \boldsymbol{X} \boldsymbol{X}^\top\right)^{k+1} \boldsymbol{w}^* \right\|^2 \right]$$

$$= \frac{1}{2} \mathbb{E}_{\boldsymbol{X}, \boldsymbol{w}^*} \left[ \mathrm{tr} \left(\boldsymbol{I} - \frac{\eta}{n} \boldsymbol{X} \boldsymbol{X}^\top\right)^{2k+2} \right] \qquad (\mathbb{E}[\boldsymbol{w}^* \boldsymbol{w}^{*\top}] = \boldsymbol{I}.)$$

$$= \frac{1}{2} \mathrm{tr} \, \mathbb{E}_{\boldsymbol{X}, \boldsymbol{w}^*} \left[ \left(\boldsymbol{I} - \frac{\eta}{n} \boldsymbol{X} \boldsymbol{X}^\top\right)^{2k+2} \right]$$

$$= \frac{1}{2} \mathrm{tr}((1 - \eta)^k (1 + \delta) \boldsymbol{I}) \qquad \text{(By Lemma D.4)}$$

$$\leq d(1 - \eta)^k \leq d^{-C \log\left(\frac{1}{1-\eta}\right)}.$$

$\square$

## C  Proof of Theorem 4.1

### C.1  Gradient computation of the full model over the CoT objective

In this appendix, we compute the gradient of the full model given the Assumption 4.1 and prove the equivalence between the dynamics of the full model and a simplified model. Throughout the appendix, we denote the $\boldsymbol{S} = \frac{1}{n} \boldsymbol{X} \boldsymbol{X}^\top$ for simplicity. And recall the $i$-th step of the linear classifier is $\boldsymbol{w}_i = (\boldsymbol{I} - (\boldsymbol{I} - \eta \boldsymbol{S})^i) \boldsymbol{w}^*$.

In Section 2.2, we have the full attention model

$$f_{\mathrm{LSA}}(\boldsymbol{Z}; \boldsymbol{V}, \boldsymbol{W})_{[:,-1]} = \boldsymbol{Z}_{[:,-1]} + \boldsymbol{V} \boldsymbol{Z} \cdot \frac{\boldsymbol{Z}^\top \boldsymbol{W} \boldsymbol{Z}_{[:,-1]}}{n}$$

and the Chain of Thought (CoT) objective

$$\mathcal{L}^{\mathrm{CoT}}(\boldsymbol{V}, \boldsymbol{W}) = \mathbb{E}_{\boldsymbol{X}, \boldsymbol{w}^*} \left[ \frac{1}{2} \sum_{i=0}^k \left\| f_{\mathrm{LSA}}(\boldsymbol{Z}_i)_{[:,-1]} - (\mathbf{0}_d, 0, \boldsymbol{w}_{i+1}, 1) \right\|^2 \right]$$

We define the error for the $i$-th step

$$\mathcal{L}_i := \frac{1}{2} \mathbb{E}_{\boldsymbol{X}, \boldsymbol{w}^*} \left\| f_{\mathrm{LSA}}(\boldsymbol{Z}_i)_{[:,-1]} - (\mathbf{0}_d, 0, \boldsymbol{w}_{i+1}, 1) \right\|^2$$

By linearity of expectation, we know the gradient of the CoT objective is the sum of gradients of all CoT steps: $\nabla \mathcal{L}^{\mathrm{CoT}} = \sum_{i=1}^k \nabla \mathcal{L}_i$. Now we can calculate the gradients of $\boldsymbol{V}, \boldsymbol{W}$ based on the loss of each CoT step:

**Lemma C.1** (Gradients of the full model). *The gradient of $\boldsymbol{V}, \boldsymbol{W}$ are given by the following equations:*

$$\nabla_{\boldsymbol{V}} \mathcal{L} = \frac{1}{n} \mathbb{E}_{\boldsymbol{X}, \boldsymbol{w}^*} \sum_{i=0}^k \left( \boldsymbol{V} \boldsymbol{Z}_i \cdot \frac{\boldsymbol{Z}_i^\top \boldsymbol{W} \boldsymbol{Z}_{i[:,-1]}}{n} - (\mathbf{0}_d, 0, \boldsymbol{w}_{i+1} - \boldsymbol{w}_i, 0)^\top \right) \boldsymbol{Z}_{i[:,-1]}^\top \boldsymbol{W}^\top \boldsymbol{Z}_i \boldsymbol{Z}_i^\top$$

$$\nabla_{\boldsymbol{W}} \mathcal{L} = \frac{1}{n} \mathbb{E}_{\boldsymbol{X}, \boldsymbol{w}^*} \sum_{i=0}^k \boldsymbol{Z}_i \boldsymbol{Z}_i^\top \boldsymbol{V}^\top \left( \boldsymbol{V} \boldsymbol{Z}_i \cdot \frac{\boldsymbol{Z}_i^\top \boldsymbol{W} \boldsymbol{Z}_{i[:,-1]}}{n} - (\mathbf{0}_d, 0, \boldsymbol{w}_{i+1} - \boldsymbol{w}_i, 0)^\top \right) \boldsymbol{Z}_{i[:,-1]}^\top$$

*Proof.* The loss is given by eq. (6):

$$\mathcal{L}^{\mathrm{CoT}}(\boldsymbol{V}, \boldsymbol{W}) = \mathbb{E}_{\boldsymbol{X}, \boldsymbol{w}^*} \left[ \frac{1}{2} \sum_{i=0}^k \left\| f_{\mathrm{LSA}}(\boldsymbol{Z}_i)_{[:,-1]} - (\mathbf{0}_d, 0, \boldsymbol{w}_{i+1}, 1) \right\|^2 \right] = \sum_{i=1}^k \mathcal{L}_i$$

Take differential of the loss for the $i$-th step $\mathcal{L}_i$ and we have

$$d\mathcal{L}_i = \mathbb{E}_{\boldsymbol{X},\boldsymbol{w}^*}\left(f_{\text{LSA}}(\boldsymbol{Z}_i)_{[:,-1]} - (\boldsymbol{0}_d, 0, \boldsymbol{w}_{i+1}, 1)\right)^\top df_{\text{LSA}}(\boldsymbol{Z}_i)_{[:,-1]}$$

$$= \mathbb{E}_{\boldsymbol{X},\boldsymbol{w}^*}\left(f_{\text{LSA}}(\boldsymbol{Z}_i)_{[:,-1]} - (\boldsymbol{0}_d, 0, \boldsymbol{w}_{i+1}, 1)\right)^\top d\left(\boldsymbol{V}\boldsymbol{Z}_i \cdot \frac{\boldsymbol{Z}_i^\top \boldsymbol{W} \boldsymbol{Z}_{i[:,-1]}}{n}\right)$$

$$= \mathbb{E}_{\boldsymbol{X},\boldsymbol{w}^*}\left(f_{\text{LSA}}(\boldsymbol{Z}_i)_{[:,-1]} - (\boldsymbol{0}_d, 0, \boldsymbol{w}_{i+1}, 1)\right)^\top d(\boldsymbol{V})\boldsymbol{Z}_i \cdot \frac{\boldsymbol{Z}_i^\top \boldsymbol{W} \boldsymbol{Z}_{i[:,-1]}}{n}$$

$$+ \mathbb{E}_{\boldsymbol{X},\boldsymbol{w}^*}\left(f_{\text{LSA}}(\boldsymbol{Z}_i)_{[:,-1]} - (\boldsymbol{0}_d, 0, \boldsymbol{w}_{i+1}, 1)\right)^\top \boldsymbol{V}\boldsymbol{Z}_i \cdot \frac{\boldsymbol{Z}_i^\top d\boldsymbol{W} \boldsymbol{Z}_{i[:,-1]}}{n}$$

Then the gradients of $\boldsymbol{W}, \boldsymbol{V}$ of the $\mathcal{L}_i$ are:

$$\nabla_{\boldsymbol{V}}\mathcal{L}_i = \frac{1}{n}\mathbb{E}_{\boldsymbol{X},\boldsymbol{w}^*}\left(f_{\text{LSA}}(\boldsymbol{Z}_i)_{[:,-1]} - (\boldsymbol{0}_d, 0, \boldsymbol{w}_{i+1}, 1)\right)\boldsymbol{Z}_{i[:,-1]}^\top \boldsymbol{W}^\top \boldsymbol{Z}_i \boldsymbol{Z}_i^\top$$

$$= \frac{1}{n}\mathbb{E}_{\boldsymbol{X},\boldsymbol{w}^*}\left(\boldsymbol{V}\boldsymbol{Z}_i \cdot \frac{\boldsymbol{Z}_i^\top \boldsymbol{W} \boldsymbol{Z}_{i[:,-1]}}{n} - (\boldsymbol{0}_d, 0, \boldsymbol{w}_{i+1} - \boldsymbol{w}_i, 0)^\top\right)\boldsymbol{Z}_{i[:,-1]}^\top \boldsymbol{W}^\top \boldsymbol{Z}_i \boldsymbol{Z}_i^\top$$

$$\nabla_{\boldsymbol{W}}\mathcal{L}_i = \frac{1}{n}\mathbb{E}_{\boldsymbol{X},\boldsymbol{w}^*}\boldsymbol{Z}_i \boldsymbol{Z}_i^\top \boldsymbol{V}^\top \left(f_{\text{LSA}}(\boldsymbol{Z}_i)_{[:,-1]} - (\boldsymbol{0}_d, 0, \boldsymbol{w}_{i+1}, 1)\right)\boldsymbol{Z}_{i[:,-1]}^\top$$

$$= \frac{1}{n}\mathbb{E}_{\boldsymbol{X},\boldsymbol{w}^*}\boldsymbol{Z}_i \boldsymbol{Z}_i^\top \boldsymbol{V}^\top \left(\boldsymbol{V}\boldsymbol{Z}_i \cdot \frac{\boldsymbol{Z}_i^\top \boldsymbol{W} \boldsymbol{Z}_{i[:,-1]}}{n} - (\boldsymbol{0}_d, 0, \boldsymbol{w}_{i+1} - \boldsymbol{w}_i, 0)^\top\right)\boldsymbol{Z}_{i[:,-1]}^\top$$

Take the sum of the two equations above from $i = 0$ to $k$, and we finish the proof. $\qquad\square$

Now we consider the gradient flow (GF) trajectory (note that $w_{24}$ is fixed under Assumption 4.1):

$$\frac{d\boldsymbol{\theta}}{dt} = -\nabla\mathcal{L}^{\text{CoT}}(\boldsymbol{\theta}), \quad \boldsymbol{\theta} := (\boldsymbol{V}, \boldsymbol{W}\backslash\{w_{24}\}).$$

Recall the block matrix form of $\boldsymbol{V}, \boldsymbol{W}$:

$$\boldsymbol{V} = \begin{bmatrix} \boldsymbol{V}_{11} & \boldsymbol{V}_{12} & \boldsymbol{V}_{13} & \boldsymbol{V}_{14} \\ \boldsymbol{V}_{21} & v_{22} & \boldsymbol{V}_{23} & v_{24} \\ \boldsymbol{V}_{31} & \boldsymbol{V}_{32} & \boldsymbol{V}_{33} & \boldsymbol{V}_{34} \\ \boldsymbol{V}_{41} & v_{42} & \boldsymbol{V}_{43} & v_{44} \end{bmatrix}, \boldsymbol{W} = \begin{bmatrix} \boldsymbol{W}_{11} & \boldsymbol{W}_{12} & \boldsymbol{W}_{13} & \boldsymbol{W}_{14} \\ \boldsymbol{W}_{21} & w_{22} & \boldsymbol{W}_{23} & w_{24} \\ \boldsymbol{W}_{31} & \boldsymbol{W}_{32} & \boldsymbol{W}_{33} & \boldsymbol{W}_{34} \\ \boldsymbol{W}_{41} & w_{42} & \boldsymbol{W}_{43} & w_{44} \end{bmatrix}$$

According to the construction in Theorem 3.2, the blocks $\boldsymbol{W}_{13}, \boldsymbol{V}_{31}, w_{24}$ are the only relevant parameter blocks, while the others should be zeroed out. Next, we prove that if we initialize those irrelevant blocks to 0, then they will stay at 0 along the gradient descent trajectory.

**Lemma C.2.** *Under the Assumption 4.1, when the linear transformer is trained under GF, we have for all $t > 0$, the parameters $\boldsymbol{V}(t), \boldsymbol{W}(t)$ have the following form:*

$$\boldsymbol{V}(t) = \begin{bmatrix} \boldsymbol{0} & \boldsymbol{0} & \boldsymbol{0} & \boldsymbol{0} \\ \boldsymbol{0} & 0 & 0 & 0 \\ \boldsymbol{V}_{31}(t) & \boldsymbol{0} & \boldsymbol{0} & \boldsymbol{0} \\ \boldsymbol{0} & 0 & 0 & 0 \end{bmatrix}, \boldsymbol{W}(t) = \begin{bmatrix} \boldsymbol{0} & 0 & \boldsymbol{W}_{13}(t) & 0 \\ 0 & 0 & \boldsymbol{0} & -1 \\ \boldsymbol{0} & 0 & \boldsymbol{0} & 0 \\ \boldsymbol{0} & 0 & \boldsymbol{0} & 0 \end{bmatrix}$$

*Proof.* To prove this lemma, we prove that when the irrelevant blocks are 0, the gradients $\nabla_{\boldsymbol{V}}\mathcal{L}_i, \nabla_{\boldsymbol{W}}\mathcal{L}_i$ for those blocks are always 0 and they never update the corresponding parameter block. Also, note that $w_{24} = -1$ for all $t > 0$.

First, we calculate the output of the linear self-attention $\boldsymbol{V}\boldsymbol{Z}_i \cdot \frac{\boldsymbol{Z}_i^\top \boldsymbol{W} \boldsymbol{Z}_{i[:,-1]}}{n}$:

$$\boldsymbol{V}\boldsymbol{Z}_i \cdot \frac{\boldsymbol{Z}_i^\top \boldsymbol{W} \boldsymbol{Z}_{i[:,-1]}}{n}$$

$$= \frac{1}{n}\begin{bmatrix} \boldsymbol{0} & \boldsymbol{0} & \boldsymbol{0} & \boldsymbol{0} \\ \boldsymbol{0} & 0 & 0 & 0 \\ \boldsymbol{V}_{31}(t) & \boldsymbol{0} & \boldsymbol{0} & \boldsymbol{0} \\ \boldsymbol{0} & 0 & 0 & 0 \end{bmatrix}\begin{bmatrix} \boldsymbol{X} & \boldsymbol{0} & \boldsymbol{0} & \cdots & \boldsymbol{0} \\ \boldsymbol{y} & 0 & 0 & \cdots & 0 \\ \boldsymbol{0}_{d\times n} & \boldsymbol{w}_0 & \boldsymbol{w}_1 & \cdots & \boldsymbol{w}_i \\ \boldsymbol{0}_{1\times n} & 1 & 1 & \cdots & 1 \end{bmatrix}\boldsymbol{Z}_i^\top \boldsymbol{W} \begin{bmatrix} \boldsymbol{0} \\ 0 \\ \boldsymbol{w}_i \\ 1 \end{bmatrix}$$

$$
= \frac{1}{n} \begin{bmatrix} \mathbf{0}_{d\times n} & \mathbf{0}_d & \cdots & \mathbf{0}_d \\ \mathbf{0}_{1\times n} & 0 & \cdots & 0 \\ \boldsymbol{V}_{31}(t)\boldsymbol{X} & \mathbf{0}_d & \cdots & \mathbf{0}_d \\ \mathbf{0}_{1\times n} & 0 & \cdots & 0 \end{bmatrix} \begin{bmatrix} \boldsymbol{X} & \mathbf{0} & \mathbf{0} & \cdots & \mathbf{0} \\ \boldsymbol{y} & 0 & 0 & \cdots & 0 \\ \mathbf{0}_{d\times n} & \boldsymbol{w}_0 & \boldsymbol{w}_1 & \cdots & \boldsymbol{w}_i \\ \mathbf{0}_{1\times n} & 1 & 1 & \cdots & 1 \end{bmatrix}^\top \begin{bmatrix} \boldsymbol{W}_{13}(t)\boldsymbol{w}_i \\ -1 \\ \mathbf{0}_d \\ 0 \end{bmatrix}
$$

$$
= \frac{1}{n} \begin{bmatrix} \mathbf{0}_{d\times n} & \mathbf{0}_d & \cdots & \mathbf{0}_d \\ \mathbf{0}_{1\times n} & 0 & \cdots & 0 \\ \boldsymbol{V}_{31}(t)\boldsymbol{X} & \mathbf{0}_d & \cdots & \mathbf{0}_d \\ \mathbf{0}_{1\times n} & 0 & \cdots & 0 \end{bmatrix} \begin{bmatrix} \boldsymbol{X}^\top\boldsymbol{W}_{13}(t)\boldsymbol{w}_i - \boldsymbol{y}^\top \\ \mathbf{0}_{i+1} \end{bmatrix} = \frac{1}{n} \begin{bmatrix} \mathbf{0}_d \\ 0 \\ \boldsymbol{V}_{31}(t)\boldsymbol{X}\boldsymbol{X}^\top(\boldsymbol{W}_{13}(t)\boldsymbol{w}_i - \boldsymbol{w}^*) \\ 0 \end{bmatrix}
$$

The last line is because $\boldsymbol{y}^\top = \boldsymbol{X}^\top\boldsymbol{w}^*$. Now, we consider the gradient for $\boldsymbol{V}$:

$$
\nabla_{\boldsymbol{V}}\mathcal{L}_i = \frac{1}{n}\mathbb{E}_{\boldsymbol{X},\boldsymbol{w}^*}\left[\left(\boldsymbol{V}\boldsymbol{Z}_i \cdot \frac{\boldsymbol{Z}_i^\top\boldsymbol{W}\boldsymbol{Z}_{i[:,-1]}}{n} - (\mathbf{0}_d, 0, \boldsymbol{w}_{i+1} - \boldsymbol{w}_i, 0)^\top\right)\boldsymbol{Z}_{i[:,-1]}^\top\boldsymbol{W}^\top\boldsymbol{Z}_i\boldsymbol{Z}_i^\top\right]
$$

$$
= \frac{1}{n^2}\begin{bmatrix} \mathbf{0}_d \\ 0 \\ \boldsymbol{V}_{31}(t)\boldsymbol{X}\boldsymbol{X}^\top(\boldsymbol{W}_{13}(t)\boldsymbol{w}_i - \boldsymbol{w}^*) - n(\boldsymbol{w}_{i+1} - \boldsymbol{w}_i) \\ 0 \end{bmatrix}\boldsymbol{Z}_{i[:,-1]}^\top\boldsymbol{W}^\top\boldsymbol{Z}_i\boldsymbol{Z}_i^\top
$$

$$
= \frac{1}{n^2}\mathbb{E}_{\boldsymbol{X},\boldsymbol{w}^*}\begin{bmatrix} \mathbf{0}_d \\ 0 \\ \boldsymbol{V}_{31}(t)\boldsymbol{X}\boldsymbol{X}^\top(\boldsymbol{W}_{13}(t)\boldsymbol{w}_i - \boldsymbol{w}^*) - n(\boldsymbol{w}_{i+1} - \boldsymbol{w}_i) \\ 0 \end{bmatrix}\begin{bmatrix} \boldsymbol{w}_i^\top\boldsymbol{W}_{13}^\top(t) \\ -1 \\ \mathbf{0}_d \\ 0 \end{bmatrix}^\top\boldsymbol{Z}_i\boldsymbol{Z}_i^\top
$$

$$
= \frac{1}{n^2}\mathbb{E}_{\boldsymbol{X},\boldsymbol{w}^*}\begin{bmatrix} \mathbf{0}_d \\ 0 \\ \boldsymbol{V}_{31}(t)\boldsymbol{X}\boldsymbol{X}^\top(\boldsymbol{W}_{13}(t)\boldsymbol{w}_i - \boldsymbol{w}^*) - n(\boldsymbol{w}_{i+1} - \boldsymbol{w}_i) \\ 0 \end{bmatrix}\begin{bmatrix} \boldsymbol{w}_i^\top\boldsymbol{W}_{13}^\top(t)\boldsymbol{X}\boldsymbol{X}^\top - \boldsymbol{y}\boldsymbol{X}^\top \\ \boldsymbol{w}_i^\top\boldsymbol{W}_{13}^\top(t)\boldsymbol{X}\boldsymbol{y}^\top - \boldsymbol{y}\boldsymbol{y}^\top \\ \mathbf{0}_d \\ 0 \end{bmatrix}^\top
$$

$$
= \begin{bmatrix} \mathbf{0} & \mathbf{0} & \mathbf{0} & \mathbf{0} \\ \mathbf{0} & 0 & \mathbf{0} & 0 \\ \nabla_{\boldsymbol{V}_{31}}\mathcal{L}_i(t) & \nabla_{\boldsymbol{V}_{32}}\mathcal{L}_i(t) & \mathbf{0} & \mathbf{0} \\ \mathbf{0} & 0 & \mathbf{0} & 0 \end{bmatrix}
$$

Therefore, we know all blocks of the gradient are zero except the positions of $\boldsymbol{V}_{31}$ and $\boldsymbol{V}_{32}$.

Now look at $\nabla_{\boldsymbol{V}_{32}}\mathcal{L}_i$:

$$
\nabla_{\boldsymbol{V}_{32}}\mathcal{L}_i = \frac{1}{n^2}\mathbb{E}_{\boldsymbol{X},\boldsymbol{w}^*}\left[\left(\boldsymbol{V}_{31}(t)\boldsymbol{X}\boldsymbol{X}^\top(\boldsymbol{W}_{13}(t)\boldsymbol{w}_i - \boldsymbol{w}^*) - n(\boldsymbol{w}_{i+1} - \boldsymbol{w}_i)\right)\right.
$$
$$
\left.\left(\boldsymbol{w}_i^\top\boldsymbol{W}_{13}^\top(t)\boldsymbol{X}\boldsymbol{y}^\top - \boldsymbol{y}\boldsymbol{y}^\top\right)\right]
$$
$$
= \frac{1}{n^2}\mathbb{E}_{\boldsymbol{X},\boldsymbol{w}^*}\left[\left(\boldsymbol{V}_{31}(t)\boldsymbol{X}\boldsymbol{X}^\top(\boldsymbol{W}_{13}(t)\boldsymbol{w}_i - \boldsymbol{w}^*) - n(\boldsymbol{w}_{i+1} - \boldsymbol{w}_i)\right)\right.
$$
$$
\left.\left(\boldsymbol{w}_i^\top\boldsymbol{W}_{13}^\top(t)\boldsymbol{X}\boldsymbol{X}^\top\boldsymbol{w}^* - \boldsymbol{w}^{*\top}\boldsymbol{X}\boldsymbol{X}^\top\boldsymbol{w}^*\right)\right]
$$

Note that $\boldsymbol{w}_i = \left(\boldsymbol{I} - (\boldsymbol{I} - \eta\boldsymbol{S})^i\right)\boldsymbol{w}^*$ for all $i \in [k]$, and $\boldsymbol{w}_{k+1} = \boldsymbol{w}^*$. Therefore, for all $i \in \{0, 1, \cdots, k+1\}$ the formula inside the expectation is an odd function of $\boldsymbol{w}^*$. Since $\boldsymbol{w}^* \sim \mathcal{N}(0, \boldsymbol{I}_d)$, the expectation should be $\mathbf{0}_d$.

Similarly, we calculate the gradient of the $\boldsymbol{W}$:

$$
\nabla_{\boldsymbol{W}}\mathcal{L}_i = \frac{1}{n}\mathbb{E}_{\boldsymbol{X},\boldsymbol{w}^*}\left[\boldsymbol{Z}_i\boldsymbol{Z}_i^\top\boldsymbol{V}^\top\left(\boldsymbol{V}\boldsymbol{Z}_i \cdot \frac{\boldsymbol{Z}_i^\top\boldsymbol{W}\boldsymbol{Z}_{i[:,-1]}}{n} - (\mathbf{0}_d, 0, \boldsymbol{w}_{i+1} - \boldsymbol{w}_i, 0)^\top\right)\boldsymbol{Z}_{i[:,-1]}^\top\right]
$$

$$
= \frac{1}{n^2}\mathbb{E}_{\boldsymbol{X},\boldsymbol{w}^*}\left[\boldsymbol{Z}_i\boldsymbol{Z}_i^\top\boldsymbol{V}^\top\begin{bmatrix} \mathbf{0}_d \\ 0 \\ \boldsymbol{V}_{31}(t)\boldsymbol{X}\boldsymbol{X}^\top(\boldsymbol{W}_{13}(t)\boldsymbol{w}_i - \boldsymbol{w}^*) - n(\boldsymbol{w}_{i+1} - \boldsymbol{w}_i) \\ 0 \end{bmatrix}\boldsymbol{Z}_{i[:,-1]}^\top\right]
$$

$$
= \frac{1}{n^2}\mathbb{E}_{\boldsymbol{X},\boldsymbol{w}^*}\left[\begin{bmatrix} \mathbf{0}_{d\times d} & 0 & \boldsymbol{X}\boldsymbol{X}^\top\boldsymbol{V}_{31}(t)^\top & 0 \\ \mathbf{0}_{d\times d} & 0 & \boldsymbol{y}\boldsymbol{X}^\top\boldsymbol{V}_{31}(t)^\top & 0 \\ \mathbf{0}_{d\times d} & 0 & \mathbf{0}_{d\times d} & 0 \\ \mathbf{0}_{d\times d} & 0 & \mathbf{0}_{d\times d} & 0 \end{bmatrix}\begin{bmatrix} \mathbf{0}_d \\ 0 \\ \boldsymbol{V}_{31}(t)\boldsymbol{X}\boldsymbol{X}^\top(\boldsymbol{W}_{13}(t)\boldsymbol{w}_i - \boldsymbol{w}^*) - n(\boldsymbol{w}_{i+1} - \boldsymbol{w}_i) \\ 0 \end{bmatrix}\boldsymbol{Z}_{i[:,-1]}^\top\right]
$$

$$= \frac{1}{n^2}\mathbb{E}_{\boldsymbol{X},\boldsymbol{w}^*}\left[\begin{bmatrix} \boldsymbol{X}\boldsymbol{X}^\top\boldsymbol{V}_{31}(t)^\top\boldsymbol{V}_{31}(t)\boldsymbol{X}\boldsymbol{X}^\top(\boldsymbol{W}_{13}(t)\boldsymbol{w}_i-\boldsymbol{w}^*)-n(\boldsymbol{w}_{i+1}-\boldsymbol{w}_i) \\ \boldsymbol{y}\boldsymbol{X}^\top\boldsymbol{V}_{31}(t)^\top\boldsymbol{V}_{31}(t)\boldsymbol{X}\boldsymbol{X}^\top(\boldsymbol{W}_{13}(t)\boldsymbol{w}_i-\boldsymbol{w}^*)-n(\boldsymbol{w}_{i+1}-\boldsymbol{w}_i) \\ \boldsymbol{0}_d \\ 0 \end{bmatrix}\begin{bmatrix} \boldsymbol{0}_d \\ 0 \\ \boldsymbol{w}_i \\ 1 \end{bmatrix}^\top\right]$$

$$= \begin{bmatrix} \boldsymbol{0} & \boldsymbol{0} & \nabla_{\boldsymbol{W}_{13}}\mathcal{L}_i(t) & \nabla_{\boldsymbol{W}_{14}}\mathcal{L}_i(t) \\ \boldsymbol{0} & \boldsymbol{0} & \nabla_{\boldsymbol{W}_{23}}\mathcal{L}_i(t) & \nabla_{\boldsymbol{w}_{24}}\mathcal{L}_i(t) \\ \boldsymbol{0} & \boldsymbol{0} & \boldsymbol{0} & \boldsymbol{0} \\ \boldsymbol{0} & \boldsymbol{0} & \boldsymbol{0} & \boldsymbol{0} \end{bmatrix}$$

Since we fix $w_{24}$, we only consider the remaining three blocks. First, we consider the gradient of the vector block $\boldsymbol{W}_{14}$:

$$\nabla_{\boldsymbol{W}_{14}}\mathcal{L}_i(t) = \frac{1}{n^2}\mathbb{E}_{\boldsymbol{X},\boldsymbol{w}^*}\left[\boldsymbol{X}\boldsymbol{X}^\top\boldsymbol{V}_{31}(t)^\top\boldsymbol{V}_{31}(t)\boldsymbol{X}\boldsymbol{X}^\top(\boldsymbol{W}_{13}(t)\boldsymbol{w}_i-\boldsymbol{w}^*)-n(\boldsymbol{w}_{i+1}-\boldsymbol{w}_i)\right].$$

Notice that the $\boldsymbol{X}\boldsymbol{X}^\top\boldsymbol{V}_{31}(t)^\top\boldsymbol{V}_{31}(t)\boldsymbol{X}\boldsymbol{X}^\top(\boldsymbol{W}_{13}(t)\boldsymbol{w}_i-\boldsymbol{w}^*)-n(\boldsymbol{w}_{i+1}-\boldsymbol{w}_i)$ is odd in $\boldsymbol{w}^*$. Therefore the expectation is $\boldsymbol{0}_d$. Similarly, we consider the other block $\boldsymbol{W}_{23}$:

$$\nabla_{\boldsymbol{W}_{23}}\mathcal{L}_i(t) = \frac{1}{n^2}\mathbb{E}_{\boldsymbol{X},\boldsymbol{w}^*}\left[\left(\boldsymbol{y}\boldsymbol{X}^\top\boldsymbol{V}_{31}(t)^\top\boldsymbol{V}_{31}(t)\boldsymbol{X}\boldsymbol{X}^\top(\boldsymbol{W}_{13}(t)\boldsymbol{w}_i-\boldsymbol{w}^*)-n(\boldsymbol{w}_{i+1}-\boldsymbol{w}_i)\right)\boldsymbol{w}_i^\top\right]$$

$$= \frac{1}{n^2}\mathbb{E}_{\boldsymbol{X},\boldsymbol{w}^*}\left[\left(\boldsymbol{w}^{*\top}\boldsymbol{X}\boldsymbol{X}^\top\boldsymbol{V}_{31}(t)^\top\boldsymbol{V}_{31}(t)\boldsymbol{X}\boldsymbol{X}^\top(\boldsymbol{W}_{13}(t)\boldsymbol{w}_i-\boldsymbol{w}^*)-n(\boldsymbol{w}_{i+1}-\boldsymbol{w}_i)\right)\boldsymbol{w}_i^\top\right]$$

$$= \boldsymbol{0}_{1\times d}.$$

In conclusion, all the blocks have zero gradient except $\boldsymbol{V}_{31}, \boldsymbol{W}_{13}$ given that they are all zero matrices. Under Assumption 4.1, all the irrelevant blocks remain zero matrices for all $t \geq 0$. □

By Lemma C.2, we prove that along the gradient flow trajectory under Assumption 4.1, the objective of the linear self-attention model with residual connection can be equivalently transform to the following simplified form.

**Lemma C.3.** *Under Assumption 4.1, we have the training objective*

$$\mathcal{L}^{\text{CoT}}(\boldsymbol{V},\boldsymbol{W}) = \frac{1}{2}\mathbb{E}_{\boldsymbol{X},\boldsymbol{w}^*}\left[\sum_{i=0}^k \|\boldsymbol{V}_{31}(\boldsymbol{S}\boldsymbol{W}_{13}\boldsymbol{w}_i-\boldsymbol{S}\boldsymbol{w}^*)-\Delta\boldsymbol{w}_i\|^2\right]$$

*where $\boldsymbol{S} = \frac{1}{n}\boldsymbol{X}\boldsymbol{X}^\top$ and $\Delta\boldsymbol{w}_i := \boldsymbol{w}_{i+1}-\boldsymbol{w}_i$, $i = 0, 1..., k$ is the residual for each step $i$.*

*Proof.* Given the following CoT objective,

$$\mathcal{L}^{\text{CoT}}(\boldsymbol{V},\boldsymbol{W}) = \frac{1}{2}\mathbb{E}_{\boldsymbol{X},\boldsymbol{w}^*}\left[\sum_{i=0}^k \left\|f_{\text{LSA}}(\boldsymbol{Z}_i)_{[:,-1]}-(\boldsymbol{0}_d, 0, \boldsymbol{w}_{i+1}, 1)\right\|^2\right]$$

By Lemma C.2, we plug in the $\boldsymbol{V}, \boldsymbol{W}$ expressions and get:

$$f_{\text{LSA}}(\boldsymbol{Z}_i)_{[:,-1]}-(\boldsymbol{0}_d, 0, \boldsymbol{w}_{i+1}, 1) = \boldsymbol{V}\boldsymbol{Z}_i\cdot\frac{\boldsymbol{Z}_i^\top\boldsymbol{W}\boldsymbol{Z}_{i[:,-1]}}{n}-(\boldsymbol{0}_d, 0, \boldsymbol{w}_{i+1}-\boldsymbol{w}_i, 0)^\top$$

$$= \left(\boldsymbol{0}_d, 0, \frac{1}{n}\boldsymbol{V}_{31}(\boldsymbol{X}\boldsymbol{X}^\top\boldsymbol{W}_{13}\boldsymbol{w}_i-\boldsymbol{X}\boldsymbol{y}^\top)-\Delta\boldsymbol{w}_i, 0\right)$$

Since $\boldsymbol{y}^\top = \boldsymbol{X}^\top\boldsymbol{w}^*$, we have

$$f_{\text{LSA}}(\boldsymbol{Z}_i)_{[:,-1]}-(\boldsymbol{0}_d, 0, \boldsymbol{w}_{i+1}, 1) = \left(\boldsymbol{0}_d, 0, \boldsymbol{V}_{31}(\boldsymbol{S}^\top\boldsymbol{W}_{13}\boldsymbol{w}_i-\boldsymbol{S}\boldsymbol{w}^*)-\Delta\boldsymbol{w}_i, 0\right)^\top$$

Put it back to the loss expression and we complete the proof. □

Now the chain of thought loss can be rewritten into the form by Lemma C.3, we can directly calculate the gradient update using the simplified loss for clarity. We denote the only relevant blocks $\widetilde{\boldsymbol{W}} := \boldsymbol{W}_{13}$ and $\widetilde{\boldsymbol{V}} := \boldsymbol{V}_{31}$. Moreover, we can further expand the CoT loss with $\Delta\boldsymbol{w}_i = -\eta\cdot\frac{\boldsymbol{X}\boldsymbol{X}^\top}{n}(\boldsymbol{w}_i-$

$\boldsymbol{w}^*$) for $i \in \{0, 1, \cdots, k-1\}$, and $\Delta \boldsymbol{w}_k = \boldsymbol{w}^* - \boldsymbol{w}_k$. That leads to the following expression of the CoT loss:

$$\mathcal{L}^{\text{CoT}}(\boldsymbol{\theta}) = \frac{1}{2} \mathbb{E}_{\boldsymbol{X}, \boldsymbol{w}^*} \sum_{i=0}^{k-1} \left\| \boldsymbol{w}_i + \widetilde{\boldsymbol{V}} \boldsymbol{S} \left( \widetilde{\boldsymbol{W}} \boldsymbol{w}_i - \boldsymbol{w}^* \right) - \boldsymbol{w}_{i+1} \right\|_2^2$$
$$+ \frac{1}{2} \mathbb{E}_{\boldsymbol{X}, \boldsymbol{w}^*} \left\| \boldsymbol{w}_k + \widetilde{\boldsymbol{V}} \boldsymbol{S} \left( \widetilde{\boldsymbol{W}} \boldsymbol{w}_k - \boldsymbol{w}^* \right) - \boldsymbol{w}^* \right\|_2^2 \tag{13}$$

Observe that the final loss only depends on the $(d+2)$ to $(2d+2)$ entries of the transformer's output, indicating we can simplify the model a bit and prune out the irrelevant part. We can define a simplified one-layer transformer to get the loss form above, where the dynamics of the equivalent model is exactly the same with the original dynamics of $\boldsymbol{W}_{13}$ and $\boldsymbol{V}_{31}$. Accordingly, the last token input of the transformer for $i$-th step becomes $\boldsymbol{w}_i$ and the label becomes $\boldsymbol{w}_{i+1}$ since the other entries in the original input/label $(\boldsymbol{0}, 0, \boldsymbol{w}_i, 1)$ do not affect prediction.

**Definition C.1** (Reduced transformer). *Let $\boldsymbol{\theta} = (\widetilde{\boldsymbol{V}}, \widetilde{\boldsymbol{W}})$. Define*

$$f_{\boldsymbol{\theta}}(\boldsymbol{X}, \boldsymbol{Z}_i) = \boldsymbol{w}_i + \widetilde{\boldsymbol{V}} \boldsymbol{S} \left( \widetilde{\boldsymbol{W}} \boldsymbol{w}_i - \boldsymbol{w}^* \right)$$

*to be the reduced model of the one-layer transformer in Equation (3). For ease of presentation, we denote $f_{\boldsymbol{\theta}}(\boldsymbol{w}_i) := f_{\boldsymbol{\theta}}(\boldsymbol{X}, \boldsymbol{Z}_i)$.*

In the following sections, we will consider the equivalent form of transformer. Here we present the gradient with regard to the reduced model. For clarification, throughout this section we will denote $\boldsymbol{w}_{k+1} := \left( \boldsymbol{I} - (\boldsymbol{I} - \eta \boldsymbol{S})^{k+1} \right) \boldsymbol{w}^*$ as the $(k+1)$-th update, and $\boldsymbol{w}^*$ is the ground-truth.

**Lemma C.4.** *The gradient of $\widetilde{\boldsymbol{V}}$ and $\widetilde{\boldsymbol{W}}$ are given by the following expectations:*

$$\frac{\partial \mathcal{L}}{\partial \widetilde{\boldsymbol{V}}} = \sum_{i=0}^{k} \mathbb{E} \left[ (f_{\boldsymbol{\theta}}(\boldsymbol{w}_i) - \boldsymbol{w}_{i+1}) \left( \boldsymbol{w}_i^\top \widetilde{\boldsymbol{W}}^\top - \boldsymbol{w}^{*T} \right) \boldsymbol{S} \right] + \mathbb{E} \left[ (\boldsymbol{w}_{k+1} - \boldsymbol{w}^*) \left( \boldsymbol{w}_k^\top \widetilde{\boldsymbol{W}}^\top - \boldsymbol{w}^{*T} \right) \boldsymbol{S} \right],$$

$$\frac{\partial \mathcal{L}}{\partial \widetilde{\boldsymbol{W}}} = \sum_{i=0}^{k} \mathbb{E} \left[ \boldsymbol{S} \widetilde{\boldsymbol{V}}^\top (f_{\boldsymbol{\theta}}(\boldsymbol{w}_i) - \boldsymbol{w}_{i+1}) \boldsymbol{w}_i^\top \right] + \mathbb{E} \left[ \boldsymbol{S} \widetilde{\boldsymbol{V}}^\top (\boldsymbol{w}_{k+1} - \boldsymbol{w}^*) \boldsymbol{w}_k^\top \right].$$

*Proof.* Given the equivalent CoT loss in Equation (13), we take the gradient with regard to $\widetilde{\boldsymbol{V}}$ of the loss and we have

$$\frac{\partial \mathcal{L}}{\partial \widetilde{\boldsymbol{V}}} = \sum_{i=0}^{k-1} \mathbb{E} \left[ (f_{\boldsymbol{\theta}}(\boldsymbol{w}_i) - \boldsymbol{w}_{i+1}) \left( \boldsymbol{w}_i^\top \widetilde{\boldsymbol{W}}^\top - \boldsymbol{w}^{*T} \right) \boldsymbol{S} \right] + \mathbb{E} \left[ (f_{\boldsymbol{\theta}}(\boldsymbol{w}_k) - \boldsymbol{w}^*) \left( \boldsymbol{w}_k^\top \widetilde{\boldsymbol{W}}^\top - \boldsymbol{w}^{*T} \right) \boldsymbol{S} \right]$$

$$= \sum_{i=0}^{k} \mathbb{E} \left[ (f_{\boldsymbol{\theta}}(\boldsymbol{w}_i) - \boldsymbol{w}_{i+1}) \left( \boldsymbol{w}_i^\top \widetilde{\boldsymbol{W}}^\top - \boldsymbol{w}^{*T} \right) \boldsymbol{S} \right] + \mathbb{E} \left[ (\boldsymbol{w}_{k+1} - \boldsymbol{w}^*) \left( \boldsymbol{w}_k^\top \widetilde{\boldsymbol{W}}^\top - \boldsymbol{w}^{*T} \right) \boldsymbol{S} \right]$$

The second step is because we subtract $\mathbb{E} \left[ (f_{\boldsymbol{\theta}}(\boldsymbol{w}_k) - \boldsymbol{w}_{k+1}) \left( \boldsymbol{w}_k^\top \widetilde{\boldsymbol{W}}^\top - \boldsymbol{w}^{*T} \right) \boldsymbol{S} \right]$ from the second term and put it into the summation. Similarly, the partial derivative of $\widetilde{\boldsymbol{W}}$ should be:

$$\frac{\partial \mathcal{L}}{\partial \widetilde{\boldsymbol{W}}} = \sum_{i=0}^{k-1} \mathbb{E} \left[ \boldsymbol{S} \widetilde{\boldsymbol{V}}^\top (f_{\boldsymbol{\theta}}(\boldsymbol{w}_i) - \boldsymbol{w}_{i+1}) \boldsymbol{w}_i^\top \right] + \mathbb{E} \left[ \boldsymbol{S} \widetilde{\boldsymbol{V}}^\top (f_{\boldsymbol{\theta}}(\boldsymbol{w}_k) - \boldsymbol{w}^*) \boldsymbol{w}_k^\top \right]$$

$$= \sum_{i=0}^{k} \mathbb{E} \left[ \boldsymbol{S} \widetilde{\boldsymbol{V}}^\top (f_{\boldsymbol{\theta}}(\boldsymbol{w}_i) - \boldsymbol{w}_{i+1}) \boldsymbol{w}_i^\top \right] + \mathbb{E} \left[ \boldsymbol{S} \widetilde{\boldsymbol{V}}^\top (\boldsymbol{w}_{k+1} - \boldsymbol{w}^*) \boldsymbol{w}_k^\top \right]$$

Therefore we complete the proof. $\square$

## C.2 Gradient characterization over the CoT objective

In this section, we compute the exact gradient for the reduced model parameters to facilitate analysis on the dynamics. For clarification, throughout this section we will denote $\boldsymbol{w}_{k+1} := \left( \boldsymbol{I} - (\boldsymbol{I} - \eta \boldsymbol{S})^{k+1} \right) \boldsymbol{w}^*$ as the $(k+1)$-th update, and $\boldsymbol{w}^*$ is the ground-truth.

We first compute our gradients for the simplfied model defined in Definition C.1, which is equivalent to the full model's dynamics. Recall that under assumption 4.1, we have $\widetilde{V}, \widetilde{W}$ are simultaneously diagonalizable, with the orthonormal basis $\{u_i\}_{i=1}^d$. We denote the orthogonal matrix formed by the basis as $U$. We will observe that $u_i$ are always the eigenvector of $\widetilde{V}, \widetilde{W}$, so we denote $\widetilde{V} = U\Lambda^{\widetilde{V}}U^\top, \widetilde{W} = U\Lambda^{\widetilde{W}}U^\top$. For clarity, we ignore the timestamp when calculating the gradients and dynamics.

We present an accurate estimate of the gradient in the following Lemma C.5. We intensively use the concentration lemma in Appendix D to separate the main terms dominating the gradient flow dynamics, and some bounded error terms that may complicate the analysis. We also call the error terms as 'interaction terms', since they contain the interactions between two subspaces $u_i u_i^\top$ and $u_j u_j^\top$. The structure of the interaction terms $\Delta^{\widetilde{V}}, \Delta^{\widetilde{W}}$ are further characterized in this lemma, which is essential for the final local convergence analysis.

**Lemma C.5.** *Suppose $n = \Theta(d\log^5 d)$, $\eta \in (0.1, 0.9)$, $k = \lceil c\log d\rceil$. Under Assumption 4.1, if we run gradient flow on the population loss in Equation (6), then the gradient of $\widetilde{V}$ and $\widetilde{W}$ are characterized by the following equations:*

$$U^\top \frac{\partial\mathcal{L}}{\partial\widetilde{V}} U = \left[\left(k+1-\frac{2}{\eta}+\frac{1}{\eta(2-\eta)}\right)\Lambda^{\widetilde{W}^2} - 2\left(k+1-\frac{1}{\eta}\right)\Lambda^{\widetilde{W}} + (k+1)I\right]\Lambda^{\widetilde{V}}$$
$$-\frac{1-\eta}{2-\eta}\Lambda^{\widetilde{W}} + I + \Delta^{\widetilde{V}},$$

$$U^\top \frac{\partial\mathcal{L}}{\partial\widetilde{W}} U = \left(k+1-\frac{2}{\eta}+\frac{1}{\eta(2-\eta)}\right)\Lambda^{\widetilde{V}^2}\Lambda^{\widetilde{W}} - \left(k+1-\frac{1}{\eta}\right)\Lambda^{\widetilde{V}^2} - \frac{1-\eta}{2-\eta}\Lambda^{\widetilde{V}} + \Delta^{\widetilde{W}}.$$

*where the error terms (interaction terms) $\left\|\Delta^{\widetilde{V}}\right\|_{op} \leq O\left(\frac{1}{\log^2 d}\right), \left\|\Delta^{\widetilde{W}}\right\|_{op} \leq O\left(\frac{1}{\log^2 d}\right)$. Moreover, there exist diagonal matrices $A^{\widetilde{V}}, B^{\widetilde{V}}, A^{\widetilde{W}}, B^{\widetilde{W}}$ with $O\left(\frac{1}{\log^2 d}\right)$-operator norm, $C^{\widetilde{V}}, D^{\widetilde{V}}, C^{\widetilde{W}}, D^{\widetilde{W}}, E^{\widetilde{W}}$ with $O\left(\frac{1}{d\log^2 d}\right)$-operator norm and $E^{\widetilde{V}}, F^{\widetilde{W}}$ with $O\left((1-\eta)^k\right)$-operator norm s.t. the error terms $\Delta^{\widetilde{V}}, \Delta^{\widetilde{W}}$ can be written as*

$$\Delta^{\widetilde{V}} = \left(\Lambda^{\widetilde{V}}+\eta I\right)A^{\widetilde{V}} + \left(I-\Lambda^{\widetilde{W}}\right)B^{\widetilde{V}} + \mathrm{tr}\left(\left(I-\Lambda^{\widetilde{W}}\right)\Lambda^{\widetilde{W}}\right)C^{\widetilde{V}} + \mathrm{tr}\left(I-\Lambda^{\widetilde{W}}\right)D^{\widetilde{V}} + E^{\widetilde{V}},$$

$$\Delta^{\widetilde{W}} = \left(\Lambda^{\widetilde{V}}+\eta I\right)A^{\widetilde{W}} + \left(I-\Lambda^{\widetilde{W}}\right)B^{\widetilde{W}} + \mathrm{tr}\left(I-\Lambda^{\widetilde{W}}\right)C^{\widetilde{W}} + \mathrm{tr}\left((\Lambda^{\widetilde{V}}+\eta I)\Lambda^{\widetilde{V}}\right)D^{\widetilde{W}}$$
$$+ \mathrm{tr}\left((I-\Lambda^{\widetilde{W}})\Lambda^{\widetilde{V}^2}\right)E^{\widetilde{W}} + F^{\widetilde{W}}.$$

*Proof.* Recall the gradients formula of $\widetilde{V}$ and $\widetilde{W}$ by Lemma C.4:

$$\frac{\partial\mathcal{L}}{\partial\widetilde{V}} = \sum_{i=0}^k \mathbb{E}\left[(f_\theta(w_i)-w_{i+1})\left(w_i^\top\widetilde{W}^\top - w^{*T}\right)S\right] + \mathbb{E}\left[(w_{k+1}-w^*)\left(w_k^\top\widetilde{W}^\top - w^{*T}\right)S\right]$$

$$\frac{\partial\mathcal{L}}{\partial\widetilde{W}} = \sum_{i=0}^k \mathbb{E}\left[S\widetilde{V}^\top(f_\theta(w_i)-w_{i+1})w_i^\top\right] + \mathbb{E}\left[S\widetilde{V}^\top(w_{k+1}-w^*)w_k^\top\right]$$

We expand the reduced model $f_\theta(w_i)$ in Definition C.1, and get the residual term

$$f_\theta(w_i) - w_{i+1} = w_i + \widetilde{V}S\left(\widetilde{W}w_i - w^*\right) - w_{i+1}$$
$$= \widetilde{V}S\left(\widetilde{W}w_i - w^*\right) + \eta S(w_i - w^*)$$
$$= \left(\widetilde{V}S\widetilde{W} + \eta S\right)w_i - \left(\widetilde{V}+\eta I\right)Sw^*$$

Substitute $f_\theta(w_i) - w_{i+1}$ term in the dynamics by the equation above, we have

$$\frac{\partial\mathcal{L}}{\partial\widetilde{V}} = \sum_{i=0}^k \mathbb{E}\left[\left(\widetilde{V}S\widetilde{W} + \eta S\right)\left(I-(I-\eta S)^i\right)^2\widetilde{W}^\top S\right] - \sum_{i=0}^k \mathbb{E}\left[\left(\widetilde{V}S\widetilde{W} + \eta S\right)\left(I-(I-\eta S)^i\right)S\right]$$

$$- \sum_{i=0}^{k} \mathbb{E}\left[\left(\widetilde{V} + \eta I\right) S\left(I - (I - \eta S)^i\right)\widetilde{W}^\top S\right] + \sum_{i=0}^{k} \mathbb{E}\left[\left(\widetilde{V} + \eta I\right) S^2\right]$$

$$- \mathbb{E}\left[(I - \eta S)^{k+1}\left(\left(I - (I - \eta S)^k\right)\widetilde{W}^\top - I\right)\right]$$

$$= \sum_{i=0}^{k} \left(\widetilde{V} + \eta I\right)\mathbb{E}\left[S\widetilde{W}\left(I - (I - \eta S)^i\right)^2\widetilde{W}^\top S\right] \qquad \text{(Term 1)}$$

$$+ \eta \sum_{i=0}^{k} \mathbb{E}\left[S\left(I - \widetilde{W}\right)\left(I - (I - \eta S)^i\right)^2\widetilde{W}^\top S\right] \qquad \text{(Term 2)}$$

$$- \sum_{i=0}^{k} \left(\widetilde{V} + \eta I\right)\mathbb{E}\left[S\widetilde{W}\left(I - (I - \eta S)^i\right)S\right] \qquad \text{(Term 3)}$$

$$- \eta \sum_{i=0}^{k} \mathbb{E}\left[S\left(I - \widetilde{W}\right)\left(I - (I - \eta S)^i\right)S\right] \qquad \text{(Term 4)}$$

$$- \sum_{i=0}^{k} \left(\widetilde{V} + \eta I\right)\mathbb{E}\left[S\left(I - (I - \eta S)^i\right)\widetilde{W}^\top S\right] \qquad \text{(Term 5)}$$

$$+ \sum_{i=0}^{k} \left(\widetilde{V} + \eta I\right)\mathbb{E}\left[S^2\right] \qquad \text{(Term 6)}$$

$$- \mathbb{E}\left[(I - \eta S)^{k+1}\left(\left(I - (I - \eta S)^k\right)\widetilde{W}^\top - I\right)\right]. \qquad \text{(Term 7)}$$

To get an accurate estimate of the gradient, we apply Lemma C.14, Lemma C.15 respectively to each of the terms (Term 1 to Term 7) and separate the interaction terms introduced by the moments of Wishart matrix, which is bounded by $O\left(\frac{1}{\log^3 d}\right)$.

Consider Term 7 and the $i$-th term in the summation of Term 1 to Term 6. By Lemma C.14 and Lemma C.15, there exist diagonal matrices $\boldsymbol{\xi}_j, j \in [6]$ satisfying $\|\boldsymbol{\xi}_j\|_{op} \leq O\left(\frac{1}{\log^3 d}\right)$ such that

$$\mathbb{E}\left[S\widetilde{W}\left(I - (I - \eta S)^i\right)^2\widetilde{W}^\top S\right] = U\left[\left(1 - (1 - \eta)^k\right)^2\Lambda^{\widetilde{W}^2} + \boldsymbol{\xi}_1\right]U^\top$$

$$\mathbb{E}\left[S\left(I - \widetilde{W}\right)\left(I - (I - \eta S)^i\right)^2\widetilde{W}^\top S\right] = U\left[\left(1 - (1 - \eta)^k\right)^2\left(I - \Lambda^{\widetilde{W}}\right)\Lambda^{\widetilde{W}} + \boldsymbol{\xi}_2\right]U^\top$$

$$\mathbb{E}\left[S\widetilde{W}\left(I - (I - \eta S)^i\right)S\right] = U\left[\left(1 - (1 - \eta)^k\right)\Lambda^{\widetilde{W}} + \boldsymbol{\xi}_3\right]U^\top$$

$$\mathbb{E}\left[S\left(I - \widetilde{W}\right)\left(I - (I - \eta S)^i\right)S\right] = U\left[\left(1 - (1 - \eta)^k\right)\left(I - \Lambda^{\widetilde{W}}\right) + \boldsymbol{\xi}_4\right]U^\top$$

$$\mathbb{E}\left[S\left(I - (I - \eta S)^i\right)\widetilde{W}^\top S\right] = U\left[\left(1 - (1 - \eta)^k\right)\Lambda^{\widetilde{W}} + \boldsymbol{\xi}_5\right]U^\top$$

$$\mathbb{E}\left[S^2\right] = U(I + \boldsymbol{\xi}_6)U^\top$$

By Lemma D.4, there exists diagonal matrix $\boldsymbol{\xi}_7$ satisfying $\|\boldsymbol{\xi}_7\|_{op} \leq O\left((1 - \eta)^k\right)$ such that

$$\mathbb{E}\left[(I - \eta S)^{k+1}\left(\left(I - (I - \eta S)^k\right)\widetilde{W}^\top - I\right)\right] = U\boldsymbol{\xi}_7 U^\top.$$

Moreover, there exist $\alpha_1, \alpha_2 \leq O\left(\frac{1}{\log^3 d}\right)$, $\alpha_3, \alpha_4, \alpha_5 \leq O\left(\frac{1}{d\log^3 d}\right)$ such that

$$\boldsymbol{\xi}_2 = \left(\alpha_1\Lambda^{\widetilde{W}} + \alpha_2 I\right)\left(I - \Lambda^{\widetilde{W}}\right) + \text{tr}\left(I - \Lambda^{\widetilde{W}}\right)\left(\alpha_3\Lambda^{\widetilde{W}} + \alpha_4 I\right) + \alpha_5 \text{tr}\left(\left(I - \Lambda^{\widetilde{W}}\right)\Lambda^{\widetilde{W}}\right)I,$$

and exist $\beta_1 \leq O\left(\frac{1}{\log^3 d}\right)$, $\beta_2 \leq O\left(\frac{1}{d\log^3 d}\right)$ such that

$$\boldsymbol{\xi}_4 = \beta_1\left(I - \Lambda^{\widetilde{W}}\right) + \beta_2 \text{tr}\left(I - \Lambda^{\widetilde{W}}\right)I.$$

We define $\Delta_i^{\widetilde{V}}$ as the sum of all the interaction terms $\left(\boldsymbol{\Lambda}^{\widetilde{V}} + \eta \boldsymbol{I}\right)(\boldsymbol{\xi}_1 - \boldsymbol{\xi}_3 - \boldsymbol{\xi}_5 + \boldsymbol{\xi}_6) + \eta(\boldsymbol{\xi}_2 - \boldsymbol{\xi}_4)$ for the $i$-th term in the summation of dynamics of $\widetilde{V}$. From the analysis above, there exist diagonal matrices $\boldsymbol{A}_i^{\widetilde{V}}$, $\boldsymbol{B}_i^{\widetilde{V}}$, $\boldsymbol{C}_i^{\widetilde{V}}$, $\boldsymbol{D}_i^{\widetilde{V}}$ with their operator norm $O\left(\frac{1}{\log^3 d}\right)$, such that (note every matrix is diagonal, so they commute)

$$\Delta_i^{\widetilde{V}} = \left(\boldsymbol{\Lambda}^{\widetilde{V}} + \eta \boldsymbol{I}\right)\boldsymbol{A}_i^{\widetilde{V}} + O\left(\frac{1}{d}\right)\operatorname{tr}\left(\left(\boldsymbol{I} - \boldsymbol{\Lambda}^{\widetilde{W}}\right)\boldsymbol{\Lambda}^{\widetilde{W}}\right)\boldsymbol{B}_i^{\widetilde{V}}$$
$$+ \left(\boldsymbol{I} - \boldsymbol{\Lambda}^{\widetilde{W}}\right)\boldsymbol{C}_i^{\widetilde{V}} + O\left(\frac{1}{d}\right)\operatorname{tr}\left(\boldsymbol{I} - \boldsymbol{\Lambda}^{\widetilde{W}}\right)\boldsymbol{D}_i^{\widetilde{V}}.$$

We define $\Delta_{-1}^{\widetilde{V}}$ as the interaction term brought by Term 7 since there is no summation in Term 7. It is obvious that $\left\|\Delta_{-1}^{\widetilde{V}}\right\|_{op} \leq O\left((1-\eta)^k\right)$.

Now we denote

$$\widehat{\Delta}^{\widetilde{V}} = \sum_{i=0}^{k} \Delta_i^{\widetilde{V}} - \Delta_{-1}^{\widetilde{V}}$$

to be the sum of all interaction term of the dynamics of $\boldsymbol{\Lambda}^{\widetilde{V}}$. From the definition of $\Delta_i^{\widetilde{V}}$ and $\Delta_{-1}^{\widetilde{V}}$ above, there exist diagonal matrices $\boldsymbol{A}^{\widetilde{V}}$, $\boldsymbol{B}^{\widetilde{V}}$, $\boldsymbol{C}^{\widetilde{V}}$, $\boldsymbol{D}^{\widetilde{V}}$ and $\boldsymbol{E}_0^{\widetilde{V}}$ satisfying $\left\|\boldsymbol{A}^{\widetilde{V}}\right\|, \left\|\boldsymbol{C}^{\widetilde{V}}\right\| \leq O\left(\frac{1}{\log^2 d}\right)$, $\left\|\boldsymbol{B}^{\widetilde{V}}\right\|, \left\|\boldsymbol{D}^{\widetilde{V}}\right\| \leq O\left(\frac{1}{d \log^2 d}\right)$ and $\left\|\boldsymbol{E}_0^{\widetilde{V}}\right\| \leq O\left((1-\eta)^k\right)$ such that (because $k = \Theta(\log d)$)

$$\widehat{\Delta}^{\widetilde{V}} = \left(\boldsymbol{\Lambda}^{\widetilde{V}} + \eta \boldsymbol{I}\right)\boldsymbol{A}^{\widetilde{V}} + \operatorname{tr}\left(\left(\boldsymbol{I} - \boldsymbol{\Lambda}^{\widetilde{W}}\right)\boldsymbol{\Lambda}^{\widetilde{W}}\right)\boldsymbol{B}^{\widetilde{V}} + \left(\boldsymbol{I} - \boldsymbol{\Lambda}^{\widetilde{W}}\right)\boldsymbol{C}^{\widetilde{V}} + \operatorname{tr}\left(\boldsymbol{I} - \boldsymbol{\Lambda}^{\widetilde{W}}\right)\boldsymbol{D}^{\widetilde{V}} + \boldsymbol{E}_0^{\widetilde{V}}$$

Sum up all the seven terms together and we have

$$\frac{\partial \mathcal{L}}{\partial \widetilde{V}} = \boldsymbol{U}\left[\left(k + 1 - \frac{2\left(1 - (1-\eta)^{k+1}\right)}{\eta} + \frac{1 - (1-\eta)^{2k+2}}{\eta(2-\eta)}\right)\boldsymbol{\Lambda}^{\widetilde{W}}\left(\boldsymbol{\Lambda}^{\widetilde{V}}\boldsymbol{\Lambda}^{\widetilde{W}} + \eta \boldsymbol{I}\right)\right]\boldsymbol{U}^{\top}$$
$$- \boldsymbol{U}\left[\left(k + 1 - \frac{1 - (1-\eta)^{k+1}}{\eta}\right)\left(\boldsymbol{\Lambda}^{\widetilde{V}}\boldsymbol{\Lambda}^{\widetilde{W}} + \eta \boldsymbol{I}\right)\right]\boldsymbol{U}^{\top}$$
$$- \boldsymbol{U}\left[\left(k + 1 - \frac{1 - (1-\eta)^{k+1}}{\eta}\right)\boldsymbol{\Lambda}^{\widetilde{W}}\left(\boldsymbol{\Lambda}^{\widetilde{V}} + \eta \boldsymbol{I}\right)\right]\boldsymbol{U}^{\top}$$
$$+ \boldsymbol{U}\left[(k+1)\left(\boldsymbol{\Lambda}^{\widetilde{V}} + \eta \boldsymbol{I}\right)\right]\boldsymbol{U}^{\top} + \boldsymbol{U}\widehat{\Delta}^{\widetilde{V}}\boldsymbol{U}^{\top}$$

Denote $\boldsymbol{E}_1^{\widetilde{V}}$ to be the sum of all $O\left((1-\eta)^k\right)$ terms in the dynamics of $\widetilde{V}$:

$$\boldsymbol{E}_1^{\widetilde{V}} = \left(\frac{2(1-\eta)^{k+1}}{\eta} - \frac{(1-\eta)^{2k+2}}{\eta(2-\eta)}\right)\boldsymbol{\Lambda}^{\widetilde{W}}\left(\boldsymbol{\Lambda}^{\widetilde{V}}\boldsymbol{\Lambda}^{\widetilde{W}} + \eta \boldsymbol{I}\right) - \frac{(1-\eta)^{k+1}}{\eta}\left(2\boldsymbol{\Lambda}^{\widetilde{V}}\boldsymbol{\Lambda}^{\widetilde{W}} + \eta\boldsymbol{\Lambda}^{\widetilde{W}} + \eta \boldsymbol{I}\right)$$

Denote $\boldsymbol{E}^{\widetilde{V}} = \boldsymbol{E}_0^{\widetilde{V}} + \boldsymbol{E}_1^{\widetilde{V}}$ and denote $\Delta^{\widetilde{V}} = \widehat{\Delta}^{\widetilde{V}} + \boldsymbol{E}_1^{\widetilde{V}}$, we have

$$\boldsymbol{U}^{\top}\frac{\partial \mathcal{L}}{\partial \widetilde{V}}\boldsymbol{U} = \left[\left(k + 1 - \frac{2}{\eta} + \frac{1}{\eta(2-\eta)}\right)\boldsymbol{\Lambda}^{\widetilde{W}^2} - 2\left(k + 1 - \frac{1}{\eta}\right)\boldsymbol{\Lambda}^{\widetilde{W}} + (k+1)\boldsymbol{I}\right]\boldsymbol{\Lambda}^{\widetilde{V}}$$
$$- \frac{1-\eta}{2-\eta}\boldsymbol{\Lambda}^{\widetilde{W}} + \boldsymbol{I} + \Delta^{\widetilde{V}}$$

Moreover, $\Delta^{\widetilde{V}}$ has the form

$$\Delta^{\widetilde{V}} = \left(\boldsymbol{\Lambda}^{\widetilde{V}} + \eta \boldsymbol{I}\right)\boldsymbol{A}^{\widetilde{V}} + \operatorname{tr}\left(\left(\boldsymbol{I} - \boldsymbol{\Lambda}^{\widetilde{W}}\right)\boldsymbol{\Lambda}^{\widetilde{W}}\right)\boldsymbol{B}^{\widetilde{V}} + \left(\boldsymbol{I} - \boldsymbol{\Lambda}^{\widetilde{W}}\right)\boldsymbol{C}^{\widetilde{V}} + \operatorname{tr}\left(\boldsymbol{I} - \boldsymbol{\Lambda}^{\widetilde{W}}\right)\boldsymbol{D}^{\widetilde{V}} + \boldsymbol{E}^{\widetilde{V}}$$

Similar to the calculation of the dynamics of $\widetilde{V}$, we can also have

$$
\begin{aligned}
\frac{\partial \mathcal{L}}{\partial \widetilde{W}} &= \sum_{i=0}^{k} \mathbb{E}\Big[ S\widetilde{V}^{\top}(f_{\theta}(\boldsymbol{w}_i) - \boldsymbol{w}_{i+1})\boldsymbol{w}_i^{\top}\Big] + \mathbb{E}\Big[ S\widetilde{V}^{\top}(\boldsymbol{w}_{k+1} - \boldsymbol{w}^{*})\boldsymbol{w}_k^{\top}\Big] \\
&= \sum_{i=0}^{k} \mathbb{E}\Big[ S\widetilde{V}^{\top}(\widetilde{V}S\widetilde{W} + \eta S)\Big(I - (I - \eta S)^{i}\Big)^{2}\Big] - \sum_{i=0}^{k} \mathbb{E}\Big[ S\widetilde{V}^{\top}\Big(\widetilde{V} + \eta I\Big)S\Big(I - (I - \eta S)^{i}\Big)\Big] \\
&\quad - \mathbb{E}\Big[ S\widetilde{V}^{\top}(I - \eta S)^{k+1}\Big(I - (I - \eta S)^{k}\Big)\Big] \\
&= \sum_{i=0}^{k} \mathbb{E}\Big[ S\widetilde{V}^{\top}\Big(\widetilde{V}S\big(\widetilde{W} - I\big) + (V + \eta I)S\Big)\Big(I - (I - \eta S)^{i}\Big)^{2}\Big] \\
&\quad - \sum_{i=0}^{k} \mathbb{E}\Big[ S\widetilde{V}^{\top}\Big(\widetilde{V} + \eta I\Big)S\Big(I - (I - \eta S)^{i}\Big)\Big] - \mathbb{E}\Big[ S\widetilde{V}^{\top}(I - \eta S)^{k+1}\Big(I - (I - \eta S)^{k}\Big)\Big] \\
&= \sum_{i=0}^{k} \mathbb{E}\Big[ S\widetilde{V}^{\top}\widetilde{V}S\big(\widetilde{W} - I\big)\Big(I - (I - \eta S)^{i}\Big)^{2}\Big] + \sum_{i=0}^{k} \mathbb{E}\Big[ S\widetilde{V}^{\top}\Big(\widetilde{V} + \eta I\Big)S\Big(I - (I - \eta S)^{i}\Big)^{2}\Big] \\
&\quad - \sum_{i=0}^{k} \mathbb{E}\Big[ S\widetilde{V}^{\top}\Big(\widetilde{V} + \eta I\Big)S\Big(I - (I - \eta S)^{i}\Big)\Big] - \mathbb{E}\Big[ S\widetilde{V}^{\top}(I - \eta S)^{k+1}\Big(I - (I - \eta S)^{k}\Big)\Big]
\end{aligned}
$$

We apply Lemma C.16 and Lemma C.17 to each term, similarly define $\Delta_i^{\widetilde{W}}$ for $i \in [k] \cup \{0\}$ as the sum of all interaction terms for the $i$-th term in the smmation of dynamics of $\widetilde{W}$. There exists diagonal matrices $A_i^{\widetilde{W}}, B_i^{\widetilde{W}}, C_i^{\widetilde{W}}, D_i^{\widetilde{W}}, E_i^{\widetilde{W}}$ with their operator norm $O\Big(\frac{1}{\log^3 d}\Big)$, such that

$$
\begin{aligned}
\Delta_i^{\widetilde{W}} &= \Big(\Lambda^{\widetilde{V}} + \eta I\Big)A_i^{\widetilde{W}} + \Big(I - \Lambda^{\widetilde{W}}\Big)B_i^{\widetilde{W}} + O\Big(\frac{1}{d}\Big)\operatorname{tr}\Big(I - \Lambda^{\widetilde{W}}\Big)C_i^{\widetilde{W}} \\
&\quad + O\Big(\frac{1}{d}\Big)\operatorname{tr}\Big((\Lambda^{\widetilde{V}} + \eta I)\Lambda^{\widetilde{V}}\Big)D_i^{\widetilde{W}} + O\Big(\frac{1}{d}\Big)\operatorname{tr}\Big((I - \Lambda^{\widetilde{W}})\Lambda^{\widetilde{V}^2}\Big)E_i^{\widetilde{W}}
\end{aligned}
$$

We define $\Delta_{-1}^{\widetilde{W}}$ as the interaction term brought by the last term

$$
\mathbb{E}\Big[ S\widetilde{V}^{\top}(I - \eta S)^{k+1}\Big(I - (I - \eta S)^{k}\Big)\Big].
$$

It is clear that $\left\|\Delta_{-1}^{\widetilde{W}}\right\|_{op} \le O\Big((1 - \eta)^{k}\Big)$. Similarly denote

$$
\widehat{\Delta}^{\widetilde{W}} = \sum_{i=0}^{k} \Delta_i^{\widetilde{W}} - \Delta_{-1}^{\widetilde{W}},
$$

then there exist diagonal matrices $A^{\widetilde{W}}, B^{\widetilde{W}}, C^{\widetilde{W}}, D^{\widetilde{W}}, E^{\widetilde{W}}, F_0^{\widetilde{W}}$ satisfying $\left\|A^{\widetilde{W}}\right\|, \left\|B^{\widetilde{W}}\right\| \le O\Big(\frac{1}{\log^2 d}\Big), \left\|C^{\widetilde{W}}\right\|, \left\|D^{\widetilde{W}}\right\|, \left\|E^{\widetilde{W}}\right\| \le O\Big(\frac{1}{d\log^2 d}\Big), \left\|F_0^{\widetilde{W}}\right\| \le O\Big((1 - \eta)^{k}\Big)$ such that

$$
\begin{aligned}
\widehat{\Delta}^{\widetilde{W}} &= \Big(\Lambda^{\widetilde{V}} + \eta I\Big)A^{\widetilde{W}} + \Big(I - \Lambda^{\widetilde{W}}\Big)B^{\widetilde{W}} + \operatorname{tr}\Big(I - \Lambda^{\widetilde{W}}\Big)C^{\widetilde{W}} \\
&\quad + \operatorname{tr}\Big((\Lambda^{\widetilde{V}} + \eta I)\Lambda^{\widetilde{V}}\Big)D^{\widetilde{W}} + \operatorname{tr}\Big((I - \Lambda^{\widetilde{W}})\Lambda^{\widetilde{V}^2}\Big)E^{\widetilde{W}} + F_0^{\widetilde{W}}.
\end{aligned}
$$

Denote $F_1^{\widetilde{W}}$ to be the sum of all $O\Big((1 - \eta)^{k}\Big)$ terms in the dynamics of $\widetilde{W}$, $F^{\widetilde{W}} = F_0^{\widetilde{W}} + F_1^{\widetilde{W}}$ and $\Delta^{\widetilde{W}} = \widehat{\Delta}^{\widetilde{W}} + F_1^{\widetilde{W}}$. Thus we have

$$
\frac{\partial \mathcal{L}}{\partial \widetilde{W}} = \sum_{i=0}^{k} U\Big(1 - (1 - \eta)^{i}\Big)^{2}\Lambda^{\widetilde{V}}\Big(\Lambda^{\widetilde{V}}\Lambda^{\widetilde{W}} + \eta I\Big)U^{\top} - \sum_{i=0}^{k} U\Big(1 - (1 - \eta)^{i}\Big)\Lambda^{\widetilde{V}}\Big(\Lambda^{\widetilde{V}} + \eta I\Big)U^{\top} + U\Delta^{\widetilde{W}}U^{\top}
$$

$$= U\left[\left(k+1-\frac{2\left(1-(1-\eta)^{k+1}\right)}{\eta}+\frac{1-(1-\eta)^{2k+2}}{\eta(2-\eta)}\right)\boldsymbol{\Lambda}^{\tilde{V}}\left(\boldsymbol{\Lambda}^{\tilde{V}}\boldsymbol{\Lambda}^{\widetilde{W}}+\eta\boldsymbol{I}\right)\right]\boldsymbol{U}^{\top}$$

$$-U\left[\left(k+1-\frac{1-(1-\eta)^{k+1}}{\eta}\right)\boldsymbol{\Lambda}^{\tilde{V}}\left(\boldsymbol{\Lambda}^{\tilde{V}}+\eta\boldsymbol{I}\right)\right]\boldsymbol{U}^{\top}+\boldsymbol{U}\widehat{\Delta}^{\widetilde{W}}\boldsymbol{U}^{\top}$$

$$= U\left[\left(k+1-\frac{2}{\eta}+\frac{1}{\eta(2-\eta)}\right)\boldsymbol{\Lambda}^{\tilde{V}2}\boldsymbol{\Lambda}^{\widetilde{W}}-\left(k+1-\frac{1}{\eta}\right)\boldsymbol{\Lambda}^{\tilde{V}2}-\frac{1-\eta}{2-\eta}\boldsymbol{\Lambda}^{\tilde{V}}+\Delta^{\widetilde{W}}\right]\boldsymbol{U}^{\top}$$

Moreover, $\Delta^{\widetilde{W}}$ has the form

$$\widehat{\Delta}^{\widetilde{W}}=\left(\boldsymbol{\Lambda}^{\tilde{V}}+\eta\boldsymbol{I}\right)\boldsymbol{A}^{\widetilde{W}}+\left(\boldsymbol{I}-\boldsymbol{\Lambda}^{\widetilde{W}}\right)\boldsymbol{B}^{\widetilde{W}}+\operatorname{tr}\left(\boldsymbol{I}-\boldsymbol{\Lambda}^{\widetilde{W}}\right)\boldsymbol{C}^{\widetilde{W}}$$
$$+\operatorname{tr}\left((\boldsymbol{\Lambda}^{\tilde{V}}+\eta\boldsymbol{I})\boldsymbol{\Lambda}^{\tilde{V}}\right)\boldsymbol{D}^{\widetilde{W}}+\operatorname{tr}\left((\boldsymbol{I}-\boldsymbol{\Lambda}^{\widetilde{W}})\boldsymbol{\Lambda}^{\tilde{V}2}\right)\boldsymbol{E}^{\widetilde{W}}+\boldsymbol{F}^{\widetilde{W}}.$$

Since $\|\boldsymbol{A}+\boldsymbol{B}\|_{op}\leq\|\boldsymbol{A}\|_{op}+\|\boldsymbol{B}\|_{op}$ and $\|\boldsymbol{AB}\|_{op}\leq\|\boldsymbol{A}\|_{op}\|\boldsymbol{B}\|_{op}$, it is obvious that

$$\left\|\Delta^{\tilde{V}}\right\|_{op}\leq O\left(\frac{1}{\log^2 d}\right),\quad\left\|\Delta^{\widetilde{W}}\right\|_{op}\leq O\left(\frac{1}{\log^2 d}\right)$$

$\square$

After obtaining the estimation of the gradient by lemma C.5, we can decompose the gradient updates into the dynamics along each eigenspace $\boldsymbol{u}_i$, which can be characterized by the following lemma.

**Lemma C.6.** *Suppose* $\widetilde{V}=\sum_{j=1}^d\lambda_j^{\tilde{V}}\boldsymbol{u}_j\boldsymbol{u}_j^{\top},\widetilde{W}=\sum_{j=1}^d\lambda_j^{\widetilde{W}}\boldsymbol{u}_j\boldsymbol{u}_j^{\top}.$ *The dynamics of the eigenvalues of* $\widetilde{V}$ *and* $\widetilde{W}$ *are given by the following equations:*

$$\frac{\mathrm{d}\lambda_j^{\tilde{V}}}{\mathrm{d}t}=-\left[(k+1)\left(1-\lambda_j^{\widetilde{W}}\right)^2+\frac{2}{\eta}\lambda_j^{\widetilde{W}}\left(1-\lambda_j^{\widetilde{W}}\right)+\frac{1}{\eta(2-\eta)}\lambda_j^{\widetilde{W}2}\right]\lambda_j^{\tilde{V}}+\frac{1-\eta}{2-\eta}\lambda_j^{\widetilde{W}}-1+\delta_j^{\tilde{V}}$$

$$\frac{\mathrm{d}\lambda_j^{\widetilde{W}}}{\mathrm{d}t}=\left(k+1-\frac{1}{\eta}\right)\lambda_j^{\tilde{V}2}\left(1-\lambda_j^{\widetilde{W}}\right)+\frac{1-\eta}{\eta(2-\eta)}\lambda_j^{\tilde{V}2}\lambda_j^{\widetilde{W}}+\frac{1-\eta}{2-\eta}\lambda_j^{\tilde{V}}-\delta_j^{\widetilde{W}}$$

*where* $\left|\delta_j^{\tilde{V}}\right|\leq O\left(\frac{1}{\log^2 d}\right),\left|\delta_j^{\widetilde{W}}\right|\leq O\left(\frac{1}{\log^2 d}\right).$

*Proof.* This is directly obtained from Lemma C.5. $\square$

## C.3 Proof of the main Theorem 4.1

In this section, we prove Theorem 4.1, which characterizes the CoT loss of the trained transformer. First, we restate the theorem.

**Theorem C.1** (Global Convergence). *Suppose* $n=\Theta(d\log^5 d)$, $\eta\in(0.1,0.9)$, $k=\lceil c\log d\rceil$, $c\log\left(\frac{1}{1-\eta}\right)>2$. *Under Assumption 4.1 with some constant* $\sigma>\frac{3(1-\eta)}{(2-\eta)}\frac{1}{k+1}$, *if we run gradient flow on the population loss in Equation* (6), *then after time* $t=O\left(\log d+\log\frac{1}{\epsilon}\right)$, *we have* $\mathcal{L}^{\mathrm{CoT}}(t)\leq\epsilon$ *for any* $\epsilon\geq\Theta\left(\frac{\log d}{d^{c\log\left(\frac{1}{1-\eta}\right)-2}}\right).$

*Proof.* According to the previous sections, we can reduce the original optimization problem to Equation (13), and consider the equivalent reduced model (Definition C.1). By Lemma C.5, we fully characterized the gradient expression, which decomposes the gradient of $\widetilde{V}$ and $\widetilde{W}$ into main signal terms with large norm at initialization (terms before $\Delta^{\tilde{V}},\Delta^{\widetilde{W}}$) and interaction terms ($\Delta^{\tilde{V}},\Delta^{\widetilde{W}}$) with bounded norm $O(\frac{1}{\log^2 d})$ for all $t>0$.

The decomposition motivates us to conduct a stage-wise analysis. We first analyze the dynamics in **Stage 1** when the distance between the parameters $\widetilde{V},\widetilde{W}$ and the ground-truth is larger than

$O(\frac{1}{\log^2 d})$. In this stage, the bounded error can be dominated by the signal terms in the gradient, leading to nearly independent dynamics along each direction $u_i$. After this stage, we enter **Stage 2** as a local convergence phase. We describe the dynamics below in detail.

**Stage 1** In the first stage, the dynamics are dominated by the main terms, and the interaction terms $\Delta^{\widetilde{V}}, \Delta^{\widetilde{W}}$ can be somehow be ignored. Specifically, by Lemma C.6, given the dynamics of $\lambda_j^{\widetilde{V}}, \lambda_j^{\widetilde{W}}$:

$$\frac{d\lambda_j^{\widetilde{V}}}{dt} = -\left[(k+1)\left(1 - \lambda_j^{\widetilde{W}}\right)^2 + \frac{2}{\eta}\lambda_j^{\widetilde{W}}\left(1 - \lambda_j^{\widetilde{W}}\right) + \frac{1}{\eta(2-\eta)}\lambda_j^{\widetilde{W}2}\right]\lambda_j^{\widetilde{V}} + \frac{1-\eta}{2-\eta}\lambda_j^{\widetilde{W}} - 1 + \delta_j^{\widetilde{V}}$$

$$\frac{d\lambda_j^{\widetilde{W}}}{dt} = \left(k+1 - \frac{1}{\eta}\right)\lambda_j^{\widetilde{V}2}\left(1 - \lambda_j^{\widetilde{W}}\right) + \frac{1-\eta}{\eta(2-\eta)}\lambda_j^{\widetilde{V}2}\lambda_j^{\widetilde{W}} + \frac{1-\eta}{2-\eta}\lambda_j^{\widetilde{V}} - \delta_j^{\widetilde{W}}$$

we can conclude that the dynamics of the eigenvalue $\lambda_j^{\widetilde{V}}, \lambda_j^{\widetilde{W}}$ mainly depend on themselves when the main term (terms before $\delta_j^{\widetilde{W}}, \delta_j^{\widetilde{V}}$) are larger than $O(\frac{1}{\log^2 d})$, which is within the stage 1. That is, the dynamics within the subspace $u_i u_i^\top$ for $\widetilde{V}, \widetilde{W}$ are almost independent with other subspaces. In this stage, we focus on the analysis of $\lambda_j^{\widetilde{V}}, \lambda_j^{\widetilde{W}}$ depending on their own value.

The first stage can be further divided into two phases.

**Stage 1, Phase 1.** At the beginning of training, we have

$$\lambda_j^{\widetilde{V}}(0) + \frac{3(1-\eta)}{2(2-\eta)}\frac{1}{(k+1)\left(1 - \lambda_j^{\widetilde{W}}(0)\right)} < -\sigma + \frac{3(1-\eta)}{(2-\eta)}\frac{1}{k+1} < 0$$

then by Lemma C.8, we can prove an upper bound of $\lambda_j^{\widetilde{V}}$ when $\lambda_j^{\widetilde{W}} \leq 1 - (k+1)^{-\frac{7}{12}}$,

$$\lambda_j^{\widetilde{V}} < -\frac{3(1-\eta)}{2(2-\eta)}\frac{1}{(k+1)\left(1 - \lambda_j^{\widetilde{W}}\right)}$$

according to the dynamics for both sides. With this upper bound, we prove $\frac{d\lambda_j^{\widetilde{W}}}{dt} \geq O\left(\frac{1}{k}\right)$. Therefore, $\lambda_j^{\widetilde{W}}$ will converge to $1 - (k+1)^{-\frac{7}{12}}$ in $t_1 = O(\log d)$ time (Lemma C.9).

**Stage 1, Phase 2.** After time $t_1$, we have $\lambda_j^{\widetilde{W}}$ very close to the ground-truth value 1. Meanwhile, the lower bound for $\lambda_j^{\widetilde{V}}$ still holds, and it will further decrease. Specifically,

$$\lambda_j^{\widetilde{W}}(t_1) = 1 - (k+1)^{-\frac{7}{12}} \qquad \lambda_j^{\widetilde{V}}(t_1) < -\frac{3(1-\eta)}{2(2-\eta)}\frac{1}{(k+1)\left(1 - \lambda_j^{\widetilde{W}}(t_1)\right)}$$

By Lemma C.10, we can prove that $\lambda_j^{\widetilde{W}}$ will stay close to $1 - o(1)$:

$$1 - 2(k+1)^{-\frac{7}{12}} < \lambda_j^{\widetilde{W}}(t) < 1 + (k+1)^{-\frac{7}{12}}$$

for any $t \geq t_1$. With this condition, a converging condition for $(\lambda_j^{\widetilde{V}} + \eta)$ can be deducted from Lemma C.11:

$$\frac{d\left(\lambda_j^{\widetilde{V}} + \eta\right)^2}{dt} \leq -\frac{1}{2\eta(2-\eta)}\left(\lambda_j^{\widetilde{V}} + \eta\right)^2$$

Lemma C.11 shows that $\left|\lambda_j^{\widetilde{V}} + \eta\right|$ converges to $(k+1)^{-\frac{1}{12}}$ in $t_2 = O(\log\log d)$ time.

**Stage 2.** Now the eigenvalues are already close to ground-truth:

$$\left|\lambda_j^{\widetilde{V}}(t_1 + t_2) + \eta\right| = O\left((k+1)^{-\frac{1}{12}}\right), \quad \left|\lambda_j^{\widetilde{W}}(t_1 + t_2) - 1\right| \leq 2(k+1)^{-\frac{7}{12}}.$$

According to the expansion of the error terms in Lemma C.5, we notice that $\delta_j^{\widetilde{W}}$ and $\delta_j^{\widetilde{V}}$ are always coupled with some individual residual like $(\mathbf{\Lambda}^{\widetilde{V}} + \eta \mathbf{I})$, $(\mathbf{\Lambda}^{\widetilde{W}} - \mathbf{I})$, or some weighted average $\frac{1}{d} \operatorname{tr}\left((\mathbf{\Lambda}^{\widetilde{V}} + \eta \mathbf{I})\mathbf{\Lambda}^{\widetilde{V}}\right)$. Meanwhile, the coefficient of this kind of residual in the interaction terms is still upper bounded by $O(1/\log^2 d)$. That helps us to derive the PL-condition like gradient lower bound (Lemma C.12):

$$\frac{\mathrm{d} \operatorname{tr}\left[\left(\mathbf{\Lambda}^{\widetilde{V}} + \eta \mathbf{I}\right)^2\right]}{\mathrm{d}t} + \frac{\mathrm{d} \operatorname{tr}\left[\left(\mathbf{I} - \mathbf{\Lambda}^{\widetilde{W}}\right)^2\right]}{\mathrm{d}t}$$

$$\leq -\frac{1}{2\eta(2-\eta)} \operatorname{tr}\left[\left(\mathbf{\Lambda}^{\widetilde{V}} + \eta \mathbf{I}\right)^2\right] - \frac{\eta^2}{2}(k+1)\operatorname{tr}\left[\left(\mathbf{I} - \mathbf{\Lambda}^{\widetilde{W}}\right)^2\right] + \alpha$$

where $\alpha = O\left((1-\eta)^k\right) \geq 0$.

By Lemma C.12, we know $\left|\lambda_j^{\widetilde{V}} + \eta\right|$ and $\left|1 - \lambda_j^{\widetilde{W}}\right|$ converge to $\delta \in \left(\Theta\left(d^{\frac{c}{2}\log(1-\eta)+\frac{1}{2}}\right), 1\right)$ in $t_3 = O\left(\log\frac{1}{\delta}\right)$ time. At this time, there exist diagonal matrices $\mathbf{A}$ and $\mathbf{B}$ satisfying $\|\mathbf{A}\|_{op} \leq \Theta(1)$ and $\|\mathbf{B}\|_{op} \leq \Theta(1)$ such that

$$\mathbf{\Lambda}^{\widetilde{V}} = -\eta \mathbf{I} + \delta \cdot \mathbf{A} \qquad \mathbf{\Lambda}^{\widetilde{W}} = \mathbf{I} + \delta \cdot \mathbf{B}.$$

Now we consider the CoT loss given by Lemma C.3

$$\mathcal{L}^{\mathrm{CoT}}(\boldsymbol{\theta}) = \frac{1}{2}\mathbb{E}_{\mathbf{X},\boldsymbol{w}^*} \sum_{i=0}^{k-1} \left\|(\widetilde{\mathbf{V}}\mathbf{S}\widetilde{\mathbf{W}} + \eta\mathbf{S})\boldsymbol{w}_i - (\widetilde{\mathbf{V}} + \eta\mathbf{I})\mathbf{S}\boldsymbol{w}^*\right\|_2^2$$

$$+ \frac{1}{2}\mathbb{E}_{\mathbf{X},\boldsymbol{w}^*}\left\|(\mathbf{I} + \widetilde{\mathbf{V}}\mathbf{S}\widetilde{\mathbf{W}})\boldsymbol{w}_k - (\widetilde{\mathbf{V}}\mathbf{S} + \mathbf{I})\boldsymbol{w}^*\right\|_2^2.$$

Apply Lemma C.13, we directly obtain that

$$\mathcal{L}^{\mathrm{CoT}}(\boldsymbol{\theta}) = O\left(\delta^2 d \log d\right).$$

Since $\delta \in \left(\Theta\left(d^{\frac{c}{2}\log(1-\eta)+\frac{1}{2}}\right), 1\right)$, the CoT loss is smaller than $\epsilon = \Theta\left(d^{c\log(1-\eta)+2}\log d\right)$. The local convergence takes $t_3 = O\left(\log\frac{1}{\delta}\right) = O\left(\log\frac{1}{\epsilon}\right)$. Considering all stages, at time $t = t_1 + t_2 + t_3 = O(\log d) + O\left(\frac{1}{\epsilon}\right)$, we have

$$\mathcal{L}^{\mathrm{CoT}}(\boldsymbol{\theta}) \leq \epsilon.$$

$\square$

### C.3.1 Technical Lemma in Appendix C.3

**Lemma C.7.** *Assume* $\lambda_j^{\widetilde{W}} \leq 1 - (k+1)^{-\frac{7}{12}}$, *if* $-\frac{3(1-\eta)}{2(2-\eta)}\frac{1}{(k+1)\left(1-\lambda_j^{\widetilde{W}}\right)} \leq \lambda_j^{\widetilde{V}} < 0$, *it holds that*

$$\frac{\mathrm{d}\left(\lambda_j^{\widetilde{V}} + \frac{3(1-\eta)}{2(2-\eta)}\frac{1}{(k+1)\left(1-\lambda_j^{\widetilde{W}}\right)}\right)}{\mathrm{d}t} < 0 \tag{14}$$

*Proof.* Directly consider the derivative

$$\frac{\mathrm{d}\left(\lambda_j^{\widetilde{V}} + \frac{3(1-\eta)}{2(2-\eta)}\frac{1}{(k+1)\left(1-\lambda_j^{\widetilde{W}}\right)}\right)}{\mathrm{d}t} = \frac{\mathrm{d}\lambda_j^{\widetilde{V}}}{\mathrm{d}t} + \frac{3(1-\eta)}{2(k+1)(2-\eta)}\frac{1}{\left(1-\lambda_j^{\widetilde{W}}\right)^2}\frac{\mathrm{d}\lambda_j^{\widetilde{W}}}{\mathrm{d}t}$$

Substitute the derivatives with the equations in Lemma C.6, we have

$$\frac{\mathrm{d}\lambda_j^{\widetilde{V}}}{\mathrm{d}t} + \frac{3(1-\eta)}{2(k+1)(2-\eta)} \frac{1}{\left(1-\lambda_j^{\widetilde{W}}\right)^2} \frac{\mathrm{d}\lambda_j^{\widetilde{W}}}{\mathrm{d}t}$$

$$= -\left[(k+1)\left(1-\lambda_j^{\widetilde{W}}\right)^2 + \frac{2}{\eta}\lambda_j^{\widetilde{W}}\left(1-\lambda_j^{\widetilde{W}}\right) + \frac{1}{\eta(2-\eta)}\lambda_j^{\widetilde{W}^2}\right]\lambda_j^{\widetilde{V}} + \frac{1-\eta}{2-\eta}\lambda_j^{\widetilde{W}} - \left(1+\delta_j^{\widetilde{V}}\right)$$

$$+ \frac{3(1-\eta)}{2(k+1)(2-\eta)} \frac{1}{\left(1-\lambda_j^{\widetilde{W}}\right)^2}\left[\left(k+1-\frac{1}{\eta}\right)\lambda_j^{\widetilde{V}^2}\left(1-\lambda_j^{\widetilde{W}}\right) + \frac{1-\eta}{\eta(2-\eta)}\lambda_j^{\widetilde{V}^2}\lambda_j^{\widetilde{W}} + \frac{1-\eta}{2-\eta}\lambda_j^{\widetilde{V}} - \delta_j^{\widetilde{W}}\right]$$

Since $-\frac{3(1-\eta)}{2(2-\eta)}\frac{1}{(k+1)\left(1-\lambda_j^{\widetilde{W}}\right)} \le \lambda_j^{\widetilde{V}} < 0$, we have

$$\frac{\mathrm{d}\lambda_j^{\widetilde{V}}}{\mathrm{d}t} + \frac{3(1-\eta)}{2(k+1)(2-\eta)} \frac{1}{\left(1-\lambda_j^{\widetilde{W}}\right)^2} \frac{\mathrm{d}\lambda_j^{\widetilde{W}}}{\mathrm{d}t}$$

$$\le \left[(k+1)\left(1-\lambda_j^{\widetilde{W}}\right)^2 + \frac{2}{\eta}\lambda_j^{\widetilde{W}}\left(1-\lambda_j^{\widetilde{W}}\right) + \frac{1}{\eta(2-\eta)}\lambda_j^{\widetilde{W}^2}\right]\frac{3(1-\eta)}{2(2-\eta)}\frac{1}{(k+1)\left(1-\lambda_j^{\widetilde{W}}\right)}$$

$$+ \frac{1-\eta}{2-\eta}\lambda_j^{\widetilde{W}} - \left(1+\delta_j^{\widetilde{V}}\right)$$

$$+ \frac{3(1-\eta)}{2(k+1)(2-\eta)} \frac{1}{\left(1-\lambda_j^{\widetilde{W}}\right)^2}\left[(k+1)\left(\frac{3(1-\eta)}{2(2-\eta)}\frac{1}{(k+1)\left(1-\lambda_j^{\widetilde{W}}\right)}\right)^2\left(1-\lambda_j^{\widetilde{W}}\right)\right.$$

$$\left.+ \frac{1-\eta}{\eta(2-\eta)}\left(\frac{3(1-\eta)}{2(2-\eta)}\frac{1}{(k+1)\left(1-\lambda_j^{\widetilde{W}}\right)}\right)^2\lambda_j^{\widetilde{W}} - \delta_j^{\widetilde{W}}\right]$$

$$= \frac{3(1-\eta)}{2(2-\eta)}\left(1-\lambda_j^{\widetilde{W}}\right) + \frac{1}{k+1}\frac{3(1-\eta)}{\eta(2-\eta)}\lambda_j^{\widetilde{W}} + \frac{1}{(k+1)\left(1-\lambda_j^{\widetilde{W}}\right)}\frac{3(1-\eta)}{2\eta(2-\eta)^2}\lambda_j^{\widetilde{W}^2}$$

$$+ \frac{1-\eta}{2-\eta}\lambda_j^{\widetilde{W}} - \left(1+\delta_j^{\widetilde{V}}\right) + \left[\frac{3(1-\eta)}{2(2-\eta)}\right]^3\frac{1}{(k+1)^2\left(1-\lambda_j^{\widetilde{W}}\right)^3}$$

$$+ \left[\frac{3(1-\eta)}{2(2-\eta)}\right]^3\frac{1-\eta}{\eta(2-\eta)}\lambda_j^{\widetilde{W}}\frac{1}{(k+1)^3\left(1-\lambda_j^{\widetilde{W}}\right)^4} - \frac{3(1-\eta)}{2(2-\eta)}\frac{1}{(k+1)\left(1-\lambda_j^{\widetilde{W}}\right)^2}\delta_j^{\widetilde{W}}$$

$$= \left[\frac{1-\eta}{2(2-\eta)}\left(1-\lambda_j^{\widetilde{W}}\right) - \frac{1}{2-\eta}\right] + \frac{1}{k+1}\frac{3(1-\eta)}{2-\eta}\lambda_j^{\widetilde{W}}$$

$$+ \frac{1}{(k+1)\left(1-\lambda_j^{\widetilde{W}}\right)}\frac{3(1-\eta)}{2\eta(2-\eta)^2}\lambda_j^{\widetilde{W}^2} - \delta_j^{\widetilde{V}} + \left[\frac{3(1-\eta)}{2(2-\eta)}\right]^3\frac{1}{(k+1)^2\left(1-\lambda_j^{\widetilde{W}}\right)^3}$$

$$+ \left[\frac{3(1-\eta)}{2(2-\eta)}\right]^3\frac{1-\eta}{\eta(2-\eta)}\lambda_j^{\widetilde{W}}\frac{1}{(k+1)^3\left(1-\lambda_j^{\widetilde{W}}\right)^4} - \frac{3(1-\eta)}{2(2-\eta)}\frac{1}{(k+1)\left(1-\lambda_j^{\widetilde{W}}\right)^2}\delta_j^{\widetilde{W}}$$

Put in the assumption on $\lambda_j^{\widetilde{W}}$ that $\lambda_j^{\widetilde{W}} \le 1 - (k+1)^{-\frac{7}{12}}$, we have

$$\frac{\mathrm{d}\lambda_j^{\widetilde{V}}}{\mathrm{d}t} + \frac{3(1-\eta)}{2(k+1)(2-\eta)} \frac{1}{\left(1-\lambda_j^{\widetilde{W}}\right)^2} \frac{\mathrm{d}\lambda_j^{\widetilde{W}}}{\mathrm{d}t}$$

$$\le -\frac{1+\eta}{2(2-\eta)} + \frac{1}{k+1}\frac{3(1-\eta)}{\eta(2-\eta)} + \frac{1}{(k+1)^{\frac{5}{12}}}\frac{3(1-\eta)}{2\eta(2-\eta)^2} + \left|\delta_j^{\widetilde{V}}\right| + \left[\frac{3(1-\eta)}{2(2-\eta)}\right]^3\frac{1}{(k+1)^{\frac{1}{4}}}$$

$$+ \left[ \frac{3(1-\eta)}{2(2-\eta)} \right]^3 \frac{1-\eta}{\eta(2-\eta)} \frac{1}{(k+1)^{\frac{2}{3}}} + \frac{3(1-\eta)}{2(2-\eta)} \left| \delta_j^{\widetilde{W}} \right|$$

$$= -\frac{1+\eta}{2(2-\eta)} + O\left( \frac{1}{\log^{\frac{1}{4}} d} \right)$$

$\square$

**Lemma C.8** (Upper bound of $\lambda_j^{\widetilde{V}}$). *Under Assumption 4.1, if $\lambda_j^{\widetilde{W}} \leq 1 - (k+1)^{-\frac{7}{12}}$, it holds that*

$$\lambda_j^{\widetilde{V}} < -\frac{3(1-\eta)}{2(2-\eta)} \frac{1}{(k+1)\left(1 - \lambda_j^{\widetilde{W}}\right)} \tag{15}$$

*Proof.* We prove by induction. First, check the initialization $\lambda_j^{\widetilde{V}}(0) \leq -\sigma$, $\sigma \leq \lambda_j^{\widetilde{W}}(0) \leq \frac{1}{2}$. If $\sigma \geq \frac{3(1-\eta)}{2-\eta} \frac{1}{k+1}$, then we have

$$\lambda_j^{\widetilde{V}}(0) + \frac{3(1-\eta)}{2(2-\eta)} \frac{1}{(k+1)\left(1 - \lambda_j^{\widetilde{W}}(0)\right)} < -\sigma + \frac{3(1-\eta)}{(2-\eta)} \frac{1}{k+1} \leq 0$$

If the inequality holds until some time $t_1$, that is for any $t < t_1$, we have

$$\lambda_j^{\widetilde{V}}(t) < -\frac{3(1-\eta)}{2(2-\eta)} \frac{1}{(k+1)\left(1 - \lambda_j^{\widetilde{W}}(t)\right)}$$

but

$$\lambda_j^{\widetilde{V}}(t_1) \geq -\frac{3(1-\eta)}{2(2-\eta)} \frac{1}{(k+1)\left(1 - \lambda_j^{\widetilde{W}}(t_1)\right)}$$

By Lemma C.7, we have

$$\frac{d\left( \lambda_j^{\widetilde{V}} + \frac{3(1-\eta)}{2(2-\eta)} \frac{1}{(k+1)\left(1-\lambda_j^{\widetilde{W}}\right)} \right)}{dt} \Bigg|_{t=t_1} < 0$$

Therefore, there exists some time $t' < t_1$ such that

$$\lambda_j^{\widetilde{V}}(t') \geq -\frac{3(1-\eta)}{2(2-\eta)} \frac{1}{(k+1)\left(1 - \lambda_j^{\widetilde{W}}(t')\right)}$$

which is a contradiction. Hence, the proof is complete. $\square$

**Lemma C.9** ($\lambda_j^{\widetilde{W}}$ converges to near optimal). *Under Assumption 4.1, it takes $O(\log d)$ time for $\lambda_j^{\widetilde{W}}$ to converge to $1 - (k+1)^{-\frac{7}{12}}$.*

*Proof.* Recall the gradient of $\lambda_j^{\widetilde{W}}$ in Lemma C.6

$$\frac{d\lambda_j^{\widetilde{W}}}{dt} = \left( k+1 - \frac{1}{\eta} \right) \lambda_j^{\widetilde{V}^2} \left( 1 - \lambda_j^{\widetilde{W}} \right) + \frac{1-\eta}{\eta(2-\eta)} \lambda_j^{\widetilde{V}^2} \lambda_j^{\widetilde{W}} + \frac{1-\eta}{2-\eta} \lambda_j^{\widetilde{V}} - \delta_j^{\widetilde{W}}$$

Substitute $\lambda_j^{\widetilde{V}}$ with Lemma C.8, we have

$$\frac{d\lambda_j^{\widetilde{W}}}{dt} \geq \left( k+1 - \frac{1}{\eta} \right) \left( \frac{3(1-\eta)}{2(2-\eta)} \frac{1}{(k+1)\left(1 - \lambda_j^{\widetilde{W}}\right)} \right)^2 \left( 1 - \lambda_j^{\widetilde{W}} \right)$$

$$+ \frac{1-\eta}{\eta(2-\eta)} \left( \frac{3(1-\eta)}{2(2-\eta)} \frac{1}{(k+1)\left(1-\lambda_j^{\widetilde{W}}\right)} \right)^2 \lambda_j^{\widetilde{W}}$$

$$- \frac{1-\eta}{2-\eta} \left( \frac{3(1-\eta)}{2(2-\eta)} \frac{1}{(k+1)\left(1-\lambda_j^{\widetilde{W}}\right)} \right) - \delta_j^{\widetilde{W}}$$

$$\geq \frac{4}{5}(k+1) \left( \frac{3(1-\eta)}{2(2-\eta)} \frac{1}{(k+1)\left(1-\lambda_j^{\widetilde{W}}\right)} \right)^2 \left(1-\lambda_j^{\widetilde{W}}\right)$$

$$- \frac{1-\eta}{2-\eta} \left( \frac{3(1-\eta)}{2(2-\eta)} \frac{1}{(k+1)\left(1-\lambda_j^{\widetilde{W}}\right)} \right) - \left|\delta_j^{\widetilde{W}}\right|$$

$$= \frac{3}{10} \frac{(1-\eta)^2}{(2-\eta)^2} \frac{1}{(k+1)\left(1-\lambda_j^{\widetilde{W}}\right)} - \left|\delta_j^{\widetilde{W}}\right|$$

$$\geq \frac{1}{5} \frac{(1-\eta)^2}{(2-\eta)^2} \frac{1}{(k+1)\left(1-\lambda_j^{\widetilde{W}}\right)}$$

$$\geq \frac{1}{5} \frac{(1-\eta)^2}{(2-\eta)^2} \frac{1}{k+1}$$

In $O(\log d)$ time, $\lambda_j^{\widetilde{W}}$ can converge to $1 - (k+1)^{-\frac{7}{12}}$. $\qquad\square$

**Lemma C.10.** *Assume* $\lambda_j^{\widetilde{W}}(t_1) = 1 - (k+1)^{-\frac{7}{12}}$ *and* $\lambda_j^{\widetilde{V}}(t_1) < -\frac{3(1-\eta)}{2(2-\eta)} \frac{1}{(k+1)\left(1-\lambda_j^{\widetilde{W}}(t_1)\right)}$, *for any* $t \geq t_1$ *it holds that*

$$1 - 2(k+1)^{-\frac{7}{12}} < \lambda_j^{\widetilde{W}}(t) < 1 + (k+1)^{-\frac{7}{12}}.$$

*Proof.* First, it is clear that the inequality holds at time $t_1$. If the inequality doesn't hold, then there exists $t' > t_1$ such that

$$1 - 2(k+1)^{-\frac{7}{12}} < \lambda_j^{\widetilde{W}}(t) < 1 + (k+1)^{-\frac{7}{12}} \qquad \text{for any } t_1 \leq t < t'$$
$$\lambda_j^{\widetilde{W}}(t') = 1 - 2(k+1)^{-\frac{7}{12}}$$

or

$$1 - 2(k+1)^{-\frac{7}{12}} < \lambda_j^{\widetilde{W}}(t) < 1 + (k+1)^{-\frac{7}{12}} \qquad \text{for any } t_1 \leq t < t'$$
$$\lambda_j^{\widetilde{W}}(t') = 1 + (k+1)^{-\frac{7}{12}}$$

In the first case, it suffices to prove

$$\lambda_j^{\widetilde{V}}(t') \leq -\frac{3(1-\eta)}{2(2-\eta)}(k+1)^{-\frac{5}{12}} < -\frac{3(1-\eta)}{2(2-\eta)} \frac{1}{(k+1)\left(1-\lambda_j^{\widetilde{W}}(t')\right)}$$

to show

$$\left. \frac{d\lambda_j^{\widetilde{W}}}{dt} \right|_{t=t'} > 0$$

which says there exists $t_1 \leq t'' < t'$ such that

$$\lambda_j^{\widetilde{W}}(t'') \leq 1 - 2(k+1)^{-\frac{7}{12}}$$

and leads to a contradiction. Recall the gradient of $\lambda_j^{\widetilde{V}}$ in Lemma C.6

$$\frac{d\lambda_j^{\widetilde{V}}}{dt} = - \left[ (k+1)\left(1-\lambda_j^{\widetilde{W}}\right)^2 + \frac{2}{\eta}\lambda_j^{\widetilde{W}}\left(1-\lambda_j^{\widetilde{W}}\right) + \frac{1}{\eta(2-\eta)}\lambda_j^{\widetilde{W}^2} \right] \lambda_j^{\widetilde{V}} + \frac{1-\eta}{2-\eta}\lambda_j^{\widetilde{W}} - \left(1+\delta_j^{\widetilde{V}}\right)$$

$$\leq -\left[4(k+1)^{-\frac{1}{6}} + \frac{4}{\eta}\left[(k+1)^{-\frac{7}{12}} + (k+1)^{-\frac{7}{6}}\right] + \frac{1}{\eta(2-\eta)}\left[1 + 2(k+1)^{-\frac{7}{12}} + (k+1)^{-\frac{7}{6}}\right]\right]\lambda_j^{\widetilde{V}}$$

$$-\frac{1}{2-\eta} + \frac{1-\eta}{2-\eta}(k+1)^{-\frac{7}{12}} + \left|\delta_j^{\widetilde{V}}\right|$$

$$\leq -\frac{2}{\eta(2-\eta)}\lambda_j^{\widetilde{V}} - \frac{1}{2(2-\eta)}$$

and thus we have

$$\lambda_j^{\widetilde{V}}(t) \leq Ce^{-\frac{2}{\eta(2-\eta)}(t-t_1)} - \frac{\eta}{4}$$

If $C \leq 0$, then

$$\lambda_j^{\widetilde{V}}(t') \leq -\frac{\eta}{4} \leq -\frac{3(1-\eta)}{2(2-\eta)}(k+1)^{-\frac{5}{12}}$$

else

$$\lambda_j^{\widetilde{V}}(t') \leq \lambda_j^{\widetilde{V}}(t_1) = -\frac{3(1-\eta)}{2(2-\eta)}(k+1)^{-\frac{5}{12}}$$

In the second case,

$$\lambda_j^{\widetilde{V}}(t) \leq -\frac{3(1-\eta)}{2(2-\eta)}(k+1)^{-\frac{5}{12}}$$

still holds for any $t_1 \leq t \leq t'$. Recall the gradient of $\lambda_j^{\widetilde{W}}$ in Lemma C.6

$$\left.\frac{d\lambda_j^{\widetilde{W}}}{dt}\right|_{t=t'} = \left(k+1-\frac{1}{\eta}\right)\lambda_j^{\widetilde{V}2}\left(1-\lambda_j^{\widetilde{W}}\right) + \frac{1-\eta}{\eta(2-\eta)}\lambda_j^{\widetilde{V}2}\lambda_j^{\widetilde{W}} + \frac{1-\eta}{2-\eta}\lambda_j^{\widetilde{V}} - \delta_j^{\widetilde{W}}$$

$$= -\left(k+1-\frac{1}{\eta}\right)\lambda_j^{\widetilde{V}2}(k+1)^{-\frac{7}{12}} + \frac{1-\eta}{\eta(2-\eta)}\lambda_j^{\widetilde{V}2}\left[1+(k+1)^{-\frac{7}{12}}\right] + \frac{1-\eta}{2-\eta}\lambda_j^{\widetilde{V}} - \delta_j^{\widetilde{W}}$$

$$\leq -\frac{(k+1)^{\frac{5}{12}}}{2}\lambda_j^{\widetilde{V}2} + \frac{1-\eta}{2(2-\eta)}\lambda_j^{\widetilde{V}} + \left|\delta_j^{\widetilde{W}}\right|$$

$$\leq -\frac{9(1-\eta)^2}{8(2-\eta)^2}(k+1)^{-\frac{5}{12}} - \frac{3(1-\eta)^2}{4(2-\eta)^2}(k+1)^{-\frac{5}{12}} + \left|\delta_j^{\widetilde{W}}\right|$$

$$\leq -\frac{(1-\eta)^2}{(2-\eta)^2}(k+1)^{-\frac{5}{12}}$$

There exists $t_1 \leq t'' < t'$ such that

$$\lambda_j^{\widetilde{W}}(t'') \geq 1 + (k+1)^{-\frac{7}{12}}$$

which is a contradiction. Hence, the proof is complete. $\qquad\square$

**Lemma C.11** ($\lambda_j^{\widetilde{V}}$ converges to near optimal). *Assume*

$$1 - 2(k+1)^{-\frac{7}{12}} < \lambda_j^{\widetilde{W}}(t) < 1 + (k+1)^{-\frac{7}{12}}$$

*then it takes $O(\log\log d)$ time for $\left|\lambda_j^{\widetilde{V}} + \eta\right|$ to converge to $(k+1)^{-\frac{1}{12}}$.*

*Proof.* From Lemma C.10, we know

$$\frac{d\lambda_j^{\widetilde{V}}}{dt} = -\left[(k+1)\left(1-\lambda_j^{\widetilde{W}}\right)^2 + \frac{2}{\eta}\lambda_j^{\widetilde{W}}\left(1-\lambda_j^{\widetilde{W}}\right) + \frac{1}{\eta(2-\eta)}\lambda_j^{\widetilde{W}2}\right]\lambda_j^{\widetilde{V}} + \frac{1-\eta}{2-\eta}\lambda_j^{\widetilde{W}} - \left(1+\delta_j^{\widetilde{V}}\right)$$

$$\leq -\left[4(k+1)^{-\frac{1}{6}} + \frac{4}{\eta}\left[(k+1)^{-\frac{7}{12}} + (k+1)^{-\frac{7}{6}}\right] + \frac{1}{\eta(2-\eta)}\left[1 + 2(k+1)^{-\frac{7}{12}} + (k+1)^{-\frac{7}{6}}\right]\right]\lambda_j^{\widetilde{V}}$$

$$-\frac{1}{2-\eta} + \frac{1-\eta}{2-\eta}(k+1)^{-\frac{7}{12}} + \left|\delta_j^{\widetilde{V}}\right|$$

$$= -\left[\frac{1}{\eta(2-\eta)} + O\left((k+1)^{-\frac{1}{6}}\right)\right]\left(\lambda_j^{\widetilde{V}} + \eta\right) + O\left((k+1)^{-\frac{1}{6}}\right)$$

and

$$\frac{d\lambda_j^{\widetilde{V}}}{dt} = -\left[(k+1)\left(1 - \lambda_j^{\widetilde{W}}\right)^2 + \frac{2}{\eta}\lambda_j^{\widetilde{W}}\left(1 - \lambda_j^{\widetilde{W}}\right) + \frac{1}{\eta(2-\eta)}\lambda_j^{\widetilde{W}^2}\right]\lambda_j^{\widetilde{V}} + \frac{1-\eta}{2-\eta}\lambda_j^{\widetilde{W}} - \left(1 + \delta_j^{\widetilde{V}}\right)$$

$$\geq -\left[-\frac{2}{\eta}\left[(k+1)^{-\frac{7}{12}} + (k+1)^{-\frac{7}{6}}\right] + \frac{1}{\eta(2-\eta)}\left[1 - 4(k+1)^{-\frac{7}{12}} + 4(k+1)^{-\frac{7}{6}}\right]\right]\lambda_j^{\widetilde{V}}$$

$$\quad - \frac{1}{2-\eta} - \frac{2(1-\eta)}{2-\eta}(k+1)^{-\frac{7}{12}} - \left|\delta_j^{\widetilde{V}}\right|$$

$$= -\left[\frac{1}{\eta(2-\eta)} + O\left((k+1)^{-\frac{1}{6}}\right)\right]\left(\lambda_j^{\widetilde{V}} + \eta\right) + O\left((k+1)^{-\frac{1}{6}}\right)$$

Therefore,

$$\frac{d\left(\lambda_j^{\widetilde{V}} + \eta\right)^2}{dt} = 2\left(\lambda_j^{\widetilde{V}} + \eta\right)\frac{d\lambda_j^{\widetilde{V}}}{dt}$$

$$\leq -\left[\frac{1}{\eta(2-\eta)} + O\left((k+1)^{-\frac{1}{6}}\right)\right]\left(\lambda_j^{\widetilde{V}} + \eta\right)^2 + O\left((k+1)^{-\frac{1}{6}}\right)\left(\lambda_j^{\widetilde{V}} + \eta\right)$$

If $\left|\lambda_j^{\widetilde{V}} + \eta\right|$ converges to $\epsilon = (k+1)^{-\frac{1}{12}}$, then

$$\frac{d\left(\lambda_j^{\widetilde{V}} + \eta\right)^2}{dt} \leq -\frac{1}{2\eta(2-\eta)}\left(\lambda_j^{\widetilde{V}} + \eta\right)^2$$

Thus, there exists $c \leq \Theta(1)$ such that

$$\epsilon^2 = \left(\lambda_j^{\widetilde{V}} + \eta\right)^2 \leq c^2 \exp\left(-\frac{1}{2\eta(2-\eta)}t\right)$$

In $O\left(\log\left(\frac{1}{\epsilon}\right)\right) = O(\log\log d)$ time, $\left|\lambda_j^{\widetilde{V}} + \eta\right|$ can converge to $\epsilon$. $\qquad\square$

**Lemma C.12** (Local convergence). *Suppose $k = \lceil c\log d\rceil$. Assume*

$$\left|\lambda_j^{\widetilde{W}}(t) - 1\right| \leq 2(k+1)^{-\frac{7}{12}} \qquad \left|\lambda_j^{\widetilde{V}}(t) + \eta\right| = O\left((k+1)^{-\frac{1}{12}}\right),$$

*then there exists $\alpha = O\left((1-\eta)^k\right) \geq 0$ such that $\boldsymbol{\Lambda}^{\widetilde{V}}$ and $\boldsymbol{\Lambda}^{\widetilde{W}}$ comply with*

$$\frac{d\,\mathrm{tr}\left[\left(\boldsymbol{\Lambda}^{\widetilde{V}} + \eta\boldsymbol{I}\right)^2\right]}{dt} + \frac{d\,\mathrm{tr}\left[\left(\boldsymbol{I} - \boldsymbol{\Lambda}^{\widetilde{W}}\right)^2\right]}{dt}$$

$$\leq -\frac{1}{2\eta(2-\eta)}\,\mathrm{tr}\left[\left(\boldsymbol{\Lambda}^{\widetilde{V}} + \eta\boldsymbol{I}\right)^2\right] - \frac{\eta^2}{2}(k+1)\,\mathrm{tr}\left[\left(\boldsymbol{I} - \boldsymbol{\Lambda}^{\widetilde{W}}\right)^2\right] + \alpha,$$

*thus $\left|\lambda_j^{\widetilde{V}} + \eta\right|$ and $\left|1 - \lambda_j^{\widetilde{W}}\right|$ can converge to $\epsilon \in \left(d^{-\frac{c}{2}\log\left(\frac{1}{1-\eta}\right)+\frac{1}{2}}, 1\right)$ in $O\left(\log\frac{1}{\epsilon}\right)$ time.*

*Proof.* Consider the error term in Lemma C.6 more carefully, we have

$$\frac{d\lambda_j^{\widetilde{V}}}{dt} = -\left[(k+1)\left(1 - \lambda_j^{\widetilde{W}}\right)^2 + \frac{2}{\eta}\lambda_j^{\widetilde{W}}\left(1 - \lambda_j^{\widetilde{W}}\right) + \frac{1}{\eta(2-\eta)}\lambda_j^{\widetilde{W}^2}\right]\left(\lambda_j^{\widetilde{V}} + \eta\right)$$

$$\quad + \eta(k+1)\left(1 - \lambda_j^{\widetilde{W}}\right)^2 + \left(\frac{3-2\eta}{2-\eta}\lambda_j^{\widetilde{W}} - 1\right)\left(1 - \lambda_j^{\widetilde{W}}\right)$$

$$\quad + \left(\lambda_j^{\widetilde{V}} + \eta\right)O\left(\frac{1}{\log^2 d}\right) + \left(1 - \lambda_j^{\widetilde{W}}\right)O\left(\frac{1}{\log^2 d}\right)$$

$$+ \operatorname{tr}\left(\boldsymbol{I} - \boldsymbol{\Lambda}^{\widetilde{\boldsymbol{W}}}\right) O\left(\frac{1}{d \log^2 d}\right) + \operatorname{tr}\left(\left(\boldsymbol{I} - \boldsymbol{\Lambda}^{\widetilde{\boldsymbol{W}}}\right)\boldsymbol{\Lambda}^{\widetilde{\boldsymbol{W}}}\right) O\left(\frac{1}{d \log^2 d}\right)$$

$$+ O\left((1-\eta)^k\right)$$

and

$$\frac{\mathrm{d}\lambda_j^{\widetilde{\boldsymbol{W}}}}{\mathrm{d}t} = \left(k + 1 - \frac{2}{\eta} + \frac{1}{\eta(2-\eta)}\right)\lambda_j^{\widetilde{\boldsymbol{V}}^2}\left(1 - \lambda_j^{\widetilde{\boldsymbol{W}}}\right) + \frac{1-\eta}{\eta(2-\eta)}\lambda_j^{\widetilde{\boldsymbol{V}}}\left(\lambda_j^{\widetilde{\boldsymbol{V}}} + \eta\right)$$

$$+ \left(\lambda_j^{\widetilde{\boldsymbol{V}}} + \eta\right) O\left(\frac{1}{\log^2 d}\right) + \left(1 - \lambda_j^{\widetilde{\boldsymbol{W}}}\right) O\left(\frac{1}{\log^2 d}\right)$$

$$+ \operatorname{tr}\left(\boldsymbol{I} - \boldsymbol{\Lambda}^{\widetilde{\boldsymbol{W}}}\right) O\left(\frac{1}{d \log^2 d}\right) + \operatorname{tr}\left(\left(\boldsymbol{\Lambda}^{\widetilde{\boldsymbol{V}}} + \eta\boldsymbol{I}\right)\boldsymbol{\Lambda}^{\widetilde{\boldsymbol{V}}}\right) O\left(\frac{1}{d \log^2 d}\right)$$

$$+ \operatorname{tr}\left(\left(\boldsymbol{I} - \boldsymbol{\Lambda}^{\widetilde{\boldsymbol{W}}}\right)\boldsymbol{\Lambda}^{\widetilde{\boldsymbol{V}}^2}\right) O\left(\frac{1}{d \log^2 d}\right) + O\left((1-\eta)^k\right)$$

Now we consider the decay rate of the distance between $\lambda_j^{\widetilde{\boldsymbol{V}}}$, $\lambda_j^{\widetilde{\boldsymbol{W}}}$ and their ground truth.

$$\frac{\mathrm{d}\left(\lambda_j^{\widetilde{\boldsymbol{V}}} + \eta\right)^2}{\mathrm{d}t} + \frac{\mathrm{d}\left(\lambda_j^{\widetilde{\boldsymbol{W}}} - 1\right)^2}{\mathrm{d}t}$$

$$= -\left[(k+1)\left(1 - \lambda_j^{\widetilde{\boldsymbol{W}}}\right)^2 + \frac{2}{\eta}\lambda_j^{\widetilde{\boldsymbol{W}}}\left(1 - \lambda_j^{\widetilde{\boldsymbol{W}}}\right) + \frac{1}{\eta(2-\eta)}\lambda_j^{\widetilde{\boldsymbol{W}}^2} + O\left(\frac{1}{\log^2 d}\right)\right]\left(\lambda_j^{\widetilde{\boldsymbol{V}}} + \eta\right)^2$$

$$+ \left(\frac{3 - 2\eta}{2-\eta}\lambda_j^{\widetilde{\boldsymbol{W}}} - \frac{1-\eta}{\eta(2-\eta)}\lambda_j^{\widetilde{\boldsymbol{V}}} - 1 + O\left(\frac{1}{\log^2 d}\right)\right)\left(1 - \lambda_j^{\widetilde{\boldsymbol{W}}}\right)\left(\lambda_j^{\widetilde{\boldsymbol{V}}} + \eta\right)$$

$$- \left[\left(k + 1 - \frac{2}{\eta} + \frac{1}{\eta(2-\eta)}\right)\lambda_j^{\widetilde{\boldsymbol{V}}^2} - \eta(k+1)\left(\lambda_j^{\widetilde{\boldsymbol{V}}} + \eta\right) + O\left(\frac{1}{\log^2 d}\right)\right]\left(1 - \lambda_j^{\widetilde{\boldsymbol{W}}}\right)^2$$

$$+ \left(\lambda_j^{\widetilde{\boldsymbol{V}}} + \eta\right)\operatorname{tr}\left(\boldsymbol{I} - \boldsymbol{\Lambda}^{\widetilde{\boldsymbol{W}}}\right) O\left(\frac{1}{d \log^2 d}\right) + \left(\lambda_j^{\widetilde{\boldsymbol{V}}} + \eta\right)\operatorname{tr}\left(\left(\boldsymbol{I} - \boldsymbol{\Lambda}^{\widetilde{\boldsymbol{W}}}\right)\boldsymbol{\Lambda}^{\widetilde{\boldsymbol{W}}}\right) O\left(\frac{1}{d \log^2 d}\right)$$

$$+ \left(1 - \lambda_j^{\widetilde{\boldsymbol{W}}}\right)\operatorname{tr}\left(\boldsymbol{I} - \boldsymbol{\Lambda}^{\widetilde{\boldsymbol{W}}}\right) O\left(\frac{1}{d \log^2 d}\right) + \left(1 - \lambda_j^{\widetilde{\boldsymbol{W}}}\right)\operatorname{tr}\left(\left(\boldsymbol{\Lambda}^{\widetilde{\boldsymbol{V}}} + \eta\boldsymbol{I}\right)\boldsymbol{\Lambda}^{\widetilde{\boldsymbol{V}}}\right) O\left(\frac{1}{d \log^2 d}\right)$$

$$+ \left(1 - \lambda_j^{\widetilde{\boldsymbol{W}}}\right)\operatorname{tr}\left(\left(\boldsymbol{I} - \boldsymbol{\Lambda}^{\widetilde{\boldsymbol{W}}}\right)\boldsymbol{\Lambda}^{\widetilde{\boldsymbol{V}}^2}\right) O\left(\frac{1}{d \log^2 d}\right) + O\left((1-\eta)^k\right)$$

$$= -\left[\frac{1}{\eta(2-\eta)} + O\left(\frac{1}{\log^{\frac{1}{6}} d}\right)\right]\left(\lambda_j^{\widetilde{\boldsymbol{V}}} + \eta\right)^2$$

$$+ \left(\frac{2(1-\eta)}{2-\eta} + O\left(\frac{1}{\log^{\frac{1}{12}} d}\right)\right)\left(1 - \lambda_j^{\widetilde{\boldsymbol{W}}}\right)\left(\lambda_j^{\widetilde{\boldsymbol{V}}} + \eta\right)$$

$$- \left[\eta^2(k+1) + O\left(k^{\frac{11}{12}}\right)\right]\left(1 - \lambda_j^{\widetilde{\boldsymbol{W}}}\right)^2$$

$$+ \left(\lambda_j^{\widetilde{\boldsymbol{V}}} + \eta\right)\operatorname{tr}\left(\boldsymbol{I} - \boldsymbol{\Lambda}^{\widetilde{\boldsymbol{W}}}\right) O\left(\frac{1}{d \log^2 d}\right) + \left(\lambda_j^{\widetilde{\boldsymbol{V}}} + \eta\right)\operatorname{tr}\left(\left(\boldsymbol{I} - \boldsymbol{\Lambda}^{\widetilde{\boldsymbol{W}}}\right)\boldsymbol{\Lambda}^{\widetilde{\boldsymbol{W}}}\right) O\left(\frac{1}{d \log^2 d}\right)$$

$$+ \left(1 - \lambda_j^{\widetilde{\boldsymbol{W}}}\right)\operatorname{tr}\left(\boldsymbol{I} - \boldsymbol{\Lambda}^{\widetilde{\boldsymbol{W}}}\right) O\left(\frac{1}{d \log^2 d}\right) + \left(1 - \lambda_j^{\widetilde{\boldsymbol{W}}}\right)\operatorname{tr}\left(\left(\boldsymbol{\Lambda}^{\widetilde{\boldsymbol{V}}} + \eta\boldsymbol{I}\right)\boldsymbol{\Lambda}^{\widetilde{\boldsymbol{V}}}\right) O\left(\frac{1}{d \log^2 d}\right)$$

$$+ \left(1 - \lambda_j^{\widetilde{\boldsymbol{W}}}\right)\operatorname{tr}\left(\left(\boldsymbol{I} - \boldsymbol{\Lambda}^{\widetilde{\boldsymbol{W}}}\right)\boldsymbol{\Lambda}^{\widetilde{\boldsymbol{V}}^2}\right) O\left(\frac{1}{d \log^2 d}\right) + O\left((1-\eta)^k\right)$$

Utilizing Mean Inequality, we have

$$\left|\left(1 - \lambda_j^{\widetilde{\boldsymbol{W}}}\right)\left(\lambda_j^{\widetilde{\boldsymbol{V}}} + \eta\right)\right| \leq \frac{1}{2\sqrt{k}}\left(\lambda_j^{\widetilde{\boldsymbol{V}}} + \eta\right)^2 + \frac{\sqrt{k}}{2}\left(1 - \lambda_j^{\widetilde{\boldsymbol{W}}}\right)^2$$

Insert the inequality into the equation and we have

$$
\begin{aligned}
&\frac{\mathrm{d}\left(\lambda_j^{\widetilde{V}} + \eta\right)^2}{\mathrm{d}t} + \frac{\mathrm{d}\left(\lambda_j^{\widetilde{W}} - 1\right)^2}{\mathrm{d}t} \\
&\leq -\left[\frac{1}{\eta(2-\eta)} + O\left(\frac{1}{\log^{\frac{1}{6}} d}\right)\right]\left(\lambda_j^{\widetilde{V}} + \eta\right)^2 - \left[\eta^2(k+1) + O\left(k^{\frac{11}{12}}\right)\right]\left(1 - \lambda_j^{\widetilde{W}}\right)^2 \\
&\quad + \left(\lambda_j^{\widetilde{V}} + \eta\right)\operatorname{tr}\left(\boldsymbol{I} - \boldsymbol{\Lambda}^{\widetilde{W}}\right)O\left(\frac{1}{d\log^2 d}\right) + \left(\lambda_j^{\widetilde{V}} + \eta\right)\operatorname{tr}\left(\left(\boldsymbol{I} - \boldsymbol{\Lambda}^{\widetilde{W}}\right)\boldsymbol{\Lambda}^{\widetilde{W}}\right)O\left(\frac{1}{d\log^2 d}\right) \\
&\quad + \left(1 - \lambda_j^{\widetilde{W}}\right)\operatorname{tr}\left(\boldsymbol{I} - \boldsymbol{\Lambda}^{\widetilde{W}}\right)O\left(\frac{1}{d\log^2 d}\right) + \left(1 - \lambda_j^{\widetilde{W}}\right)\operatorname{tr}\left(\left(\boldsymbol{\Lambda}^{\widetilde{V}} + \eta\boldsymbol{I}\right)\boldsymbol{\Lambda}^{\widetilde{V}}\right)O\left(\frac{1}{d\log^2 d}\right) \\
&\quad + \left(1 - \lambda_j^{\widetilde{W}}\right)\operatorname{tr}\left(\left(\boldsymbol{I} - \boldsymbol{\Lambda}^{\widetilde{W}}\right)\boldsymbol{\Lambda}^{\widetilde{V}^2}\right)O\left(\frac{1}{d\log^2 d}\right) + O\left((1-\eta)^k\right)
\end{aligned}
$$

There exist $\alpha_1, \alpha_2, \alpha_3, \alpha_4, \alpha_5 = O\left(\frac{1}{d\log^2 d}\right) \geq 0$ and $\alpha_6 = O\left((1-\eta)^k\right) \geq 0$ such that

$$
\begin{aligned}
\frac{\mathrm{d}\left(\lambda_j^{\widetilde{V}} + \eta\right)^2}{\mathrm{d}t} + \frac{\mathrm{d}\left(\lambda_j^{\widetilde{W}} - 1\right)^2}{\mathrm{d}t} &\leq -\left[\frac{1}{\eta(2-\eta)} + O\left(\frac{1}{\log^{\frac{1}{6}} d}\right)\right]\left(\lambda_j^{\widetilde{V}} + \eta\right)^2 \\
&\quad - \left[\eta^2(k+1) + O\left(k^{\frac{11}{12}}\right)\right]\left(1 - \lambda_j^{\widetilde{W}}\right)^2 \\
&\quad + \alpha_1\left|\lambda_j^{\widetilde{V}} + \eta\right|\operatorname{tr}\left(\left|\boldsymbol{I} - \boldsymbol{\Lambda}^{\widetilde{W}}\right|\right) + \alpha_2\left|\lambda_j^{\widetilde{V}} + \eta\right| \cdot \left|\operatorname{tr}\left(\left(\boldsymbol{I} - \boldsymbol{\Lambda}^{\widetilde{W}}\right)\boldsymbol{\Lambda}^{\widetilde{W}}\right)\right| \\
&\quad + \alpha_3\left|1 - \lambda_j^{\widetilde{W}}\right|\operatorname{tr}\left(\left|\boldsymbol{I} - \boldsymbol{\Lambda}^{\widetilde{W}}\right|\right) + \alpha_4\left|1 - \lambda_j^{\widetilde{W}}\right| \cdot \left|\operatorname{tr}\left(\left(\boldsymbol{\Lambda}^{\widetilde{V}} + \eta\boldsymbol{I}\right)\boldsymbol{\Lambda}^{\widetilde{V}}\right)\right| \\
&\quad + \alpha_5\left|1 - \lambda_j^{\widetilde{W}}\right| \cdot \left|\operatorname{tr}\left(\left(\boldsymbol{I} - \boldsymbol{\Lambda}^{\widetilde{W}}\right)\boldsymbol{\Lambda}^{\widetilde{V}^2}\right)\right| + \alpha_6.
\end{aligned}
$$

Notice that for diagonal matrices $\boldsymbol{A}$ and $\boldsymbol{B}$, we have

$$
\operatorname{tr}\left(\boldsymbol{A}\boldsymbol{B}\right) \leq \left|\operatorname{tr}\left(\boldsymbol{A}\boldsymbol{B}\right)\right| = \left|\sum_i a_{ii}b_{ii}\right| \leq \sum_i |a_{ii}||b_{ii}| \leq \sum_i |a_{ii}|\|B\| = \operatorname{tr}(|A|)\|B\|
$$

Plug in the inequality and we have

$$
\begin{aligned}
&\frac{\mathrm{d}\left(\lambda_j^{\widetilde{V}} + \eta\right)^2}{\mathrm{d}t} + \frac{\mathrm{d}\left(\lambda_j^{\widetilde{W}} - 1\right)^2}{\mathrm{d}t} \\
&\leq -\left[\frac{1}{\eta(2-\eta)} + O\left(\frac{1}{\log^{\frac{1}{6}} d}\right)\right]\left(\lambda_j^{\widetilde{V}} + \eta\right)^2 - \left[\eta^2(k+1) + O\left(k^{\frac{11}{12}}\right)\right]\left(1 - \lambda_j^{\widetilde{W}}\right)^2 \\
&\quad + \alpha_1\left|\lambda_j^{\widetilde{V}} + \eta\right|\operatorname{tr}\left(\left|\boldsymbol{I} - \boldsymbol{\Lambda}^{\widetilde{W}}\right|\right) + \alpha_2\left|\lambda_j^{\widetilde{V}} + \eta\right|\operatorname{tr}\left(\left|\boldsymbol{I} - \boldsymbol{\Lambda}^{\widetilde{W}}\right|\right) \cdot \left\|\boldsymbol{\Lambda}^{\widetilde{W}}\right\| \\
&\quad + \alpha_3\left|1 - \lambda_j^{\widetilde{W}}\right|\operatorname{tr}\left(\left|\boldsymbol{I} - \boldsymbol{\Lambda}^{\widetilde{W}}\right|\right) + \alpha_4\left|1 - \lambda_j^{\widetilde{W}}\right|\operatorname{tr}\left(\left|\boldsymbol{\Lambda}^{\widetilde{V}} + \eta\boldsymbol{I}\right|\right) \cdot \left\|\boldsymbol{\Lambda}^{\widetilde{V}}\right\| \\
&\quad + \alpha_5\left|1 - \lambda_j^{\widetilde{W}}\right|\operatorname{tr}\left(\left|\boldsymbol{I} - \boldsymbol{\Lambda}^{\widetilde{W}}\right|\right) \cdot \left\|\boldsymbol{\Lambda}^{\widetilde{V}^2}\right\| + \alpha_6 \\
&= -\left[\frac{1}{\eta(2-\eta)} + O\left(\frac{1}{\log^{\frac{1}{6}} d}\right)\right]\left(\lambda_j^{\widetilde{V}} + \eta\right)^2 - \left[\eta^2(k+1) + O\left(k^{\frac{11}{12}}\right)\right]\left(1 - \lambda_j^{\widetilde{W}}\right)^2 \\
&\quad + \left(\alpha_1 + \alpha_2\left\|\boldsymbol{\Lambda}^{\widetilde{W}}\right\|\right) \cdot \left|\lambda_j^{\widetilde{V}} + \eta\right| \cdot \operatorname{tr}\left(\left|\boldsymbol{I} - \boldsymbol{\Lambda}^{\widetilde{W}}\right|\right) + \alpha_4\left\|\boldsymbol{\Lambda}^{\widetilde{V}}\right\| \cdot \left|1 - \lambda_j^{\widetilde{W}}\right| \cdot \operatorname{tr}\left(\left|\boldsymbol{\Lambda}^{\widetilde{V}} + \eta\boldsymbol{I}\right|\right) \\
&\quad + \left(\alpha_3 + \alpha_5\left\|\boldsymbol{\Lambda}^{\widetilde{V}^2}\right\|\right) \cdot \left|1 - \lambda_j^{\widetilde{W}}\right| \cdot \operatorname{tr}\left(\left|\boldsymbol{I} - \boldsymbol{\Lambda}^{\widetilde{W}}\right|\right) + \alpha_6
\end{aligned}
$$

Take the sum of both sides separately, we have

$$
\frac{\mathrm{d}\operatorname{tr}\left[\left(\boldsymbol{\Lambda}^{\widetilde{V}}+\eta\boldsymbol{I}\right)^2\right]}{\mathrm{d}t} + \frac{\mathrm{d}\operatorname{tr}\left[\left(\boldsymbol{I}-\boldsymbol{\Lambda}^{\widetilde{W}}\right)^2\right]}{\mathrm{d}t}
$$

$$
\leq -\left[\frac{1}{\eta(2-\eta)}+O\left(\frac{1}{\log^{\frac{1}{6}}d}\right)\right]\operatorname{tr}\left[\left(\boldsymbol{\Lambda}^{\widetilde{V}}+\eta\boldsymbol{I}\right)^2\right] - \left[\eta^2(k+1)+O\left(k^{\frac{11}{12}}\right)\right]\operatorname{tr}\left[\left(\boldsymbol{I}-\boldsymbol{\Lambda}^{\widetilde{W}}\right)^2\right]
$$

$$
+\left(\alpha_1+\alpha_2\left\|\boldsymbol{\Lambda}^{\widetilde{W}}\right\|+\alpha_4\left\|\boldsymbol{\Lambda}^{\widetilde{V}}\right\|\right)\cdot\operatorname{tr}\left(\left|\boldsymbol{\Lambda}^{\widetilde{V}}+\eta\boldsymbol{I}\right|\right)\cdot\operatorname{tr}\left(\left|\boldsymbol{I}-\boldsymbol{\Lambda}^{\widetilde{W}}\right|\right)
$$

$$
+\left(\alpha_3+\alpha_5\left\|\boldsymbol{\Lambda}^{\widetilde{V}^2}\right\|\right)\cdot\operatorname{tr}^2\left(\left|\boldsymbol{I}-\boldsymbol{\Lambda}^{\widetilde{W}}\right|\right)+\alpha_6
$$

From Jensen's Inequality with $f(x)=x^2$, we have

$$
\left(\frac{\sum_{i=1}^d\lambda_i}{d}\right)^2 \leq \frac{\sum_{i=1}^d\lambda_i^2}{d}.
$$

Therefore, it holds for diagonal matrix $\boldsymbol{\Lambda}\in\mathbb{R}^{d\times d}$ that

$$
\operatorname{tr}^2\left(\boldsymbol{\Lambda}\right)\leq d\operatorname{tr}\left(\boldsymbol{\Lambda}^2\right)
$$

Plug in the inequality and we have

$$
\frac{\mathrm{d}\operatorname{tr}\left[\left(\boldsymbol{\Lambda}^{\widetilde{V}}+\eta\boldsymbol{I}\right)^2\right]}{\mathrm{d}t} + \frac{\mathrm{d}\operatorname{tr}\left[\left(\boldsymbol{I}-\boldsymbol{\Lambda}^{\widetilde{W}}\right)^2\right]}{\mathrm{d}t}
$$

$$
\leq -\frac{1}{2\eta(2-\eta)}\operatorname{tr}\left[\left(\boldsymbol{\Lambda}^{\widetilde{V}}+\eta\boldsymbol{I}\right)^2\right] - \frac{\eta^2}{2}(k+1)\operatorname{tr}\left[\left(\boldsymbol{I}-\boldsymbol{\Lambda}^{\widetilde{W}}\right)^2\right]+\alpha_6
$$

Because $k=\lceil c\log d\rceil$, we have $O\left(d(1-\eta)^k\right)=d^{-c\log\left(\frac{1}{1-\eta}\right)+1}$. So in $O\left(\log\frac{1}{\epsilon}\right)$ time, $\left|\lambda_j^{\widetilde{V}}+\eta\right|$ and $\left|1-\lambda_j^{\widetilde{V}}\right|$ converge to $\epsilon\in\left(\Theta\left(d^{\frac{c}{2}\log(1-\eta)+\frac{1}{2}}\right),1\right)$. $\qquad\square$

**Lemma C.13.** *Suppose* $\delta\in\left(\Theta\left(d^{\frac{c}{2}\log(1-\eta)+\frac{1}{2}}\right),1\right)$ *and there exist diagonal matrices* $\boldsymbol{A}$ *and* $\boldsymbol{B}$ *satisfying* $\|\boldsymbol{A}\|_{op}\leq\Theta(1)$ *and* $\|\boldsymbol{B}\|_{op}\leq\Theta(1)$ *such that*

$$
\boldsymbol{\Lambda}^{\widetilde{V}}=-\eta\boldsymbol{I}+\delta\cdot\boldsymbol{A}\qquad\boldsymbol{\Lambda}^{\widetilde{W}}=\boldsymbol{I}+\delta\cdot\boldsymbol{B},
$$

*then it holds that*

$$
\mathcal{L}^{\mathrm{CoT}}(\boldsymbol{\theta})=O\left(\delta^2 d\log d\right).
$$

*Proof.* Now we consider the CoT loss given by Lemma C.3

$$
\mathcal{L}^{\mathrm{CoT}}(\boldsymbol{\theta})=\frac{1}{2}\mathbb{E}_{\boldsymbol{X},\boldsymbol{w}^*}\sum_{i=0}^{k-1}\left\|(\widetilde{\boldsymbol{V}}\boldsymbol{S}\widetilde{\boldsymbol{W}}+\eta\boldsymbol{S})\boldsymbol{w}_i-(\widetilde{\boldsymbol{V}}+\eta\boldsymbol{I})\boldsymbol{S}\boldsymbol{w}^*\right\|_2^2
$$

$$
+\frac{1}{2}\mathbb{E}_{\boldsymbol{X},\boldsymbol{w}^*}\left\|(\boldsymbol{I}+\widetilde{\boldsymbol{V}}\boldsymbol{S}\widetilde{\boldsymbol{W}})\boldsymbol{w}_k-(\widetilde{\boldsymbol{V}}\boldsymbol{S}+\boldsymbol{I})\boldsymbol{w}^*\right\|_2^2.
$$

Plug in the expression of $\boldsymbol{\Lambda}^{\widetilde{V}}$ and $\boldsymbol{\Lambda}^{\widetilde{W}}$, we get

$$
\mathcal{L}^{\mathrm{CoT}}(\boldsymbol{\theta})=\frac{\delta^2}{2}\mathbb{E}\sum_{i=0}^{k-1}\left\|(\boldsymbol{A}\boldsymbol{S}-\eta\boldsymbol{S}\boldsymbol{B}+\delta\boldsymbol{A}\boldsymbol{S}\boldsymbol{B})\left(\boldsymbol{I}-(\boldsymbol{I}-\eta\boldsymbol{S})^i\right)-\boldsymbol{A}\boldsymbol{S}\right\|_F^2 \tag{16}
$$

$$
+\frac{1}{2}\mathbb{E}\left\|-(\boldsymbol{I}-\eta\boldsymbol{S})^k+\boldsymbol{\Lambda}^{\widetilde{V}}\boldsymbol{S}\left[-(\boldsymbol{I}-\eta\boldsymbol{S})^k+\delta\boldsymbol{B}\left(\boldsymbol{I}-(\boldsymbol{I}-\eta\boldsymbol{S})^k\right)\right]\right\|_F^2. \tag{17}
$$

We first consider the term in the summation:

$$\mathbb{E}\left\|(\boldsymbol{AS} - \eta \boldsymbol{SB} + \delta \boldsymbol{ASB})\left(\boldsymbol{I} - (\boldsymbol{I} - \eta \boldsymbol{S})^i\right) - \boldsymbol{AS}\right\|_F^2$$

$$= \mathbb{E}\left\|(-\eta \boldsymbol{SB} + \delta \boldsymbol{ASB})\left(\boldsymbol{I} - (\boldsymbol{I} - \eta \boldsymbol{S})^i\right) - \boldsymbol{AS}(\boldsymbol{I} - \eta \boldsymbol{S})^i\right\|_F^2$$

$$= \operatorname{tr}\mathbb{E}\left[(-\eta \boldsymbol{SB} + \delta \boldsymbol{ASB})\left(\boldsymbol{I} - (\boldsymbol{I} - \eta \boldsymbol{S})^i\right)^2(-\eta \boldsymbol{BS} + \delta \boldsymbol{BSA})\right]$$

$$\quad - 2\operatorname{tr}\mathbb{E}\left[(-\eta \boldsymbol{SB} + \delta \boldsymbol{ASB})\left(\boldsymbol{I} - (\boldsymbol{I} - \eta \boldsymbol{S})^i\right)(\boldsymbol{I} - \eta \boldsymbol{S})^i \boldsymbol{SA}\right] + \operatorname{tr}\mathbb{E}\left[\boldsymbol{AS}(\boldsymbol{I} - \eta \boldsymbol{S})^{2i}\boldsymbol{SA}\right]$$

$$= \operatorname{tr}\left((-\eta \boldsymbol{I} + \delta \boldsymbol{A})\mathbb{E}\left[\boldsymbol{SB}\left(\boldsymbol{I} - (\boldsymbol{I} - \eta \boldsymbol{S})^i\right)^2 \boldsymbol{BS}\right](-\eta \boldsymbol{I} + \delta \boldsymbol{A})\right) \qquad \text{(Term 1)}$$

$$\quad - 2\operatorname{tr}\left((-\eta \boldsymbol{I} + \delta \boldsymbol{A})\mathbb{E}\left[\boldsymbol{SB}\left(\boldsymbol{I} - (\boldsymbol{I} - \eta \boldsymbol{S})^i\right)(\boldsymbol{I} - \eta \boldsymbol{S})^i \boldsymbol{S}\right]\boldsymbol{A}\right) \qquad \text{(Term 2)}$$

$$\quad + \operatorname{tr}\left(\boldsymbol{A}\mathbb{E}\left[\boldsymbol{S}(\boldsymbol{I} - \eta \boldsymbol{S})^{2i}\boldsymbol{S}\right]\boldsymbol{A}\right) \qquad \text{(Term 3)}$$

Apple Lemma C.15 to the expectation in Term 1, we have

$$\mathbb{E}\left[\boldsymbol{SB}\left(\boldsymbol{I} - (\boldsymbol{I} - \eta \boldsymbol{S})^i\right)^2 \boldsymbol{BS}\right]$$

$$= \left(1 - (1 - \eta)^i\right)^2 \boldsymbol{B}^2 + O\left(\frac{1}{\log^3 d}\right)\left[\boldsymbol{B}^2 + O\left(\frac{1}{d}\right)\operatorname{tr}(\boldsymbol{B})\boldsymbol{B} + O\left(\frac{1}{d}\right)\operatorname{tr}(\boldsymbol{B}^2)\boldsymbol{I} + O\left(\frac{1}{d^2}\right)\operatorname{tr}^2(\boldsymbol{B})\boldsymbol{I}\right].$$

It is obvious that

$$\left\|\mathbb{E}\left[\boldsymbol{SB}\left(\boldsymbol{I} - (\boldsymbol{I} - \eta \boldsymbol{S})^i\right)^2 \boldsymbol{BS}\right]\right\|_{op} \leq \Theta(1).$$

Therefore, for Term 1 we have

$$\operatorname{tr}\left((-\eta \boldsymbol{I} + \delta \boldsymbol{A})\mathbb{E}\left[\boldsymbol{SB}\left(\boldsymbol{I} - (\boldsymbol{I} - \eta \boldsymbol{S})^i\right)^2 \boldsymbol{BS}\right](-\eta \boldsymbol{I} + \delta \boldsymbol{A})\right)$$

$$\leq d\|-\eta \boldsymbol{I} + \delta \boldsymbol{A}\|_{op}^2 \cdot \left\|\mathbb{E}\left[\boldsymbol{SB}\left(\boldsymbol{I} - (\boldsymbol{I} - \eta \boldsymbol{S})^i\right)^2 \boldsymbol{BS}\right]\right\|_{op} \leq O(d).$$

(all matrices in the inequality are diagonal matrices.)

Similarly, for Term 2 and Term 3, we have

$$\left|\operatorname{tr}\left((-\eta \boldsymbol{I} + \delta \boldsymbol{A})\mathbb{E}\left[\boldsymbol{SB}\left(\boldsymbol{I} - (\boldsymbol{I} - \eta \boldsymbol{S})^i\right)(\boldsymbol{I} - \eta \boldsymbol{S})^i \boldsymbol{S}\right]\boldsymbol{A}\right)\right| \leq O(d)$$

$$\operatorname{tr}\left(\boldsymbol{A}\mathbb{E}\left[\boldsymbol{S}(\boldsymbol{I} - \eta \boldsymbol{S})^{2i}\boldsymbol{S}\right]\boldsymbol{A}\right) \qquad\qquad \leq O(d).$$

Add Term 1, 2, 3 together and we have

$$\mathbb{E}\left\|(\boldsymbol{AS} - \eta \boldsymbol{SB} + \delta \boldsymbol{ASB})\left(\boldsymbol{I} - (\boldsymbol{I} - \eta \boldsymbol{S})^i\right) - \boldsymbol{AS}\right\|_F^2 \leq O(d).$$

We then consider the second term in Equation (17):

$$\mathbb{E}\left\|-(\boldsymbol{I} - \eta \boldsymbol{S})^k + \boldsymbol{\Lambda}^{\widetilde{V}}\boldsymbol{S}\left[-(\boldsymbol{I} - \eta \boldsymbol{S})^k + \delta \boldsymbol{B}\left(\boldsymbol{I} - (\boldsymbol{I} - \eta \boldsymbol{S})^k\right)\right]\right\|_F^2$$

$$= \mathbb{E}\left\|-\left(\boldsymbol{I} + \boldsymbol{\Lambda}^{\widetilde{V}}\boldsymbol{S}\right)(\boldsymbol{I} - \eta \boldsymbol{S})^k + \delta \boldsymbol{\Lambda}^{\widetilde{V}}\boldsymbol{SB}\left(\boldsymbol{I} - (\boldsymbol{I} - \eta \boldsymbol{S})^k\right)\right\|_F^2$$

$$= \operatorname{tr}\left(\mathbb{E}\left[\left(\boldsymbol{I} + \boldsymbol{\Lambda}^{\widetilde{V}}\boldsymbol{S}\right)(\boldsymbol{I} - \eta \boldsymbol{S})^{2k}\left(\boldsymbol{I} + \boldsymbol{S}\boldsymbol{\Lambda}^{\widetilde{V}}\right)\right]\right)$$

$$\quad - 2\delta \operatorname{tr}\left(\mathbb{E}\left[\left(\boldsymbol{I} + \boldsymbol{\Lambda}^{\widetilde{V}}\boldsymbol{S}\right)(\boldsymbol{I} - \eta \boldsymbol{S})^k\left(\boldsymbol{I} - (\boldsymbol{I} - \eta \boldsymbol{S})^k\right)\boldsymbol{BS}\right]\boldsymbol{\Lambda}^{\widetilde{V}}\right)$$

$$\quad + \delta^2 \operatorname{tr}\left(\boldsymbol{\Lambda}^{\widetilde{V}}\mathbb{E}\left[\boldsymbol{SB}\left(\boldsymbol{I} - (\boldsymbol{I} - \eta \boldsymbol{S})^k\right)^2 \boldsymbol{BS}\right]\boldsymbol{\Lambda}^{\widetilde{V}}\right)$$

$$\leq O(\delta^2 d). \qquad\qquad ((1 - \eta)^k \leq \delta)$$

Recall the CoT loss in Equation (16) and Equation (17). By the analysis above, we directly obtain that

$$\mathcal{L}^{\mathrm{CoT}}(\boldsymbol{\theta}) = O\big(\delta^2 d \log d\big).$$

Hence, the proof is complete. □

**Lemma C.14.** *Suppose* $\boldsymbol{S} = \frac{1}{n}\sum_{i=1}^n \boldsymbol{x}_i \boldsymbol{x}_i^\top$, $n = \Theta(d \log^5 d)$, $k = O(\log d)$, $\eta = \Theta(1) \in (0.1, 0.9)$, $\|\boldsymbol{\Lambda}\|_{op} \leq \Theta(1)$, $\|\boldsymbol{\Gamma}\|_{op} \leq \Theta(1)$. *Then the expectation*

$$\mathbb{E}\Big[\boldsymbol{S}\boldsymbol{\Lambda}\Big(\boldsymbol{I} - (\boldsymbol{I} - \eta\boldsymbol{S})^k\Big)\boldsymbol{\Gamma}\boldsymbol{S}\Big] = \Big(1 - (1-\eta)^k\Big)\boldsymbol{\Lambda}\boldsymbol{\Gamma} + \Delta,$$

*where* $\|\Delta\|_{op} = O(\frac{k^2 d}{n}) \leq O\Big(\frac{1}{\log^3 d}\Big)$. *Moreover, the error is in the form*

$$\Delta = \alpha_1 \boldsymbol{\Lambda}\boldsymbol{\Gamma} + \alpha_2 \operatorname{tr}(\boldsymbol{\Lambda})\boldsymbol{\Gamma} + \alpha_3 \operatorname{tr}(\boldsymbol{\Gamma})\boldsymbol{\Lambda} + \alpha_4 \operatorname{tr}(\boldsymbol{\Lambda})\operatorname{tr}(\boldsymbol{\Gamma})\boldsymbol{I} + \alpha_5 \operatorname{tr}(\boldsymbol{\Lambda}\boldsymbol{\Gamma})\boldsymbol{I}$$

*where* $\alpha_1 = O\Big(\frac{k^2 d}{n}\Big), \alpha_2, \alpha_3, \alpha_5 = O\Big(\frac{k^2}{n}\Big), \alpha_4 = O(\frac{k^2}{nd})$.

*Proof.* We can directly get the lemma by applying Lemma D.2 to $\mathbb{E}[\boldsymbol{S}\boldsymbol{\Lambda}\boldsymbol{\Gamma}\boldsymbol{S}]$, $\mathbb{E}\Big[\boldsymbol{S}\boldsymbol{\Lambda}(\boldsymbol{I} - \eta\boldsymbol{S})^k \boldsymbol{\Gamma}\boldsymbol{S}\Big]$.
□

**Lemma C.15.** *Suppose* $\boldsymbol{S} = \frac{1}{n}\sum_{i=1}^n \boldsymbol{x}_i \boldsymbol{x}_i^\top$, $n = \Theta(d \log^5 d)$, $k = O(\log d)$, $\eta = \Theta(1) \in (0.1, 0.9)$, $\|\boldsymbol{\Lambda}\|_{op} \leq \Theta(1)$, $\|\boldsymbol{\Gamma}\|_{op} \leq \Theta(1)$. *Then the expectation*

$$\mathbb{E}\Big[\boldsymbol{S}\boldsymbol{\Lambda}\Big(\boldsymbol{I} - (\boldsymbol{I} - \eta\boldsymbol{S})^k\Big)^2 \boldsymbol{\Gamma}\boldsymbol{S}\Big] = \Big(1 - (1-\eta)^k\Big)^2 \boldsymbol{\Lambda}\boldsymbol{\Gamma} + \Delta,$$

*where* $\|\Delta\|_{op} = O(\frac{k^2 d}{n}) \leq O\Big(\frac{1}{\log^3 d}\Big)$. *Moreover, the error is in the form*

$$\Delta = \alpha_1 \boldsymbol{\Lambda}\boldsymbol{\Gamma} + \alpha_2 \operatorname{tr}(\boldsymbol{\Lambda})\boldsymbol{\Gamma} + \alpha_3 \operatorname{tr}(\boldsymbol{\Gamma})\boldsymbol{\Lambda} + \alpha_4 \operatorname{tr}(\boldsymbol{\Lambda})\operatorname{tr}(\boldsymbol{\Gamma})\boldsymbol{I} + \alpha_5 \operatorname{tr}(\boldsymbol{\Lambda}\boldsymbol{\Gamma})\boldsymbol{I}$$

*where* $\alpha_1 = O\Big(\frac{k^2 d}{n}\Big), \alpha_2, \alpha_3, \alpha_5 = O\Big(\frac{k^2}{n}\Big), \alpha_4 = O(\frac{k^2}{nd})$.

*Proof.* We can directly get the lemma by applying Lemma D.2 to $\mathbb{E}[\boldsymbol{S}\boldsymbol{\Lambda}\boldsymbol{\Gamma}\boldsymbol{S}]$, $\mathbb{E}\Big[\boldsymbol{S}\boldsymbol{\Lambda}(\boldsymbol{I} - \eta\boldsymbol{S})^k \boldsymbol{\Gamma}\boldsymbol{S}\Big]$ and $\mathbb{E}\Big[\boldsymbol{S}\boldsymbol{\Lambda}(\boldsymbol{I} - \eta\boldsymbol{S})^{2k} \boldsymbol{\Gamma}\boldsymbol{S}\Big]$.
□

**Lemma C.16.** *Suppose* $\boldsymbol{S} = \frac{1}{n}\sum_{i=1}^n \boldsymbol{x}_i \boldsymbol{x}_i^\top$, $n = \Theta(d \log^5 d)$, $k = O(\log d)$, $\eta = \Theta(1) \in (0.1, 0.9)$, $\|\boldsymbol{\Lambda}\|_{op} \leq \Theta(1)$, $\|\boldsymbol{\Gamma}\|_{op} \leq \Theta(1)$. *Then the expectation*

$$\mathbb{E}\Big[\boldsymbol{S}\boldsymbol{\Lambda}\boldsymbol{S}\boldsymbol{\Gamma}\Big(\boldsymbol{I} - (\boldsymbol{I} - \eta\boldsymbol{S})^k\Big)\Big] = \Big(1 - (1-\eta)^k\Big)\boldsymbol{\Lambda}\boldsymbol{\Gamma} + \Delta,$$

*where* $\|\Delta\|_{op} = O(\frac{k^2 d}{n}) \leq O\Big(\frac{1}{\log^3 d}\Big)$. *Moreover, the error is in the form*

$$\Delta = \alpha_1 \boldsymbol{\Lambda}\boldsymbol{\Gamma} + \alpha_2 \operatorname{tr}(\boldsymbol{\Lambda})\boldsymbol{\Gamma} + \alpha_3 \operatorname{tr}(\boldsymbol{\Gamma})\boldsymbol{\Lambda} + \alpha_4 \operatorname{tr}(\boldsymbol{\Lambda})\operatorname{tr}(\boldsymbol{\Gamma})\boldsymbol{I} + \alpha_5 \operatorname{tr}(\boldsymbol{\Lambda}\boldsymbol{\Gamma})\boldsymbol{I}$$

*where* $\alpha_1 = O\Big(\frac{k^2 d}{n}\Big), \alpha_2, \alpha_3, \alpha_5 = O\Big(\frac{k^2}{n}\Big), \alpha_4 = O(\frac{k^2}{nd})$.

*Proof.* We can directly get the lemma by applying Lemma D.3 to $\mathbb{E}[\boldsymbol{S}\boldsymbol{\Lambda}\boldsymbol{S}\boldsymbol{\Gamma}]$, $\mathbb{E}\Big[\boldsymbol{S}\boldsymbol{\Lambda}\boldsymbol{S}\boldsymbol{\Gamma}(\boldsymbol{I} - \eta\boldsymbol{S})^k\Big]$.
□

**Lemma C.17.** *Suppose* $\boldsymbol{S} = \frac{1}{n}\sum_{i=1}^n \boldsymbol{x}_i \boldsymbol{x}_i^\top$, $n = \Theta(d \log^5 d)$, $k = O(\log d)$, $\eta = \Theta(1) \in (0.1, 0.9)$, $\|\boldsymbol{\Lambda}\|_{op} \leq \Theta(1)$, $\|\boldsymbol{\Gamma}\|_{op} \leq \Theta(1)$. *Then the expectation*

$$\mathbb{E}\Big[\boldsymbol{S}\boldsymbol{\Lambda}\boldsymbol{S}\boldsymbol{\Gamma}\Big(\boldsymbol{I} - (\boldsymbol{I} - \eta\boldsymbol{S})^k\Big)^2\Big] = \Big(1 - (1-\eta)^k\Big)^2 \boldsymbol{\Lambda}\boldsymbol{\Gamma} + \Delta,$$

*where* $\|\Delta\|_{op} = O(\frac{k^2 d}{n}) \leq O\Big(\frac{1}{\log^3 d}\Big)$. *Moreover, the error is in the form*

$$\Delta = \alpha_1 \boldsymbol{\Lambda}\boldsymbol{\Gamma} + \alpha_2 \operatorname{tr}(\boldsymbol{\Lambda})\boldsymbol{\Gamma} + \alpha_3 \operatorname{tr}(\boldsymbol{\Gamma})\boldsymbol{\Lambda} + \alpha_4 \operatorname{tr}(\boldsymbol{\Lambda})\operatorname{tr}(\boldsymbol{\Gamma})\boldsymbol{I} + \alpha_5 \operatorname{tr}(\boldsymbol{\Lambda}\boldsymbol{\Gamma})\boldsymbol{I}$$

*where* $\alpha_1 = O\Big(\frac{k^2 d}{n}\Big), \alpha_2, \alpha_3, \alpha_5 = O\Big(\frac{k^2}{n}\Big), \alpha_4 = O(\frac{k^2}{nd})$.

*Proof.* We can directly get the lemma by applying Lemma D.3 to $\mathbb{E}[\boldsymbol{S}\boldsymbol{\Lambda}\boldsymbol{S}\boldsymbol{\Gamma}]$, $\mathbb{E}\left[\boldsymbol{S}\boldsymbol{\Lambda}\boldsymbol{S}\boldsymbol{\Gamma}(\boldsymbol{I}-\eta\boldsymbol{S})^k\right]$ and $\mathbb{E}\left[\boldsymbol{S}\boldsymbol{\Lambda}\boldsymbol{S}\boldsymbol{\Gamma}(\boldsymbol{I}-\eta\boldsymbol{S})^{2k}\right]$. $\square$

### C.4 OUT-OF-DISTRIBUTION GENERALIZATION

We restate the formal theorem here. We still denote $\boldsymbol{S} := \frac{1}{n}\boldsymbol{X}\boldsymbol{X}^\top$ for simplicity. Note that the number of steps $k$ can be different/larger compared to the step number in the previous training theorem.

**Theorem C.2.** *Suppose $n = \Theta(d\log^5 d)$, $\eta \in (0.1, 0.9)$, $k = C\log d$. Assume the out-of-distribution input data $\boldsymbol{x}_i \sim \mathcal{N}(\boldsymbol{0}_d, \boldsymbol{\Sigma}), i \in [n]$ where $\frac{\delta}{\eta} \leq \lambda_{\min}(\boldsymbol{\Sigma}) \leq \lambda_{\max}(\boldsymbol{\Sigma}) \leq \frac{2-\delta}{\eta}$ for some constant $\delta > 0.1$, and $\boldsymbol{w}^* \sim \mathcal{N}(\boldsymbol{0}_d, \boldsymbol{I})$. Then the trained transformer in Theorem 4.1 satisfies that $\mathcal{L}_{\boldsymbol{\Sigma}}^{\mathrm{Eval}}(t) \leq \epsilon$ for any $\epsilon \in \left(d^{-C\log\left(\min\{\frac{1}{1-\eta}, \frac{1}{1-\delta}\}\right)+1}\log^2 d, 1\right)$.*

*Proof.* Recall the definition of the evaluation loss and our reduced transformer (Definition C.1)

$$\mathcal{L}^{\mathrm{Eval}}(\boldsymbol{V}, \boldsymbol{W}) = \frac{1}{2}\mathbb{E}_{\boldsymbol{X}, \boldsymbol{w}^*}\left[\left\|f_{\mathrm{LSA}}(\hat{\boldsymbol{Z}}_k)_{[:,-1]} - (\boldsymbol{0}_d, 0, \boldsymbol{w}^*, 1)\right\|^2\right]$$

$$= \frac{1}{2}\mathbb{E}\left[\|f_{\boldsymbol{\theta}}(\hat{\boldsymbol{w}}_k) - \boldsymbol{w}^*\|^2\right]$$

where $\hat{\boldsymbol{Z}}_k$ is the generated sequence after $k$ steps and $\hat{\boldsymbol{w}}_k := f_{\boldsymbol{\theta}}(\hat{\boldsymbol{w}}_{k-1})$ is the $k$-th generated intermediate weight vector. Note that each step the transformer is inputted with the last step prediction. We define the prediction error at each step $i$ is $\Delta\boldsymbol{w}_i := \hat{\boldsymbol{w}}_i - \boldsymbol{w}_i = f_{\boldsymbol{\theta}}(\hat{\boldsymbol{w}}_{i-1}) - \boldsymbol{w}_i$. We expand the term $f_{\boldsymbol{\theta}}(\hat{\boldsymbol{w}}_k) - \boldsymbol{w}^*$ and sum up the error accumulation as follows:

$$f_{\boldsymbol{\theta}}(\hat{\boldsymbol{w}}_k) - \boldsymbol{w}^* \leq (\boldsymbol{w}_{k+1} - \boldsymbol{w}^*) + (f_{\boldsymbol{\theta}}(\hat{\boldsymbol{w}}_k) - \boldsymbol{w}_{k+1})$$

$$= (\boldsymbol{w}_{k+1} - \boldsymbol{w}^*) + \hat{\boldsymbol{w}}_k + \widetilde{\boldsymbol{V}}\boldsymbol{S}(\widetilde{\boldsymbol{W}}\hat{\boldsymbol{w}}_k - \boldsymbol{w}^*) - \boldsymbol{w}_{k+1}$$

$$\leq (\boldsymbol{w}_{k+1} - \boldsymbol{w}^*) + \left(\boldsymbol{w}_k + \widetilde{\boldsymbol{V}}\boldsymbol{S}(\widetilde{\boldsymbol{W}}\boldsymbol{w}_k - \boldsymbol{w}^*) - \boldsymbol{w}_{k+1}\right) + \left(\boldsymbol{I} + \widetilde{\boldsymbol{V}}\boldsymbol{S}\widetilde{\boldsymbol{W}}\right)\Delta\boldsymbol{w}_k.$$

After one step of decomposition, we notice that the error $\Delta\boldsymbol{w}_{k+1}$ can be decomposed into two parts: (1) The approximation error predicting $\boldsymbol{w}_{k+1}$ with ground-truth input $\boldsymbol{w}_k$. We define it

$$\Delta_{k+1}^{\mathrm{pred}} := \boldsymbol{w}_k + \widetilde{\boldsymbol{V}}\boldsymbol{S}(\widetilde{\boldsymbol{W}}\boldsymbol{w}_k - \boldsymbol{w}^*) - \boldsymbol{w}_{k+1}$$

(2) The accumulated error from the last inference step: $\left(\boldsymbol{I} + \widetilde{\boldsymbol{V}}\boldsymbol{S}\widetilde{\boldsymbol{W}}\right)\Delta\boldsymbol{w}_k$. Therefore, we can inductively calculate the sum of the error:

$$f_{\boldsymbol{\theta}}(\hat{\boldsymbol{w}}_k) - \boldsymbol{w}^* \leq (\boldsymbol{w}_{k+1} - \boldsymbol{w}^*) + \left(\boldsymbol{w}_k + \widetilde{\boldsymbol{V}}\boldsymbol{S}(\widetilde{\boldsymbol{W}}\boldsymbol{w}_k - \boldsymbol{w}^*) - \boldsymbol{w}_{k+1}\right) + \left(\boldsymbol{I} + \widetilde{\boldsymbol{V}}\boldsymbol{S}\widetilde{\boldsymbol{W}}\right)\Delta\boldsymbol{w}_k.$$

$$= (\boldsymbol{w}_{k+1} - \boldsymbol{w}^*) + \Delta_{k+1}^{\mathrm{pred}} + \left(\boldsymbol{I} + \widetilde{\boldsymbol{V}}\boldsymbol{S}\widetilde{\boldsymbol{W}}\right)\Delta\boldsymbol{w}_k$$

$$= (\boldsymbol{w}_{k+1} - \boldsymbol{w}^*) + \Delta_{k+1}^{\mathrm{pred}} + \left(\boldsymbol{I} + \widetilde{\boldsymbol{V}}\boldsymbol{S}\widetilde{\boldsymbol{W}}\right)\Delta_k^{\mathrm{pred}} + \left(\boldsymbol{I} + \widetilde{\boldsymbol{V}}\boldsymbol{S}\widetilde{\boldsymbol{W}}\right)^2\Delta\boldsymbol{w}_{k-1}$$

$$= (\boldsymbol{w}_{k+1} - \boldsymbol{w}^*) + \sum_{i=0}^{k}\left(\boldsymbol{I} + \widetilde{\boldsymbol{V}}\boldsymbol{S}\widetilde{\boldsymbol{W}}\right)^i\Delta_{k-i+1}^{\mathrm{pred}} \qquad (\Delta\boldsymbol{w}_0 = 0 \text{ by definition.})$$

Then we have our evaluation loss upper bounded:

$$\frac{1}{2}\mathbb{E}\left[\|f_{\boldsymbol{\theta}}(\hat{\boldsymbol{w}}_k) - \boldsymbol{w}^*\|^2\right] = \frac{1}{2}\mathbb{E}\left\|(\boldsymbol{w}_{k+1} - \boldsymbol{w}^*) + \sum_{i=0}^{k}\left(\boldsymbol{I} + \widetilde{\boldsymbol{V}}\boldsymbol{S}\widetilde{\boldsymbol{W}}\right)^i\Delta_{k-i+1}^{\mathrm{pred}}\right\|^2$$

$$\leq \frac{k+2}{2}\left(\mathbb{E}\|(\boldsymbol{w}_{k+1} - \boldsymbol{w}^*)\|^2 + \sum_{i=0}^{k}\mathbb{E}\left\|(\boldsymbol{I} + \widetilde{\boldsymbol{V}}\boldsymbol{S}\widetilde{\boldsymbol{W}})^i\Delta_{k-i+1}^{\mathrm{pred}}\right\|^2\right) \quad (*)$$

We first consider the first term: $\mathbb{E}\left\|(\boldsymbol{w}_{k+1} - \boldsymbol{w}^*)\right\|^2$:

$$\mathbb{E}\left\|\boldsymbol{w}_{k+1} - \boldsymbol{w}^*\right\|^2 = \mathbb{E}\left\|\left(\boldsymbol{I} - (\boldsymbol{I} - \eta\boldsymbol{S})^{k+1}\right)\boldsymbol{w}^* - \boldsymbol{w}^*\right\|^2 = \operatorname{tr}\left(\mathbb{E}(\boldsymbol{I} - \eta\boldsymbol{S})^{2k+2}\right)$$

$$\leq 2d(1 - \delta)^{2k+2} \leq 2d^{-2c\log\left(\frac{1}{1-\delta}\right)+1}. \qquad \text{(Lemma D.5)}$$

Then we consider the second summation term. Since the parameters of the reduced model $\widetilde{\boldsymbol{V}} = -\eta\boldsymbol{I} + \boldsymbol{A}, \widetilde{\boldsymbol{W}} = \boldsymbol{I} + \boldsymbol{B}$, where $\|\boldsymbol{A}\|_{op}, \|\boldsymbol{B}\|_{op} \leq d^{-\frac{1}{2}C\log\left(\frac{1}{1-\eta}\right)+\frac{1}{2}}$ for some constant $c > 0$, we want to bound the prediction error given the ground-truth input. By Lemma D.6, we have

$$\mathbb{E}\sum_{i=0}^{k}\left\|(\boldsymbol{I} + \widetilde{\boldsymbol{V}}\boldsymbol{S}\widetilde{\boldsymbol{W}})^i\Delta_{k-i+1}^{\text{pred}}\right\|^2$$

$$= \mathbb{E}\sum_{i=0}^{k}\left\|(\boldsymbol{I} + \widetilde{\boldsymbol{V}}\boldsymbol{S}\widetilde{\boldsymbol{W}})^i(\boldsymbol{w}_{k-i} + \widetilde{\boldsymbol{V}}\boldsymbol{S}(\widetilde{\boldsymbol{W}}\boldsymbol{w}_{k-i} - \boldsymbol{w}^*) - \boldsymbol{w}_{k-i+1})\right\|^2$$

$$\leq O\left(d^{-C\log\left(\frac{1}{1-\eta}\right)+1} \cdot k\right).$$

Therefore, plug those back to Equation (*), the total evaluation loss should be upper bounded by

$$\mathcal{L}^{\text{Eval}}(\boldsymbol{\theta}) \leq O\left(d^{-C\log\left(\min\{\frac{1}{1-\eta}, \frac{1}{1-\delta}\}\right)+1} \cdot k^2\right) = O(d^{-C\log\left(\min\{\frac{1}{1-\eta}, \frac{1}{1-\delta}\}\right)+1}\log^2 d)$$

$\square$

# D  SUPPLEMENTARY LEMMAS

## D.1  CONCENTRATION LEMMAS

In this appendix, we prove some concentration lemmas to estimate the expected gradient more accurately. Throughout the proof, $\boldsymbol{\Lambda}, \boldsymbol{\Gamma}$ are both symmetric matrices with orthonormal eigenbasis $\{\boldsymbol{u}_i\}_{i=1}^{d}$.

**Lemma D.1.** *Suppose* $\boldsymbol{S} = \frac{1}{n}\sum_{i=1}^{n}\boldsymbol{x}_i\boldsymbol{x}_i^\top$, $n = \Theta(d\log^5 d), k = O(\log d), \eta = \Theta(1) \in (0.1, 0.9), \|\boldsymbol{\Lambda}\|_{op} \leq \Theta(1).$ *Then the expectation*

$$\mathbb{E}\left[\boldsymbol{S}\boldsymbol{\Lambda}(\boldsymbol{I} - \eta\boldsymbol{S})^k\boldsymbol{S}\right] = (1-\eta)^k(\boldsymbol{\Lambda} + \Delta),$$

*where* $\|\Delta\|_{op} \leq O(\frac{k^2d}{n}) = O\left(\frac{1}{\log^3 d}\right).$ *Moreover, the error is in the form* $\Delta = \alpha_1\boldsymbol{\Lambda} + \alpha_2\operatorname{tr}(\boldsymbol{\Lambda})\boldsymbol{I},$ *where* $\alpha_1 = O\left(\frac{k^2d}{n}\right), \alpha_2 = O\left(\frac{k^2}{n}\right).$

*Proof.* Denote $\delta\boldsymbol{S} := \boldsymbol{S} - \boldsymbol{I}$. Then we expand the term $\boldsymbol{S}\boldsymbol{\Lambda}(\boldsymbol{I} - \eta\boldsymbol{S})^k\boldsymbol{S}$:

$$\boldsymbol{S}\boldsymbol{\Lambda}(\boldsymbol{I} - \eta\boldsymbol{S})^k\boldsymbol{S}$$

$$= (\boldsymbol{I} + \delta\boldsymbol{S})\boldsymbol{\Lambda}((1-\eta)\boldsymbol{I} - \eta\delta\boldsymbol{S})^k(\boldsymbol{I} + \delta\boldsymbol{S})$$

$$= (1-\eta)^k(\boldsymbol{I} + \delta\boldsymbol{S})\boldsymbol{\Lambda}\left(\boldsymbol{I} - \frac{\eta}{(1-\eta)}\delta\boldsymbol{S}\right)^k(\boldsymbol{I} + \delta\boldsymbol{S})$$

$$= (1-\eta)^k(\boldsymbol{I} + \delta\boldsymbol{S})\boldsymbol{\Lambda}\left(\boldsymbol{I} - \frac{k\eta}{(1-\eta)}\delta\boldsymbol{S} + \binom{k}{2}\left(\frac{\eta}{1-\eta}\right)^2\delta\boldsymbol{S}^2 + \sum_{j=3}^{k}\binom{k}{j}\left(\frac{-\eta}{1-\eta}\right)^j\delta\boldsymbol{S}^j\right)$$

$$+ (1-\eta)^k(\boldsymbol{I} + \delta\boldsymbol{S})\boldsymbol{\Lambda}\left(\boldsymbol{I} - \frac{k\eta}{(1-\eta)}\delta\boldsymbol{S} + \binom{k}{2}\left(\frac{\eta}{1-\eta}\right)^2\delta\boldsymbol{S}^2 + \sum_{j=3}^{k}\binom{k}{j}\left(\frac{-\eta}{1-\eta}\right)^j\delta\boldsymbol{S}^j\right)\delta\boldsymbol{S}$$

Now take expectation to both sides. Note that $\mathbb{E}[\delta\boldsymbol{S}] = 0$, so all the terms only contain first order $\delta\boldsymbol{S}$ vanish. We denote

$$(1-\eta)^k\widetilde{\Delta} = \boldsymbol{S}\boldsymbol{\Lambda}(\boldsymbol{I} - \eta\boldsymbol{S})^k\boldsymbol{S} - (1-\eta)^k\left(\boldsymbol{\Lambda} + \delta\boldsymbol{S}\boldsymbol{\Lambda} + \boldsymbol{\Lambda}\delta\boldsymbol{S} - \frac{k\eta}{1-\eta}\boldsymbol{\Lambda}\delta\boldsymbol{S}\right),$$

which denotes all the higher order terms (the degree of $\delta \boldsymbol{S} \geq 2$.)

Since we have the tail bound for $\delta \boldsymbol{S}$ in Theorem 4.6.1 Vershynin (2018) (In this lemma $\|\cdot\|$ is operator norm if without specification):

$$\Pr\left(\|\delta \boldsymbol{S}\| > \max\left(\delta, \delta^2\right)\right) \leq 2\exp\{-s^2\}, \quad \text{where } \delta = C\left(\sqrt{\frac{d}{n}} + \frac{s}{\sqrt{n}}\right) \tag{18}$$

We can estimate the expectation using this property. First, given $s = \sqrt{d}$ and $\|\delta \boldsymbol{S}\| \leq \max\left(\delta, \delta^2\right) = C\sqrt{\frac{d}{n}}$ (since $n = \Theta(d\log^5 d)$), we can upper bound the operator norm of $\widetilde{\Delta}$:

$$\|\widetilde{\Delta}\|_{op} \leq \left\|\boldsymbol{\Lambda}\left(\binom{k}{2}\left(\frac{\eta}{1-\eta}\right)^2\delta \boldsymbol{S}^2 + \sum_{j=3}^{k}\binom{k}{j}\left(\frac{-\eta}{1-\eta}\right)^j\delta \boldsymbol{S}^j\right)\right\|_{op}$$

$$+ \left\|\delta \boldsymbol{S}\boldsymbol{\Lambda}\left(-\frac{k\eta}{(1-\eta)}\delta \boldsymbol{S} + \binom{k}{2}\left(\frac{\eta}{1-\eta}\right)^2\delta \boldsymbol{S}^2 + \sum_{j=3}^{k}\binom{k}{j}\left(\frac{-\eta}{1-\eta}\right)^j\delta \boldsymbol{S}^j\right)\right\|_{op}$$

$$+ \left\|\delta \boldsymbol{S}\boldsymbol{\Lambda}\left(\boldsymbol{I} - \frac{k\eta}{(1-\eta)}\delta \boldsymbol{S} + \binom{k}{2}\left(\frac{\eta}{1-\eta}\right)^2\delta \boldsymbol{S}^2 + \sum_{j=3}^{k}\binom{k}{j}\left(\frac{-\eta}{1-\eta}\right)^j\delta \boldsymbol{S}^j\right)\delta \boldsymbol{S}\right\|_{op}$$

$$+ \left\|\boldsymbol{\Lambda}\left(-\frac{k\eta}{(1-\eta)}\delta \boldsymbol{S} + \binom{k}{2}\left(\frac{\eta}{1-\eta}\right)^2\delta \boldsymbol{S}^2 + \sum_{j=3}^{k}\binom{k}{j}\left(\frac{-\eta}{1-\eta}\right)^j\delta \boldsymbol{S}^j\right)\delta \boldsymbol{S}\right\|_{op}$$

Now upper bound all matrices with their operator norm and combine all terms with the same degree of $\delta \boldsymbol{S}$. We have

$$\|\widetilde{\Delta}\|_{op} \leq \sum_{j=2}^{k+2}\|\boldsymbol{\Lambda}\|\left(\binom{k}{j}\left(\frac{\eta}{1-\eta}\right)^j + 2\binom{k}{j-1}\left(\frac{\eta}{1-\eta}\right)^{j-1} + \binom{k}{j-2}\left(\frac{\eta}{1-\eta}\right)^{j-2}\right)\|\delta \boldsymbol{S}\|^j$$

$$\leq \sum_{j=2}^{k+2}\|\boldsymbol{\Lambda}\|\left((9k)^j + 2(9k)^{j-1} + (9k)^{j-2}\right)\|\delta \boldsymbol{S}\|^j \qquad \left(\frac{\eta}{1-\eta} \leq 9, \binom{k}{j} \leq k^j.\right)$$

$$\leq 4\sum_{j=2}^{k+2}\|\boldsymbol{\Lambda}\| \cdot (9k)^j\left(C\sqrt{\frac{d}{n}}\right)^j \qquad \left(\|\delta \boldsymbol{S}\| \leq C\sqrt{\frac{d}{n}}.\right)$$

$$\leq 4\|\boldsymbol{\Lambda}\| \cdot \frac{81C^2k^2d}{n} \cdot \frac{1}{1 - \left(\frac{9kd^{1/2}}{n^{1/2}}\right)} \leq C'\frac{k^2d}{n} \leq O\left(\frac{1}{\log^3 d}\right). \tag{*1}$$

Given this upper bound, we can now upper bound the operator norm of the error term $\Delta := \mathbb{E}[\widetilde{\Delta}]$. Suppose $\boldsymbol{u} := \arg\max_{\boldsymbol{u}:\|\boldsymbol{u}\|=1}\frac{\|\Delta \boldsymbol{u}\|}{\|\boldsymbol{u}\|}$, then the operator norm becomes:

$$\|\Delta\| = \left|\boldsymbol{u}^\top\mathbb{E}[\widetilde{\Delta}]\boldsymbol{u}\right|$$

$$= \mathbb{E}\left[\left|\boldsymbol{u}^\top\widetilde{\Delta}\boldsymbol{u}\right|\left(\mathbb{1}\left\{\|\widetilde{\Delta}\| \leq C'\frac{k^2d}{n}\right\} + \mathbb{1}\left\{\|\widetilde{\Delta}\| > C'\frac{k^2d}{n}\right\}\right)\right]$$

$$\leq C'\frac{k^2d}{n} + \int_{\frac{C'k^2d}{n}}^{\infty}\Pr\left[\|\widetilde{\Delta}\| \geq s\right]\mathrm{d}s$$

When $\|\widetilde{\Delta}\| \geq s$ where $s \geq \frac{C'k^2d}{n}$, we can first upper bound the $\|\widetilde{\Delta}\|$ with $\|\delta \boldsymbol{S}\|$ using the second row of eq. (*1): there exists some constant $C_1 > 0$ s.t.

$$\|\widetilde{\Delta}\| \leq 4\sum_{j=2}^{k+2}\|\boldsymbol{\Lambda}\| \cdot (9k\|\delta \boldsymbol{S}\|)^j \leq \max\left((C_1k\|\delta \boldsymbol{S}\|)^2, (C_1k\|\delta \boldsymbol{S}\|)^{k+2}\right).$$

Therefore, when $\|\widetilde{\Delta}\| \geq s$, $\|\delta\boldsymbol{S}\| \geq \min\{\frac{s^{1/2}}{C_1 k}, \frac{s^{1/(k+2)}}{C_1 k}\}$. To apply the tail bound, we need to make sure we pick some $s'$ such that $\max(\delta, \delta^2) \leq \min\{\frac{s^{1/2}}{C_1 k}, \frac{s^{1/(k+2)}}{C_1 k}\}$ to upper bound the integral of probability, where $\delta = C(\sqrt{\frac{d}{n}} + \frac{s'}{\sqrt{n}})$. Now since $s > \frac{C' k^2 d}{n}$, $\min\{\frac{s^{1/2}}{C_1 k}, \frac{s^{1/(k+2)}}{C_1 k}\} \geq C_\alpha \sqrt{\frac{d}{n}}$ for some constant $C_\alpha$. Therefore, we just need $\max\{\frac{s'}{\sqrt{n}}, \frac{s'^2}{n}\} \leq \min\{\frac{s^{1/2}}{C_1 k}, \frac{s^{1/(k+2)}}{C_1 k}\}$, i.e. $s' \leq \min\left\{C_2 \frac{s^{1/(k+2)}\sqrt{n}}{k}, C_3 \frac{s^{1/(2k+4)}\sqrt{n}}{\sqrt{k}}, C_4 \frac{\sqrt{sn}}{k}, C_5 \frac{s^{1/4}\sqrt{n}}{\sqrt{k}}\right\}$.

Applying the tail bound (18) with $s' = \min\left\{C_2 \frac{s^{1/(k+2)}\sqrt{n}}{k}, C_3 \frac{s^{1/(2k+4)}\sqrt{n}}{\sqrt{k}}, C_4 \frac{\sqrt{sn}}{k}, C_5 \frac{s^{1/4}\sqrt{n}}{\sqrt{k}}\right\}$ where $C_2, C_3, C_4, C_5$ are some constant, we have the error term for the tail expectation,

$$
\int_{\frac{C' k^2 d}{n}}^{\infty} \Pr\Big[\|\widetilde{\Delta}\| \geq s\Big] \mathrm{d}s \leq \int_{\frac{C' k^2 d}{n}}^{\infty} \Pr\Big[\|\delta\boldsymbol{S}\| \geq \min\Big\{\frac{s^{1/2}}{C_1 k}, \frac{s^{1/(k+2)}}{C_1 k}\Big\}\Big] \mathrm{d}s
$$
$$
\leq 2\int_{\frac{C' k^2 d}{n}}^{\infty} \exp\{-s'^2\} \mathrm{d}s.
$$

Now we estimate the upper bound of error with

$$
s'^2 = \min\left\{C_2^2 \cdot \frac{s^{2/(k+2)}}{k^2}n, C_3^2 \cdot \frac{s^{1/(k+2)}}{k}n, C_4^2 \cdot \frac{sn}{k^2}, C_5^2 \cdot \frac{\sqrt{sn}}{k}\right\}.
$$

For the first term, let $x = \frac{C_2^2 n}{k^2} s^{2/(k+2)}$:

$$
2\int_{\frac{C' k^2 d}{n}}^{\infty} \exp\left\{-C_2^2 \cdot \frac{s^{2/(k+2)}}{k^2}n\right\} \mathrm{d}s
$$
$$
= (k+2)\int_{\frac{C_2^2 n}{k^2}\left(\frac{C' k^2 d}{n}\right)^{\frac{2}{k+2}}}^{\infty} \left(\frac{k^2}{C_2^2 n}\right)^{(k+2)/2} \exp\{-x\} x^{k/2} \mathrm{d}x
$$
$$
\leq (k+2) \cdot \left(\frac{k^2}{C_2^2 n}\right)^{(k+2)/2} \cdot \left(\frac{C_2^2 n}{k^2}\left(\frac{C' k^2 d}{n}\right)^{\frac{2}{k+2}}\right)^{k/2} \exp\left\{-\frac{C_2^2 n}{k^2}\left(\frac{C' k^2 d}{n}\right)^{\frac{2}{k+2}}\right\} \leq \frac{k^2 d}{n}.
$$

The second term, let $x = C_3^2 \cdot \frac{s^{1/(k+2)}}{k}n$:

$$
2\int_{\frac{C' k^2 d}{n}}^{\infty} \exp\left\{-C_3^2 \cdot \frac{s^{1/(k+2)}}{k}n\right\} \mathrm{d}s
$$
$$
= 2(k+2)\int_{\frac{C_3 n}{k}\left(\frac{C' k^2 d}{n}\right)^{\frac{1}{k+2}}}^{\infty} \left(\frac{k}{C_3^2 n}\right)^{k+2} \exp\{-x\} x^{k+1} \mathrm{d}x
$$
$$
\leq 2(k+2) \cdot \left(\frac{k}{C_3^2 n}\right)^{k+2} \cdot \left(\frac{C_3^2 n}{k}\left(\frac{C' k^2 d}{n}\right)^{\frac{1}{k+2}}\right)^{k+1} \exp\left\{-\frac{C_3^2 n}{k}\left(\frac{C' k^2 d}{n}\right)^{\frac{1}{k+2}}\right\} \leq \frac{k^2 d}{n}.
$$

For the third term, let $x = \frac{C_4^2 sn}{k^2}$:

$$
2\int_{\frac{C' k^2 d}{n}}^{\infty} \exp\left\{-\frac{C_4^2 sn}{k^2}\right\} \mathrm{d}s = \int_{\frac{C' k^2 d}{n} \cdot \frac{C_4^2 n}{k^2}}^{\infty} \frac{k^2}{C_4^2 n} \exp\{-x\} \mathrm{d}x
$$
$$
\leq \frac{k^2}{C_4^2 n} \exp\left\{-\frac{C' k^2 d}{n} \cdot \frac{C_4^2 n}{k^2}\right\} \leq \frac{k^2 d}{n}.
$$

The fourth term, let $x = C_5^2 \cdot \frac{s^{1/2}}{k}n$:

$$
2\int_{\frac{C' k^2 d}{n}}^{\infty} \exp\left\{-C_5^2 \cdot \frac{s^{1/2}}{k}n\right\} \mathrm{d}s
$$

$$= \frac{4k^2}{n^2 C_5^4} \int_{C_5^2 \frac{n}{k} \left( \frac{C' k^2 d}{n} \right)^{1/2}}^{\infty} \exp\{-x\} x \mathrm{d}x$$

$$\leq \frac{4k^2}{n^2 C_5^4} \cdot C_5^2 \frac{n}{k} \left( \frac{C' k^2 d}{n} \right)^{1/2} \exp\left\{ -C_5^2 \frac{n}{k} \left( \frac{C' k^2 d}{n} \right)^{1/2} \right\} \leq \frac{k^2 d}{n}.$$

Therefore, we plug this error back to the upper bound of $\|\Delta\|$:

$$\|\Delta\| \leq C' \frac{k^2 d}{n} + \int_{\frac{C' k^2 d}{n}}^{\infty} \Pr\left[ \|\widetilde{\Delta}\| \geq s \right] \mathrm{d}s$$

$$\leq C' \frac{k^2 d}{n} + \int_{\frac{C' k^2 d}{n}}^{\infty} \Pr\left[ \|\delta \boldsymbol{S}\| \geq \min\{ \frac{s^{1/2}}{C_1 k}, \frac{s^{1/(k+2)}}{C_1 k} \} \right] \mathrm{d}s = O\left( \frac{k^2 d}{n} \right) \leq O\left( \frac{1}{\log^3 d} \right).$$

Finally, since by Lemma D.8, we know the error is in the form $\Delta = \alpha_1 \boldsymbol{\Lambda} + \alpha_2 \operatorname{tr}(\boldsymbol{\Lambda}) \boldsymbol{I}$ for all $\boldsymbol{\Lambda}$. Therefore $\alpha_1 = O\left( \frac{k^2 d}{n} \right), \alpha_2 = O\left( \frac{k^2}{n} \right)$. $\qquad\square$

**Lemma D.2.** *Suppose* $\boldsymbol{S} = \frac{1}{n} \sum_{i=1}^{n} \boldsymbol{x}_i \boldsymbol{x}_i^{\top}$, $n = \Theta(d \log^5 d), k = O(\log d), \eta = \Theta(1) \in (0.1, 0.9), \|\boldsymbol{\Lambda}\|_{op} \leq \Theta(1), \|\boldsymbol{\Gamma}\|_{op} \leq \Theta(1)$. *Then the expectation*

$$\mathbb{E}\left[ \boldsymbol{S} \boldsymbol{\Lambda} (\boldsymbol{I} - \eta \boldsymbol{S})^k \boldsymbol{\Gamma} \boldsymbol{S} \right] = (1 - \eta)^k (\boldsymbol{\Lambda} \boldsymbol{\Gamma} + \Delta),$$

*where* $\|\Delta\|_{op} = O(\frac{k^2 d}{n}) \leq O\left( \frac{1}{\log^3 d} \right)$. *Moreover, the error is in the form*

$$\Delta = \alpha_1 \boldsymbol{\Lambda} \boldsymbol{\Gamma} + \alpha_2 \operatorname{tr}(\boldsymbol{\Lambda}) \boldsymbol{\Gamma} + \alpha_3 \operatorname{tr}(\boldsymbol{\Gamma}) \boldsymbol{\Lambda} + \alpha_4 \operatorname{tr}(\boldsymbol{\Lambda}) \operatorname{tr}(\boldsymbol{\Gamma}) \boldsymbol{I} + \alpha_5 \operatorname{tr}(\boldsymbol{\Lambda} \boldsymbol{\Gamma}) \boldsymbol{I}$$

*where* $\alpha_1 = O\left( \frac{k^2 d}{n} \right), \alpha_2, \alpha_3, \alpha_5 = O\left( \frac{k^2}{n} \right), \alpha_4 = O(\frac{k^2}{nd})$.

*Proof.* Denote $\delta \boldsymbol{S} := \boldsymbol{S} - \boldsymbol{I}$. Then we expand the term $\boldsymbol{S} \boldsymbol{\Lambda} (\boldsymbol{I} - \eta \boldsymbol{S})^k \boldsymbol{\Gamma} \boldsymbol{S}$:

$$\boldsymbol{S} \boldsymbol{\Lambda} (\boldsymbol{I} - \eta \boldsymbol{S})^k \boldsymbol{\Gamma} \boldsymbol{S}$$

$$= (\boldsymbol{I} + \delta \boldsymbol{S}) \boldsymbol{\Lambda} ((1 - \eta) \boldsymbol{I} - \eta \delta \boldsymbol{S})^k \boldsymbol{\Gamma} (\boldsymbol{I} + \delta \boldsymbol{S})$$

$$= (1 - \eta)^k (\boldsymbol{I} + \delta \boldsymbol{S}) \boldsymbol{\Lambda} \left( \boldsymbol{I} - \frac{\eta}{(1 - \eta)} \delta \boldsymbol{S} \right)^k \boldsymbol{\Gamma} (\boldsymbol{I} + \delta \boldsymbol{S})$$

$$= (1 - \eta)^k (\boldsymbol{I} + \delta \boldsymbol{S}) \boldsymbol{\Lambda} \left( \boldsymbol{I} - \frac{k\eta}{(1 - \eta)} \delta \boldsymbol{S} + \binom{k}{2} \left( \frac{\eta}{1 - \eta} \right)^2 \delta \boldsymbol{S}^2 + \sum_{j=3}^{k} \binom{k}{j} \left( \frac{-\eta}{1 - \eta} \right)^j \delta \boldsymbol{S}^j \right) \boldsymbol{\Gamma}$$

$$+ (1 - \eta)^k (\boldsymbol{I} + \delta \boldsymbol{S}) \boldsymbol{\Lambda} \left( \boldsymbol{I} - \frac{k\eta}{(1 - \eta)} \delta \boldsymbol{S} + \binom{k}{2} \left( \frac{\eta}{1 - \eta} \right)^2 \delta \boldsymbol{S}^2 + \sum_{j=3}^{k} \binom{k}{j} \left( \frac{-\eta}{1 - \eta} \right)^j \delta \boldsymbol{S}^j \right) \boldsymbol{\Gamma} \delta \boldsymbol{S}$$

Take expectation to both sides. Note that $\mathbb{E}[\delta \boldsymbol{S}] = 0$, so all the first order term vanish. We denote

$$(1 - \eta)^k \widetilde{\Delta} = \boldsymbol{S} \boldsymbol{\Lambda} (\boldsymbol{I} - \eta \boldsymbol{S})^k \boldsymbol{\Gamma} \boldsymbol{S} - (1 - \eta)^k \left( \boldsymbol{\Lambda} + \delta \boldsymbol{S} \cdot \boldsymbol{\Lambda} \boldsymbol{\Gamma} + \boldsymbol{\Lambda} \boldsymbol{\Gamma} \cdot \delta \boldsymbol{S} - \frac{k\eta}{1 - \eta} \boldsymbol{\Lambda} \cdot \delta \boldsymbol{S} \boldsymbol{\Gamma} \right),$$

which denotes all the higher order terms (the degree of $\delta \boldsymbol{S} \geq 2$.)

We can estimate the expectation using similar technique as in Lemma D.1. First, given $s = \sqrt{d}$ and $\|\delta \boldsymbol{S}\| \leq \max\left( \delta, \delta^2 \right) = C \sqrt{\frac{d}{n}}$ (since $n = \Theta(d \log^5 d)$), we upper bound the operator norm of $\widetilde{\Delta}$:

$$\|\widetilde{\Delta}\|_{op} \leq \left\| \boldsymbol{\Lambda} \left( \binom{k}{2} \left( \frac{\eta}{1 - \eta} \right)^2 \delta \boldsymbol{S}^2 + \sum_{j=3}^{k} \binom{k}{j} \left( \frac{-\eta}{1 - \eta} \right)^j \delta \boldsymbol{S}^j \right) \boldsymbol{\Gamma} \right\|_{op}$$

$$+ \left\| \delta \boldsymbol{S} \boldsymbol{\Lambda} \left( -\frac{k\eta}{(1 - \eta)} \delta \boldsymbol{S} + \binom{k}{2} \left( \frac{\eta}{1 - \eta} \right)^2 \delta \boldsymbol{S}^2 + \sum_{j=3}^{k} \binom{k}{j} \left( \frac{-\eta}{1 - \eta} \right)^j \delta \boldsymbol{S}^j \right) \boldsymbol{\Gamma} \right\|_{op}$$

$$+ \left\| \delta \boldsymbol{S} \boldsymbol{\Lambda} \left( \boldsymbol{I} - \frac{k\eta}{(1-\eta)} \delta \boldsymbol{S} + \binom{k}{2} \left( \frac{\eta}{1-\eta} \right)^2 \delta \boldsymbol{S}^2 + \sum_{j=3}^{k} \binom{k}{j} \left( \frac{-\eta}{1-\eta} \right)^j \delta \boldsymbol{S}^j \right) \boldsymbol{\Gamma} \delta \boldsymbol{S} \right\|_{op}$$

$$+ \left\| \boldsymbol{\Lambda} \left( -\frac{k\eta}{(1-\eta)} \delta \boldsymbol{S} + \binom{k}{2} \left( \frac{\eta}{1-\eta} \right)^2 \delta \boldsymbol{S}^2 + \sum_{j=3}^{k} \binom{k}{j} \left( \frac{-\eta}{1-\eta} \right)^j \delta \boldsymbol{S}^j \right) \boldsymbol{\Gamma} \delta \boldsymbol{S} \right\|_{op}$$

Now upper bound all matrices with their operator norm and combine all terms with the same degree of $\delta \boldsymbol{S}$. We have

$$\|\widetilde{\Delta}\|_{op} \leq \sum_{j=2}^{k+2} \|\boldsymbol{\Gamma}\| \|\boldsymbol{\Lambda}\| \left( \binom{k}{j} \left( \frac{\eta}{1-\eta} \right)^j + 2 \binom{k}{j-1} \left( \frac{\eta}{1-\eta} \right)^{j-1} + \binom{k}{j-2} \left( \frac{\eta}{1-\eta} \right)^{j-2} \right) \|\delta \boldsymbol{S}\|^j$$

$$\leq \sum_{j=2}^{k+2} \|\boldsymbol{\Gamma}\| \|\boldsymbol{\Lambda}\| \left( (9k)^j + 2(9k)^{j-1} + (9k)^{j-2} \right) \|\delta \boldsymbol{S}\|^j \qquad (\tfrac{\eta}{1-\eta} \leq 9, \binom{k}{j} \leq k^j.)$$

$$\leq 4 \sum_{j=2}^{k+2} \|\boldsymbol{\Gamma}\| \|\boldsymbol{\Lambda}\| \cdot (9k)^j \left( C\sqrt{\frac{d}{n}} \right)^j \qquad (\|\delta \boldsymbol{S}\| \leq C\sqrt{\tfrac{d}{n}}.)$$

$$\leq 4 \|\boldsymbol{\Lambda}\| \|\boldsymbol{\Gamma}\| \cdot \frac{81 C^2 k^2 d}{n} \cdot \frac{1}{1 - \left( \frac{9k d^{1/2}}{n^{1/2}} \right)} \leq C' \frac{k^2 d}{n} \leq O\left( \frac{1}{\log^3 d} \right)$$

Now upper bound the operator norm of the error term $\Delta := \mathbb{E}[\widetilde{\Delta}]$. Suppose $\boldsymbol{u} := \arg\max_{\boldsymbol{u}:\|\boldsymbol{u}\|=1} \frac{\|\Delta \boldsymbol{u}\|}{\|\boldsymbol{u}\|}$, then the operator norm becomes:

$$\|\Delta\| = \left| \boldsymbol{u}^\top \mathbb{E}[\widetilde{\Delta}] \boldsymbol{u} \right|$$

$$= \mathbb{E}\left[ \left| \boldsymbol{u}^\top \widetilde{\Delta} \boldsymbol{u} \right| \left( \mathbb{1}\left\{ \|\widetilde{\Delta}\| \leq C' \frac{k^2 d}{n} \right\} + \mathbb{1}\left\{ \|\widetilde{\Delta}\| > C' \frac{k^2 d}{n} \right\} \right) \right]$$

$$\leq C' \frac{k^2 d}{n} + \int_{\frac{C' k^2 d}{n}}^{\infty} \Pr\left[ \|\widetilde{\Delta}\| \geq s \right] \mathrm{d}s$$

When $\|\widetilde{\Delta}\| \geq s$ where $s \geq \frac{C' k^2 d}{n}$, there exists some constant $C_1 > 0$ s.t.

$$\|\widetilde{\Delta}\| \leq \max\left( (C_1 k \|\delta \boldsymbol{S}\|)^2, (C_1 k \|\delta \boldsymbol{S}\|)^{k+2} \right).$$

Therefore, when $\|\widetilde{\Delta}\| \geq s$, $\|\delta \boldsymbol{S}\| \geq \min\{ \frac{s^{1/2}}{C_1 k}, \frac{s^{1/(k+2)}}{C_1 k} \}$. Like Lemma D.1, applying the tail bound (18) with $s' \leq \min\left\{ C_2 \frac{s^{1/(k+2)}\sqrt{n}}{k}, C_3 \frac{s^{1/(2k+4)}\sqrt{n}}{\sqrt{k}}, C_4 \frac{\sqrt{sn}}{k}, C_5 \frac{s^{1/4}\sqrt{n}}{\sqrt{k}} \right\}$ where $C_2, C_3, C_4, C_5$ are some constant, we have the error term for the tail expectation

$$\int_{\frac{C' k^2 d}{n}}^{\infty} \Pr\left[ \|\widetilde{\Delta}\| \geq s \right] \mathrm{d}s \leq \int_{\frac{C' k^2 d}{n}}^{\infty} \Pr\left[ \|\delta \boldsymbol{S}\| \geq \min\{ \frac{s^{1/2}}{C_1 k}, \frac{s^{1/(k+2)}}{C_1 k} \} \right] \mathrm{d}s$$

$$\leq 2 \int_{\frac{C' k^2 d}{n}}^{\infty} \exp\{ -s'^2 \} \mathrm{d}s.$$

Use the exact same argument, $2 \int_{\frac{C' k^2 d}{n}}^{\infty} \exp\{ -s'^2 \} \mathrm{d}s \leq \frac{k^2 d}{n}$. Thus, the upper bound of $\|\Delta\|$ is:

$$\|\Delta\| \leq C' \frac{k^2 d}{n} + \int_{\frac{C' k^2 d}{n}}^{\infty} \Pr\left[ \|\widetilde{\Delta}\| \geq s \right] \mathrm{d}s$$

$$\leq C' \frac{k^2 d}{n} + \int_{\frac{C' k^2 d}{n}}^{\infty} \Pr\left[ \|\delta \boldsymbol{S}\| \geq \min\{ \frac{s^{1/2}}{C_1 k}, \frac{s^{1/(k+2)}}{C_1 k} \} \right] \mathrm{d}s = O\left( \frac{k^2 d}{n} \right) \leq O\left( \frac{1}{\log^3 d} \right).$$

Finally by Lemma D.8, we know the error is in the form $\Delta = \alpha_1 \boldsymbol{\Lambda}\boldsymbol{\Gamma} + \alpha_2 \operatorname{tr}(\boldsymbol{\Lambda})\boldsymbol{\Gamma} + \alpha_3 \operatorname{tr}(\boldsymbol{\Gamma})\boldsymbol{\Lambda} + \alpha_4 \operatorname{tr}(\boldsymbol{\Lambda})\operatorname{tr}(\boldsymbol{\Gamma})\boldsymbol{I}$ for all $\boldsymbol{\Lambda}, \boldsymbol{\Gamma}$. Therefore $\alpha_1 = O\left( \frac{k^2 d}{n} \right), \alpha_2, \alpha_3 = O\left( \frac{k^2}{n} \right), \alpha_4 = O(\frac{k^2}{nd})$. $\qquad \square$

**Lemma D.3.** *Suppose* $S = \frac{1}{n}\sum_{i=1}^{n} x_i x_i^{\top}$, $n = \Theta(d\log^5 d), k = O(\log d), \eta = \Theta(1) \in (0.1, 0.9), \|\Lambda\|_{op} \leq \Theta(1), \|\Gamma\|_{op} \leq \Theta(1)$. *Then the expectation*

$$\mathbb{E}\big[S\Lambda S\Gamma(I - \eta S)^k\big] = (1-\eta)^k(\Lambda\Gamma + \Delta),$$

*where* $\|\Delta\|_{op} \leq O\left(\frac{1}{\log^3 d}\right)$. *Moreover, the error is in the form*

$$\Delta = \alpha_1 \Lambda\Gamma + \alpha_2 \operatorname{tr}(\Lambda)\Gamma + \alpha_3 \operatorname{tr}(\Gamma)\Lambda + \alpha_4 \operatorname{tr}(\Lambda)\operatorname{tr}(\Gamma)I + \alpha_5 \operatorname{tr}(\Lambda\Gamma)I$$

*where* $\alpha_1 = O\left(\frac{k^2 d}{n}\right), \alpha_2, \alpha_3, \alpha_5 = O\left(\frac{k^2}{n}\right), \alpha_4 = O(\frac{k^2}{nd})$.

*Proof.* Denote $\delta S := S - I$. Then we expand the term $S\Lambda S\Gamma(I - \eta S)^k$:

$$S\Lambda S\Gamma(I - \eta S)^k$$

$$= (1-\eta)^k(I + \delta S)\Lambda(I + \delta S)\Gamma\left(I - \frac{\eta}{(1-\eta)}\delta S\right)^k$$

$$= (1-\eta)^k(I + \delta S)\Lambda\Gamma\left(I - \frac{k\eta}{(1-\eta)}\delta S + \binom{k}{2}\left(\frac{\eta}{1-\eta}\right)^2\delta S^2 + \sum_{j=3}^{k}\binom{k}{j}\left(\frac{-\eta}{1-\eta}\right)^j\delta S^j\right)$$

$$+ (1-\eta)^k(I + \delta S)\Lambda\delta S\Gamma\left(I - \frac{k\eta}{(1-\eta)}\delta S + \binom{k}{2}\left(\frac{\eta}{1-\eta}\right)^2\delta S^2 + \sum_{j=3}^{k}\binom{k}{j}\left(\frac{-\eta}{1-\eta}\right)^j\delta S^j\right)$$

Take expectation to both sides. Note that $\mathbb{E}[\delta S] = 0$, so all the first order term vanish. We denote

$$(1-\eta)^k\widetilde{\Delta} = S\Lambda(I - \eta S)^k\Gamma S - (1-\eta)^k\left(\Lambda + \delta S \cdot \Lambda\Gamma + \Lambda\delta S\Gamma - \frac{k\eta}{1-\eta}\Lambda\Gamma\cdot\delta S\right),$$

which denotes all the higher order terms (the degree of $\delta S \geq 2$.)

We can estimate the expectation using similar technique as in Lemma D.1. Given $s = \sqrt{d}$ and $\|\delta S\| \leq \max\left(\delta, \delta^2\right) = C\sqrt{\frac{d}{n}}$ (since $n = \Theta(d\log^5 d)$), we upper bound the operator norm of $\widetilde{\Delta}$. We directly expand the formula and upper bound all matrices with their operator norm and combine all terms with the same degree of $\delta S$. We have

$$\|\widetilde{\Delta}\|_{op} \leq \sum_{j=2}^{k+2}\|\Gamma\|\|\Lambda\|\left(\binom{k}{j}\left(\frac{\eta}{1-\eta}\right)^j + 2\binom{k}{j-1}\left(\frac{\eta}{1-\eta}\right)^{j-1} + \binom{k}{j-2}\left(\frac{\eta}{1-\eta}\right)^{j-2}\right)\|\delta S\|^j$$

$$\leq \sum_{j=2}^{k+2}\|\Gamma\|\|\Lambda\|\big((9k)^j + 2(9k)^{j-1} + (9k)^{j-2}\big)\|\delta S\|^j \qquad (\frac{\eta}{1-\eta} \leq 9, \binom{k}{j} \leq k^j.)$$

$$\leq 4\sum_{j=2}^{k+2}\|\Gamma\|\|\Lambda\|\cdot(9k)^j\left(C\sqrt{\frac{d}{n}}\right)^j \leq C'\frac{k^2 d}{n} \leq O\left(\frac{1}{\log^3 d}\right) \qquad (\|\delta S\| \leq C\sqrt{\frac{d}{n}}.)$$

Now upper bound the operator norm of $\Delta := \mathbb{E}[\widetilde{\Delta}]$. Suppose $u := \arg\max_{u:\|u\|=1}\frac{\|\Delta u\|}{\|u\|}$, then

$$\|\Delta\| = \mathbb{E}\left[\left|u^{\top}\widetilde{\Delta}u\right|\left(\mathbb{1}\left\{\|\widetilde{\Delta}\| \leq C'\frac{k^2 d}{n}\right\} + \mathbb{1}\left\{\|\widetilde{\Delta}\| > C'\frac{k^2 d}{n}\right\}\right)\right]$$

$$\leq C'\frac{k^2 d}{n} + \int_{\frac{C'k^2 d}{n}}^{\infty}\Pr\left[\|\widetilde{\Delta}\| \geq s\right]\mathrm{d}s$$

When $\|\widetilde{\Delta}\| \geq s$ where $s \geq \frac{C'k^2 d}{n}$, there exists some constant $C_1 > 0$ s.t.

$$\|\widetilde{\Delta}\| \leq \max\left((C_1 k\|\delta S\|)^2, (C_1 k\|\delta S\|)^{k+2}\right).$$

Therefore, when $\|\widetilde{\Delta}\| \geq s$, $\|\delta S\| \geq \min\{\frac{s^{1/2}}{C_1 k}, \frac{s^{1/(k+2)}}{C_1 k}\}$. Like Lemma D.1, applying the tail bound (18) with $s' \leq \min\left\{C_2 \frac{s^{1/(k+2)}\sqrt{n}}{k}, C_3 \frac{s^{1/(2k+4)}\sqrt{n}}{\sqrt{k}}, C_4 \frac{\sqrt{sn}}{k}, C_5 \frac{s^{1/4}\sqrt{n}}{\sqrt{k}}\right\}$ where $C_2, C_3, C_4, C_5$ are some constant, we have

$$\int_{\frac{C'k^2 d}{n}}^{\infty} \Pr\left[\|\widetilde{\Delta}\| \geq s\right] \mathrm{d}s \leq \int_{\frac{C'k^2 d}{n}}^{\infty} \Pr\left[\|\delta S\| \geq \min\{\frac{s^{1/2}}{C_1 k}, \frac{s^{1/(k+2)}}{C_1 k}\}\right] \mathrm{d}s \leq 2\int_{\frac{C'k^2 d}{n}}^{\infty} \exp\{-s'^2\} \mathrm{d}s.$$

Use the exact same argument, $2\int_{\frac{C'k^2 d}{n}}^{\infty} \exp\{-s'^2\} \mathrm{d}s \leq \frac{k^2 d}{n}$. Thus, the upper bound of $\|\Delta\|$ is:

$$\|\Delta\| \leq C'\frac{k^2 d}{n} + \int_{\frac{C'k^2 d}{n}}^{\infty} \Pr\left[\|\delta S\| \geq \min\{\frac{s^{1/2}}{C_1 k}, \frac{s^{1/(k+2)}}{C_1 k}\}\right] \mathrm{d}s = O\left(\frac{k^2 d}{n}\right) \leq O\left(\frac{1}{\log^3 d}\right).$$

Finally by Lemma D.8, we know the error is in the form $\Delta = \alpha_1 \Lambda\Gamma + \alpha_2 \operatorname{tr}(\Lambda)\Gamma + \alpha_3 \operatorname{tr}(\Gamma)\Lambda + \alpha_4 \operatorname{tr}(\Lambda)\operatorname{tr}(\Gamma)I$ for all $\Lambda, \Gamma$. Therefore $\alpha_1 = O\left(\frac{k^2 d}{n}\right), \alpha_2, \alpha_3 = O\left(\frac{k^2}{n}\right), \alpha_4 = O(\frac{k^2}{nd})$. $\qquad\square$

**Lemma D.4.** *Suppose* $S = \frac{1}{n}\sum_{i=1}^{n} x_i x_i^\top$, $n = \Theta(d\log^5 d), k = O(\log d), \eta = \Theta(1) \in (0.1, 0.9), \|\Lambda\|_{op} \leq \Theta(1)$. *Then there exists* $\delta = O\left(\frac{k^2 d}{n}\right) \leq O\left(\frac{1}{\log^3 d}\right)$, *the expectation is*

$$\mathbb{E}\left[\Lambda(I - \eta S)^k\right] = (1-\eta)^k(1+\delta)\Lambda,$$

*Proof.* Denote $\delta S := S - I$. Then we expand the term $\Lambda(I - \eta S)^k$:

$$\Lambda(I - \eta S)^k = (1-\eta)^k \Lambda\left(I - \frac{\eta}{(1-\eta)}\delta S\right)^k$$

$$= (1-\eta)^k \Lambda\left(I - \frac{k\eta}{(1-\eta)}\delta S + \binom{k}{2}\left(\frac{\eta}{1-\eta}\right)^2 \delta S^2 + \sum_{j=3}^{k}\binom{k}{j}\left(\frac{-\eta}{1-\eta}\right)^j \delta S^j\right)$$

Take expectation to both sides. Note that $\mathbb{E}[\delta S] = 0$, so all the first order term vanish. We denote

$$(1-\eta)^k \widetilde{\Delta} = \Lambda(I - \eta S)^k - (1-\eta)^k\left(\Lambda - \frac{k\eta}{1-\eta}\Lambda \cdot \delta S\right),$$

which denotes all the higher order terms (the degree of $\delta S \geq 2$.)

We can estimate the expectation using similar technique as in Lemma D.1. First, given $s = \sqrt{d}$ and $\|\delta S\| \leq \max\left(\delta, \delta^2\right) = C\sqrt{\frac{d}{n}}$ (since $n = \Theta(d\log^5 d)$), we upper bound the operator norm of $\widetilde{\Delta}$:

$$\|\widetilde{\Delta}\|_{op} \leq \left\|\Lambda\left(\binom{k}{2}\left(\frac{\eta}{1-\eta}\right)^2 \delta S^2 + \sum_{j=3}^{k}\binom{k}{j}\left(\frac{-\eta}{1-\eta}\right)^j \delta S^j\right)\right\|_{op}$$

Now upper bound all matrices by operator norm and combine all terms with the same degree of $\delta S$:

$$\|\widetilde{\Delta}\|_{op} \leq \sum_{j=2}^{k}\|\Lambda\|\left(\binom{k}{j}\left(\frac{\eta}{1-\eta}\right)^j\right)\|\delta S\|^j \leq \sum_{j=2}^{k+2}\|\Lambda\|(9k)^j\|\delta S\|^j \qquad (\tfrac{\eta}{1-\eta} \leq 9, \binom{k}{j} \leq k^j.)$$

$$\leq \|\Lambda\| \cdot \frac{81C^2 k^2 d}{n} \cdot \frac{1}{1 - (\frac{9kd^{1/2}}{n^{1/2}})} \leq C'\frac{k^2 d}{n} \leq O\left(\frac{1}{\log^3 d}\right) \qquad (\|\delta S\| \leq C\sqrt{\tfrac{d}{n}}.)$$

Now upper bound the operator norm of the error. Suppose $u := \arg\max_{u:\|u\|=1}\frac{\|\Delta u\|}{\|u\|}$, we have

$$\|\Delta\| = \left|u^\top \mathbb{E}[\widetilde{\Delta}]u\right| = \mathbb{E}\left[\left|u^\top \widetilde{\Delta}u\right|\left(\mathbb{1}\left\{\|\widetilde{\Delta}\| \leq C'\frac{k^2 d}{n}\right\} + \mathbb{1}\left\{\|\widetilde{\Delta}\| > C'\frac{k^2 d}{n}\right\}\right)\right]$$

$$\leq C'\frac{k^2 d}{n} + \int_{\frac{C'k^2 d}{n}}^{\infty} \Pr\left[\|\widetilde{\Delta}\| \geq s\right]\mathrm{d}s$$

When $\|\widetilde{\Delta}\| \geq s$ where $s \geq \frac{C'k^2d}{n}$, there exists some constant $C_1 > 0$ s.t.

$$\|\widetilde{\Delta}\| \leq \max\left((C_1 k\|\delta \boldsymbol{S}\|)^2, (C_1 k\|\delta \boldsymbol{S}\|)^{k+2}\right).$$

Therefore, when $\|\widetilde{\Delta}\| \geq s$, $\|\delta \boldsymbol{S}\| \geq \min\{\frac{s^{1/2}}{C_1 k}, \frac{s^{1/(k+2)}}{C_1 k}\}$. Like Lemma D.1, applying the tail bound (18) with $s' \leq \min\left\{C_2 \frac{s^{1/(k+2)}\sqrt{n}}{k}, C_3 \frac{s^{1/(2k+4)}\sqrt{n}}{\sqrt{k}}, C_4 \frac{\sqrt{sn}}{k}, C_5 \frac{s^{1/4}\sqrt{n}}{\sqrt{k}}\right\}$ where $C_2, C_3, C_4, C_5$ are some constant, we have the error term for the tail expectation

$$\int_{\frac{C'k^2d}{n}}^{\infty} \Pr\left[\|\widetilde{\Delta}\| \geq s\right] \mathrm{d}s \leq \int_{\frac{C'k^2d}{n}}^{\infty} \Pr\left[\|\delta \boldsymbol{S}\| \geq \min\{\frac{s^{1/2}}{C_1 k}, \frac{s^{1/(k+2)}}{C_1 k}\}\right] \mathrm{d}s \leq 2 \int_{\frac{C'k^2d}{n}}^{\infty} \exp\{-s'^2\} \mathrm{d}s.$$

Use the exact same argument, $2 \int_{\frac{C'k^2d}{n}}^{\infty} \exp\{-s'^2\} \mathrm{d}s \leq \frac{k^2d}{n}$. Thus, the upper bound of $\|\Delta\|$ is:

$$\|\Delta\| \leq C'\frac{k^2d}{n} + \int_{\frac{C'k^2d}{n}}^{\infty} \Pr\left[\|\delta \boldsymbol{S}\| \geq \min\{\frac{s^{1/2}}{C_1 k}, \frac{s^{1/(k+2)}}{C_1 k}\}\right] \mathrm{d}s = O\left(\frac{k^2d}{n}\right) \leq O\left(\frac{1}{\log^3 d}\right).$$

Finally by Lemma D.8, we know the error is in the form $\Delta = \alpha_1 \boldsymbol{\Lambda}$ for all $\boldsymbol{\Lambda}$. So $\alpha_1 = O\left(\frac{k^2d}{n}\right)$. $\qquad\square$

### D.2 Concentration lemmas for out-of-distribution data

For non-isotropic covariance Gaussian data input, we also have the concentration around the covariance $\boldsymbol{\Sigma}$ when $n = \Theta(d \log^c d)$ for $c > 0$. We still denote $\boldsymbol{S} = \frac{1}{n}\boldsymbol{X}\boldsymbol{X}^\top$. The following lemmas are involved in the calculation for the evaluation process, for in-distribution and out-of-distribution input examples $\boldsymbol{X}$.

**Lemma D.5.** *Suppose $\boldsymbol{S} = \frac{1}{n}\sum_{i=1}^n \boldsymbol{x}_i \boldsymbol{x}_i^\top$ where $\boldsymbol{x}_i \sim \mathcal{N}(\boldsymbol{0}_d, \boldsymbol{\Sigma})$, $\frac{\delta}{\eta} \leq \lambda_{\min}(\boldsymbol{\Sigma}) \leq \lambda_{\max}(\boldsymbol{\Sigma}) \leq \frac{2-\delta}{\eta}$ for some constant $\delta > 0.1$, $n = \Theta(d \log^5 d), k = O(\log d), \eta = \Theta(1) \in (0.1, 0.9)$. Then the expectation*

$$\mathrm{tr}\left(\mathbb{E}(\boldsymbol{I} - \eta \boldsymbol{S})^k\right) \leq 2d(1 - \delta)^k.$$

*Proof.* Denote $\delta \boldsymbol{S} := \boldsymbol{S} - \boldsymbol{\Sigma}$. Then we expand the term $\boldsymbol{\Lambda}(\boldsymbol{I} - \eta \boldsymbol{S})^k$:

$$(\boldsymbol{I} - \eta \boldsymbol{S})^k = (1 - \delta)^k \left(\frac{\boldsymbol{I} - \eta\boldsymbol{\Sigma}}{1 - \delta} - \frac{\eta}{1 - \delta}\delta \boldsymbol{S}\right)^k$$

$$= (1 - \delta)^k \left(\left(\frac{\boldsymbol{I} - \eta\boldsymbol{\Sigma}}{1 - \delta}\right)^k - \frac{k\eta}{(1 - \delta)}\left(\frac{\boldsymbol{I} - \eta\boldsymbol{\Sigma}}{1 - \delta}\right)^{k-1}\delta \boldsymbol{S} + \sum_{j=2}^k \binom{k}{j}\left(\frac{\boldsymbol{I} - \eta\boldsymbol{\Sigma}}{1 - \delta}\right)^{k-j}\left(\frac{-\eta}{1 - \delta}\right)^j \delta \boldsymbol{S}^j\right)$$

Take expectation to both sides. Note that $\mathbb{E}[\delta \boldsymbol{S}] = 0$, so all the first order term vanish. We denote

$$(1 - \delta)^k \widetilde{\Delta} = (\boldsymbol{I} - \eta \boldsymbol{S})^k - (1 - \delta)^k \left(\left(\frac{\boldsymbol{I} - \eta\boldsymbol{\Sigma}}{1 - \delta}\right)^k - \frac{k\eta}{(1 - \delta)}\left(\frac{\boldsymbol{I} - \eta\boldsymbol{\Sigma}}{1 - \delta}\right)^{k-1}\delta \boldsymbol{S}\right),$$

which denotes all the higher order terms (the degree of $\delta \boldsymbol{S} \geq 2$). Note $\left\|\frac{\boldsymbol{I} - \eta\boldsymbol{\Sigma}}{1 - \delta}\right\|_{op} \leq 1$.

We can estimate the expectation using similar technique as in Lemma D.1. First, given $s = \sqrt{d}$ and $\|\delta \boldsymbol{S}\| \leq \max\left(\delta, \delta^2\right) = C\sqrt{\frac{d}{n}}$ (since $n = \Theta(d \log^5 d)$), we upper bound the operator norm of $\widetilde{\Delta}$:

$$\|\widetilde{\Delta}\|_{op} \leq \left\|\sum_{j=2}^k \binom{k}{j}\left(\frac{\boldsymbol{I} - \eta\boldsymbol{\Sigma}}{1 - \delta}\right)^{k-j}\left(\frac{-\eta}{1 - \delta}\right)^j \delta \boldsymbol{S}^j\right\|_{op}$$

Now upper bound all matrices by operator norm and combine all terms with the same degree of $\delta \boldsymbol{S}$:

$$\|\widetilde{\Delta}\|_{op} \leq \sum_{j=2}^k \left(\binom{k}{j}\left(\frac{\eta}{1 - \delta}\right)^j\right)\|\delta \boldsymbol{S}\|^j \leq \sum_{j=2}^{k+2}(9k)^j\|\delta \boldsymbol{S}\|^j \qquad (\frac{\eta}{1-\eta} \leq 9, \binom{k}{j} \leq k^j.)$$

$$\leq \frac{81C^2 k^2 d}{n} \cdot \frac{1}{1 - \left(\frac{9kd^{1/2}}{n^{1/2}}\right)} \leq C' \frac{k^2 d}{n} \leq O\left(\frac{1}{\log^3 d}\right) \qquad (\|\delta\boldsymbol{S}\| \leq C\sqrt{\tfrac{d}{n}}.)$$

Now upper bound the operator norm of the error. Suppose $\boldsymbol{u} := \arg\max_{\boldsymbol{u}:\|\boldsymbol{u}\|=1} \frac{\|\Delta\boldsymbol{u}\|}{\|\boldsymbol{u}\|}$, we have

$$\|\Delta\| = \left|\boldsymbol{u}^\top \mathbb{E}[\widetilde{\Delta}]\boldsymbol{u}\right| = \mathbb{E}\left[\left|\boldsymbol{u}^\top \widetilde{\Delta}\boldsymbol{u}\right|\left(\mathbb{1}\left\{\|\widetilde{\Delta}\| \leq C'\frac{k^2 d}{n}\right\} + \mathbb{1}\left\{\|\widetilde{\Delta}\| > C'\frac{k^2 d}{n}\right\}\right)\right]$$

$$\leq C'\frac{k^2 d}{n} + \int_{\frac{C' k^2 d}{n}}^{\infty} \Pr\left[\|\widetilde{\Delta}\| \geq s\right]\mathrm{d}s$$

When $\|\widetilde{\Delta}\| \geq s$ where $s \geq \frac{C' k^2 d}{n}$, there exists some constant $C_1 > 0$ s.t.

$$\|\widetilde{\Delta}\| \leq \max\left((C_1 k\|\delta\boldsymbol{S}\|)^2, (C_1 k\|\delta\boldsymbol{S}\|)^{k+2}\right).$$

Therefore, when $\|\widetilde{\Delta}\| \geq s$, $\|\delta\boldsymbol{S}\| \geq \min\{\frac{s^{1/2}}{C_1 k}, \frac{s^{1/(k+2)}}{C_1 k}\}$. Like Lemma D.1, applying the tail bound (18) with $s' \leq \min\left\{C_2 \frac{s^{1/(k+2)}\sqrt{n}}{k}, C_3 \frac{s^{1/(2k+4)}\sqrt{n}}{\sqrt{k}}, C_4 \frac{\sqrt{sn}}{k}, C_5 \frac{s^{1/4}\sqrt{n}}{\sqrt{k}}\right\}$ where $C_2, C_3, C_4, C_5$ are some constant, we have the error term for the tail expectation

$$\int_{\frac{C' k^2 d}{n}}^{\infty} \Pr\left[\|\widetilde{\Delta}\| \geq s\right]\mathrm{d}s \leq \int_{\frac{C' k^2 d}{n}}^{\infty} \Pr\left[\|\delta\boldsymbol{S}\| \geq \min\{\frac{s^{1/2}}{C_1 k}, \frac{s^{1/(k+2)}}{C_1 k}\}\right]\mathrm{d}s \leq 2\int_{\frac{C' k^2 d}{n}}^{\infty} \exp\{-s'^2\}\mathrm{d}s.$$

Use the exact same argument, $2\int_{\frac{C' k^2 d}{n}}^{\infty} \exp\{-s'^2\}\mathrm{d}s \leq \frac{k^2 d}{n}$. Thus, the upper bound of $\|\Delta\|$ is:

$$\|\Delta\| \leq C'\frac{k^2 d}{n} + \int_{\frac{C' k^2 d}{n}}^{\infty} \Pr\left[\|\delta\boldsymbol{S}\| \geq \min\{\frac{s^{1/2}}{C_1 k}, \frac{s^{1/(k+2)}}{C_1 k}\}\right]\mathrm{d}s = O\left(\frac{k^2 d}{n}\right) \leq O\left(\frac{1}{\log^3 d}\right) < \frac{1}{2}.$$

Finally, the absolute value of the trace should be upper bounded by

$$\mathrm{tr}\left((1-\delta)^k\left(\left(\frac{\boldsymbol{I} - \eta\boldsymbol{\Sigma}}{1-\delta}\right)^k + \Delta\right)\right) \leq 2d(1-\delta)^k.$$

$\square$

The next lemma deals with the prediction error.

**Lemma D.6.** *Suppose $\boldsymbol{S} = \frac{1}{n}\sum_{i=1}^n \boldsymbol{x}_i \boldsymbol{x}_i^\top$ where $\boldsymbol{x}_i \sim \mathcal{N}(\boldsymbol{0}_d, \boldsymbol{\Sigma})$, and the covariance matrix satisfies $\frac{\delta}{\eta} \leq \lambda_{\min}(\boldsymbol{\Sigma}) \leq \lambda_{\max}(\boldsymbol{\Sigma}) \leq \frac{2-\delta}{\eta}$ for some constant $\delta > 0$. Assume $n = \Theta(d\log^5 d), k = O(\log d), \eta = \Theta(1) \in (0.1, 0.9)$. Denote that $\boldsymbol{A} := \widetilde{\boldsymbol{V}} + \eta\boldsymbol{I}, \boldsymbol{B} := \widetilde{\boldsymbol{W}} - \boldsymbol{I}, \|\boldsymbol{A}\|_{op}, \|\boldsymbol{B}\|_{op} \leq \Theta(d^{-c})$. Then for any $i < k$,*

$$\mathbb{E}\left\|(\boldsymbol{I} + \widetilde{\boldsymbol{V}}\boldsymbol{S}\widetilde{\boldsymbol{W}})^i(\boldsymbol{w}_{k-i} + \widetilde{\boldsymbol{V}}\boldsymbol{S}(\widetilde{\boldsymbol{W}}\boldsymbol{w}_{k-i} - \boldsymbol{w}^*) - \boldsymbol{w}_{k-i+1}))\right\|^2 \leq O\left(\frac{(1-\delta)^{2i}}{d^{-2c+1}}\right)$$

*Proof.* We will adopt a similar method as we did throughout Lemma D.1 to Lemma D.4.

First, we expand the left hand side loss:

$$\mathbb{E}\left\|(\boldsymbol{I} + \widetilde{\boldsymbol{V}}\boldsymbol{S}\widetilde{\boldsymbol{W}})^i(\boldsymbol{w}_{k-i} + \widetilde{\boldsymbol{V}}\boldsymbol{S}(\widetilde{\boldsymbol{W}}\boldsymbol{w}_{k-i} - \boldsymbol{w}^*) - \boldsymbol{w}_{k-i+1}))\right\|^2$$

$$= \mathbb{E}\left\|(\boldsymbol{I} + \widetilde{\boldsymbol{V}}\boldsymbol{S}\widetilde{\boldsymbol{W}})^i\left((\widetilde{\boldsymbol{V}}\boldsymbol{S}\widetilde{\boldsymbol{W}} + \eta\boldsymbol{S})(\boldsymbol{I} - (\boldsymbol{I} - \eta\boldsymbol{S})^{k-i})\boldsymbol{w}^* - (\widetilde{\boldsymbol{V}} + \eta\boldsymbol{I})\boldsymbol{S}\boldsymbol{w}^*\right)\right\|^2$$

$$= \mathbb{E}\left\|(\boldsymbol{I} + \widetilde{\boldsymbol{V}}\boldsymbol{S}\widetilde{\boldsymbol{W}})^i\left((\widetilde{\boldsymbol{V}}\boldsymbol{S}\widetilde{\boldsymbol{W}} + \eta\boldsymbol{S})(\boldsymbol{I} - (\boldsymbol{I} - \eta\boldsymbol{S})^{k-i}) - (\widetilde{\boldsymbol{V}} + \eta\boldsymbol{I})\boldsymbol{S}\right)\right\|_F^2$$

$$\leq d \cdot \mathbb{E}\left\|(\boldsymbol{I} + \widetilde{\boldsymbol{V}}\boldsymbol{S}\widetilde{\boldsymbol{W}})^i\left((\widetilde{\boldsymbol{V}}\boldsymbol{S}\widetilde{\boldsymbol{W}} + \eta\boldsymbol{S})(\boldsymbol{I} - (\boldsymbol{I} - \eta\boldsymbol{S})^{k-i}) - (\widetilde{\boldsymbol{V}} + \eta\boldsymbol{I})\boldsymbol{S}\right)\right\|_{op}^2$$

The second equation is due to $\boldsymbol{w}_i = (\boldsymbol{I} - (\boldsymbol{I} - \eta\boldsymbol{S})^i)$, and we arranged to stress the error terms. The third line is because $\mathbb{E}[\boldsymbol{w}^*\boldsymbol{w}^{*\top}] = \boldsymbol{I}$. The last line is $\|\cdot\|_F \leq \sqrt{d}\|\cdot\|_{op}$.

Now we expand each term of the expression within the operator norm into $\boldsymbol{A}, \boldsymbol{B}, \boldsymbol{I}$, and $\boldsymbol{S}$:

$$\boldsymbol{I} + \widetilde{\boldsymbol{V}}\boldsymbol{S}\widetilde{\boldsymbol{W}} = \boldsymbol{I} - \eta\boldsymbol{S} + \boldsymbol{A}\boldsymbol{S}\boldsymbol{B} - \eta\boldsymbol{S}\boldsymbol{B} + \boldsymbol{A}\boldsymbol{S}.$$

$$\widetilde{\boldsymbol{V}}\boldsymbol{S}\widetilde{\boldsymbol{W}} + \eta\boldsymbol{S} = \boldsymbol{A}\boldsymbol{S}\boldsymbol{B} - \eta\boldsymbol{S}\boldsymbol{B} + \boldsymbol{A}\boldsymbol{S}, \widetilde{\boldsymbol{V}} + \eta\boldsymbol{I} = \boldsymbol{A}.$$

Therefore the formula becomes (consider each term separately)

$$\left(\boldsymbol{I} + \widetilde{\boldsymbol{V}}\boldsymbol{S}\widetilde{\boldsymbol{W}}\right)^i = (\boldsymbol{I} - \eta\boldsymbol{S} + \boldsymbol{A}\boldsymbol{S}\boldsymbol{B} - \eta\boldsymbol{S}\boldsymbol{B} + \boldsymbol{A}\boldsymbol{S})^i$$

$$\left((\widetilde{\boldsymbol{V}}\boldsymbol{S}\widetilde{\boldsymbol{W}} + \eta\boldsymbol{S})(\boldsymbol{I} - (\boldsymbol{I} - \eta\boldsymbol{S})^{k-i}) - (\widetilde{\boldsymbol{V}} + \eta\boldsymbol{I})\boldsymbol{S}\right)$$

$$= (-\eta\boldsymbol{S}\boldsymbol{B} + \boldsymbol{A}\boldsymbol{S}\boldsymbol{B})(\boldsymbol{I} - (\boldsymbol{I} - \eta\boldsymbol{S})^{k-i}) - \boldsymbol{A}\boldsymbol{S}(\boldsymbol{I} - \eta\boldsymbol{S})^{k-i}$$

We still denote $\delta\boldsymbol{S} = \boldsymbol{S} - \boldsymbol{\Sigma}$. We first consider when the concentration holds, a.k.a $\|\delta\boldsymbol{S}\| \leq C\sqrt{\frac{d}{n}}$. Since $\|\boldsymbol{A}\|, \|\boldsymbol{B}\| \leq O(d^{-c})$, their error are dominated by $C\sqrt{\frac{d}{n}}$. We reduce this case to the previous Lemma D.5. Therefore we can upper bound the expression by

$$\left\|\boldsymbol{I} + \widetilde{\boldsymbol{V}}\boldsymbol{S}\widetilde{\boldsymbol{W}}\right\|^i = \|\boldsymbol{I} - \eta\boldsymbol{S} + \boldsymbol{A}\boldsymbol{S}\boldsymbol{B} - \eta\boldsymbol{S}\boldsymbol{B} + \boldsymbol{A}\boldsymbol{S}\|^i \leq \frac{3}{2}(1-\delta)^i$$

$$\left\|(-\eta\boldsymbol{S}\boldsymbol{B} + \boldsymbol{A}\boldsymbol{S}\boldsymbol{B})(\boldsymbol{I} - (\boldsymbol{I} - \eta\boldsymbol{S})^{k-i}) - \boldsymbol{A}\boldsymbol{S}(\boldsymbol{I} - \eta\boldsymbol{S})^{k-i}\right\| \leq O(d^{-c}).$$

That means this part of the expectation is upper bounded by $d \cdot \frac{9}{4}(1-\delta)^2 i \cdot O(d^{-2c}) = O\left(\frac{(1-\delta)^{2i}}{d^{-2c+1}}\right)$

Then we estimate the tail expectation. We first upper bound the above formula by $\|\delta\boldsymbol{S}\|$:

$$\left\|\boldsymbol{I} + \widetilde{\boldsymbol{V}}\boldsymbol{S}\widetilde{\boldsymbol{W}}\right\|^i = \|\boldsymbol{I} - \eta\boldsymbol{S} + \boldsymbol{A}\boldsymbol{S}\boldsymbol{B} - \eta\boldsymbol{S}\boldsymbol{B} + \boldsymbol{A}\boldsymbol{S}\|^i \leq O(k(1-\delta)^i \min\{\|\delta\boldsymbol{S}\|, 1\}^i)$$

$$\left\|(-\eta\boldsymbol{S}\boldsymbol{B} + \boldsymbol{A}\boldsymbol{S}\boldsymbol{B})(\boldsymbol{I} - (\boldsymbol{I} - \eta\boldsymbol{S})^{k-i}) - \boldsymbol{A}\boldsymbol{S}(\boldsymbol{I} - \eta\boldsymbol{S})^{k-i}\right\| \leq O(kd^{-c}\min\{1, \|\delta\boldsymbol{S}\|^{k-i}\}).$$

Use the same argument as in Lemma D.1 to calculate the integral of tail bound, the tail expectation can also be upper bounded by $O\left(\frac{(1-\delta)^{2i}}{d^{-2c+1}}\right)$. Combine those two part and we finish the proof.

$\square$

## D.3 THE FORM OF EXPECTATION

**Lemma D.7.** *Suppose* $\boldsymbol{S} = \frac{1}{n}\sum_{i=1}^n \boldsymbol{x}_i\boldsymbol{x}_i^\top$, *then the expectation is in the following form for any k:*

$$\mathbb{E}\left[\boldsymbol{S}\boldsymbol{u}_s\boldsymbol{u}_s^\top\boldsymbol{S}^k\boldsymbol{u}_t\boldsymbol{u}_t^\top\boldsymbol{S}^{k'}\right] = \alpha_1\boldsymbol{u}_s\boldsymbol{u}_s^\top + \alpha_2\boldsymbol{u}_t\boldsymbol{u}_t^\top + \alpha_3\boldsymbol{I} \text{ for any } s \neq t.$$

$$\mathbb{E}\left[\boldsymbol{S}\boldsymbol{u}_s\boldsymbol{u}_s^\top\boldsymbol{S}^k\boldsymbol{u}_s\boldsymbol{u}_s^\top\boldsymbol{S}^{k'}\right] = \alpha_4\boldsymbol{u}_s\boldsymbol{u}_s^\top + \alpha_5\boldsymbol{I}.$$

*Proof.* We notice that by changing the basis to $\{\boldsymbol{u}_s\}_{s=1}^d$,

$$\mathbb{E}\left[\boldsymbol{S}\boldsymbol{u}_s\boldsymbol{u}_s^\top\boldsymbol{S}^k\boldsymbol{u}_t\boldsymbol{u}_t^\top\boldsymbol{S}^{k'}\right] = \boldsymbol{U}\mathbb{E}\left[(\boldsymbol{U}^\top\boldsymbol{S}\boldsymbol{U})\boldsymbol{e}_s\boldsymbol{e}_s^\top(\boldsymbol{U}^\top\boldsymbol{S}\boldsymbol{U})^k\boldsymbol{e}_t\boldsymbol{e}_t^\top(\boldsymbol{U}^\top\boldsymbol{S}\boldsymbol{U})^{k'}\right]\boldsymbol{U}^\top. \tag{19}$$

Define $\hat{\boldsymbol{x}}_i = \boldsymbol{U}^\top\boldsymbol{x}_i$. Since gaussian is isotropic, we have $\mathbb{E}[\hat{\boldsymbol{x}}_i] = \boldsymbol{U}^\top\mathbb{E}[\boldsymbol{x}_i] = \boldsymbol{0}$. After we change the basis, the covariance matrix of $\hat{\boldsymbol{x}}_i$ should also be the same:

$$\text{Cov}(\hat{\boldsymbol{x}}_i) = \boldsymbol{U}^\top\text{Cov}(\boldsymbol{x}_i)\boldsymbol{U} = \boldsymbol{I}.$$

Therefore $\hat{\boldsymbol{x}}_i$ has the same distribution as $\boldsymbol{x}_i$ and we have

$$\boldsymbol{U}\mathbb{E}\left[(\boldsymbol{U}^\top\boldsymbol{S}\boldsymbol{U})\boldsymbol{e}_s\boldsymbol{e}_s^\top(\boldsymbol{U}^\top\boldsymbol{S}\boldsymbol{U})^k\boldsymbol{e}_t\boldsymbol{e}_t^\top(\boldsymbol{U}^\top\boldsymbol{S}\boldsymbol{U})^{k'}\right]\boldsymbol{U}^\top = \boldsymbol{U}\mathbb{E}\left[\boldsymbol{S}\boldsymbol{e}_s\boldsymbol{e}_s^\top\boldsymbol{S}^k\boldsymbol{e}_t\boldsymbol{e}_t^\top\boldsymbol{S}^{k'}\right]\boldsymbol{U}^\top.$$

Subsequently, we only need to consider the expectation of $\boldsymbol{S}\boldsymbol{e}_s\boldsymbol{e}_s^\top \boldsymbol{S}^k \boldsymbol{e}_t \boldsymbol{e}_t^\top \boldsymbol{S}^{k'}$. Decompose $\boldsymbol{x}_i$ into the sum of basis vectors and we get $\boldsymbol{x}_i = \sum_{j=1}^d x_{ij}\boldsymbol{e}_j$.

Plug in the decomposition into the expectation and we have

$$
n^{k+2}\mathbb{E}\Big[\boldsymbol{S}\boldsymbol{e}_s\boldsymbol{e}_s^\top \boldsymbol{S}^k \boldsymbol{e}_t \boldsymbol{e}_t^\top \boldsymbol{S}^{k'}\Big]
$$

$$
= \mathbb{E}\Bigg[ \Bigg( \sum_{i_0=1}^n \sum_{j_0,j_1\in[d]} x_{i_0 j_0} x_{i_0 j_1} \boldsymbol{e}_{j_0}\boldsymbol{e}_{j_1}^\top \Bigg) \boldsymbol{e}_s\boldsymbol{e}_s^\top \prod_{l=1}^{k'} \Bigg( \sum_{i_l=1}^n \sum_{j_{2l},j_{2l+1}\in[d]} x_{i_l j_{2l}} x_{i_l j_{2l+1}} \boldsymbol{e}_{j_{2l}}\boldsymbol{e}_{j_{2l+1}}^\top \Bigg)
$$

$$
\boldsymbol{e}_t\boldsymbol{e}_t^\top \prod_{l=1}^{k} \Bigg( \sum_{i_l=1}^n \sum_{j_{2l},j_{2l+1}\in[d]} x_{i_l j_{2l}} x_{i_l j_{2l+1}} \boldsymbol{e}_{j_{2l}}\boldsymbol{e}_{j_{2l+1}}^\top \Bigg) \Bigg]
$$

$$
= \mathbb{E}\Bigg[ \sum_{i_0,\cdots,i_{k+k'}\in[n]} \sum_{j_0,\cdots,j_{2(k+k')+1}\in[d]} x_{i_0 j_0} x_{i_0 j_1} \cdots x_{i_{k+k'} j_{2(k+k')}} x_{i_{k+k'} j_{2(k+k')+1}}
$$

$$
\boldsymbol{e}_{j_0}\boldsymbol{e}_{j_1}^\top \boldsymbol{e}_s\boldsymbol{e}_s^\top \boldsymbol{e}_{j_2}\boldsymbol{e}_{j_3}^\top \cdots \boldsymbol{e}_{j_{2k}}\boldsymbol{e}_{j_{2k+1}}^\top \boldsymbol{e}_t\boldsymbol{e}_t^\top \boldsymbol{e}_{j_{2k+2}}\boldsymbol{e}_{j_{2k+3}}^\top \cdots \boldsymbol{e}_{j_{2(k+k')}}\boldsymbol{e}_{j_{2(k+k')+1}}^\top \Bigg]
$$

$$
= \sum_{i_0,\cdots,i_{k+k'}\in[n]} \sum_{j_0,\cdots,j_{2(k+k')+1}\in[d]} \mathbb{E}\Big[ x_{i_0 j_0} x_{i_0 j_1} \cdots x_{i_{k+k'} j_{2(k+k')}} x_{i_{k+k'} j_{2(k+k')+1}} \Big]
$$

$$
\boldsymbol{e}_{j_0}\boldsymbol{e}_{j_1}^\top \boldsymbol{e}_s\boldsymbol{e}_s^\top \boldsymbol{e}_{j_2}\boldsymbol{e}_{j_3}^\top \cdots \boldsymbol{e}_{j_{2k}}\boldsymbol{e}_{j_{2k+1}}^\top \boldsymbol{e}_t\boldsymbol{e}_t^\top \boldsymbol{e}_{j_{2k+2}}\boldsymbol{e}_{j_{2k+3}}^\top \cdots \boldsymbol{e}_{j_{2(k+k')}}\boldsymbol{e}_{j_{2(k+k')+1}}^\top.
$$

Note that $\boldsymbol{e}_a^\top \boldsymbol{e}_b \neq 0$ only when $a=b$, so $\boldsymbol{e}_a^\top \boldsymbol{e}_s\boldsymbol{e}_s^\top \boldsymbol{e}_b \neq 0$ only when $a=b=s$. Therefore, we only need to consider the case where $j_{2q-1} = j_{2q}$ for any $q \in [1, k+k']$. By symmetry, we know $\mathbb{E}\Big[\boldsymbol{S}\boldsymbol{e}_s\boldsymbol{e}_s^\top \boldsymbol{S}^k \boldsymbol{e}_t \boldsymbol{e}_t^\top \boldsymbol{S}^{k'}\Big]$ is a diagonal matrix, so we have $j_0 = j_{2(k+k')+1}$. We denote

$$
\boldsymbol{E}_{j_0} = \boldsymbol{e}_{j_0}\boldsymbol{e}_{j_1}^\top \boldsymbol{e}_s\boldsymbol{e}_s^\top \boldsymbol{e}_{j_1}\boldsymbol{e}_{j_2}^\top \cdots \boldsymbol{e}_{j_k}\boldsymbol{e}_{j_{k+1}}^\top \boldsymbol{e}_t\boldsymbol{e}_t^\top \boldsymbol{e}_{j_{k+1}}\boldsymbol{e}_{j_{k+2}}^\top \cdots \boldsymbol{e}_{j_{k+k'}}\boldsymbol{e}_{j_0}^\top
$$

to be one of the standard basis in $\mathbb{R}^{d\times d}$ space. It is a non-zero matrix when $j_1 = s$ and $j_{k+1} = t$. By the analysis above, we have

$$
n^{k+2}\mathbb{E}\Big[\boldsymbol{S}\boldsymbol{e}_s\boldsymbol{e}_s^\top \boldsymbol{S}^k \boldsymbol{e}_t \boldsymbol{e}_t^\top \boldsymbol{S}^{k'}\Big] = \sum_{i_0,\cdots,i_{k+k'}\in[n]} \sum_{j_0,\cdots,j_{k+k'}\in[d]} \mathbb{E}\Big[ x_{i_0 j_0} x_{i_0 j_1} \cdots x_{i_{k+k'} j_{k+k'}} x_{i_{k+k'} j_0} \Big] \boldsymbol{E}_{j_0}.
$$

Let $\mathcal{P}(2k+2)$ be the set of all distinct ways of partitioning $\{i_0 j_0, i_0 j_1 \cdots, i_{k+k'} j_{k+k'}, i_{k+k'} j_0\}$ into $k+1$ unordered pairs $p = ((p_1, p_2), \cdots, (p_{2k+1}, p_{2k+2}))$. From Isserlis' theorem, we have

$$
\mathbb{E}\Big[ x_{i_0 j_0} x_{i_0 j_1} \cdots x_{i_{k+k'} j_{k+k'}} x_{i_{k+k'} j_0} \Big] = \sum_{p\in\mathcal{P}(2k+2)} \prod_{i=0}^{k+k'} \mathbb{E}\big[ x_{p_{2i}} x_{p_{2i+1}} \big].
$$

Plug it in the expectation and we have

$$
n^{k+2}\mathbb{E}\Big[\boldsymbol{S}\boldsymbol{e}_s\boldsymbol{e}_s^\top \boldsymbol{S}^k \boldsymbol{e}_t \boldsymbol{e}_t^\top \boldsymbol{S}^{k'}\Big] = \sum_{i_0,\cdots,i_{k+k'}\in[n]} \sum_{j_0,\cdots,j_{k+k'}\in[d]} \sum_{p\in\mathcal{P}(2k+2)} \prod_{i=0}^{k+k'} \mathbb{E}\big[ x_{p_{2i}} x_{p_{2i+1}} \big] \boldsymbol{E}_{j_0}
$$

$$
= \sum_{p\in\mathcal{P}(2k+2)} \sum_{i_0,\cdots,i_k\in[n]} \sum_{j_0,\cdots,j_k\in[d]} \prod_{i=0}^{k+k'} \mathbb{E}\big[ x_{p_{2i}} x_{p_{2i+1}} \big] \boldsymbol{E}_{j_0}.
$$

To make sure the term in the summation is non-zero, $p_{2q-1} = p_{2q}$ should hold for any $1 \leq q \leq k+1$. Now consider the graph $\mathcal{G}_p$ and $\mathcal{G}'_p$ with vertices $\{0, 1, \cdots, k+k'\}$. If $i_{u_1}j_{v_1}$ is paired with $i_{u_2}j_{v_2}$, then we put an edge between $u_1$ and $u_2$ into $\mathcal{G}_p$ and put an edge between $v_1$ and $v_2$ into $\mathcal{G}'_p$, which means $i_{u_1} = i_{u_2}$ and $j_{v_1} = j_{v_2}$. Therefore, for a cycle $C = (u_1, u_2, \cdots, u_r)$ in $\mathcal{G}_p$ or $\mathcal{G}'_p$, we have $i_{u_1} = i_{u_2} = \cdots = i_{u_r}$ or $j_{u_1} = j_{u_2} = \cdots = j_{u_r}$. Note that we have $n$ or $d$ choices for the value of the circle. Here we use $\mathrm{C}(\cdot)$ to denote the set of circles in the graph and use $|\mathrm{C}(\cdot)|$ to denote the number of circles in the graph. Let $c^*$ be the cycle in $\mathcal{G}'_p$ which includes the vertex $j_0$.

**Case 1:** $s \neq t$. For the partition $p$ where $j_1 \in c^*$ and $j_{k+1} \in c \neq c^*$, there is only one choice for $c$ and $c^*$ to take. So the term in the summation should be $n^{|\,C(\mathcal{G}_p)|}d^{|\,C(\mathcal{G}'_p)|-2}\boldsymbol{e}_s\boldsymbol{e}_s^\top$. Similarly, for the partition $p$ where $j_{k+1} \in c^*$ and $j_1 \in c \neq c^*$, the term in the summation should be $n^{|\,C(\mathcal{G}_p)|}d^{|\,C(\mathcal{G}'_p)|-2}\boldsymbol{e}_t\boldsymbol{e}_t^\top$. For the partition $p$ where $j_1 \in c' \neq c^*$ and $j_{k+1} \in c'' \neq c^*$, there is only one choice for $c'$ and $c''$ to take. Therefore, the expectation should be

$$
n^{k+2}\mathbb{E}\Big[\boldsymbol{S}\boldsymbol{e}_s\boldsymbol{e}_s^\top\boldsymbol{S}^k\boldsymbol{e}_t\boldsymbol{e}_t^\top\boldsymbol{S}^{k'}\Big]
$$
$$
= \sum_{\mathcal{P}:j_1\in c^*,j_{k+1}\notin c^*} n^{|\,C(\mathcal{G}_p)|}d^{|\,C(\mathcal{G}'_p)|-2}\boldsymbol{e}_s\boldsymbol{e}_s^\top + \sum_{\mathcal{P}:j_{k+1}\in c^*,j_1\notin c^*} n^{|\,C(\mathcal{G}_p)|}d^{|\,C(\mathcal{G}'_p)|-2}\boldsymbol{e}_t\boldsymbol{e}_t^\top
$$
$$
+ \sum_{\mathcal{P}:j_1,j_{k+1}\notin c^*} n^{|\,C(\mathcal{G}_p)|}d^{|\,C(\mathcal{G}'_p)|-2}\boldsymbol{e}_{j_0}\boldsymbol{e}_{j_0}^\top
$$
$$
= \sum_{\mathcal{P}:j_1\in c^*,j_{k+1}\notin c^*} n^{|\,C(\mathcal{G}_p)|}d^{|\,C(\mathcal{G}'_p)|-2}\boldsymbol{e}_s\boldsymbol{e}_s^\top + \sum_{\mathcal{P}:j_{k+1}\in c^*,j_1\notin c^*} n^{|\,C(\mathcal{G}_p)|}d^{|\,C(\mathcal{G}'_p)|-2}\boldsymbol{e}_t\boldsymbol{e}_t^\top
$$
$$
+ \sum_{\mathcal{P}:j_1,j_{k+1}\notin c^*} n^{|\,C(\mathcal{G}_p)|}d^{|\,C(\mathcal{G}'_p)|-3}\boldsymbol{I}.
$$

Recall Equation (19), we prove that

$$
\mathbb{E}\Big[\boldsymbol{S}\boldsymbol{u}_s\boldsymbol{u}_s^\top\boldsymbol{S}^k\boldsymbol{u}_t\boldsymbol{u}_t^\top\boldsymbol{S}^{k'}\Big] = \alpha_1\boldsymbol{u}_s\boldsymbol{u}_s^\top + \alpha_2\boldsymbol{u}_t\boldsymbol{u}_t^\top + \alpha_3\boldsymbol{I}.
$$

**Case 2:** $s = t$. For the partition $p$ where $j_1, j_{k+1} \in c^*$, there is only one choice for $c^*$ to take. So the term in the summation should be $n^{|\,C(\mathcal{G}_p)|}d^{|\,C(\mathcal{G}'_p)|-1}\boldsymbol{e}_s\boldsymbol{e}_s^\top$. For the partition $p$ where $j_1 \in c^*$ and $j_{k+1} \in c \neq c^*$, there is only one choice for $c$ and $c^*$ to take. So the term in the summation should be $n^{|\,C(\mathcal{G}_p)|}d^{|\,C(\mathcal{G}'_p)|-2}\boldsymbol{e}_s\boldsymbol{e}_s^\top$. Similarly, for the partition $p$ where $j_{k+1} \in c^*$ and $j_1 \in c \neq c^*$, the term in the summation should be $n^{|\,C(\mathcal{G}_p)|}d^{|\,C(\mathcal{G}'_p)|-2}\boldsymbol{e}_s\boldsymbol{e}_s^\top$. For the partition $p$ where $j_1 \in c' \neq c^*$ and $j_{k+1} \in c'' \neq c^*$, there is only one choice for $c'$ and $c''$ to take. Therefore, the expectation should be

$$
n^{k+2}\mathbb{E}\Big[\boldsymbol{S}\boldsymbol{e}_s\boldsymbol{e}_s^\top\boldsymbol{S}^k\boldsymbol{e}_s\boldsymbol{e}_s^\top\boldsymbol{S}^{k'}\Big]
$$
$$
= \sum_{\mathcal{P}:j_1,j_{k+1}\in c^*} n^{|\,C(\mathcal{G}_p)|}d^{|\,C(\mathcal{G}'_p)|-1}\boldsymbol{e}_s\boldsymbol{e}_s^\top + \sum_{\mathcal{P}:j_1\in c^*,j_{k+1}\notin c^*} n^{|\,C(\mathcal{G}_p)|}d^{|\,C(\mathcal{G}'_p)|-2}\boldsymbol{e}_s\boldsymbol{e}_s^\top
$$
$$
+ \sum_{\mathcal{P}:j_{k+1}\in c^*,j_1\notin c^*} n^{|\,C(\mathcal{G}_p)|}d^{|\,C(\mathcal{G}'_p)|-2}\boldsymbol{e}_s\boldsymbol{e}_s^\top + \sum_{\mathcal{P}:j_1,j_{k+1}\notin c^*} n^{|\,C(\mathcal{G}_p)|}d^{|\,C(\mathcal{G}'_p)|-2}\boldsymbol{e}_{j_0}\boldsymbol{e}_{j_0}^\top
$$
$$
= \Bigg[\sum_{\mathcal{P}:j_1,j_{k+1}\in c^*} n^{|\,C(\mathcal{G}_p)|}d^{|\,C(\mathcal{G}'_p)|-1} + \sum_{\mathcal{P}:j_1\in c^*,j_{k+1}\notin c^*} n^{|\,C(\mathcal{G}_p)|}d^{|\,C(\mathcal{G}'_p)|-2}
$$
$$
+ \sum_{\mathcal{P}:j_{k+1}\in c^*,j_1\notin c^*} n^{|\,C(\mathcal{G}_p)|}d^{|\,C(\mathcal{G}'_p)|-2}\Bigg]\boldsymbol{e}_s\boldsymbol{e}_s^\top + \sum_{\mathcal{P}:j_1,j_{k+1}\notin c^*} n^{|\,C(\mathcal{G}_p)|}d^{|\,C(\mathcal{G}'_p)|-3}\boldsymbol{I}.
$$

Recall Equation (19), we prove that

$$
\mathbb{E}\Big[\boldsymbol{S}\boldsymbol{u}_s\boldsymbol{u}_s^\top\boldsymbol{S}^k\boldsymbol{u}_s\boldsymbol{u}_s^\top\boldsymbol{S}^{k'}\Big] = \alpha_4\boldsymbol{u}_s\boldsymbol{u}_s^\top + \alpha_5\boldsymbol{I}.
$$

Hence, the proof is complete. $\qquad\square$

**Lemma D.8.** *Suppose* $\boldsymbol{S} = \frac{1}{n}\sum_{i=1}^n \boldsymbol{x}_i\boldsymbol{x}_i^\top$, *then the expectation is in the following form for any $k$:*

$$
\mathbb{E}\Big[\boldsymbol{S}\boldsymbol{\Lambda}\boldsymbol{S}^k\boldsymbol{\Gamma}\boldsymbol{S}^{k'}\Big] = \beta_1\boldsymbol{\Lambda}\boldsymbol{\Gamma} + \beta_2\operatorname{tr}(\boldsymbol{\Lambda})\boldsymbol{\Gamma} + \beta_3\operatorname{tr}(\boldsymbol{\Gamma})\boldsymbol{\Lambda} + \beta_4\operatorname{tr}(\boldsymbol{\Lambda})\operatorname{tr}(\boldsymbol{\Gamma})\boldsymbol{I} + \beta_5\operatorname{tr}(\boldsymbol{\Lambda}\boldsymbol{\Gamma})\boldsymbol{I}.
$$

*where* $\boldsymbol{\Lambda} = \sum_{j=1}^d \lambda_j^{\boldsymbol{\Lambda}}\boldsymbol{u}_j\boldsymbol{u}_j^\top, \boldsymbol{\Gamma} = \sum_{j=1}^d \lambda_j^{\boldsymbol{\Gamma}}\boldsymbol{u}_j\boldsymbol{u}_j^\top.$

*Proof.* By lemma D.7, we have:

$$\mathbb{E}\left[\boldsymbol{S}\boldsymbol{\Lambda}\boldsymbol{S}^k\boldsymbol{\Gamma}\boldsymbol{S}^{k'}\right]$$

$$= \sum_{j=1}^{d}\sum_{i\neq j}^{d}\lambda_i^{\boldsymbol{\Lambda}}\lambda_j^{\boldsymbol{\Gamma}}\left(\alpha_1\boldsymbol{u}_i\boldsymbol{u}_i^{\top} + \alpha_2\boldsymbol{u}_j\boldsymbol{u}_j^{\top} + \alpha_3\boldsymbol{I}\right) + \sum_{i=1}^{d}\lambda_i^{\boldsymbol{\Lambda}}\lambda_i^{\boldsymbol{\Gamma}}\left(\alpha_4\boldsymbol{u}_i\boldsymbol{u}_i^{\top} + \alpha_5\boldsymbol{I}\right)$$

The first term here can be expand into the following form:

$$\sum_{j=1}^{d}\sum_{i\neq j}^{d}\lambda_i^{\boldsymbol{\Lambda}}\lambda_j^{\boldsymbol{\Gamma}}\left(\alpha_1\boldsymbol{u}_i\boldsymbol{u}_i^{\top} + \alpha_2\boldsymbol{u}_j\boldsymbol{u}_j^{\top} + \alpha_3\boldsymbol{I}\right)$$

$$= \alpha_1\operatorname{tr}(\boldsymbol{\Gamma})\boldsymbol{\Lambda} + \alpha_2\operatorname{tr}(\boldsymbol{\Lambda})\boldsymbol{\Gamma} + \alpha_3\operatorname{tr}(\boldsymbol{\Lambda})\operatorname{tr}(\boldsymbol{\Gamma})\boldsymbol{I} - (\alpha_1 + \alpha_2)\boldsymbol{\Lambda}\boldsymbol{\Gamma} - \alpha_3\operatorname{tr}(\boldsymbol{\Lambda}\boldsymbol{\Gamma})\boldsymbol{I}$$

Meanwhile, the second term is directly $\alpha_4\boldsymbol{\Lambda}\boldsymbol{\Gamma} + \alpha_5\operatorname{tr}(\boldsymbol{\Lambda}\boldsymbol{\Gamma})\boldsymbol{I}$. We pick $\beta_2 = \alpha_1, \beta_3 = \alpha_2, \beta_4 = \alpha_3, \beta_1 = \alpha_4 - \alpha_1 - \alpha_2, \beta_5 = \alpha_5 - \alpha_3$, and we complete the proof.

$\square$

# E    EXPERIMENTAL DETAILS

For all our experiments, we use pytorch Paszke et al. (2019) and models are trained on an NVIDIA RTX A6000. Each experiment takes about 1 hour.

**Setup**    In all our experiments, we choose $d = 10$, $n = 20$ and $\eta = 0.4$. The architecture is

$$f_{\text{LSA}}(\boldsymbol{Z}; \boldsymbol{V}, \boldsymbol{W})_{[:,-1]} = \boldsymbol{Z}_{[:,-1]} + \boldsymbol{V}\boldsymbol{Z} \cdot \frac{\boldsymbol{Z}^\top \boldsymbol{W} \boldsymbol{Z}_{[:,-1]}}{n}$$

and data is drawn from the distribution in Equation (1). The batch size $B$ is 1000 and the learning rate $\alpha$ is 0.001. The total time is $\tau = 750$ iterations. In the first experiment, $k$ is chosen as 20 while $k = 10, 20, 30, 40$ in the second experiment. The baseline (evaluation loss of transformers without CoT) is given by Corollary 3.1 where $\eta^* = \frac{n}{n+d+1}$:

$$\mathcal{L}^{\text{Eval}}(\boldsymbol{V}, \boldsymbol{W}) \geq \frac{1}{2}\left(d - 2\eta^* d + \frac{\eta^{*2}}{n}(n + d + 1)d\right)$$

**In-distribution Generalization**    We empirically verify the evaluation loss gap between transformers with and without CoT shown by Theorem 3.1 and Theorem 3.2. Our experiments in Figure 2 demonstrate that the evaluation loss of transformers with CoT converges to near zero even when $k = 10$. See Section 5 for details.

**Out-of-distribution Generalization**    In addition, we empirically verify the OOD generalization result shown by Theorem 4.2. We sample 10 different covariance matrices from the distribution which complies to

$$\frac{\delta}{\eta} \leq \lambda_{\min}(\boldsymbol{\Sigma}) \leq \lambda_{\max}(\boldsymbol{\Sigma}) \leq \frac{2 - \delta}{\eta}$$

where $\eta = 0.4$ and $\eta = 0.4$. 10 experiments are taken to show the generality of our results for each set of experiment. Our experiment in Figure 3 exhibits that the OOD loss of transformers with CoT converges to near zero when $k = 10, 20, 30, 40$ as the training loss/in-distribution loss converges to zero. The final loss also drops when the number of reasoning steps increases.

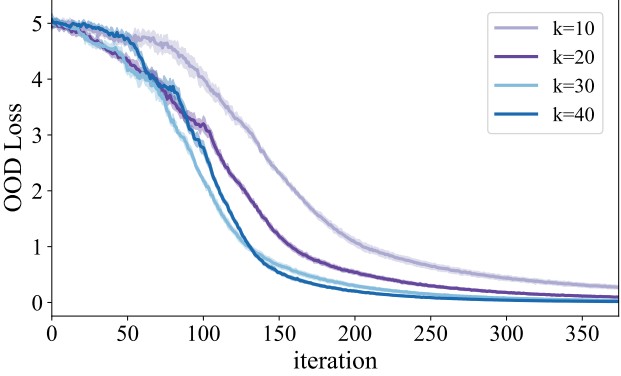

Figure 3: **OOD Generalization:** We plot the OOD loss $\mathcal{L}_{\boldsymbol{\Sigma}}^{\text{Eval}}$ when $n = 20$, $d = 10$. Each set of experiments sampled 10 different $\boldsymbol{\Sigma}$. The mean results are presented as line charts, with variance represented by shaded areas. As shown, OOD loss will converge to near zero.

Given all experiments above, we conclude that transformers with CoT can converge to our construction (Theorem 4.1), surpass those without CoT (Corollary 3.1, Theorem 3.2) and generalize well to unseen data (Theorem 4.2).

