# OpenReview forum: "Transformers Learn to Implement Multi-step Gradient Descent with Chain of Thought"
_ICLR.cc/2025/Conference — ICLR 2025 Spotlight_

### Official Review · Reviewer_D4BQ · 2024-11-03

**Soundness:** 3
**Presentation:** 3
**Contribution:** 3
**Rating:** 8
**Confidence:** 4

**Summary:**

This paper introduces a novel in-context learning task, termed in-context weight prediction for linear regression. The authors demonstrate that single-layer transformers exhibit limitations in this task when working with limited number of examples, and they show that incorporating a chain-of-thought mechanism significantly enhances performance. Through analysis of gradient dynamics in single-layer transformers using chain-of-thought, they establish convergence results that align well with their empirical findings.

**Strengths:**

While existing literature always try to understand the expressive power of CoT from a computational complexity perspective, this work offers a fresh analytical framework, and they provide convergence analysis and sufficient  experiments to validate their findings. The approach is both innovative and provides valuable insights for future research directions.

**Weaknesses:**

Although this work presents interesting findings, the construction and proof techniques largely adapt existing methodologies. Furthermore, similar results might be achievable using looped transformers as demonstrated in [1], where:
$$
Z_0 = \begin{pmatrix}
x_0 & x_n& x_{query} \\\\
y_0 & y_n& 0
\end{pmatrix}, Z_k = f(Z_{k-1})
$$
where $f$ is a linear transformer with fixed parameters. In other words, while existing work demonstrate looped transformer (inference a (single layer) fixed parameter transformer with $k$ times can get a similar form $w^{k} x_{query}$, where $w^k$  defined like theorem 3.2 in this work, the theoretical contribution that directly analysis $w^k$ seems limited. So a more detailed discussion between this work and analysis in [1], and a meanfull explaination why such a in-context learning task framework is necessary, is desired.

[1] Ding, N., Levinboim, T., Wu, J., Goodman, S., & Soricut, R. (2023). CausalLM is not optimal for in-context learning. arXiv preprint arXiv:2308.06912.

Note that [1] analysis a multilayer transformer with the same parameters as each layer, so I refer this work as looped transformers, other works like [2,3] also analysis similar transformers.

[2] Ahn, K., Cheng, X., Daneshmand, H., & Sra, S. (2023). Transformers learn to implement preconditioned gradient descent for in-context learning. Advances in Neural Information Processing Systems, 36, 45614-45650.

[3] Gatmiry, K., Saunshi, N., Reddi, S. J., Jegelka, S., & Kumar, S. (2024). Can Looped Transformers Learn to Implement Multi-step Gradient Descent for In-context Learning?. arXiv preprint arXiv:2410.08292.

**Questions:**

Please refer to the weakness section, the reviewer would like to better understand how the authors reconcile the apparent disconnect between CoT steps in practical applications (where more steps don't necessarily yield better performance) and their theoretical analysis. Additionally, how can the insights from this work inform the design of effective step-by-step instructions for improved CoT performance?

---

> ### Author Response · Authors · 2024-11-18
> **Response to Reviewer D4BQ (Part 1)**
>
> Dear Reviewer, we sincerely appreciate your constructive and insightful feedback on our work. Your efforts in helping us enhance our paper are greatly valued, and we would be happy to provide clarification if you have any additional questions.
> > ***Weakness 1: Although this work presents interesting findings, the construction and proof techniques largely adapt existing methodologies. Furthermore, similar results might be achievable using looped transformers as demonstrated in [1], ...***
>
> **A1:** Thanks the reviewer for pointing out the related work [1] and we updated it in the current version. We agree that the construction that serves as a warm-up definitely adapts existing methodologies, but the setting of CoT, the stage-wise training dynamics analysis and final inference time analysis are novel contributions. Moreover, our technique involves a novel characterization of the higher moments of
> Wishart matrices to estimate the population gradient more accurately.
>
> First, we considered this more practical auto-regressive CoT loss instead of the common "sequence to token" loss in previous ICL works. The previous multi-layer transformer setting for ICL cannot support auto-regressive generation/reasoning, which is disconnected from the practice. The in-context weight prediction task is also a novel setting compared to all previous linear ICL works. Altogether, they introduce additional hardness in analyzing the gradient flow training dynamics, so we applied a stage-wise dynamics analysis compared to the previous analyses based on landscape property (e.g. [3], [4]).
>
> We agree that a multi-layer/looped transformer can probably also *express* the multi-step GD on our task, as implicit (preconditioned) GD can be applied in [1][2]. However, there is no guarantee that any gradient-based algorithm can learn this solution. Our work and [3] both try to overcome this gap in a different way. [3] adopts the looped transformer to implicitly do multi-step GD, which is not auto-regressive and more similar to a multi-layer setting. Compared to them, we use a simpler model but utilize the auto-regressive generation, which is closer to practice.
>
> As we pointed out in the discussion (Appendix A.2), [3] cannot theoretically outperform the one-step GD due to technical issues. Our novel analysis makes our paper the first to theoretically exhibit that transformers can learn to implement multi-step GD with CoT and outperform the model without CoT (where one-step GD is optimal). With our novel technique estimating the higher moments of Wishart matrices, we believe results in [3] can also be improved.
>
>
>
> In all, we believe our theoretical contribution does not rely on existing methodologies and we try to take one step closer to the practice.
>
> [4] Zhang R, Frei S, Bartlett P L. Trained transformers learn linear models in-context[J]. arXiv preprint arXiv:2306.09927, 2023.
>
> > ***Weakness2: A more detailed discussion between this work and analysis in [1], and a meanfull explanation why such a in-context learning task framework is necessary, is desired.***
>
> A2: Thanks the reviewer for the constructive suggestions. We added a more comprehensive discussion section in the appendix. We adopt the in-context learning task framework as a testbed of understanding the transformers' reasoning capability by utilizing chain-of-thought, especially on those "iterative tasks" like gradient descent. We believe it is a necessary initial step to understand the practical applications of CoT.
>
> (to be continued)

---

> ### Author Response · Authors · 2024-11-18
> **Response to Reviewer D4BQ (Part 2)**
>
> > ***Question 1: ...the reviewer would like to better understand how the authors reconcile the apparent disconnect between CoT steps in practical applications...and their theoretical analysis. Additionally, how can the insights from this work inform the design of effective step-by-step instructions for improved CoT performance?***
>
> **A3:** We thank the reviewer for bringing up such an insightful question. In practical LLMs, reasoning is a very complicated process that requires all kinds of skills in dealing with languages. We think there exist both cases, where more steps yield better performance or not. For those iterative tasks where the same skill is applied, we believe more steps lead to better performance. For example, one can prompt GPT-4o or o1-preview to do gradient descent/power method/numerical algorithms, and more steps definitely get better accuracy.
>
> **Prompt:** Now I will give you $X$ and $y$ which satisfy $y=X^\top w$. I want you to do 10 steps of GD with $w_0=0$ and $\eta=0.1$ not using coding, just using reasoning. Please output every $w$ you get.
>
> Using the prompt above, we designate one pair of $X$ and $y$ to get the following result for reference:
>
> | Iteration |   1  |   2  |   3  |   4  | 5    | 6    | 7    | 8    | 9    | 10   |
> |:---------:|:----:|:----:|:----:|:----:|------|------|------|------|------|------|
> | **MSE Loss**  | 2.68 | 2.37 | 2.09 | 1.84 | 1.61 | 1.40 | 1.21 | 1.04 | 0.88 | 0.73 |
> | $\lVert w_i-w^* \rVert$     | 2.48 | 2.28 | 2.11 | 1.94 | 1.79 | 1.65 | 1.53 | 1.41 | 1.30 | 1.20 |
>
> As the result shows, LLM clearly performs better on the GD task with more steps.
>
> In this work, we mainly focus on these kinds of "iterative" tasks, and we believe our setting is one of the simplest forms where multi-step CoT can help yield better performance. That serves as the initial step towards understanding why CoT helps reasoning following the first principle. It is potentially possible that other language models cannot do those tasks accurately with more steps. That is probably because there are too many confounding effects and LLMs cannot perform perfectly on some specific tasks. In comparison, transformers trained on synthetic datasets can do those perfectly because they are task-specific.
>
> We also believe that CoT can empower the transformer to acquire compositional reasoning capability instead of doing the same iterative step. It is a much harder question beyond our paper's scope, so we won't claim our work can reconcile these cases where composing different skills is required. However, as pointed out in our conclusion, it is a very important future direction and definitely worth further exploring.
>
> As for the final question, our main focus is to theoretically draw a learnable separation between the transformer with and without CoT, rigorously showing the CoT prompting method can empower multi-step reasoning and lead to improved performance in some cases. Though we don't think the theoretical understanding can directly apply to the practice (which is beyond this paper's scope), our work suggests precise characterization of the intermediate steps of CoT is necessary. For example, if intermediate steps are wrong or biased, the transformer may collapse and fail to learn the algorithm even with CoT.
>
>
> We are eager to hear about your further comments. Please let us know if there is anything else we can clarify.

---

> > ### Comment · Reviewer_D4BQ · 2024-11-18
> >
> > I thank the authors for their responses and additional experiments. I will increase my rating from 6 to 8 as I support accepting this work.

---

> > > ### Author Response · Authors · 2024-11-22
> > >
> > > Thank you very much for the thoughtful feedback and suggestions! We truly appreciate it.

---

### Official Review · Reviewer_Kfz9 · 2024-11-04

**Soundness:** 3
**Presentation:** 2
**Contribution:** 2
**Rating:** 8
**Confidence:** 3

**Summary:**

The authors investigate the training dynamics of transformers trained with a Chain of Thought (CoT) objective, specifically in an in-context weight prediction task for linear regression. Under certain assumptions, it proves that a one-layer transformer without CoT training can only perform a single gradient descent step, resulting in suboptimal recovery of the weight vector. However, with CoT training, the transformer can execute multi-step gradient descent, enabling near-exact recovery and some out-of-distribution generalization. The authors provide theoretical results demonstrating the global convergence of the training via gradient flow and empirical evidence showcasing the superior performance of transformers trained with CoT compared to those without.

**Strengths:**

This paper provides some theoretical analysis on CoT, which is lacking and deserves much more attention. These seem to be new results on CoT, albeit in the setting of a one-layer linear transformer.

**Weaknesses:**

-I’m not sure about the significance of these results. The comparison here involves a transformer trained on a CoT objective, which seems to be unusual. It’s entirely possible (and likely) that CoT data is contained in the large pretraining corpuses of today’s LLMs, but it remains a next-token prediction task. There is no difference between the training objectives of an LLM trained on text with or without CoT.

-The results are for one-layer linear transformers. This seems to be a common model choice in the literature for obvious reasons, but it deviates so far from the setup of an LLM that the utility of these results is questionable.

-What is the significance of assuming the number of samples to be bounded by dlog^5d? It seems like a contrived assumption to make the proofs work. For example, couldn’t Corollary 3.1 instead be framed as saying at least a quadratic number of samples are needed to control the evaluation error?

-The main theorem 4.1 relies on seemingly restrictive assumptions. Even though these are settings used in previous works, can you say anything in a more general setting?

-While the theorems may be new, it is hard to tell whether there is any novelty in the methods or whether these are straightforward extensions of techniques used in other works the authors mentioned such as Bai et al.. The manuscript would benefit from clearly outlining where novel ideas are needed.

-Theorem 4.2 is confusing. Can the authors provide an intuitive explanation for why the model exhibits OOD generalization within the provided bounds of the spectrum of the covariance matrix? Also, I don’t think L^{eval}_{\Sigma} is defined anywhere.

-The manuscript would benefit from empirical analysis for Theorem 4.2.

-In general, the manuscript would benefit from some intuitive explanations of the assumptions in the theorems. Furthermore, while a theoretical analysis of CoT is interesting, I don’t think the results are very impactful.

**Questions:**

See the weaknesses.

---

> ### Author Response · Authors · 2024-11-18
> **Response to Reviewer Kfz9 (Part 1)**
>
> Dear reviewer, thank you very much for your constructive and insightful comments on our work. We appreciate your effort to help us improve our paper, and we are glad to clarify if you have any further questions.
> > ***Weakness 1: I’m not sure about the significance of these results ... There is no difference between the training objectives of an LLM trained on text with or without CoT.***
>
> **A1:** Thanks for the question! The reviewer seems to assume we are comparing with or without CoT both under the auto-regressive generation setting. But the actual comparison in the paper is between training "with CoT" under a more practical next-token prediction setting, and the previous theoretical ICL papers without considering auto-regression. As mentioned in ([1], [2], etc.), the chain-of-thought/scratchpad method exactly shows the capability of the auto-regressive paradigm compared to direct output from an expressiveness perspective, and this work generalizes the insights from expressivity to optimization. Otherwise, the number of serial computations is intuitively bounded by the depth of the transformer [1]. Here, we follow their terminology and call "CoT" a method that allows transformers to auto-regressively generate intermediate steps at inference time instead of directly outputting the answers. Similarly, training on CoT objectives is defined as
> training transformers on some auto-regressive objectives with CoT data (data with ground-truth intermediate reasoning steps) instead of data without intermediate steps.
>
> Therefore, we totally agree with the reviewer's opinion on the similarity between our CoT loss and the practical auto-regressive pretraining loss on next-token prediction tasks. Actually, what the reviewer pointed out is exactly our main theoretical contribution: compared to all the previous ICL papers where the next-token prediction concept is missing, we propose using CoT loss with practical next-token prediction to train the model on auto-regressive generation. Then we also evaluate its ability to auto-regressively generate $k$ steps, until it reaches some final prediction that is close to the ground-truth. It is provably better than the previous ICL output without auto-regressive generation. Therefore, we believe our theoretical contribution is highly nontrivial and we try to take one step closer to the practice.
>
> [1] Li Z, Liu H, Zhou D, et al. Chain of thought empowers transformers to solve inherently serial problems[J]. arXiv preprint arXiv:2402.12875, 2024.
>
> [2] Malach E. Auto-regressive next-token predictors are universal learners[J]. arXiv preprint arXiv:2309.06979, 2023.
>
> > ***Weakness 2: The results are for one-layer linear transformers. This seems to be a common model choice in the literature for obvious reasons, but it deviates so far from the setup of an LLM that the utility of these results is questionable.***
>
> **A2:** We agree with the reviewer that using a simple linear transformer is indeed a limitation. However, it is essential to understand complex systems like LLMs from the first principle in a simple, controlled way. Analyzing the linear counterpart of the model before targeting the more difficult practical models is common in the development of learning theory. As for linear attention, the connection between linear attention and softmax attention is also partially justified by the empirical observations in [3]. Furthermore, our work was among the first to provide rigorous learnable separation between the auto-regressive CoT method and directly outputting the ICL solution on some specific reasoning task. Though it remains unclear how the reasoning process in the real-world LLM setup works, we take a first step toward understanding why training on carefully designed instructions (e.g. CoT data) can help the model do multi-step reasoning (e.g. multi-step GD) compared to those methods without auto-regressive reasoning process.
>
> [3] Ahn K, Cheng X, Song M, et al. Linear attention is (maybe) all you need (to understand transformer optimization)[J]. arXiv preprint arXiv:2310.01082, 2023.
>
> (to be continued)

---

> ### Author Response · Authors · 2024-11-18
> **Response to Reviewer Kfz9 (Part 2)**
>
> > ***Weakness 3: What is the significance of assuming the number of samples to be bounded by dlog^5d? ... couldn’t Corollary 3.1 instead be framed as saying at least a quadratic number of samples are needed to control the evaluation error?***
>
> **A3:** Thanks for your criticism of the presentation and sorry for the confusion. For Corollary 3.1, actually we don't need the condition $n=\Omega(d\log^5 d)$, because this condition is only useful in the proof of training dynamics. The number of examples is plugged in because Corollary is meant to argue the optimal loss of ICL without CoT (auto-regressive generation reasoning) is far worse than the multi-step GD with the implementation of CoT. We improved the presentation for this corollary in the updated version.
>
> For the training dynamics analysis, we meant to assume that  $n=\Omega(d\log^5 d)$ as the lower bound for the examples, so $n=\Theta(d^2)$ definitely works for our proof because ${d^2}$ is asymptotically much larger than $d\log^5 d$. This assumption is technical to ensure the concentration of the Wishart matrices. It is a limitation that $\log^5 d$ might not be the tightest bound, but it is asymptotically optimal up to log factors (i.e. $\widetilde{O}(d)$ with logarithmic factors hidden). Actually, even if we switch to $n=\Omega(d^2)$, we can still ensure the separation between with/without CoT. Empirically, we also observe that only $\Theta(d)$ examples are needed, which means the assumption is only for technical reasons and can be generalized to a more practical setting.
>
> > ***Weakness 4: The main theorem 4.1 relies on seemingly restrictive assumptions. Even though these are settings used in previous works, can you say anything in a more general setting?***
>
> **A4:** Thanks for the question. All the seemingly restrictive assumptions are just technical guarantees to prove the training dynamics rigorously, but the main messages can be generalized to a more general setting. Empirically, we randomly initialize the weights, use mini-batch gradients, and only need $\Theta(d)$ examples for input sequences. Still, we can easily show that the multi-step solution with CoT can outperform ICL without CoT prompting.
>
> > ***Weakness 5: While the theorems may be new, it is hard to tell whether there is any novelty in the methods or whether these are straightforward extensions of techniques used in other works the authors mentioned such as Bai et al.. The manuscript would benefit from clearly outlining where novel ideas are needed.***
>
> **A5:** We thank the reviewer suggestion on outlining the novel ideas, and we highlight those novel ideas in our updated version. First, we want to point out that Bai et al is a theoretical work in 'expressiveness' where they constructed some multi-layer transformer that can do multi-step gradient descent. However, there is no guarantee that any trained transformer can obtain that constructed solution, and the proof does not require any training dynamics analysis, which is our work's focus. Also, the stronger expressiveness of the network could imply that one needs to put more effort into finding the solution algorithmically. Therefore, we use a completely different proof technique compared to Bai et al., so it is apparently not a straightforward extension of the techniques used in Bai et al. and that line of work.
>
> As for the novelty compared to the related ***training dynamics*** work, we listed in the updated contribution that we considered this auto-regressive loss instead of the common "sequence to token" loss. The in-context weight prediction task is also a novel setting compared to all previous linear ICL works. Altogether, they introduce additional hardness in analyzing the gradient flow training dynamics, so we applied a novel stage-wise analysis compared to the previous analyses based on landscape property. We introduced our novel training techniques in detail in section 4.
>
> > ***Weakness 6: Theorem 4.2 is confusing. Can the authors provide an intuitive explanation for why the model exhibits OOD generalization within the provided bounds of the spectrum of the covariance matrix? Also, I don’t think $L^{eval}_{\Sigma}$ is defined anywhere.***
>
> **A6:** Thanks the reviewer for the constructive feedback! We apologize for the confusion and we have already added the missing definitions and clarifications. We added the definition of $\mathcal{L}^{\mathrm{eval}}_{\Sigma}$, which is exactly the evaluation loss with $x$ sampled from the OOD distribution with covariance $\Sigma$. The assumptions mean that the covariance matrix should not be too ill-conditioned (e.g. the smallest eigenvalue is infinitesimally close to 0, or the largest eigenvalue is too large that leads to divergence).
>
> > ***Weakness 7: The manuscript would benefit from empirical analysis for Theorem 4.2.***
>
> **A7:** Thanks to the reviewer for the great suggestion. We have already included the empirical analysis in Appendix E.
>
> (to be continued)

---

> ### Author Response · Authors · 2024-11-18
> **Response to Reviewer Kfz9 (Part 3)**
>
> > ***Weakness 8: In general, the manuscript would benefit from some intuitive explanations of the assumptions in the theorems. Furthermore, while a theoretical analysis of CoT is interesting, I don’t think the results are very impactful.***
>
> **A8:** Thanks the reviewer for all the constructive suggestions and we improved our writing in the updated version. In general, the assumption for $n=\widetilde{\Omega}(d)$ is to guarantee one can fully recover the ground-truth $w^*$, and the extra $\log$-factors are due to technical issues. Note that all previous work require $n\to \infty$ to recover the ground-truth, which is much less sample-efficient than our assumption. $k=\Theta(\log d)$ is because we need at least $\log d$ steps to make the loss converge to small constant $\epsilon$ with constant learning rate GD. The constant-large learning rate $\eta$ is standard, while the specific value are for the ease of rigorous proof. In our updated version, we revised the theorems to be more intuitive instead of being formal, and defer all the formal theorems in the appendix.
>
> We are eager to hear about your further comments. Please let us know if there is anything else we can clarify.

---

> ### Comment · Reviewer_Kfz9 · 2024-11-22
>
> Thank you for the detailed responses and addressing my concerns and misunderstandings. I believe the manuscript should be accepted to this venue. I will adjust my score from 5 to 8.

---

> > ### Author Response · Authors · 2024-11-22
> >
> > Thank you so much for your recognition and thoughtful suggestions! We greatly value your help in improving our paper.

---

### Official Review · Reviewer_iVV1 · 2024-11-10

**Soundness:** 3
**Presentation:** 3
**Contribution:** 3
**Rating:** 6
**Confidence:** 3

**Summary:**

This paper studies the single-layer linear self-attention (LSA) model in the context of solving in-context weight prediction problems for linear regression tasks. Unlike standard ICL, where given a query feature $x_q$, the model is trained to predict the label $y=x_q^\top w^*$, this work trains the model to predict the task feature $w^*$ directly. As a result, the loss is evaluated based on the prediction of the task feature, defined as $\ell=||LSA(X,y)_{[:,-1]}-w^*||^2$.

1. Given input $X,y$, gradient descent (GD) with an appropriately chosen learning rate returns task feature predictions $w_0,w_1,\cdots,w_k$. The authors introduce CoT prompting by appending the intermediate GD steps $w_0,w_1,\cdots,w_k$ to the input, demonstrating that this approach reduces the loss compared to scenarios without CoT prompting.

2. Under certain initialization assumptions, the authors present convergence results using gradient flow analysis and further demonstrate that their findings can generalize to out-of-distribution (OOD) settings.

**Strengths:**

1. The paper is well-organized, and the theoretical analysis appears solid.
2. The paper introduces CoT prompting to enhance the expressivity of single-layer linear attention models in ICL tasks.

**Weaknesses:**

1. Limited explanation is provided about the training setting. As described, the ground-truth GD steps $w_0,w_1...$ are assumed to be available. To generate this data, the gradient of the data model, such as the linear model in this case, must be known. While, in standard ICL settings, only input-label pairs are required. Additionally, each $w^*$ is randomly sampled for each prompt, which implies that generating $M$ training samples would require $M$ gradient calculations over an $n\times d$-dimensional dataset, typically where $M\to\infty$.

2. The model's performance and loss are highly dependent on the learning rate $\eta$. In this work, $\eta$ is fixed, meaning that optimal losses can only be achieved by setting $\eta=\frac{n}{n+d+1}$. In standard ICL settings, $\eta$ is often implicitly learned. As a result, the definition of global minimization in the paper is somewhat ambiguous.

3. By making $n$ and $k$ dependent on $d$, it is unclear how varying values of $n$ and $k$ affect convergence and evaluation losses.

4. The CoT + one-layer approach in this paper appears closely related to ICL + multi-layer methods. The paper could benefit from a discussion on this connection.

**Questions:**

1. In some places, the notations are unclear:
    - The dimensions of model output $f$ and $w$ do not seem to align in Lines 199 and 209.
    - Is $\sigma<1/2$ in Assumption 4.1?
    - Could you clarify whether the loss in Theorem 4.1 corresponds to Eq. (7)?

2. Could the authors explain the reasoning behind setting $V_{31}$ and $W_{13}$ to have the same set of eigenvalues in Assumption 4.1?

3. Could the authors clarify in Theorem 3.1 what the model prediction is: $w^*$ or $w_1$?

4. See Weakness section.

---

> ### Author Response · Authors · 2024-11-18
> **Response to Reviewer iVV1 (Part 1)**
>
> Dear reviewer, thank you very much for your constructive and insightful comments on our work. We appreciate your effort in helping us improve our paper.
>
> > ***Weakness 1: Limited explanation is provided about the training setting ... which implies that generating $M$ training samples would require $M$ gradient calculations over an $n\times d$-dimensional dataset, typically where $M\rightarrow\infty$.***
>
> **A1:** Thanks for the interesting question from the data-generating perspective! We agree that extra effort needs to be made in the labeling process, which exactly aligns with the practice. In practice, chain-of-thought data are known to be expensive to generate, and the labeler must know the correct reasoning intermediate steps for the practical chain-of-thought tasks. In our settings, we assume that the labeler has the prior knowledge that the training data is on the linear regression task, which makes it possible to generate intermediate reasoning steps (GD steps in this case). It is different from the ICL settings in prior works since CoT data requires data with more effort in labeling. Furthermore, the data generating process for the linear regression task is rather simple since it has a simple closed-form formula to calculate the GD steps, indicating that not much additional compute is required.
>
> For the issue of $M\rightarrow \infty,$ I believe it is common among all related works since all the related works use population loss to simplify the analysis. However, one does not necessarily need infinite data to train the model. Empirically, we show that we can learn multi-step GD with limited samples. That means the labelers won't have to pay too much for additional computation. Though it is possible to analyze the sample complexity of this problem, this analysis is beyond the scope of this work and can be seen as another interesting future direction.
>
> > ***Weakness 2: ... In this work, $\eta$ is fixed... In standard ICL settings, $\eta$ is often implicitly learned. As a result, the definition of global minimization in the paper is somewhat ambiguous.***
>
> **A2:** We are sorry for the confusion but we didn't actually assume $\eta=\frac{n}{n+d+1}$ in Theorem 3.1. To clarify, the message of the theorem is: any one-layer transformer **provably cannot perform better than the one-step GD solution output:** $(0,0,\frac{\eta}{n}Xy^\top,1)$ with the learning rate $\eta=\frac{n}{n+d+1}=\frac{\mathbb{E}[wXy^\top]}{\mathbb{E}[yX^\top Xy^\top]}$. The specific value is provably optimal for the isotropic Gaussian case. If we consider the ICL setting, it is similar to Mahankali et al. [1], which proves that **an implicit one-step GD (with this learning rate $\eta$) is optimal for any one-layer transformer**. The lower bound in our proof adopts similar techniques. If there is any further confusion, please let us know and we will be happy to clarify!
>
> [1] Mahankali A, Hashimoto T B, Ma T. One step of gradient descent is provably the optimal in-context learner with one layer of linear self-attention[J]. arXiv preprint arXiv:2307.03576, 2023.
>
> > ***Weakness 3: By making $n$ and $k$ dependent on $d$, it is unclear how varying values of $n$ and $k$ affect convergence and evaluation losses.***
>
> **A3:** Thanks to the reviewer for the criticism! We are sorry for the confusion, and we will update a new version with $k,d$ included in the evaluation loss. Since we normalized $n$ before $XX^\top$, $n$ does not affect convergence/evaluation as long as $n=\Omega(d)$ to make sure one can fully recover the ground truth $w^*$ (Otherwise, $XX^\top$ is not full rank and loses information). To be specific, we only require the lower bound for $n=\Omega(d\mathrm{poly}\log d)=\widetilde{\Omega}(d)$ to ensure the concentration. Note that all previous ICL papers need $n\rightarrow \infty$ to ensure a small loss and in contrast, we need much fewer examples. The selection of $k$ can actually be an arbitrary positive integer, but we select $k=\Omega(\log d)$ to ensure after $k$-steps of GD on the linear regression objective, the prediction is close enough to the ground-truth $w^*$.
>
> > ***Weakness 4: The CoT + one-layer approach in this paper appears closely related to ICL + multi-layer methods. The paper could benefit from a discussion on this connection.***
>
> **A4:** Thanks for the suggestion! Though we already discussed the multi-layer methods of expressive power in the related works, we updated a more comprehensive discussion in the appendix (due to space limitation). To highlight our contribution, our work focuses on the **training** of the transformer, which is one large step beyond simply expressiveness.
>
> (to be continued)

---

> ### Author Response · Authors · 2024-11-18
> **Response to Reviewer iVV1 (Part 2)**
>
> >***Question 1: In some places, the notations are unclear:***
> > - The dimensions of model output $f$ and $w$ do not seem to align in Lines 199 and 209.
> > - Is $\sigma<1/2$ in Assumption 4.1?
> > - Could you clarify whether the loss in Theorem 4.1 corresponds to Eq. (7)?
>
> **A5:** (1) Thanks for catching this typo! The output of $f$ should be in the form of $(0,0,w,1)$ as the predicted next token, though the first two entries are not important. We fixed the issue in our updated version \wnote{todo}. (2) Yes, you are correct. We assumed $\lambda_i^W \in [\sigma, \frac{1}{2}]$ so $\sigma<\frac{1}{2}$. Basically, we mean that the scale of init should be a small constant. (3) The loss in Theorem 4.1 is $\mathcal{L}^{\mathrm{CoT}}$ defined in Eq. (6) as an auto-regressive loss, which is for our training stage. The evaluation loss $\mathcal{L}^{\mathrm{Eval}}$ defined in Eq. (7) is the difference between the final prediction and the ground-truth, ignoring all intermediate steps. It is for our inference(test) stage. Those two losses exactly correspond to the practical pre-training and inference stages for LLMs. We believe we explained the two different losses in section 2.3, but we would be happy to clarify if the reviewer has more questions on the setting.
>
> >***Question 2: Could the authors explain the reasoning behind setting $V_{13}$ and $W_{13}$ to have the same set of eigenvalues in Assumption 4.1?***
>
> **A6:** Thanks for the question on our technical details. We didn't assume they have the same set of eigenvalues (we believe it is a typo in the question), but we do assume they have the same eigenvectors. The assumption is to simplify the analysis of the training dynamics of the model. We believe the assumption can be slightly weakened to *singular vectors* and our dynamical system analysis still holds. However, the dynamics will be too chaotic to analyze without the eigenvector/singular vector assumption. That is why we need the assumption for theoretical analysis. Compared to previous works, most of them only need to analyze the dynamics of a matrix and a scalar. The only related work that involves multiple matrix dynamics by Chen et al. [2] also adopts the singular vector assumption.
>
> [2] Chen S, Sheen H, Wang T, et al. Training dynamics of multi-head softmax attention for in-context learning: Emergence, convergence, and optimality[J]. arXiv preprint arXiv:2402.19442, 2024.
>
> >***Question 3: Could the authors clarify in Theorem 3.1 what the model prediction is: $w^\star$ or $w_1$?***
>
> **A7:** We aim to predict $w^*$, but any one-layer transformer cannot approximate $w^*$ in context without CoT. The optimal transformer can at most output a one-step gradient approximation of $w^*$, which corresponds to some $w_1=\frac{\eta}{n} Xy^\top$ with $\eta=\frac{\mathbb{E}[wXy^\top]}{\mathbb{E}[yX^\top Xy^\top]}(=\frac{n}{n+d+1}\text{ for isotropic data})$.
>
> We are eager to hear about your further comments. Please let us know if there is anything else we can clarify.

---

> ### Author Response · Authors · 2024-11-23
> **Looking forward to hearing your feedback**
>
> Dear Reviewer iVV1:
>
> As the discussion phase of ICLR draws to a close, we would like to kindly follow up to ensure that our responses have addressed your concerns. If you have any remaining questions or concerns, we would be delighted to provide further details or clarifications. We greatly appreciate your time and feedback and look forward to hearing your thoughts.
>
> Best regards.

---

> > ### Comment · Reviewer_iVV1 · 2024-11-24
> >
> > Thank the authors for their detailed response, which addressed most of my concerns. I would like to suggest adding further clarification in the paper regarding the notion of the "global minimizer" in the context of CoT tasks. Specifically, I believe that the choice of the parameter $\eta$ used to generate intermediate weight vectors (e.g., $w_1,w_2,...$) could affect the minimal loss for the single-layer attention model. I decide to maintain my rating and lean toward acceptance.

---

> > > ### Author Response · Authors · 2024-11-24
> > >
> > > Thank you so much for your constructive and thoughtful suggestions! We agree that the choice of $\eta$ in the data-generating process will affect the final minimal CoT training loss and we will add more discussion on this notion as you requested in the updated version. We greatly appreciate your effort in improving our paper.

---

### Author Response · Authors · 2024-11-18
**Global Response by Authors**

We sincerely thank all the reviewers for their insightful and constructive comments. Following your suggestions, we have updated and revised our paper.

In this update, we have made the following changes to the paper's content:
- We have emphasized our contributions in Section 1 to clearly outline the significance of our work.
- We added a discussion on related works and limitations in Appendix A, providing a comprehensive context for our findings.
- We conducted and included empirical analyses for Theorem 4.2 in Appendix E.
We also revised the statements of all theorems for greater clarity and resolved the typographical errors identified by Reviewers iVV1 and Kfz9.

To address concerns about the novelty of our results, we highlight our novel contributions below:
-  We are the **first to introduce auto-regressive** generation in the ICL linear regression setting, enabling CoT reasoning.  We separates the next-token prediction training and inference stages similar to the pre-training and inference stages of LLMs. In contrast, previous works on linear regression ICL are limited to one-step generation.
- Our methodology combines dynamics analysis and landscape properties in a novel **stage-wise** approach. By characterizing the higher moments of Wishart matrices, we provide more accurate estimates of the population gradient.
- Additionally, we analyze the inference time error after training **with multi-step generations**, which is novel since no previous work considered auto-regressive generation.
- We are the **first to establish the learnable separation** between transformers with and without CoT/looping under the in-context linear regression setting, further showing the novelty of our approach.

We hope that the modifications can strengthen the state of our submission. Please let us know if there are any additional points that we can clarify/modify!

---

### Meta-Review · Area_Chair_ren4 · 2024-12-20

**Metareview:**

This paper studies gradient flow dynamics and loss landscape of chain of thought with 1 layer transformer. The theoretical properties of the linear attention has attracted a lot of attention in recent in-context learning theory literature. However, much of the research focuses on simple regression setting that make connection between K-step gradient descent with K-step linear attention.  This goes beyond earlier work on in-context learning analysis which mostly focus on 1 step analysis. The reviewers and I unanimously agree that this will a strong addition to the conference program.

**Additional Comments On Reviewer Discussion:**

The reviewers raised several good points such as availability of gradient iterates as the training data (i.e. type of the chain of thought data) or sensitivity to learning rate. While some assumptions could be made weaker, it doesn't takeaway much from the contribution.

---

### Decision · Program_Chairs · 2025-01-22

Accept (Spotlight)